# Continuous-time Analysis of Anchor Acceleration

**Jaewook J. Suh**
Seoul National University
jacksuhkr@snu.ac.kr

**Jisun Park**
Seoul National University
colleenp0515@snu.ac.kr

**Ernest K. Ryu**
Seoul National University
ernestryu@snu.ac.kr

## Abstract

Recently, the anchor acceleration, an acceleration mechanism distinct from Nesterov's, has been discovered for minimax optimization and fixed-point problems, but its mechanism is not understood well, much less so than Nesterov acceleration. In this work, we analyze continuous-time models of anchor acceleration. We provide tight, unified analyses for characterizing the convergence rate as a function of the anchor coefficient $\beta(t)$, thereby providing insight into the anchor acceleration mechanism and its accelerated $\mathcal{O}(1/k^2)$-convergence rate. Finally, we present an adaptive method inspired by the continuous-time analyses and establish its effectiveness through theoretical analyses and experiments.

## 1   Introduction

Nesterov acceleration [51] is foundational to first-order optimization theory, but the mechanism and its convergence proof are not transparent. One approach to better understand the mechanism is the continuous-time analysis: derive an ODE model of the discrete-time algorithm and analyze the continuous-time dynamics [65, 66]. This approach provides insight into the accelerated dynamics and has led to a series of follow-up work [71, 62, 29].

Recently, a new acceleration mechanism, distinct from Nesterov's, has been discovered. This *anchor acceleration* for minimax optimization and fixed-point problems [35, 75, 54] has been an intense subject of study, but its mechanism is understood much less than Nesterov acceleration. The various analytic techniques developed to understand Nesterov acceleration, including continuous-time analyses, have only been applied in a very limited manner [59].

**Contribution.**   In this work, we present continuous-time analyses of anchor acceleration. The continuous-time model is the differential inclusion

$$\dot{X} \in -\mathbb{A}(X) - \beta(t)(X - X_0)$$

with initial condition $X(0) = X_0 \in \mathrm{dom}\,\mathbb{A}$, maximal monotone operator $\mathbb{A}$, and scalar-valued function $\beta(t)$. The case $\beta(t) = \frac{1}{t}$ corresponds to the prior anchor-accelerated methods APPM [35], EAG [75], and FEG [42].

We first establish that the differential inclusion is well-posed, despite the anchor coefficient $\beta(t)$ blowing up at $t = 0$. We then provide tight, unified analyses for characterizing the convergence rate as a function of the anchor coefficient $\beta(t)$. This is the first formal and rigorous treatment of this anchored dynamics, and it provides insight into the anchor acceleration mechanism and its accelerated $\mathcal{O}(1/k^2)$-convergence rate. Finally, we present an adaptive method inspired by the continuous-time analyses and establish its effectiveness through theoretical analyses and experiments.

### 1.1   Preliminaries and notation

We provide the organization for prior works in Appendix A. Here, we review standard definitions and set up the notation.

37th Conference on Neural Information Processing Systems (NeurIPS 2023).

**Monotone and set-valued operators.** We follow the standard definitions of Bauschke and Combettes [15], Ryu and Yin [58]. For the underlying space, consider $\mathbb{R}^n$ with standard inner product $\langle \cdot, \cdot \rangle$ and norm $\|\cdot\|$. Define domain of $\mathbb{A}$ as $\operatorname{dom} \mathbb{A} = \{x \in \mathbb{R}^n \mid \mathbb{A}x \neq \emptyset\}$. We say $\mathbb{A}$ is an operator on $\mathbb{R}^n$ and write $\mathbb{A} \colon \mathbb{R}^n \rightrightarrows \mathbb{R}^n$ if $\mathbb{A}$ maps a point in $\mathbb{R}^n$ to a subset of $\mathbb{R}^n$. We say $\mathbb{A} \colon \mathbb{R}^n \rightrightarrows \mathbb{R}^n$ is monotone if

$$\langle \mathbb{A}x - \mathbb{A}y, x - y \rangle \geq 0, \qquad \forall x, y \in \mathbb{R}^n,$$

where the notation means that $\langle u - v, x - y \rangle \geq 0$ for all $u \in \mathbb{A}x$ and $v \in \mathbb{A}y$. For $\mu \in (0, \infty)$, say $\mathbb{A} \colon \mathbb{R}^n \rightrightarrows \mathbb{R}^n$ is $\mu$-strongly monotone if

$$\langle \mathbb{A}x - \mathbb{A}y, x - y \rangle \geq \mu \|x - y\|^2, \qquad \forall x, y \in \mathbb{R}^n.$$

Write $\operatorname{Gra} \mathbb{A} = \{(x, u) \mid u \in \mathbb{A}x\}$ for the graph of $\mathbb{A}$. An operator $\mathbb{A}$ is maximally monotone if there is no other monotone $\mathbb{B}$ such that $\operatorname{Gra} \mathbb{A} \subset \operatorname{Gra} \mathbb{B}$ properly, and is maximally $\mu$-strongly monotone if there is no other $\mu$-strongly monotone $\mathbb{B}$ such that $\operatorname{Gra} \mathbb{A} \subset \operatorname{Gra} \mathbb{B}$ properly.

For $L \in (0, \infty)$, single-valued operator $\mathbb{T} \colon \mathbb{R}^n \to \mathbb{R}^n$ is $L$-Lipschitz if

$$\|\mathbb{T}x - \mathbb{T}y\| \leq L \|x - y\|, \qquad \forall x, y \in \mathbb{R}^n.$$

Write $\mathbb{J}_{\mathbb{A}} = (\mathbb{I} + \mathbb{A})^{-1}$ for the resolvent of $\mathbb{A}$, while $\mathbb{I} \colon \mathbb{R}^n \to \mathbb{R}^n$ is the identity operator. When $\mathbb{A}$ is maximally monotone, it is well known that $\mathbb{J}_{\mathbb{A}}$ is single-valued with $\operatorname{dom} \mathbb{J}_{\mathbb{A}} = \mathbb{R}^n$.

We say $x_\star \in \mathbb{R}^n$ is a zero of $\mathbb{A}$ if $0 \in \mathbb{A}x_\star$. We say $y_\star$ is a fixed-point of $\mathbb{T}$ if $\mathbb{T}y_\star = y_\star$. Write $\operatorname{Zer} \mathbb{A}$ for the set of all zeros of $\mathbb{A}$ and $\operatorname{Fix} \mathbb{T}$ for the set of all fixed-points of $\mathbb{T}$.

**Monotonicity with continuous curves.** We say an operator is differentiable if it is single-valued, continuous, and differentiable as a function. If a differentiable operator $\mathbb{A}$ is monotone and $X \colon [0, \infty) \to \mathbb{R}^n$ is a differentiable curve, then taking limit $h \to 0$ of

$$\frac{1}{h^2} \langle \mathbb{A}(X(t + h)) - \mathbb{A}(X(t)), X(t + h) - X(t) \rangle \geq 0$$

leads to

$$\left\langle \frac{d}{dt} \mathbb{A}(X(t)), \dot{X}(t) \right\rangle \geq 0. \tag{1}$$

Similarly if $\mathbb{A}$ is furthermore $\mu$-strongly monotone, then

$$\left\langle \frac{d}{dt} \mathbb{A}(X(t)), \dot{X}(t) \right\rangle \geq \mu \left\| \dot{X}(t) \right\|^2. \tag{2}$$

## 2 Derivation of differential inclusion model of anchor acceleration

### 2.1 Anchor ODE

Suppose $\mathbb{A} \colon \mathbb{R}^n \rightrightarrows \mathbb{R}^n$ is a maximal monotone operator and $\beta \colon (0, \infty) \to [0, \infty)$ is a twice differentiable function. Consider differential inclusion

$$\dot{X}(t) \in -\mathbb{A}(X(t)) - \beta(t)(X(t) - X_0) \tag{3}$$

with initial condition $X(0) = X_0 \in \operatorname{dom}(\mathbb{A})$. We refer to this as the *anchor ODE*. [1] We say $X \colon [0, \infty) \to \mathbb{R}^n$ is a solution, if it is absolutely continuous and satisfies (3) for $t \in (0, \infty)$ almost everywhere.

Denote $S$ as the subset of $[0, \infty)$ on which $X$ satisfies the differential inclusion. Define

$$\tilde{\mathbb{A}}(X(t)) = -\dot{X}(t) - \beta(t)(X(t) - X_0)$$

for $t \in S$. Since $\tilde{\mathbb{A}}(X(t)) \in \mathbb{A}(X(t))$ for $t \in S$, we say $\tilde{\mathbb{A}}$ is a *selection* of $\mathbb{A}$ for $t \in S$. If $\|\tilde{\mathbb{A}}(X(t))\|$ is bounded on all bounded subsets of $S$, then we can extend $\tilde{\mathbb{A}}$ to $[0, \infty)$ while retaining certain favorable properties. We discuss the technical details of this extension in Appendix E.1. The statements of Section 3 are stated with this extension.

---

[1]Strictly speaking, this is a differential inclusion, not a differential equation, but we nevertheless refer to it as an ODE.

## 2.2 Derivation from discrete methods

We now show that the following instance of the anchor ODE

$$\dot{X}(t) = -\mathbb{A}(X(t)) - \frac{1}{t}(X(t) - X_0), \tag{4}$$

where $X(0) = X_0$ is the initial condition and $\mathbb{A} \colon \mathbb{R}^n \to \mathbb{R}^n$ is a continuous operator, is a continuous-time model of APPM [35], EAG [75], and FEG [42], which are accelerated methods for monotone inclusion and minimax problems.

Consider APPM with operator $h\mathbb{A}$

$$x^k = \mathbb{J}_{h\mathbb{A}} y^{k-1}$$
$$y^k = \frac{k}{k+1}(2x^k - y^{k-1}) + \frac{1}{k+1} y^0 \tag{5}$$

with initial condition $y^0 = x^0$. Assume $h > 0$ and $\mathbb{A} \colon \mathbb{R}^n \to \mathbb{R}^n$ is a continuous monotone operator. Using $y^{k-1} = x^k + h\mathbb{A}x^k$ obtained from the first line, substituting $y^k$ and $y^{k-1}$ in the second line we get,

$$x^{k+1} + h\mathbb{A}x^{k+1} = \frac{k}{k+1}\left(x^k - h\mathbb{A}x^k\right) + \frac{1}{k+1}x^0.$$

Then reorganizing and dividing both sides by $h$, we have

$$\frac{x^{k+1} - x^k}{h} = -\mathbb{A}x^{k+1} - \frac{k}{k+1}\mathbb{A}x^k - \frac{1}{h(k+1)}(x^k - x^0).$$

Identifying $x^0 = X_0$, $2hk = t$, and $x^k = X(t)$, we have $\frac{k}{k+1} = 1 - \frac{h}{hk+h} = 1 + \mathcal{O}(h)$ and so

$$2\dot{X}(t) + \mathcal{O}(h) = -\mathbb{A}(X(t+2h)) - (1 + \mathcal{O}(h))\,\mathbb{A}(X(t)) - \frac{2}{t + \mathcal{O}(h)}(X(t) - X_0).$$

Taking limit $h \to 0^+$ and dividing both sides by 2, we get the anchor ODE (4). The correspondence with EAG and FEG are provided in Appendix D.4.

The following theorem establishes a rigorous correspondence between APPM and the anchor ODE for general maximal monotone operators.

**Theorem 2.1.** *Let $\mathbb{A}$ be a (possibly set-valued) maximal monotone operator and assume $\mathrm{Zer}\mathbb{A} \neq \emptyset$. Let $x^k$ be the sequence generated by APPM (5) and $X$ be the solution of the differential inclusion (3) with $\beta(t) = \frac{1}{t}$. For all fixed $T > 0$,*

$$\lim_{h \to 0+} \max_{0 \leq k \leq \frac{T}{2h}} \left\| x^k - X(2kh) \right\| = 0.$$

We provide the proof in Appendix D.2.

## 2.3 Existence of the solution for $\beta(t) = \frac{\gamma}{t^p}$

To get further insight into the anchor acceleration, we generalize anchor coefficient to $\beta(t) = \frac{\gamma}{t^p}$ for $p, \gamma > 0$. We first establish the uniqueness and existence of the solution.

**Theorem 2.2.** *Consider (3) with $\beta(t) = \frac{\gamma}{t^p}$, i.e.*

$$\dot{X}(t) \in -\mathbb{A}(X(t)) - \frac{\gamma}{t^p}(X(t) - X_0). \tag{6}$$

*for $p, \gamma > 0$. Then solution of (6) uniquely exists.*

We provide the proof in Appendix B.

## 2.4 Additional properties of anchor ODE

We state a regularity lemma of the differential inclusion (3), which we believe may be of independent interest. In particular, we use this result several times throughout our various proofs.

**Lemma 2.3.** *Let $X(\cdot)$ and $Y(\cdot)$ are solutions of the differential inclusion (3) respectively with initial values and anchors $X_0$ and $Y_0$. Then for all $t \in [0, \infty)$,*

$$\|X(t) - Y(t)\| \leq \|X_0 - Y_0\|.$$

We provide the proof in Appendix B.1.

Boundedness of trajectories is an immediate corollary of Lemma 2.3. Specifically, suppose $X(\cdot)$ is the solution of differential inclusion (3) with initial value $X_0$. Then for all $X_\star \in \text{Zer}\mathbb{A}$ and $t \in [0, \infty)$,

$$\|X(t) - X_\star\| \leq \|X_0 - X_\star\|.$$

This follows from setting $Y_0 = X_\star$ in Lemma 2.3.

## 3 Convergence analysis

We now analyze the convergence rate of $\left\|\tilde{\mathbb{A}}(X(t))\right\|^2$ for the anchor ODE (3) with $\beta(t) = \frac{\gamma}{t^p}$ and $\gamma, p > 0$. The results are organized in Table 1.

| Case | $p = 1,$ $\gamma \geq 1$ | $p = 1,$ $\gamma < 1$ | $p < 1$ | $p > 1$ |
|---|---|---|---|---|
| $\left\|\tilde{\mathbb{A}}(X(t))\right\|^2$ | $\mathcal{O}\left(\frac{1}{t^2}\right)$ | $\mathcal{O}\left(\frac{1}{t^{2\gamma}}\right)$ | $\mathcal{O}\left(\frac{1}{t^{2p}}\right)$ | $\mathcal{O}(1)$ |

Table 1: Convergence rates of Theorem 3.1.

Let $\beta$ be the anchor coefficient function of (3). Define $C\colon [0, \infty) \to \mathbb{R}$ as $C(t) = e^{\int_v^t \beta(s)ds}$ for some $v \in [0, \infty]$. Note that $\dot{C} = C\beta$ and $C$ is unique up to scalar multiple. We call $\mathcal{O}(\beta(t))$ the *vanishing speed* and $\mathcal{O}\left(\frac{1}{C(t)}\right)$ the *contracting speed*, and we describe their trade-off in the following.

Loosely speaking, the *contracting speed* describes how fast the anchor term alone contracts the dynamical system. Consider $\dot{X}(t) = -\beta(t)(X(t) - a)$ for $a \in \mathbb{R}^n$, a system only with the anchor. Then, $X(t) = \frac{C(0)}{C(t)}(X(0) - a) + a$ is the solution, so the flow contracts towards the anchor $a$ with rate $\frac{1}{C(t)}$. Intuitively speaking, this contracting behavior leads to stability and convergence. On the other hand, the anchor must eventually vanish, since our goal is to converge to an element in $\text{Zer}\mathbb{A}$, not the anchor. Thus the *vanishing speed* must be fast enough to not slow down the convergence of the flow to $\text{Zer}\mathbb{A}$.

This observation is captured in Figure 1. Consider a monotone linear operator $\mathbb{A} = \begin{pmatrix} 0 & 1 \\ -1 & 0 \end{pmatrix}$ on $\mathbb{R}^2$ and $\beta(t) = \frac{\gamma}{t^p}$ with $\gamma = 1$ and $p > 0$. Note if there is no anchor, the ODE reduces to $\dot{X} = -\mathbb{A}(X)$ which do not converge [31, Chapter 8.2]. Figure 1 shows that with $p > 1$, the anchor vanished too early before the flow is contracted enough to result in converging flow. With $p < 1$, the flow does converge but the anchor vanished too late, slowing down the convergence. With $p = 1$, the convergence is fastest.

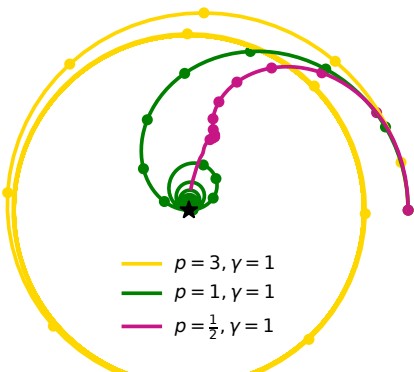

Figure 1: Flows of the solution of (6) with $\mathbb{A} = \begin{pmatrix} 0 & 1 \\ -1 & 0 \end{pmatrix}$, $\gamma = 1$, $X_0 = (1, 0)$ and various $p$. Flow is from $t = 0$ to $t = 100$. The marker is plotted every 0.8 units of time until $t = 9.6$. Note the last marker of the flow for $p = \frac{1}{2}$ is farther from the optimal point $\star$ than that of the flow for $p = 1$.

The following theorem formalizes this insight and produces the results of Table 1.

**Theorem 3.1.** *Suppose* $\mathbb{A}$ *is a maximal monotone operator with* $\mathrm{Zer}\mathbb{A} \neq \emptyset$. *Consider* (3) *with* $\beta(t) = \frac{\gamma}{t^p}$. *Let* $\tilde{\mathbb{A}}(X(t))$ *be the selection of* $\mathbb{A}(X(t))$ *as in Section 2.1. Then,*

$$\left\| \tilde{\mathbb{A}}(X(t)) \right\|^2 = \mathcal{O}\left(\frac{1}{C(t)^2}\right) + \mathcal{O}\left(\beta(t)^2\right) + \mathcal{O}\left(\dot{\beta}(t)\right).$$

Note that

$$C(t) = \begin{cases} t^\gamma & p = 1 \\ e^{\frac{\gamma}{1-p} t^{1-p}} & p \neq 1. \end{cases}$$

We expect the convergence rate of Theorem 3.1 to be optimized when the terms are balanced. When $\beta(t) = \frac{1}{t}$,

$$\frac{1}{C(t)^2} = \frac{1}{(e^{\int_1^t \frac{1}{s} ds})^2} = \frac{1}{t^2} = \beta(t)^2 = -\dot{\beta}(t)$$

and all three terms are balanced. Indeed, the choice $\beta(t) = \frac{1}{t}$ corresponds to the optimal discrete-time choice $\frac{1}{k+2}$ of APPM or other accelerated methods.

## 3.1 Proof outline of Theorem 3.1

The proof of Theorem 3.1 follows from Lemma 3.4, which we will introduce later in this section. To derive Lemma 3.4, we introduce a conservation law.

**Proposition 3.2.** *Suppose* $\tilde{\mathbb{A}}$ *is Lipschitz continuous and monotone. For* $t_0 > 0$, *define* $E : (0, \infty) \to \mathbb{R}$ *as*

$$E = \frac{C(t)^2}{2} \left( \left\| \tilde{\mathbb{A}}(X(t)) \right\|^2 + 2\beta(t) \left\langle \tilde{\mathbb{A}}(X(t)), X(t) - X_0 \right\rangle + \left(\beta(t)^2 + \dot{\beta}(t)\right) \left\| X(t) - X_0 \right\|^2 \right)$$

$$- \int_{t_0}^t \frac{d}{ds}\left(\frac{C(s)^2 \dot{\beta}(s)}{2}\right) \left\| X(s) - X_0 \right\|^2 ds + \int_{t_0}^t C(s)^2 \left\langle \frac{d}{ds} \tilde{\mathbb{A}}(X(s)), \dot{X}(s) \right\rangle ds.$$

*Then* $E$ *is a constant function.*

The proof of Proposition 3.2 uses dilated coordinate $W(t) = C(t)(X(t) - X_0)$ to derive its conservation law in the style of Suh et al. [67]. We provide the details in Appendix E.2.

Recall from (1) that $\left\langle \frac{d}{ds} \tilde{\mathbb{A}}(X(s)), \dot{X}(s) \right\rangle \geq 0$, the integrand of the last term of $E$ is nonnegative. This motivates us to define

$$V(t) = E - \int_{t_0}^t C(s)^2 \left\langle \frac{d}{ds} \tilde{\mathbb{A}}(X(s)), \dot{X}(s) \right\rangle ds$$

as our Lyapunov function.

**Corollary 3.3.** *Let* $\mathbb{A}$ *be maximal monotone and* $\beta(t) = \frac{\gamma}{t^p}$ *with* $p > 0$, $\gamma > 0$. *Let* $\tilde{\mathbb{A}}(X(t))$ *be the selection of* $\mathbb{A}(X(t))$ *as in Section 2.1. For* $t_0 \geq 0$, *define* $V : [0, \infty) \to \mathbb{R}$ *as*

$$V(t) = \frac{C(t)^2}{2} \left( \left\| \tilde{\mathbb{A}}(X(t)) \right\|^2 + 2\beta(t) \left\langle \tilde{\mathbb{A}}(X(t)), X(t) - X_0 \right\rangle + \left(\beta(t)^2 + \dot{\beta}(t)\right) \left\| X(t) - X_0 \right\|^2 \right)$$

$$- \int_{t_0}^t \frac{d}{ds}\left(\frac{C(s)^2 \dot{\beta}(s)}{2}\right) \left\| X(s) - X_0 \right\|^2 ds.$$

*for* $t > 0$ *and* $V(0) = \lim_{t \to 0+} V(t)$. *Then* $V(t) \leq V(0)$ *holds for* $t \geq 0$.

A technical detail is that all terms involving $\frac{d}{ds} \tilde{\mathbb{A}}(X(s))$ have been excluded in the definition of $V$ and this is what allows $\mathbb{A}$ to not be Lipschitz continuous. We provide the details in Appendix E.3.

**Lemma 3.4.** *Consider the setup of Corollary 3.3. Assume* $\mathrm{Zer}\mathbb{A} \neq \emptyset$. *Then for* $t > 0$ *and* $X_\star \in \mathrm{Zer}\mathbb{A}$,

$$\left\| \tilde{\mathbb{A}}(X(t)) \right\|^2 \leq 4\beta(t)^2 \left\| X_0 - X_\star \right\|^2 + \frac{4V(0)}{C(t)^2} - 2\left(\beta(t)^2 + \dot{\beta}(t)\right) \left\| X(t) - X_0 \right\|^2$$

$$+ \frac{2}{C(t)^2} \int_{t_0}^t \frac{d}{ds}\left(C(s)^2 \dot{\beta}(s)\right) \left\| X(s) - X_0 \right\|^2 ds. \tag{7}$$

*Proof outline of Lemma 3.4.* Define

$$\Phi(t) = \left\|\tilde{\mathbb{A}}(X(t))\right\|^2 + 2\beta(t)\left\langle\tilde{\mathbb{A}}(X(t)), X(t) - X_0\right\rangle.$$

Then, from monotonicity of $\tilde{\mathbb{A}}$ and Young's inequality,

$$\begin{aligned}
\Phi(t) &\geq \left\|\tilde{\mathbb{A}}(X(t))\right\|^2 + 2\beta(t)\left\langle\tilde{\mathbb{A}}(X(t)), X_\star - X_0\right\rangle \\
&\geq \left\|\tilde{\mathbb{A}}(X(t))\right\|^2 - 2\left(\left\|\frac{1}{2}\tilde{\mathbb{A}}(X(t))\right\|^2 + \left\|\beta(t)\left(X_\star - X_0\right)\right\|^2\right) \\
&= \frac{1}{2}\left\|\tilde{\mathbb{A}}(X(t))\right\|^2 - 2\beta(t)^2\left\|X_0 - X_\star\right\|^2.
\end{aligned} \tag{8}$$

By Corollary 3.3, $\frac{2V(0)}{C(t)^2} - \frac{2V(t)}{C(t)^2} + \Phi(t) \geq \Phi(t)$ for $t > 0$. Applying (8) and organizing, we can get the desired result. The details are provided in Appendix E.4. $\qquad\square$

*Proof outline of Theorem 3.1.* It remains to show that last integral term of Lemma 3.4 is $\mathcal{O}\left(\frac{1}{C(t)^2}\right) + \mathcal{O}\left(\beta(t)^2\right) + \mathcal{O}\left(\dot{\beta}(t)\right)$. The details are provided in Appendix E.5. $\qquad\square$

Before we end this section, we observe how our analysis simplifies in the special case $\beta(t) = \frac{1}{t}$. In this case,

$$V(t) = \frac{t^2}{2}\left\|\tilde{\mathbb{A}}(X(t))\right\|^2 + t\left\langle\tilde{\mathbb{A}}(X(t)), X(t) - X_0\right\rangle,$$

and this corresponds to the Lyapunov function of [59, Section 4] for the case $\gamma = 1$. As $V(0) = 0$, the conclusion of Lemma 3.4 becomes

$$\left\|\tilde{\mathbb{A}}(X(t))\right\|^2 \leq \frac{4}{t^2}\left\|X_0 - X_\star\right\|^2 = \mathcal{O}\left(\frac{1}{t^2}\right),$$

which to the best rate in Table 1.

## 3.2 Point convergence

APPM is an instance of the Halpern method [54, Lemma 3.1], which iterates converge to the element in $\text{Zer}\mathbb{A}$ closest to $X_0$ [34, 73]. The anchor ODE also exhibits this behavior.

**Theorem 3.5.** *Let $\mathbb{A}$ be a maximal monotone operator with $\text{Zer}\mathbb{A} \neq \emptyset$ and $X$ be the solution of* (3). *If $\lim_{t\to\infty}\left\|\tilde{\mathbb{A}}(X(t))\right\| = 0$ and $\lim_{t\to\infty}1/C(t) = 0$, then, as $t \to \infty$,*

$$X(t) \to \underset{z\in\text{Zer}\mathbb{A}}{\operatorname{argmin}}\left\|z - X_0\right\|.$$

We provide the proof in Appendix E.6.

# 4 Tightness of analysis

In this section, we show that the convergence rates of Table 1 are actually tight by considering the dynamics under the explicit example $\mathbb{A} = \left(\begin{smallmatrix}0 & 1\\ -1 & 0\end{smallmatrix}\right)$. Throughout this section, we denote $\mathbb{A}$ as $A$ when when the operator is linear.

## 4.1 Explicit solution for linear $A$

**Lemma 4.1.** *Let $A\colon \mathbb{R}^n \to \mathbb{R}^n$ be a linear operator and let $\beta(t) = \frac{\gamma}{t}$. The series*

$$X(t) = \sum_{n=0}^{\infty}\frac{(-tA)^n}{\Gamma(n+\gamma+1)}\Gamma(\gamma+1)X_0,$$

*where $\Gamma$ denotes the gamma function, is the solution for* (3) *with $\mathbb{A} = A$.*

Note that when $\gamma = 0$, this is the series definition of the matrix exponential and $X(t) = e^{-tA}$. The solution also has an integral form, which extends to general $\beta(t)$.

**Lemma 4.2.** *Suppose* $A \colon \mathbb{R}^n \to \mathbb{R}^n$ *is a monotone linear operator. Then*

$$X(t) = \frac{e^{-tA}}{C(t)} \left( \int_0^t e^{sA} C(s)\beta(s)ds + C(0)I \right) X_0 \tag{9}$$

*is the solution for* (3) *with* $\mathbb{A} = A$.

See Appendix F.1.1 and Appendix F.1.2 for details.

## 4.2 The rates in Table 1 are tight

First, we consider $p > 1$ for $\beta(t) = \frac{\gamma}{t^p}$.

**Theorem 4.3.** *Suppose* $\lim_{t \to \infty} \frac{1}{C(t)} \neq 0$, *i.e., suppose* $\beta(t) \in L^1[t_0, \infty)$ *for some* $t_0 > 0$. *Then there exists an operator* $\mathbb{A}$ *such that*

$$\lim_{t \to \infty} \left\| \tilde{\mathbb{A}}(X(t)) \right\| \neq 0,$$

*where* $X$ *is the solution of* (3).

Note that $\frac{\gamma}{t^p} \in L^1[t_0, \infty)$ when $p > 1$. The proof of Theorem 4.3 considers $\mathbb{A} = 2\pi\xi \left( \begin{smallmatrix} 0 & 1 \\ -1 & 0 \end{smallmatrix} \right)$ for $\xi \in \mathbb{R}$ and uses the Fourier inversion formula. See Appendix F.2 for details.

Next, we consider $\beta(t) = \frac{\gamma}{t^p}$ for cases other than $p > 1$.

**Theorem 4.4.** *Let* $A = \left( \begin{smallmatrix} 0 & 1 \\ -1 & 0 \end{smallmatrix} \right)$, $\beta(t) = \frac{\gamma}{t^p}$, $0 < p \le 1$, *and* $\gamma > 0$. *Let* $X$ *be the solution given by* (9) *and* $X_0 \neq 0$. *Let*

$$r(t) = \begin{cases} t^2 & \text{for } p = 1, \gamma \ge 1 \\ t^{2\gamma} & \text{for } p = 1, \gamma < 1 \\ t^{2p} & \text{for } 0 < p < 1. \end{cases}$$

*Then,*

$$\lim_{t \to \infty} r(t) \left\| A(X(t)) \right\|^2 \neq 0.$$

We provide the proof in Appendix F.3.

## 5 Discretized algorithms

In this section, we provide discrete-time convergence results that match the continuous-time rate of Section 3.

**Theorem 5.1.** *Suppose* $\mathbb{A}$ *be a maximal monotone operator,* $p > 0$, *and* $\gamma > 0$. *Consider*

$$x^k = \mathbb{J}_{\mathbb{A}} y^{k-1}$$
$$y^k = \frac{k^p}{k^p + \gamma}(2x^k - y^{k-1}) + \frac{\gamma}{k^p + \gamma}x^0$$

*for* $k = 1, 2, \dots$, *with initial condition* $y^0 = x^0 \in \mathbb{R}^n$. *Let* $\tilde{\mathbb{A}}x^k = y^{k-1} - x^k$ *for* $k = 1, 2, \dots$. *Then this method exhibits the rates of convergence in Table 2.*

| Case | $p = 1,$ $\gamma \ge 1$ | $p = 1,$ $\gamma < 1$ | $p < 1$ | $p > 1$ |
|---|---|---|---|---|
| $\left\| \tilde{\mathbb{A}}(x^k) \right\|^2$ | $\mathcal{O}\left(\frac{1}{k^2}\right)$ | $\mathcal{O}\left(\frac{1}{k^{2\gamma}}\right)$ | $\mathcal{O}\left(\frac{1}{k^{2p}}\right)$ | $\mathcal{O}(1)$ |

Table 2: Rates for the discrete-time method of Theorem 5.1.

Note that the method of Theorem 5.1 reduces to APPM when $\gamma = 1$, $p = 1$.

*Proof outline of Theorem 5.1.* The general strategy is to find discretized counterparts of corresponding continuous-time analyses. However, directly discretizing the conservation law of Proposition 3.2 was difficult due to technical reasons. Instead, we obtain differently scaled but equivalent conservation laws using dilated coordinates and then performed the discretization. The specific dilated coordinates, inspired by [67], are $W_1(t) = X(t) - X_0$ for $p > 1$, $W_2(t) = t^p(X(t) - X_0)$ for $0 < p < 1$, $W_3(t) = t(X(t) - X_0)$ for $p = 1$, $\gamma \geq 1$ and $W_4(t) = t^\gamma(X(t) - X_0)$ for $p = 1$, $0 < \gamma < 1$.

In the discrete-time analyses, the behavior of the leading-order terms is predictable as they match the continuous-time counterpart. The difficult part is, however, controlling the higher-order terms that were not present in the continuous-time analyses. Through our detailed analyses, we bound such higher-order terms and show that they do not affect the convergence rate in the end. We provide the details in Appendix G.3. □

## 6 Convergence analysis under strong monotonicity

In this section, we analyze the dynamics of the anchor ODE (3) for $\mu$-strongly monotone $\mathbb{A}$. When $\beta(t) = \frac{1}{t}$ and $\mathbb{A} = \left(\begin{smallmatrix} \mu & 0 \\ 0 & \mu \end{smallmatrix}\right)$, Lemma 4.1 tells us that $\mathbb{A}(X(t)) = \frac{1}{t}\left(I - e^{-tA}\right)X_0$ and therefore that $\|\mathbb{A}(X(t))\|^2 = \Theta\left(\frac{1}{t^2}\right)$, which is a slow rate for the strongly monotone setup. On the other hand, we will see that $\beta(t) = \frac{2\mu}{e^{2\mu t}-1}$ is a better choice leading to a faster rate in this setup.

Our analysis of this section is also based on a conservation law, but we use a slightly modified version to exploit strong monotonicity.

**Proposition 6.1.** *Suppose $\tilde{\mathbb{A}}$ is monotone and Lipschitz continuous. Let $X$ be the solution of* (3) *and let $R : [0, \infty) \to (0, \infty)$ be a differentiable function. For $t_0 > 0$, define $E : (0, \infty) \to \mathbb{R}$ as*

$$E = \frac{C(t)^2 R(t)^2}{2}\left(\left\|\tilde{\mathbb{A}}(X(t))\right\|^2 + 2\beta(t)\langle\tilde{\mathbb{A}}(X(t)), X(t) - X_0\rangle + \left(\beta(t)^2 + \dot{\beta}(t)\right)\|X(t) - X_0\|^2\right)$$

$$-\int_{t_0}^t \frac{d}{ds}\left(\frac{C(s)^2 R(s)^2 \dot{\beta}(s)}{2}\right)\|X(t) - X_0\|^2\,ds + \int_{t_0}^t C(s)^2 R(s)^2\left(\left\langle\frac{d}{ds}\tilde{\mathbb{A}}(X(s)), \dot{X}(s)\right\rangle - \frac{\dot{R}(s)}{R(s)}\left\|\dot{X}(s)\right\|^2\right)ds.$$

*Then $E$ is a constant function for $t \in [0, \infty)$.*

Proposition 6.1 generalizes Proposition 3.2, since it corresponds to the special case with $R(t) \equiv 1$.

Recall from (2), when $\mathbb{A}$ is $\mu$-strongly monotone we have

$$\left\langle\frac{d}{ds}\mathbb{A}(X(t)), \dot{X}(t)\right\rangle - \mu\left\|\dot{X}(t)\right\|^2 \geq 0.$$

This motivates the choice $R(t) = e^{\mu t}$, since $\frac{\dot{R}(s)}{R(s)} = \mu$. From calculation provided in Appendix H.2, the choice $\beta(t) = \frac{2\mu}{e^{2\mu t}-1}$ makes $\frac{d}{ds}\left(\frac{C(s)^2 R(s)^2 \dot{\beta}(s)}{2}\right) = 0$. Plugging these choices into Proposition 6.1 and following arguments of Section 3, we arrive at the following theorem.

**Theorem 6.2.** *Let $\mathbb{A}$ be a $\mu$-strongly maximal monotone operator with $\mu > 0$ and assume $\mathrm{Zer}\mathbb{A} \neq \emptyset$. Let $X$ be a solution of the differential inclusion* (3) *with $\beta(t) = \frac{2\mu}{e^{2\mu t}-1}$, i.e. for almost all $t$,*

$$\dot{X} \in -\mathbb{A}(X) - \frac{2\mu}{e^{2\mu t} - 1}(X - X_0). \tag{10}$$

*Let $\tilde{\mathbb{A}}(X(t))$ be the selection of $\mathbb{A}(X(t))$ as in Section 2.1. Define $V : [0, \infty) \to \mathbb{R}$ as*

$$V(t) = \frac{(e^{\mu t} - e^{-\mu t})^2}{2}\left\|\tilde{\mathbb{A}}(X(t))\right\|^2 + 2\mu\left(1 - e^{-2\mu t}\right)\left(\langle\tilde{\mathbb{A}}(X(t)), X(t) - X_0\rangle - \mu\|X(t) - X_0\|^2\right).$$

*Then $V(t) \leq V(0)$ holds for $t \geq 0$. Furthermore for $X_\star \in \mathrm{Zer}\mathbb{A}$,*

$$\|\tilde{\mathbb{A}}(X(t))\|^2 \leq \left(\frac{2\mu}{e^{\mu t} - 1}\right)^2\|X_0 - X_\star\|^2 = \mathcal{O}\left(\frac{1}{e^{2\mu t}}\right).$$

In Appendix H.2.3, we show that (10) is a continuous-time model for OS-PPM of Park and Ryu [54]. In Appendix C, we show the existence and uniqueness of the solution.

Since $\beta(t) = \frac{2\mu}{e^{2\mu t}-1} \in L^1[t_0, \infty)$ for any $t_0 > 0$, Theorem 4.3 implies that $\tilde{\mathbb{A}}(X(t)) \not\to 0$ when $\mathbb{A}$ is merely monotone. This tells us that the optimal choice of $\beta(t)$ for should depend on the properties of $\mathbb{A}$. In the following section, we describe how $\beta(t)$ can be chosen to adapt to the operator's properties.

## 7 Adaptive anchor acceleration and experiments

In this section, we present an adaptive method for choosing the anchor coefficient $\beta$, and we theoretically and experimentally show that this choice allows the dynamics to adapt to the operator's properties. It achieves the optimal $\mathcal{O}(1/k^2)$-convergence rate when $\mathbb{A}$ is monotone and an exponential convergence rate when $\mathbb{A}$ is furthermore $\mu$-strongly monotone and Lipschitz continuous.

**Theorem 7.1.** *Suppose $\tilde{\mathbb{A}}$ is Lipschitz continuous and monotone. Consider the anchor ODE*

$$\dot{X} = -\tilde{\mathbb{A}}(X) + \underbrace{\frac{\left\|\tilde{\mathbb{A}}(X)\right\|^2}{2\left\langle \tilde{\mathbb{A}}(X), X - X_0 \right\rangle}}_{=-\beta(t)}(X - X_0) \tag{11}$$

*with initial condition $X(0) = X_0$ and $\left\|\tilde{\mathbb{A}}(X_0)\right\| \neq 0$. Suppose the solution exists and $\dot{X}$ is continuous at $t = 0$. Moreover, suppose $\beta\colon (0,\infty) \to \mathbb{R}$ is well-defined, i.e., no division by zero occurs in the definition of $\beta(t)$. Then for $t > 0$ and $X_\star \in \mathrm{Zer}\tilde{\mathbb{A}}$, we have $\beta(t) > 0$ and*

$$\left\|\tilde{\mathbb{A}}(X(t))\right\|^2 \leq 4\beta(t)^2 \left\|X_0 - X_\star\right\|^2$$

$$\beta(t)^2 \leq \frac{1}{t^2}.$$

*If $\tilde{\mathbb{A}}$ is furthermore $\mu$-strongly monotone, then for $t > 0$,*

$$\beta(t)^2 \leq \left(\frac{\mu/2}{e^{\mu t/2} - 1}\right)^2.$$

We provide the proof in Appendix I.1. Note that anchor coefficient (11) is chosen so that

$$\Phi(t) = \left\|\tilde{\mathbb{A}}(X(t))\right\|^2 + 2\beta(t)\left\langle \tilde{\mathbb{A}}(X(t)), X(t) - X_0 \right\rangle = 0.$$

So left-hand side of (8) is zero and a $\mathcal{O}\left(\beta(t)^2\right)$ convergence rate is immediate. An analogous discrete-time result is shown in the following theorem.

**Theorem 7.2.** *Let $\mathbb{A}$ be a maximal monotone operator. Let $x^0 = y^0 \in \mathbb{R}^n$. Consider*

$$x^k = \mathbb{J}_{\mathbb{A}} y^{k-1}$$
$$y^k = (1 - \beta_k)(2x^k - y^{k-1}) + \beta_k x^0$$

*with*

$$\beta_k = \begin{cases} \dfrac{\|\tilde{\mathbb{A}}x^k\|^2}{-\langle \tilde{\mathbb{A}}x^k, \, x^k - x^0 \rangle + \|\tilde{\mathbb{A}}x^k\|^2} & \text{if } \|\tilde{\mathbb{A}}x^k\|^2 \neq 0 \\ 0 & \text{if } \|\tilde{\mathbb{A}}x^k\|^2 = 0, \end{cases}$$

*for $k = 1, 2, \ldots$, where $\tilde{\mathbb{A}}x^k = y^{k-1} - x^k$.*

*Then $\beta_k \geq 0$ and*

$$\left\|\tilde{\mathbb{A}}x^{k+1}\right\|^2 \leq \beta_k^2 \left\|x^0 - x^\star\right\|^2$$

$$\beta_k^2 \leq \frac{1}{(k+1)^2}$$

*for $k = 1, 2, \ldots$ and $x^\star \in \mathrm{Zer}\mathbb{A}$.*

*If $\mathbb{A}$ is furthermore $\mu$-strongly monotone and $L$-Lipschitz continuous, then for $k = 1, 2, \ldots$,*

$$\beta_k^2 \leq \left(\frac{\mu/(1+L^2)}{\left(1 + \mu/(1+L^2)\right)^k - 1 + \mu/(1+L^2)}\right)^2.$$

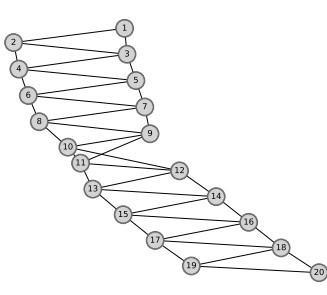
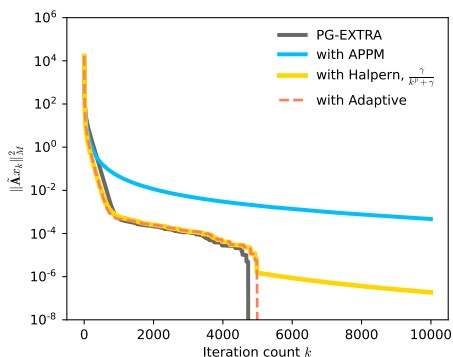

Figure 2: (Left) Network graph. (Right) Squared $M$-norm $\|\tilde{\mathbb{A}}x^k\|_M^2$ vs. $k$. Halpern corresponds to the method in Theorem 5.1, we use $p = 1.5$ and $\gamma = 2.0$.

For the monotone setup, the rate $\left\|\tilde{\mathbb{A}}x^{k+1}\right\|^2 \leq \frac{1}{(k+1)^2}\left\|x^0 - x^\star\right\|^2$ matches the exact optimal rate of APPM [54]. In the limit $\mu \to 0$, the result for the $\mu$-strongly monotone case reduces to the result for the monotone case. We provide the proof and details of Theorem 7.2 in Appendix I.3.

The method of Theorem 7.2 is a discrete-time counterpart of the ODE of (11). The extra term $\|\tilde{\mathbb{A}}x^k\|^2$ of the denominator in the definition of $\beta_k$ vanishes in the continuous-time limit. We provide further details in Appendix I.2. Analogous to the continuous-time case, a key property of the discrete-time adaptive method is that the counterpart of $\Phi(t)$ is kept nonpositive. In the proof of Lemma I.3, the fact $\beta_k < 1$ plays the key role in proving this property. The extra term $\|\tilde{\mathbb{A}}x^k\|^2$ in the denominator and the fact that $\langle\tilde{\mathbb{A}}x^k, x^k - x^0\rangle < 0$ when $\|\tilde{\mathbb{A}}x^k\|^2 \neq 0$ leads to $\beta_k < 1$.

## 7.1 Experiment details

We now show an experiment with the method of Theorem 7.2 applied to a decentralized compressed sensing problem Shi et al. [63]. We assume that we have the measurement $b_i = A_{(i)}x + e_i$, where $A_{(i)}$ is a measurement matrix available for each local agent $i$, $x$ is an unknown shared signal we hope to recover, and $e_i$ is an error in measurement. We solve this problem in a decentralized manner in which the local agents keep their measurements private and only communicate with their neighbors.

As in Shi et al. [63], we formulate the problem into an unconstrained $\ell_1$-regularized least squares problem

$$\underset{x \in \mathbb{R}^d}{\text{minimize}} \quad \frac{1}{n}\sum_{i=1}^n \left\{\frac{1}{2}\|A_{(i)}x - b_i\|^2 + \rho\|x\|_1\right\},$$

and apply PG-EXTRA. We compare vanilla PG-EXTRA with the various anchored versions of PG-EXTRA with $\beta_k$ as in Theorem 5.1 and Theorem 7.2. We show the results in Figure 2. Further details of the experiment are provided in Appendix J.

## 8 Conclusion

This work introduces a continuous-time model of anchor acceleration, the anchor ODE $\dot{X} \in -\mathbb{A}(X) - \beta(t)(X - X_0)$. We characterize the convergence rate as a function of $\beta(t)$ and thereby obtain insight into the anchor acceleration mechanism. Finally, inspired by the continuous-time analyses, we present an adaptive method and establish its effectiveness through theoretical analyses and experiments.

Prior work analyzing continuous-time models of Nesterov acceleration had inspired various follow-up research, such as analyses based on Lagrangian and Hamiltonian mechanics [71, 72, 26], high-resolution ODE model [62], and continuized framework [29]. Carrying out similar analyses for the anchor ODE are interesting directions of future work.

## Acknowledgments and Disclosure of Funding

This work was supported by the Samsung Science and Technology Foundation (Project Number SSTF-BA2101-02). We thank Jaeyeon Kim for providing valuable feedback. We thank Hangjun Cho for the helpful discussions on well-posedness of ODEs. We also thank anonymous reviewers for the constructive comments.

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

# A  Prior work

**Acceleration for smooth convex function in discrete setting.**    There had been rich amount of research on acceleration about smooth convex functions. Nesterov [51] introduced accelerated gradient method (AGM), which has a faster $\mathcal{O}(1/k^2)$ rate than $\mathcal{O}(1/k)$ rate of gradient descent [22] in reducing the function value. Optimized gradient method (OGM) [36] improved AGM's rate by a constant factor, and is proven to be optimal [27]. For smooth strongly convex setup, strongly convex AGM [52] achieves an accelerated rate, and further improvements were studied [70, 53, 68, 61]. Recently, OGM-G [37] was introduced as an accelerated method reducing squared gradient magnitude for smooth convex minimization.

**Acceleration for smooth convex function in continuous setting.**    Continuous-time analysis of Nesterov acceleration has been thoroughly studied as well. Su et al. [65, 66] introduced an ODE model of AGM $\dot{X}(t) + \frac{r}{t}X + \nabla f(X) = 0$, providing $f(X(t)) - f_\star \in \mathcal{O}\left(1/t^2\right)$ rate for $r \geq 3$. Attouch et al. [7] improved the constant of bound for $r > 3$ and proved convergence of the trajectories. Attouch et al. [9] achieved $\mathcal{O}\left(1/t^{-2r/3}\right)$ rate for $0 < r < 3$. Apidopoulos et al. [2] generalized their results to differential inclusion with non-differentiable convex function. Furthermore, wide range of variations of AGM ODE has been studied [3, 8, 12, 16, 18, 5, 10, 19]. Also, applications to monotone inclusion problem were studied by Attouch and Peypouquet [6], Attouch and László [4], Boţ and Hulett [17].

Motivated from above continuous-time analysis for accelerated methods, tools analyzing ODEs have further developed. Wibisono et al. [71], Wilson et al. [72] and Kim and Yang [38] adopted Lagrangian mechanics and introduced first, second, unified Bregman Lagrangian to provide unified analysis for generalized family of ODE, where the latter provided analysis for strongly convex AGM. Systemical approach to obtain Lyapunov functions exploiting Hamiltonian mechanics [26] and dilated coordinate system [67] were proposed, and analysis of OGM-G was provided by dilated coordinate framework. Different forms of continuous-time models such as high-resolution ODE [62] and continuized framework [29] were developed.

On the other hand, another type of acceleration called *anchor acceleration* recently gained attention. As Yoon and Ryu [76] focused, many recently discovered accelerated methods for both minimax optimization and fixed-point problems are based on anchor acceleration. More recently, practical applications of anchor acceleration to detecting infeasibility for constrained optimization problems [55], and accelerating value iteration for dynamic programming and reinforcement learning [41] were introduced as well.

**Fixed-point problem.**    The history of studies on fixed-point problem dates back to the work of Banach [14], which established that the Picard iteration with contractive operator is convergent. Kransnosel'skii-Mann iteration (KM) [40, 47] was introduced, which is a generalization of Picard iteration. Convergence of KM iteration with general nonexpansive operators was proven by Martinet [49]. For iteration of Halpern [34], convergence with wide choice of parameter were shown by Wittmann [73].

The squared norm $\|y_k - \mathbb{T}y_k\|^2$ of fixed-point residual is a common error measure for fixed-point problems. KM iteration was shown to exhibit $\mathcal{O}(1/k)$ rate [23, 44, 21] and $o(1/k)$-rate [13, 50]. For Halpern iteration, $\mathcal{O}(1/(\log k)^2)$-rate was established by Leustean [43], then improve to $\mathcal{O}(1/k)$ rate by Kohlenbach [39]. First accelerated $\mathcal{O}(1/k^2)$ rate was achieved by Sabach and Shtern [60] and the constant was improved by Lieder [45] by a factor of 16.

It is known that there is an equivalence between solving fixed-point problem and solving monotone inclusion problem [30, 28, 54]. Proximal point method (PPM) [48] achieves $\mathcal{O}\left(1/k\right)$-rate in terms of $\left\|\tilde{\mathbb{A}}x_k\right\|^2$ [33]. Accelerated proximal point method (APPM) [35] improved the rate to accelerated $\mathcal{O}\left(1/k^2\right)$-rate. Park and Ryu [54] showed APPM is exactly optimal method for this problem and provided exactly optimal method for $\mu$-strongly monotone operator named OS-PPM, which achieved $\mathcal{O}\left(1/e^{4\mu k}\right)$ rate. The optimal methods APPM and OS-PPM are based on anchor acceleration [76].

**Minimax problems.**    Minimax optimization problem of the form $\min_x \max_y \mathbf{L}(x, y)$ have recently gained attention in machine learning society. One of the commonly considered theoretical setting is smooth convex-concave setup, with squared gradient norm as error measure. In terms of $\|\partial \mathbf{L}(x, y)\|^2$, classical EG [64] and OG [56, 57, 24] was shown to achieved $\mathcal{O}\left(1/k\right)$-rate [32]. SGDA [59] achieved $\mathcal{O}\left(1/k^{2-2p}\right)$ rate for $p > 1/2$ with introducing the term *anchor*. With introducing a parameter-free Halpern type method, Diakonikolas [25] achieved $\mathcal{O}(\log k/k^2)$. Recently, EAG [75] first achieved accelerated rate $\mathcal{O}(1/k^2)$ with anchor acceleration, followed by FEG [42], anchored Popov's scheme [69] and moving anchor methods [1]. Fast ODGA [20] also achieved accelerated $o(1/k^2)$ rate. For $\partial \mathbf{L}$ is furthermore strongly monotone with condition number $\kappa$, SM-EAG+ [76] achieved accelerated $\mathcal{O}\left(1/e^{2k\kappa}\right)$ rate.

However, continuous-time analysis for anchor acceleration is, to the best of our knowledge, insufficient. Continuous-time analyses of acceleration for monotone inclusion problem were studied by Bot et al. [20], Lin and Jordan [46], but they did not consider anchor acceleration. Ryu et al. [59] considered continuous-time analysis of anchor acceleration, but only with limited cases $\dot{X}(t) = -\mathbb{A}(X(t)) - \frac{\gamma}{t}(X - X_0)$ for $\gamma \geq 1$. In this paper, we provide a unified continuous-time analysis for anchor acceleration with generalized anchor coefficient.

# B  Proof of Theorem 2.2

## B.1  Proof of uniqueness

Proof of uniqueness is immediate from Lemma 2.3, we first prove the lemma.

*Proof of Lemma 2.3.* It is trivial for $t = 0$, so we may assume $t > 0$.
By monotonicity of $\mathbb{A}$ and Young's inequality, we get the following inequality.

$$
\begin{aligned}
\frac{d}{dt} \|X(t) - Y(t)\|^2 &= 2 \left\langle \dot{X}(t) - \dot{Y}(t), X(t) - Y(t) \right\rangle \\
&= -2 \left\langle \tilde{\mathbb{A}}(X(t)) - \tilde{\mathbb{A}}(Y(t)) + \beta(t)(X(t) - Y(t)) - \beta(t)(X_0 - Y_0), X(t) - Y(t) \right\rangle \\
&\leq -2\beta(t) \|X(t) - Y(t)\|^2 + 2\beta(t) \langle X_0 - Y_0, X(t) - Y(t) \rangle \\
&\leq -2\beta(t) \|X(t) - Y(t)\|^2 + \beta(t) \left( \|X_0 - Y_0\|^2 + \|X(t) - Y(t)\|^2 \right) \\
&= -\beta(t) \|X(t) - Y(t)\|^2 + \beta(t) \|X_0 - Y_0\|^2 .
\end{aligned}
$$

Now define $C(t) = e^{\int_v^t \beta(s)ds}$ for some $v > 0$, then we see $\dot{C}(t) = C(t)\beta(t)$. Moving $-\beta(t) \|X(t) - Y(t)\|^2$ to the left hand side and multiplying both sides by $C(t)$, we have

$$
\begin{aligned}
\frac{d}{dt} \left( C(t) \|X(t) - Y(t)\|^2 \right) &= C(t) \frac{d}{dt} \|X(t) - Y(t)\|^2 + C(t)\beta(t) \|X(t) - Y(t)\|^2 \\
&\leq C(t)\beta(t) \|X_0 - Y_0\|^2 = \frac{d}{dt} C(t) \|X_0 - Y_0\|^2
\end{aligned}
$$

Integrating both sides from $\epsilon > 0$ to $t$ we have

$$
C(t) \|X(t) - Y(t)\|^2 - C(\epsilon) \|X(\epsilon) - Y(\epsilon)\|^2 \leq C(t) \|X_0 - Y_0\|^2 - C(\epsilon) \|X_0 - Y_0\|^2 .
$$

As $C$ is nonnegative and nondecreasing, $\lim_{\epsilon \to 0+} C(\epsilon)$ exists. Taking limit $\epsilon \to 0+$ both sides we have

$$
C(t) \|X(t) - Y(t)\|^2 \leq C(t) \|X_0 - Y_0\|^2 .
$$

Finally, dividing both sides by $C(t)$ we conclude

$$
\|X(t) - Y(t)\|^2 \leq \|X_0 - Y_0\|^2 .
$$

$\square$

Proof for uniqueness can be done to generalized case (3).

**Theorem B.1** (Uniqueness of solutions)**.** *If the solution for* (3) *exists, it is unique.*

*Proof.* Suppose $X_1, X_2$ are solutions of (3) with same initial value $X_0$. By Lemma 2.3, we have
$$
\|X_1(t) - X_2(t)\| \leq \|X_0 - X_0\| = 0
$$
Therefore $X_1(t) = X_2(t)$ for all $t \in [0, \infty)$, we get the desired result. $\square$

As (6) is special case of (3), uniqueness proof for Theorem 2.2 follows from Theorem B.1.

## B.2  Proof of existence

The proof of existence needs tedious work due to the singularity of $\frac{\gamma}{t^p}$ at $t = 0$, we provide our proof through subsections. Before we start, we provide a short outline of the proof. The proof is basically based on the proof provided in [65] and [11].

We first prove the case $\mathbb{A}$ is Lipschitz. The differential inclusion becomes ODE when $\mathbb{A}$ is Lipschitz, we can adopt similar argument done in [65]. We consider series of ODEs Lipschitz with respect to $X$, approximating $\dot{X}(t) = -\mathbb{A}(X(t)) - \frac{\gamma}{t^p}(X_0 - X(t))$. The approximated ODEs have solutions by classical theory of ODE, we obtain the true solution by considering proper subsequence of solutions.

As the solution for Lipschitz $\mathbb{A}$ is obtained, we can adopt similar argument done in [11]. We first consider solution $X_\lambda$ with Yosida approximation $\mathbb{A}_\lambda = \frac{1}{\lambda}(\mathbb{I} - (\mathbb{I} + \lambda\mathbb{A})^{-1})$, which is an approximation of $\mathbb{A}$ that is Lipschitz continuous. Then we can obtain a subsequence $X_{\lambda_n}$ converging to original differential inclusion.

### B.2.1 Existence proof for Lipschitz $\mathbb{A}$

Since we will approximate $\mathbb{A}$ with Lipschitz continuous functions, we first consider the ODE with Lipschitz continuous $\mathbb{A}$.

**Theorem B.2.** *(Existence of solution for Lipschitz $\mathbb{A}$)*
*Suppose $\tilde{A} : \mathbb{R}^n \to \mathbb{R}^n$ is L-Lipschitz continuous function. Consider the differential equation, with initial value condition $\tilde{X}(0) = X_0 \in dom(\tilde{A})$,*

$$\dot{\tilde{X}}(t) = -\tilde{A}(\tilde{X}(t)) - \frac{\gamma}{t^p}(\tilde{X}(t) - X_0). \tag{12}$$

*where $\gamma, p > 0$. Then there exists a unique solution $\tilde{X} \in \mathcal{C}^1([0, \infty), \mathbb{R}^n)$ that satisfies (12) for all $t \in (0, \infty)$. Moreover, for $\dot{\tilde{X}}(0)$ defined by $\dot{\tilde{X}}(0) = \lim_{t \to 0+} \frac{\tilde{X}(t) - X_0}{t}$, following is true*

$$\dot{\tilde{X}}(0) = \begin{cases} -\tilde{A}(X_0) & \text{if } 0 < p < 1 \\ -\frac{1}{\gamma+1}\tilde{A}(X_0) & \text{if } p = 1 \\ 0 & \text{if } p > 1. \end{cases}$$

The proof need some preparation. We will think of approximated solutions $\tilde{X}_\delta$, obtain a sequence that converges to solution $\tilde{X}$. Thus we first define $\tilde{X}_\delta$. From now, we will denote $\tilde{A}$ as a $L$-Lipschitz monotone function.

**Definition B.3.** Let $0 < \delta < 1$. Consider

$$\dot{\tilde{X}}_\delta(t) = \begin{cases} -\tilde{A}(\tilde{X}_\delta(t)) - \frac{\gamma}{\delta^p}(\tilde{X}_\delta(t) - X_0) & 0 \le t < \delta \\ -\tilde{A}(\tilde{X}_\delta(t)) - \frac{\gamma}{t^p}(\tilde{X}_\delta(t) - X_0) & t > \delta \end{cases} \tag{13}$$

Since right hand side above is $\left(L + \frac{\gamma}{\delta^p}\right)$-Lipschitz with respect to $\tilde{X}_\delta$, the solution uniquely exists by classical ODE theory. Define the solution as $\tilde{X}_\delta$. Then for positive sequence $\{\delta_m\}_{m\in\mathbb{N}}$ such that $\delta_m < 1$ and $\lim_{m\to\infty} \delta_m = 0$, consider sequence $\left\{\tilde{X}_{\delta_m}\right\}_{m\in\mathbb{N}}$.

Before we start, we prove a useful lemma we will widely use for the cases that operator is Lipschitz continuous.

**Lemma B.4.** *Let $\tilde{A} : \mathbb{R}^n \to \mathbb{R}^n$ a Lipschitz continuous function. Suppose $\beta : D \to [0, \infty)$ be a continuous function with $D \subset [0, \infty)$. Consider differential equation*

$$\dot{\tilde{X}} = -\tilde{A}(\tilde{X}) - \beta(t)(\tilde{X} - X_0),$$

*with $X_0 \in \mathbb{R}^n$. Let $\tilde{X} : D \to \mathbb{R}^n$ be a differentiable curve that satisfies above equation for $t \in D$. Then for all $0 \le a < b$ such that $[a, b] \subset D$, $\tilde{X}$ and the composition $\tilde{A} \circ \tilde{X} : [a, b] \to \mathbb{R}^n$ are is Lipschitz continuous.*

*Moreover if $\dot{\beta}(t)$ is well-defined and bounded for almost all $t \in [a, b]$, then $\dot{\tilde{X}}$ is Lipschitz continuous in $[a, b]$. Thus if $\beta$ is twice differentiable, $\dot{\tilde{X}}$ is Lipschitz continuous.*

*As Lipschitz continuous functions, $\tilde{X}$, $\tilde{A} \circ \tilde{X}$ and $\dot{\tilde{X}}$ are absolutely continuous functions.*

*Proof.* We first prove $\tilde{X}$ is Lipschitz continuous. As $\tilde{A}, \tilde{X}, \beta$ are continuous in $[a, b]$ we see $\tilde{A}(\tilde{X}(t)) + \beta(t)(\tilde{X}(t) - X_0)$ is continuous in $[a, b]$, thus

$$M_1 = \max_{t\in[a,b]} \left\| \dot{\tilde{X}}(t) \right\| = \max_{t\in[a,b]} \left\| \tilde{A}(\tilde{X}(t)) + \beta(t)(\tilde{X}(t) - X_0) \right\|$$

exists. Since its derivative is bounded by $M_1 < \infty$, we have $\tilde{X}$ is $M_1$-Lipshitz continuous. As composition of two Lipschitz continuous functions, $\tilde{A} \circ \tilde{X}$ is Lipschitz continuous.

First observe if $\beta$ is twice differentiable, $\dot{\beta}$ is bounded in $[a, b]$ as it is continuous. Now if $\dot{\beta}(t)$ is bounded for $t \in [a, b]$, i.e. $\left| \dot{\beta}(t) \right| \le M_2$ for some $M_2 > 0$, for $M_3 = \max_{t\in[a,b]} |\beta(t)|$ we have

$$\left\| \frac{d}{dt}\left( \beta(t)(\tilde{X}(t) - X_0) \right) \right\| = \left\| \dot{\beta}(t)(\tilde{X}(t) - X_0) \right\| + \left\| \beta(t)\dot{\tilde{X}}(t) \right\| \le M_2 M_1 |b - a| + M_3 M_1$$

Therefore $\beta(t)(\tilde{X}(t) - X_0)$ is $(M_2 M_1 |b - a| + M_3 M_1)$-Lipschitz continuous. Thus as a sum of Lipschitz continuous functions, $\dot{\tilde{X}}$ is Lispchitz continuous. $\square$

We will show for every $T > 0$, the set of derivatives $\left\{ \dot{\tilde{X}}_\delta \mid \delta \in (0,1) \right\}$ is uniformly bounded on $[0,T]$ and $\left\{ \tilde{X}_{\delta_m} \right\}_{m \in \mathbb{N}}$ converges uniformly on $[0,T]$. We first prove the boundedness of derivatives. To do so, we first prove a useful lemma.

**Lemma B.5.** *For $0 < a < b$, suppose $\tilde{X} : [a,b] \to \mathbb{R}^n$ satisfies ODE (12) for $t \in [a,b]$. Define $\tilde{U} : [a,b] \to \mathbb{R}$ as*

$$\tilde{U}_1(t) = \left\| \dot{\tilde{X}}(t) \right\|^2 + \frac{\gamma p}{t^{p+1}} \left\| \tilde{X}(t) - X_0 \right\|^2$$

$$\tilde{U}_2(t) = \frac{1}{t^{p-1}} \left\| \dot{\tilde{X}}(t) \right\|^2 + \frac{\gamma p}{t^{2p}} \left\| \tilde{X}(t) - X_0 \right\|^2$$

*Then $\tilde{U}_1$ is nonincreasing if for $0 < p \le 1$, and $\tilde{U}_2$ is nonincreasing for $p > 1$.*

*Proof.* From Lemma B.4 we can check $\tilde{U}_1$ and $\tilde{U}_2$ are absolutely continuous in $[a,b]$. Therefore it is enough to check the derivative is nonpositive for almost all $t$. From Lemma B.4, we can differentiate (12) both sides. Thus for almost all $t$ we have

$$\ddot{\tilde{X}}(t) = -\frac{d}{dt} \tilde{A}(\tilde{X}(t)) + \frac{\gamma p}{t^{p+1}} (\tilde{X}(t) - X_0) - \frac{\gamma}{t^p} \dot{\tilde{X}}(t).$$

Recall from monotonicity of $\tilde{A}$ and (1), we know $\left\langle \dot{\tilde{X}}(t), \frac{d}{dt} \tilde{A}(\tilde{X}(t)) \right\rangle \ge 0$ for almost all $t$. Therefore for almost all $t$,

(i) $0 < p \le 1$

$$\dot{\tilde{U}}_1(t) = 2 \left\langle \dot{\tilde{X}}(t), \ddot{\tilde{X}}(t) \right\rangle + \frac{2\gamma p}{t^{p+1}} \left\langle \dot{\tilde{X}}(t), \tilde{X}(t) - X_0 \right\rangle - \frac{\gamma p(p+1)}{t^{p+2}} \left\| \tilde{X}(t) - X_0 \right\|^2$$

$$= -2 \left\langle \dot{\tilde{X}}(t), \frac{d}{dt} \tilde{A}(\tilde{X}(t)) \right\rangle - \frac{2\gamma}{t^p} \left\| \dot{\tilde{X}}(t) \right\|^2 + \frac{2\gamma p}{t^{p+1}} \left\langle \dot{\tilde{X}}(t), \tilde{X}(t) - X_0 \right\rangle$$

$$+ \frac{2\gamma p}{t^{p+1}} \left\langle \dot{\tilde{X}}(t), \tilde{X}(t) - X_0 \right\rangle - \frac{2\gamma p^2}{t^{p+2}} \left\| \tilde{X}(t) - X_0 \right\|^2 - \frac{\gamma p(1-p)}{t^{p+2}} \left\| \tilde{X}(t) - X_0 \right\|^2$$

$$= -2 \left\langle \dot{\tilde{X}}(t), \frac{d}{dt} \tilde{A}(\tilde{X}(t)) \right\rangle - \frac{2\gamma}{t^p} \left\| \dot{\tilde{X}}(t) - \frac{p}{t}(\tilde{X} - X_0) \right\|^2 - \frac{\gamma p(1-p)}{t^{p+2}} \left\| \tilde{X}(t) - X_0 \right\|^2$$

$$\le 0.$$

(ii) $p > 1$

$$\dot{\tilde{U}}_2(t) = -\frac{p-1}{t^p} \left\| \dot{\tilde{X}}(t) \right\|^2 + \frac{2}{t^{p-1}} 2 \left\langle \dot{\tilde{X}}(t), \ddot{\tilde{X}}(t) \right\rangle + \frac{2\gamma p}{t^{2p}} \left\langle \dot{\tilde{X}}(t), \tilde{X} - X_0 \right\rangle - \frac{2\gamma p}{t^{2p+1}} \left\| \tilde{X}(t) - X_0 \right\|^2$$

$$= -\frac{p-1}{t^p} \left\| \dot{\tilde{X}}(t) \right\|^2 - \frac{2}{t^{p-1}} \left\langle \dot{\tilde{X}}(t), \frac{d}{dt} \tilde{A}(\tilde{X}(t)) \right\rangle$$

$$- \frac{2\gamma}{t^{2p-1}} \left\| \dot{\tilde{X}}(t) \right\|^2 + \frac{2\gamma p}{t^{2p}} \left\langle \dot{\tilde{X}}(t), \tilde{X} - X_0 \right\rangle + \frac{2\gamma p}{t^{2p}} \left\langle \dot{\tilde{X}}(t), \tilde{X} - X_0 \right\rangle - \frac{2\gamma p}{t^{2p+1}} \left\| \tilde{X}(t) - X_0 \right\|^2$$

$$= -\frac{p-1}{t^p} \left\| \dot{\tilde{X}}(t) \right\|^2 - \frac{2}{t^{p-1}} \left\langle \dot{\tilde{X}}(t), \frac{d}{dt} \tilde{A}(\tilde{X}(t)) \right\rangle - \frac{2\gamma}{t^{2p-1}} \left\| \dot{\tilde{X}}(t) - \frac{p}{t}(\tilde{X} - X_0) \right\|^2$$

$$\le 0.$$

$\square$

We now are ready to prove uniform boundedness of derivatives $\dot{\tilde{X}}_\delta$.

**Lemma B.6.** *Take $T > 0$. For all $t \in [0,T]$ and $\delta \in (0,1)$, below inequality holds.*

$$\left\| \dot{\tilde{X}}_\delta(t) \right\| \le \tilde{M}_{dot}(T) = \begin{cases} \sqrt{1 + \gamma p} \left\| \tilde{A}(X_0) \right\| & 0 < p \le 1 \\ \sqrt{T^{p-1} \left( \frac{1}{2\gamma} + 2p \left( 2L^2 + 2\gamma + 1 \right) \right)} \left\| \tilde{A}(X_0) \right\| & p \ge 1. \end{cases}$$

*Proof.* We prove first statement by considering two cases.

(1) $t \in [0, \delta]$

From Lemma B.4, we know $\dot{\tilde{X}}_\delta$ is absolutely continuous and so $\left\|\dot{\tilde{X}}_\delta\right\|^2$ is absolutely continuous as well. Differentiating (13), for almost all $t \in (0, \delta)$ we have

$$\ddot{\tilde{X}}_\delta = -\frac{d}{dt}\tilde{A}(\tilde{X}_\delta(t)) - \frac{\gamma}{\delta^p}\dot{\tilde{X}}_\delta.$$

Now for almost all $t \in (0, \delta)$,

$$\frac{d}{dt}\left\|\dot{\tilde{X}}_\delta\right\|^2 = 2\left\langle \dot{\tilde{X}}_\delta, \ddot{\tilde{X}}_\delta \right\rangle = -2\left\langle \dot{\tilde{X}}_\delta, \frac{d}{dt}\tilde{A}(\tilde{X}_\delta(t)) \right\rangle - \frac{2\gamma}{\delta^p}\left\|\dot{\tilde{X}}_\delta\right\|^2 \leq -\frac{2\gamma}{\delta^p}\left\|\dot{\tilde{X}}_\delta\right\|^2 \leq 0.$$

  (i) $0 < p \leq 1$

From above we know $\left\|\dot{\tilde{X}}_\delta\right\|$ is nonincreasing in $t \in [0, \delta]$, we have $\left\|\dot{\tilde{X}}_\delta(t)\right\| \leq \left\|\dot{\tilde{X}}_\delta(0)\right\| = \left\|\tilde{A}(X_0)\right\|$.

  (ii) $p > 1$

Integrating $\frac{d}{dt}\left\|\dot{\tilde{X}}_\delta\right\|^2 \leq -\frac{2\gamma}{\delta^p}\left\|\dot{\tilde{X}}_\delta\right\|^2$ we have

$$\left\|\dot{\tilde{X}}_\delta(t)\right\|^2 - \left\|\dot{\tilde{X}}_\delta(0)\right\|^2 \leq -\frac{2\gamma}{\delta^p}\int_0^t \left\|\dot{\tilde{X}}_\delta(s)\right\|^2 ds$$

$$\leq -\frac{2\gamma}{\delta^p}\int_0^t \left\|\dot{\tilde{X}}_\delta(t)\right\|^2 ds = -\frac{2\gamma t}{\delta^p}\left\|\dot{\tilde{X}}_\delta(t)\right\|^2.$$

Organizing with respect to $\left\|\dot{\tilde{X}}_\delta(t)\right\|$ we have

$$\left\|\dot{\tilde{X}}_\delta(t)\right\| \leq \frac{\left\|\dot{\tilde{X}}_\delta(0)\right\|}{\sqrt{1 + \frac{2\gamma t}{\delta^p}}} = \sqrt{\frac{\delta^p}{\delta^p + 2\gamma t}}\left\|\tilde{A}(X_0)\right\| \leq \sqrt{\frac{\delta^p}{2\gamma t}}\left\|\tilde{A}(X_0)\right\|.$$

(2) $t \geq \delta$

Note for $t > \delta$ (12) holds, we can apply Lemma B.5.

  (i) $0 < p \leq 1$

From (i) we know $\left\|\dot{\tilde{X}}_\delta(t)\right\| \leq \left\|\tilde{A}(X_0)\right\|$ for $t \in [0, \delta]$. Therefore $\left\|\dot{\tilde{X}}_\delta(\delta)\right\| \leq \left\|\tilde{A}(X_0)\right\|$ and we see

$$\left\|\tilde{X}_\delta(\delta) - X_0\right\| = \left\|\int_0^\delta \dot{\tilde{X}}_\delta(s)\,ds\right\| \leq \int_0^\delta \left\|\dot{\tilde{X}}_\delta(s)\right\|\,ds \leq \delta\left\|\tilde{A}(X_0)\right\|.$$

From Lemma B.5 we know $\tilde{U}_1(t) = \left\|\dot{\tilde{X}}_\delta(t)\right\|^2 + \frac{\gamma p}{t^{p+1}}\left\|\tilde{X}_\delta(t) - X_0\right\|^2$ is nonincreasing. Therefore $\left\|\dot{\tilde{X}}_\delta(t)\right\|^2 \leq \tilde{U}_1(\delta)$ for $t \geq \delta$. Since $\delta < 1$ we have

$$\left\|\dot{\tilde{X}}_\delta(t)\right\| \leq \sqrt{\tilde{U}_1(\delta)} = \sqrt{\left\|\dot{\tilde{X}}_\delta(\delta)\right\|^2 + \frac{\gamma p}{\delta^{p+1}}\left\|\tilde{X}_\delta(\delta) - X_0\right\|^2}$$

$$\leq \sqrt{\left\|\tilde{A}(X_0)\right\|^2 + \gamma p \delta^{1-p}\left\|\tilde{A}(X_0)\right\|^2} \leq \sqrt{1 + \gamma p}\left\|\tilde{A}(X_0)\right\|.$$

  (ii) $p > 1$

From (i) we know $\left\|\dot{\tilde{X}}_\delta(t)\right\| \leq \sqrt{\frac{\delta^p}{2\gamma t}}\left\|\tilde{A}(X_0)\right\|$ for $t \in [0, \delta]$. Therefore $\left\|\dot{\tilde{X}}_\delta(\delta)\right\| \leq \sqrt{\frac{\delta^{p-1}}{2\gamma}}\left\|\tilde{A}(X_0)\right\|$ and we see

$$\left\|\tilde{X}_\delta(\delta) - X_0\right\| \leq \int_0^\delta \left\|\dot{\tilde{X}}_\delta(s)\right\|\,ds \leq \int_0^\delta \sqrt{\frac{\delta^p}{2\gamma s}}\left\|\tilde{A}(X_0)\right\|\,ds = \sqrt{\frac{2\delta^{p+1}}{\gamma}}\left\|\tilde{A}(X_0)\right\|.$$

Applying (13) and recalling the fact $\tilde{A}$ is $L$-Lipschitz, we have

$$
\begin{aligned}
\frac{\gamma^2}{\delta^{2p}} \|X_\delta(\delta) - X_0\|^2 &= \left\| \tilde{A}(X_\delta(\delta)) + \dot{X}_\delta(\delta) \right\|^2 = \left\| \tilde{A}(X_\delta(\delta)) - \tilde{A}(X_0) + \tilde{A}(X_0) + \dot{X}_\delta(\delta) \right\|^2 \\
&\leq 2 \left\| \tilde{A}(X_\delta(\delta)) - \tilde{A}(X_0) \right\|^2 + 2 \left\| \tilde{A}(X_0) + \dot{X}_\delta(\delta) \right\|^2 \\
&\leq 2L^2 \left\| \tilde{X}_\delta(\delta) - X_0 \right\|^2 + 4 \left\| \tilde{A}(X_0) \right\|^2 + 4 \left\| \dot{X}_\delta(\delta) \right\|^2 \\
&\leq \left( \frac{4}{\gamma} L^2 \delta^{p+1} + 4 + \frac{2\delta^{p-1}}{\gamma} \right) \left\| \tilde{A}(X_0) \right\|^2 .
\end{aligned}
$$

From Lemma B.5 we know $\tilde{U}_2(t) = \frac{1}{t^{p-1}} \left\| \dot{\tilde{X}}(t) \right\|^2 + \frac{\gamma p}{t^{2p}} \left\| \tilde{X}(t) - X_0 \right\|^2$ is nonincreasing. Therefore $\frac{\left\| \dot{\tilde{X}}_\delta(t) \right\|^2}{t^{p-1}} \leq \tilde{U}_2(\delta)$ for $t \geq \delta$. Since $\delta < 1$ we have

$$
\begin{aligned}
\frac{\left\| \dot{\tilde{X}}_\delta(t) \right\|}{\sqrt{t^{p-1}}} &\leq \sqrt{\tilde{U}_2(\delta)} = \sqrt{ \frac{\left\| \dot{\tilde{X}}_\delta(\delta) \right\|^2}{\delta^{p-1}} + \frac{\gamma p}{\delta^{2p}} \left\| \tilde{X}_\delta(\delta) - X_0 \right\|^2 } \\
&\leq \sqrt{ \frac{\left\| \tilde{A}(X_0) \right\|^2}{2\gamma} + p \left( 4L^2 \delta^{p+1} + 4\gamma + 2\delta^{p-1} \right) \left\| \tilde{A}(X_0) \right\|^2 } \leq \sqrt{ \frac{1}{2\gamma} + 2p \left( 2L^2 + 2\gamma + 1 \right) } \left\| \tilde{A}(X_0) \right\| .
\end{aligned}
$$

Therefore for $t \in [0, T]$ we have

$$
\left\| \dot{\tilde{X}}_\delta(t) \right\| \leq \sqrt{ T^{p-1} \left( \frac{1}{2\gamma} + 2p \left( 2L^2 + 2\gamma + 1 \right) \right) } \left\| \tilde{A}(X_0) \right\|
$$

From (1) and (2), we get the desired result. $\qquad \square$

We now show the sequence $\left\{ \tilde{X}_{\delta_m} \right\}_{m \in \mathbb{N}}$ converges uniformly on $[0, T]$ for every $T > 0$. It is suffices to prove following proposition.

**Proposition B.7.** *For $T > 0$, the sequence $\left\{ \tilde{X}_{\delta_m} \right\}_{m \in \mathbb{N}}$ is a Cauchy sequence with respect to supremum norm on $[0, T]$.*

*Proof.* Take $\epsilon > 0$. We want to show, there is $N \in \mathbb{N}$ such that if $n, m > N$ then $\left\| \tilde{X}_{\delta_n}(t) - \tilde{X}_{\delta_m}(t) \right\| \leq \epsilon$ for all $t \in [0, \infty)$.

Define $d_{\delta, \nu}(t) = \frac{1}{2} \left\| \tilde{X}_\delta(t) - \tilde{X}_\nu(t) \right\|^2$. Without loss of generality, we may assume $\delta \geq \nu$. With $M_{dot}(T)$ defined in Lemma B.6, we will show $d_{\delta, \nu}(t) \leq 2\delta^2 \tilde{M}_{dot}(T)^2$.

First consider the case $0 \leq t \leq \delta$. By Lemma B.6, we have

$$
\begin{aligned}
\left\| \tilde{X}_\delta(t) - \tilde{X}_\nu(t) \right\| &\leq \int_0^t \left\| \dot{\tilde{X}}_\delta(s) - \dot{\tilde{X}}_\nu(s) \right\| ds \leq \int_0^t \left( \left\| \dot{\tilde{X}}_\delta(s) \right\| + \left\| \dot{\tilde{X}}_\nu(s) \right\| \right) ds \\
&\leq \int_0^t 2\tilde{M}_{dot}(T) ds = 2t\tilde{M}_{dot}(T)
\end{aligned}
$$

Thus for $t \in [0, \delta]$ we have

$$
d_{\delta, \nu}(t) \leq 2t^2 \tilde{M}_{dot}(T)^2 \leq 2\delta^2 \tilde{M}_{dot}(T)^2 .
$$

Now we consider the case $t \geq \delta$. By monotonicity of $\tilde{A}$, we have

$$
\begin{aligned}
\frac{d}{dt} \frac{1}{2} \left\| \tilde{X}_\delta(t) - \tilde{X}_\nu(t) \right\|^2 &= \left\langle \dot{X}_\delta(t) - \dot{X}_\nu(t), \tilde{X}_\delta(t) - \tilde{X}_\nu(t) \right\rangle \\
&= \left\langle -\left( \tilde{A}(\tilde{X}_\delta(t)) - \tilde{A}(\tilde{X}_\nu(t)) \right) - \frac{\gamma}{t^p}(X_\delta(t) - X_0) + \frac{\gamma}{t^p}(X_\nu(t) - X_0), X_\delta(t) - X_\nu(t) \right\rangle \\
&\leq \left\langle -\frac{\gamma}{t^p}(X_\delta(t) - X_0) + \frac{\gamma}{t^p}(X_\nu(t) - X_0), X_\delta(t) - X_\nu(t) \right\rangle \\
&= -\frac{\gamma}{t^p} \left\| X_\delta(t) - X_\nu(t) \right\|^2 \\
&\leq 0.
\end{aligned}
$$

Thus we have $d_{\delta,\nu}(t) \leq d_{\delta,\nu}(\delta)$ for $t \geq \delta$.

Now combining two cases, we have

$$
d_{\delta,\nu}(t) \leq 2\delta^2 \tilde{M}_{dot}(T)^2 \tag{14}
$$

Since $\lim_{k \to \infty} \delta_n = 0$, there is $N \in \mathbb{N}$ such that $m > n > N$ implies $d_{\delta_n, \delta_m} \leq \frac{\epsilon^2}{2}$, we're done. $\qquad \square$

We are now ready to prove Theorem B.2.

*Proof.* (1) *Existence of solution.*

From Proposition B.7, we know $\left\{ \tilde{X}_{\delta_m} \right\}_{m \in \mathbb{N}}$ converging uniformly on $[0, T]$ for every $T >$. Denote the limit as $\tilde{X}$, i.e. define $\tilde{X} \colon [0, \infty) \to \mathbb{R}^n$ as

$$
\lim_{m \to \infty} \tilde{X}_{\delta_m}(t) = \tilde{X}(t).
$$

We can check $\tilde{X}(0) = X_0$ easily since $\tilde{X}_{\delta_m}(0) = X_0$ for all $m \in \mathbb{N}$. It remains to show $\tilde{X}$ satisfies (12).

Take $t > 0$. We wish to show

$$
\lim_{h \to 0} \left\| \frac{\tilde{X}(t+h) - \tilde{X}(t)}{h} + \tilde{A}(\tilde{X}(t)) + \frac{\gamma}{t^p} \left( \tilde{X}(t) - X_0 \right) \right\| = 0.
$$

Consider $h, \delta, T$ such that $0 < |h| < t$, $0 < \delta < \min\{1, t - |h|\}$ and $T > t + |h|$. Then $(t - |h|, t + |h|) \in [0, T]$ and $\dot{\tilde{X}}_\delta(t) = -\tilde{A}(\tilde{X}_\delta(t)) - \frac{\gamma}{t^p} \left( \tilde{X}_\delta(t) - X_0 \right)$. Consider inequality

$$
\begin{aligned}
&\left\| \frac{\tilde{X}(t+h) - \tilde{X}(t)}{h} + \tilde{A}(\tilde{X}(t)) + \frac{\gamma}{t^p} \left( \tilde{X}(t) - X_0 \right) \right\| \\
&\leq \left\| \frac{\tilde{X}(t+h) - \tilde{X}(t)}{h} - \frac{\tilde{X}_\delta(t+h) - \tilde{X}_\delta(t)}{h} \right\| + \left\| \frac{\tilde{X}_\delta(t+h) - \tilde{X}_\delta(t)}{h} - \dot{\tilde{X}}_\delta(t) \right\| + \left\| \dot{\tilde{X}}_\delta(t) + \tilde{A}(\tilde{X}(t)) + \frac{\gamma}{t^p} \left( \tilde{X}(t) - X_0 \right) \right\|.
\end{aligned}
$$

We now show right hand side goes to zero as $h \to 0$. The point of the proof is, $\tilde{X}_\delta$ converges uniformly to $\tilde{X}$ and $\dot{\tilde{X}}_\delta$ is uniformly bounded on $[t - |h|, T]$.

From (14) we have

$$
\begin{aligned}
&\left\| \frac{\tilde{X}(t+h) - \tilde{X}(t)}{h} - \frac{\tilde{X}_\delta(t+h) - \tilde{X}_\delta(t)}{h} \right\| \\
&\leq \frac{1}{|h|} \left( \left\| \tilde{X}(t+h) - \tilde{X}_\delta(t+h) \right\| + \left\| \tilde{X}(t) - \tilde{X}_\delta(t) \right\| \right) \leq 4 \frac{\delta}{|h|} \tilde{M}_{dot}(T).
\end{aligned}
$$

Also from (14) and since $\tilde{A}$ is $L$-Lipschitz continuous,

$$
\begin{aligned}
\left\| \dot{\tilde{X}}_\delta(t) + \tilde{A}(\tilde{X}(t)) + \frac{\gamma}{t^p} \left( \tilde{X}(t) - X_0 \right) \right\| &= \left\| -\left( \tilde{A}(\tilde{X}_\delta(t)) + \frac{\gamma}{t^p} \left( \tilde{X}_\delta(t) - X_0 \right) \right) + \tilde{A}(\tilde{X}(t)) + \frac{\gamma}{t^p} \left( \tilde{X}(t) - X_0 \right) \right\| \\
&\leq L \left\| \tilde{X}_\delta(t) - \tilde{X}(t) \right\| + \frac{\gamma}{t^p} \left\| \tilde{X}_\delta(t) - \tilde{X}(t) \right\| \leq 2\delta \left( L + \frac{\gamma}{t^p} \right) \tilde{M}_{dot}(T).
\end{aligned}
$$

Now from Lemma B.4 we have $\dot{\tilde{X}}_\delta(t)$ is absolutely continuous, thus

$$\left\|\frac{\tilde{X}_\delta(t+h)-\tilde{X}_\delta(t)}{h}-\dot{\tilde{X}}_\delta(t)\right\|=\left\|\frac{\int_t^{t+h}\left(\dot{\tilde{X}}_\delta(s)-\dot{\tilde{X}}_\delta(t)\right)ds}{h}\right\|$$

$$=\frac{1}{|h|}\left\|\int_t^{t+h}\int_t^s\ddot{\tilde{X}}_\delta(u)\,du\,ds\right\|\leq\frac{1}{|h|}\left|\int_t^{t+h}\int_t^s\left\|\ddot{\tilde{X}}_\delta(u)\right\|\,du\,ds\right|$$

Observe, for almost every $u\in[t-|h|,t+|h|]\subset[0,T]$ by Lemma B.6 we have

$$\left\|\ddot{\tilde{X}}_\delta(u)\right\|=\left\|-\frac{d}{du}\tilde{A}(\tilde{X}_\delta(u))+\frac{p\gamma}{u^{p+1}}\left(\tilde{X}_\delta(u)-X_0\right)-\frac{\gamma}{u^p}\dot{X}_\delta(u)\right\|$$

$$=\left\|\frac{d}{du}\tilde{A}(\tilde{X}_\delta(u))\right\|+\left\|\frac{p\gamma}{u^{p+1}}\int_0^u\dot{X}_\delta(v)dv\right\|+\left\|\frac{\gamma}{u^p}\dot{X}_\delta(u)\right\|$$

$$\leq L\tilde{M}_{dot}(T)+\frac{p\gamma}{(t-|h|)^{p+1}}\int_0^T\tilde{M}_{dot}(T)dv+\frac{\gamma}{(t-|h|)^p}\tilde{M}_{dot}(T)$$

$$=\left(L+\frac{p\gamma T}{(t-|h|)^{p+1}}+\frac{\gamma}{(t-|h|)^p}\right)\tilde{M}_{dot}(T)=:M.$$

Note $M$ is independent of $h$ or $\delta$. While obtaining the inequality, we used the fact that $\tilde{A}(\tilde{X}_\delta(\cdot))$ is $L\tilde{M}_{dot}(T)$-Lipschitz continuous in $[0,T]$. Now

$$\frac{1}{|h|}\left|\int_t^{t+h}\int_t^s\left\|\ddot{\tilde{X}}_\delta(u)\right\|\,du\,ds\right|\leq\frac{1}{|h|}\left|\int_t^{t+h}\int_t^s M\,du\,ds\right|=\frac{1}{|h|}M\left|\int_t^{t+h}(s-t)\,ds\right|=\frac{|h|}{2}M.$$

Now consider $\delta=h^2$ with $h$ small enough that satisfies $|h|<1$ and $|h|+h^2<t$. Then the conditions $|h|<t$, $0<\delta<\min\{1,t-|h|\}$ hold, above arguments are valid. Gathering above results, we have

$$\left\|\frac{\tilde{X}(t+h)-\tilde{X}(t)}{h}+\tilde{A}(\tilde{X}(t))+\frac{\gamma}{t^p}\left(\tilde{X}(t)-X_0\right)\right\|$$

$$\leq 2\sqrt{2}|h|\tilde{M}_{dot}(T)+\sqrt{2}|h|^2\left(L+\frac{\gamma}{t^p}\right)\tilde{M}_{dot}(T)+\frac{|h|}{2}M=\mathcal{O}\left(|h|\right).$$

which implies the desired result.

(2) *The value and continuity of $\dot{\tilde{X}}(t)$ at $t=0$.*
   Define $C(t)$ as

$$C(t)=\begin{cases}t^\gamma & p=1\\ e^{\frac{\gamma}{1-p}t^{1-p}} & p>0,p\neq1.\end{cases}$$

Then we see for $t>0$

$$\frac{d}{dt}\left(C(t)\left(\tilde{X}(t)-X_0\right)\right)=C(t)\left(\dot{\tilde{X}}(t)+\frac{\gamma}{t^p}(\tilde{X}(t)-X_0)\right)=-C(t)\tilde{A}(\tilde{X}(t)).$$

Integrating both sides from $\epsilon>0$ to $t$ we have

$$C(t)(\tilde{X}(t)-X_0)-C(\epsilon)(\tilde{X}(\epsilon)-X_0)=-\int_\epsilon^t C(s)\tilde{A}(\tilde{X}(s))\,ds.$$

As $C$ is a nondecreasing function and bounded below by 0, $\lim_{\epsilon\to0+}C(\epsilon)$ exists, taking limit $\epsilon\to0+$ we have

$$C(t)(\tilde{X}(t)-X_0)=-\int_0^t C(s)\tilde{A}(\tilde{X}(s))\,ds.$$

Dividing both sides by $tC(t)$, with change of variable $v = s/t$, we have

$$\frac{\tilde{X}(t) - X_0}{t} = -\int_0^t \frac{C(s)}{C(t)}\tilde{A}(\tilde{X}(s))\frac{ds}{t} = -\int_0^1 \frac{C(tv)}{C(t)}\tilde{A}(\tilde{X}(tv))\,dv.$$

Observe

$$\frac{C(tv)}{C(t)} = \begin{cases} v^\gamma & p = 1 \\ e^{\frac{\gamma}{1-p}t^{1-p}\left(v^{1-p}-1\right)} & p \neq 1, p > 0. \end{cases}$$

Note $\frac{\gamma}{1-p}t^{1-p}\left(v^{1-p} - 1\right) \leq 0$ for $v \in [0,1]$, since $\frac{\gamma}{1-p}$ and $\left(v^{1-p} - 1\right)$ has opposite sign either $0 < p < 1$ or $p > 1$. Therefore $e^{\frac{\gamma}{1-p}t^{1-p}\left(v^{1-p}-1\right)} \leq 1$. Also, $\tilde{A}(\tilde{X}(tv))$ is bounded for $v \in [0,1]$ since $A(\tilde{X}(\cdot))$ is continuous by Lemma B.4.

So we can apply dominated convergence theorem and take limit $t \to 0+$. Since

$$\lim_{t \to 0+} \frac{C(tv)}{C(t)} = \begin{cases} 1 & 0 < p < 1 \\ v^\gamma & p = 1 \\ 0 & p > 1 \end{cases}$$

for $v \neq 0$, we have

$$\dot{\tilde{X}}(0) = \lim_{t \to 0+} \frac{\tilde{X}(t) - X_0}{t} = -\int_0^1 \lim_{t \to 0+}\left(\frac{C(tv)}{C(t)}\tilde{A}(\tilde{X}(tv))\right)dv = \begin{cases} -\tilde{A}(X_0) & 0 < p < 1 \\ -\frac{1}{\gamma+1}\tilde{A}(X_0) & p = 1 \\ 0 & p > 1. \end{cases} \tag{15}$$

We now check $\dot{\tilde{X}}(t)$ is continuous at $t = 0$.

(i) $0 < p \leq 1$

For $t > 0$, we know

$$\dot{\tilde{X}}(t) = -\tilde{A}(\tilde{X}(t)) - \frac{\gamma}{t^p}(\tilde{X}(t) - X_0).$$

Observe

$$\lim_{t \to 0+} \frac{\gamma}{t^p}(\tilde{X}(t) - X_0) = \lim_{t \to 0+}\left(\gamma t^{1-p}\cdot\frac{\tilde{X}(t)-X_0}{t}\right) = \begin{cases} 0 & 0 < p < 1 \\ -\frac{\gamma}{\gamma+1}\tilde{A}(X_0) & p = 1. \end{cases} \tag{16}$$

Now from ODE $\dot{\tilde{X}}(t) = -\tilde{A}(\tilde{X}(t)) - \frac{\gamma}{t^p}(\tilde{X}(t) - X_0)$, by taking limit $t \to 0+$ we have

$$\lim_{t \to 0+}\dot{\tilde{X}}(t) = -\tilde{A}(X_0) - \lim_{t \to 0+}\left(\frac{\gamma}{t^p}(\tilde{X}(t) - X_0)\right) = \begin{cases} -\tilde{A}(X_0) & 0 < p < 1 \\ -\frac{1}{\gamma+1}\tilde{A}(X_0) & p = 1. \end{cases}$$

Therefore $\dot{\tilde{X}}(t)$ is continuous at $t = 0$.

(ii) $p > 1$

For $p > 1$, we don't know the value of $\lim_{t \to 0}\frac{\gamma}{t^p}(\tilde{X}(t) - X_0)$. Thus we first find the limit. Let's go back to

$$C(t)(\tilde{X}(t) - X_0) = -\int_0^t C(s)\tilde{A}(\tilde{X}(s))\,ds.$$

Recall $\dot{C}(t) = \frac{\gamma}{t^p}C(t)$. By taking integration by parts for the right hand side, we have

$$\int_0^t C(s)\tilde{A}(\tilde{X}(s))\,ds = \frac{t^p}{\gamma}C(t)\tilde{A}(\tilde{X}(t)) - \int_0^t p\frac{s^{p-1}}{\gamma}C(s)\tilde{A}(\tilde{X}(s))\,ds - \int_0^t \frac{s^p}{\gamma}C(s)\frac{d}{ds}\tilde{A}(\tilde{X}(s))\,ds.$$

Where we know $\tilde{A}(\tilde{X}(s))$ is differentiable almost everywhere from Lemma B.4. Now, divide both sides by $\frac{t^p C(t)}{\gamma}$. Then for $s = tv$ we have

$$\frac{\gamma}{t^p}(\tilde{X}(t) - X_0) = -\tilde{A}(\tilde{X}(t)) + \int_0^t \frac{ps^{p-1}}{t^p}\frac{C(s)}{C(t)}\tilde{A}(\tilde{X}(s))\,ds + \int_0^t \frac{s^p}{t^p}\frac{C(s)}{C(t)}\left(\frac{d}{ds}\tilde{A}(\tilde{X}(s))\right)ds$$

$$= -\tilde{A}(\tilde{X}(t)) + \int_0^1 pv^{p-1}\frac{C(tv)}{C(t)}\tilde{A}(\tilde{X}(s))\,dv + \int_0^1 v^p\frac{C(tv)}{C(t)}\left(\frac{d}{ds}\tilde{A}(\tilde{X}(s))\right)t\,dv.$$

From Proposition B.7 and since $\tilde{X}$ satisfies (12) for $t > 0$, for $s \in (0, t]$ we have

$$\dot{\tilde{X}}(s) = \tilde{A}(\tilde{X}(s)) + \frac{\gamma}{t^p}\left(\tilde{X}(s) - X_0\right) = \lim_{m \to \infty}\left(\tilde{A}(\tilde{X}_{\delta_m}(s)) + \frac{\gamma}{t^p}\left(\tilde{X}_{\delta_m}(s) - X_0\right)\right) = \lim_{m \to \infty}\dot{\tilde{X}}_{\delta_m}(s).$$

From Lemma B.6 we know $\left\|\dot{\tilde{X}}_{\delta_m}(s)\right\| \leq \tilde{M}_{dot}(t)$ for $s \in [0, t]$, taking limit $m \to \infty$ we have $\left\|\dot{\tilde{X}}(s)\right\| \leq \tilde{M}_{dot}(t)$ for $s \in [0, t]$. Thus $\tilde{X}(s)$ becomes $\tilde{M}_{dot}(t)$-Lipschitz continuous in $s \in [0, t]$. And so $(\tilde{A} \circ \tilde{X})(s)$ becomes $L\tilde{M}_{dot}(t)$-Lipschitz continuous in $s \in [0, t]$, we have $\left\|\frac{d}{ds}\tilde{A}(\tilde{X}(s))\right\| \leq L\tilde{M}_{dot}(t)$ for almost all $s \in [0, t]$. Moreover $C(tv)/C(t)$ is bounded for $v \in [0, 1]$ since $C$ is a nonnegative nondecreasing function. Therefore we can again apply dominated convergence theorem. Reminding $\lim_{t \to 0+}\frac{C(tv)}{C(t)} = 0$ for $p > 1$, taking limit $t \to 0+$ we have

$$\lim_{t \to 0+}\frac{\gamma}{t^p}(\tilde{X}(t) - X_0) = -\tilde{A}(X_0) + 0 = -\tilde{A}(X_0). \tag{17}$$

Finally we have

$$\lim_{t \to 0+}\dot{\tilde{X}}(t) = -\tilde{A}(X_0) - \lim_{t \to 0+}\left(\frac{\gamma}{t^p}(\tilde{X}(t) - X_0)\right) = -\tilde{A}(X_0) - (-\tilde{A}(X_0)) = 0 = \dot{\tilde{X}}(0).$$

$\square$

Before we move on to original inclusion, we prove important corollaries that bound $\left\|\dot{\tilde{X}}(t)\right\|$ and $\left\|\tilde{A}(\tilde{X}(t))\right\|$ uniformly on $[0, T]$. Note the main difference between following corollary and Lemma B.6 is that the dependency on Lipschitz constant $L$ is dropped for the case $p > 1$, which will be crucial in the next section.

**Corollary B.8.** *Denote $\tilde{X}$ as the solution of* (12). *Then for $T > 0$, following inequality is true for $t \in [0, T]$.*

$$\left\|\dot{\tilde{X}}(t)\right\| \leq \begin{cases} \left\|\tilde{A}(X_0)\right\| & 0 < p < 1 \\ \frac{1}{\sqrt{\gamma + 1}}\left\|\tilde{A}(X_0)\right\| & p = 1 \\ \sqrt{\frac{p}{\gamma}T^{p-1}}\left\|\tilde{A}(X_0)\right\| & p > 1. \end{cases}$$

*Proof.*   (i) $0 < p \leq 1$
As $\tilde{X}$ is the solution for (12), from Lemma B.5 we know

$$\tilde{U}_1(t) = \left\|\dot{\tilde{X}}(t)\right\|^2 + \frac{\gamma p}{t^{p+1}}\left\|\tilde{X}(t) - X_0\right\|^2$$

is a nonincreasing function. From (15), (16) and continuity of $\dot{X}(t)$ at $t = 0$, we have

$$\lim_{\epsilon \to 0+}\tilde{U}_1(\epsilon) = \begin{cases} \left\|\tilde{A}(X_0)\right\|^2 & 0 < p < 1 \\ \frac{1}{\gamma + 1}\left\|\tilde{A}(X_0)\right\|^2 & p = 1. \end{cases} \tag{18}$$

Therefore $\left\|\dot{\tilde{X}}(t)\right\|^2 \leq \tilde{U}_1(t) \leq \lim_{\epsilon \to 0+}\tilde{U}_1(\epsilon)$ for $t > 0$, we get the desired result.

(ii) $p > 1$
As $\tilde{X}$ is the solution for (12), from Lemma B.5 we know

$$\tilde{U}_2(t) = \frac{1}{t^{p-1}}\left\|\dot{\tilde{X}}(t)\right\|^2 + \frac{\gamma p}{t^{2p}}\left\|\tilde{X}(t) - X_0\right\|^2$$

is a nonincreasing function. However, as we don't know the value of $\lim_{t \to 0+}\frac{1}{t^{p-1}}\left\|\dot{\tilde{X}}(t)\right\|^2$, we first calculate it. To do so, we consider

$$\tilde{U}_1(t) = \left\|\dot{\tilde{X}}(t)\right\|^2 + \frac{\gamma p}{t^{p+1}}\left\|\tilde{X}(t) - X_0\right\|^2.$$

In the proof of Lemma B.5, we have observed its derivative becomes

$$\dot{\tilde{U}}_1(t) = -2\left\langle \dot{\tilde{X}}(t), \frac{d}{dt}\tilde{A}(\tilde{X}(t))\right\rangle - \frac{2\gamma}{t^p}\left\|\dot{\tilde{X}}(t) - \frac{p}{t}(\tilde{X}(t) - X_0)\right\|^2 - \frac{\gamma p(1-p)}{t^{p+2}}\left\|\tilde{X}(t) - X_0\right\|^2$$

$$\leq \frac{\gamma p(p-1)}{t^{p+2}}\left\|\tilde{X}(t) - X_0\right\|^2.$$

For $\epsilon > 0$, integrating from $\epsilon$ to $t$ we have

$$\tilde{U}_1(t) \leq \tilde{U}_1(\epsilon) + \int_\epsilon^t \frac{\gamma p(p-1)}{s^{p+2}}\left\|\tilde{X}(s) - X_0\right\|^2 ds.$$

We consider taking limit $\epsilon \to 0+$. Observe from (17) and (15) we know $\lim_{t\to0+}\frac{\tilde{X}-X_0}{t^p} = -\frac{\tilde{A}(X_0)}{\gamma}$ and $\left\|\dot{\tilde{X}}(0)\right\| = 0$, and therefore we have

$$\lim_{\epsilon\to0+}\tilde{U}_1(\epsilon) = 0 + \lim_{\epsilon\to0+}\frac{\gamma p}{\epsilon^{2p}}\left\|\tilde{X}(\epsilon) - X_0\right\|^2 \cdot \epsilon^{p-1} = \frac{p}{\gamma}\left\|\tilde{A}(X_0)\right\|^2\lim_{\epsilon\to0+}\epsilon^{p-1} = 0.$$

Moreover as $\lim_{t\to0+}\frac{\|\tilde{X}(t)-X_0\|^2}{t^{2p}} = \frac{\|\tilde{A}(X_0)\|^2}{\gamma^2}$, there is $\delta > 0$ such that $0 < s < \delta$ implies $\frac{\|\tilde{X}(t)-X_0\|^2}{s^{2p}} \leq \frac{2\|\tilde{A}(X_0)\|^2}{\gamma^2}$. Recalling $p > 1$, for $0 < \epsilon < \delta$ we have

$$\int_0^\epsilon \frac{\gamma p(p-1)}{s^{p+2}}\left\|\tilde{X}(s) - X_0\right\|^2 ds \leq \int_0^\epsilon \frac{\gamma p(p-1)}{s^{2-p}}\frac{2\left\|\tilde{A}(X_0)\right\|^2}{\gamma^2}ds$$

$$= \frac{2p(p-1)\left\|\tilde{A}(X_0)\right\|^2}{\gamma}\left[\frac{1}{p-1}s^{p-1}\right]_0^\epsilon = \frac{2p\left\|\tilde{A}(X_0)\right\|^2}{\gamma}\epsilon^{p-1},$$

therefore

$$\int_0^t \frac{\gamma p(p-1)}{s^{p+2}}\left\|\tilde{X}(s) - X_0\right\|^2 ds \leq \frac{2p\left\|\tilde{A}(X_0)\right\|^2}{\gamma}\epsilon^{p-1} + \int_\epsilon^t \frac{\gamma p(p-1)}{s^{p+2}}\left\|\tilde{X}(s) - X_0\right\|^2 ds < \infty.$$

Thus the integral is well-defined when $\epsilon \to 0+$. Taking limit $\epsilon \to 0+$ we have

$$\tilde{U}_1(t) \leq \int_0^t \frac{\gamma p(p-1)}{s^{p+2}}\left\|\tilde{X}(s) - X_0\right\|^2 ds.$$

Moving $\frac{\gamma p}{t^{p+1}}\left\|\tilde{X}(t) - X_0\right\|^2$ to the right hand side, we get a inequality for $\left\|\dot{\tilde{X}}(t)\right\|^2$,

$$\left\|\dot{\tilde{X}}(t)\right\|^2 = \tilde{U}_1(t) - \frac{\gamma p}{t^{p+1}}\left\|\tilde{X}(t) - X_0\right\|^2 \leq \int_0^t \frac{\gamma p(p-1)}{s^{p+2}}\left\|\tilde{X}(s) - X_0\right\|^2 ds - \frac{\gamma p}{t^{p+1}}\left\|\tilde{X}(t) - X_0\right\|^2.$$

Observe, by L'Hôpital's rule we have

$$\lim_{t\to0+}\frac{\int_0^t \frac{\gamma p(p-1)}{s^{p+2}}\left\|\tilde{X}(s) - X_0\right\|^2 ds}{t^{p-1}} = \lim_{t\to0+}\frac{\frac{\gamma p(p-1)}{t^{p+2}}\left\|\tilde{X}(t) - X_0\right\|^2}{(p-1)t^{p-2}} = \lim_{t\to0+}\gamma p\frac{\left\|\tilde{X}(t) - X_0\right\|^2}{t^{2p}} = \frac{p}{\gamma}\left\|\tilde{A}(X_0)\right\|^2.$$

Now dividing $t^{p-1}$ to previous inequality and taking limit, we conclude

$$\lim_{t\to0+}\frac{1}{t^{p-1}}\left\|\dot{\tilde{X}}\right\|^2 \leq \lim_{t\to0+}\frac{1}{t^{p-1}}\int_0^t \frac{\gamma p(p-1)}{s^{p+2}}\left\|\tilde{X} - X_0\right\|^2 ds - \lim_{t\to0+}\frac{\gamma p}{t^{2p}}\left\|\tilde{X} - X_0\right\|^2$$

$$= \frac{p}{\gamma}\left\|\tilde{A}(X_0)\right\|^2 - \frac{p}{\gamma}\left\|\tilde{A}(X_0)\right\|^2$$

$$= 0.$$

Thus we have $\lim_{t \to 0+} \frac{1}{t^{p-1}} \left\| \dot{\tilde{X}} \right\|^2 = 0$. Therefore,

$$\lim_{\epsilon \to 0+} \tilde{U}_2(\epsilon) = \lim_{\epsilon \to 0+} \left( \frac{1}{\epsilon^{p-1}} \left\| \dot{\tilde{X}}(\epsilon) \right\|^2 + \frac{\gamma p}{\epsilon^{2p}} \left\| \tilde{X}(\epsilon) - X_0 \right\|^2 \right) = \frac{p}{\gamma} \left\| \tilde{A}(X_0) \right\|^2.$$

From $\frac{\left\| \dot{\tilde{X}} \right\|^2}{t^{p-1}} \leq \lim_{\epsilon \to 0+} \tilde{U}_2(\epsilon) = \frac{p}{\gamma} \left\| \tilde{A}(X_0) \right\|^2$, we conclude the desired result.

$\square$

**Corollary B.9.** *Denote $\tilde{X}$ as the solution of* (12). *Then for $T > 0$, following inequality is true for $t \in [0, T]$.*

$$\left\| \tilde{A}(\tilde{X}(t)) \right\| \leq \begin{cases} \sqrt{(\gamma + 1)\left(1 + \frac{T^{1-p}}{p}\right)} \left\| \tilde{A}(X_0) \right\| & 0 < p < 1 \\ \left\| \tilde{A}(X_0) \right\| & p = 1 \\ \sqrt{\frac{p(\gamma + 1)}{\gamma}\left(T^{p-1} + \frac{1}{p}\right)} \left\| \tilde{A}(X_0) \right\| & p > 1. \end{cases}$$

*Proof.* First observe,

$$\left\| \tilde{A}(\tilde{X}(t)) \right\|^2 = \left\| \dot{\tilde{X}}(t) + \frac{\gamma}{t^p}\left(\tilde{X}(t) - X_0\right) \right\|^2$$

$$= \left\| \dot{\tilde{X}}(t) \right\|^2 + 2\gamma \left\langle \dot{\tilde{X}}(t), \frac{\tilde{X}(t) - X_0}{t^p} \right\rangle + \frac{\gamma^2}{t^{2p}} \left\| \tilde{X}(t) - X_0 \right\|^2$$

$$\leq \left\| \dot{\tilde{X}}(t) \right\|^2 + \gamma \left( \left\| \dot{\tilde{X}}(t) \right\|^2 + \frac{\left\| \tilde{X}(t) - X_0 \right\|^2}{t^{2p}} \right) + \frac{\gamma^2}{t^{2p}} \left\| \tilde{X}(t) - X_0 \right\|^2$$

$$= (\gamma + 1)\left( \left\| \dot{\tilde{X}}(t) \right\|^2 + \frac{\gamma}{t^{2p}} \left\| \tilde{X}(t) - X_0 \right\|^2 \right)$$

The inequality comes from Young's inequality. Now we consider each case.

(i) $p = 1$

For this case, the terms on the right hand side exactly become $\tilde{U}_1(t)$. Therefore from (18)

$$\left\| \tilde{A}(\tilde{X}(t)) \right\|^2 \leq (\gamma + 1)\tilde{U}_1(t) \leq (\gamma + 1) \lim_{\epsilon \to 0+} \tilde{U}_1(\epsilon) = \left\| \tilde{A}(X_0) \right\|^2.$$

(ii) $0 < p < 1$

Recall from the proof of Corollary B.8 we know $\tilde{U}_1(t) \leq \lim_{\epsilon \to 0+} \tilde{U}_1(\epsilon) = \left\| \tilde{A}(X_0) \right\|^2$. Therefore we have $\frac{\gamma p}{t^{p+1}} \left\| \tilde{X}(t) - X_0 \right\|^2 \leq \tilde{U}_1(t) \leq \left\| \tilde{A}(X_0) \right\|^2$, applying Corollary B.8 we get

$$\left\| \tilde{A}(\tilde{X}(t)) \right\|^2 \leq (\gamma + 1)\left( \left\| \dot{\tilde{X}}(t) \right\|^2 + \frac{\gamma p}{t^{p+1}} \left\| \tilde{X}(t) - X_0 \right\|^2 \cdot \frac{t^{1-p}}{p} \right)$$

$$\leq (\gamma + 1) \left\| \tilde{A}(X_0) \right\|^2 \left(1 + \frac{T^{1-p}}{p}\right).$$

(iii) $p > 1$

Recall from the proof of Corollary B.8 we know $\tilde{U}_2(t) \leq \lim_{\epsilon \to 0+} \tilde{U}_2(\epsilon) = \frac{p}{\gamma} \left\| \tilde{A}(X_0) \right\|^2$. Therefore we have

$\frac{\gamma p}{t^{2p}} \left\| \tilde{X}(t) - X_0 \right\|^2 \leq \tilde{U}_2(t) \leq \frac{p}{\gamma} \left\| \tilde{A}(X_0) \right\|^2$, applying Corollary B.8 we get

$$\left\| \tilde{A}(\tilde{X}(t)) \right\|^2 \leq (\gamma + 1) \left( \left\| \dot{\tilde{X}}(t) \right\|^2 + \frac{\gamma p}{t^{2p}} \left\| \tilde{X}(t) - X_0 \right\|^2 \cdot \frac{1}{p} \right)$$
$$\leq \frac{p(\gamma + 1)}{\gamma} \left( T^{p-1} + \frac{1}{p} \right) \left\| \tilde{A}(X_0) \right\|^2 .$$

□

### B.2.2 Existence proof for general $\mathbb{A}$

Now we move on to the original inclusion. As noticed before, we will approximate $\mathbb{A}$ with a Liptschitz function $\mathbb{A}_\lambda$ called Yosida approximation. We define and state some facts about $\mathbb{A}_\lambda$ as a lemma, and use it without proof. For the ones who are interested in proofs, see [11, Chpater 3.1, Theorem 2].

**Lemma B.10.** *Define $\mathbb{A}_\lambda : \mathbb{R}^n \to \mathbb{R}^n$ as*

$$\mathbb{A}_\lambda = \frac{1}{\lambda} \left( \mathbb{I} - \mathbb{J}_{\lambda \mathbb{A}} \right) = \frac{1}{\lambda} \left( \mathbb{I} - (\mathbb{I} + \lambda \mathbb{A})^{-1} \right)$$

*This is so called* Yosida approximation *of $\mathbb{A}$. Followings are true.*

- *(i) $\mathbb{A}_\lambda$ is $\frac{1}{\lambda}$-Lipschitz continuous and maximal monotone.*

- *(ii) $\lim_{\lambda \to 0+} \mathbb{A}_\lambda x = m(\mathbb{A}(x))$.*
  *Here $m(\mathbb{A}(x))$ is defined as $m(\mathbb{A}(x)) = \Pi_{\mathbb{A}(x)}(0)$, the element of $\mathbb{A}(x)$ with minimal norm.*

- *(iii) $\|\mathbb{A}_\lambda(x)\| \leq \|m(\mathbb{A}(x))\|$.*

- *(iv) $\forall x \in \mathbb{R}^n$, $\mathbb{A}_\lambda(x) \in \mathbb{A}(\mathbb{J}_{\lambda \mathbb{A}} x)$.*

Now we can state the proposition that proves existence of Theorem 2.2

**Proposition B.11.** *For Yosida approximation $\mathbb{A}_\lambda$, consider the ODE*

$$\dot{X}_\lambda(t) = -\mathbb{A}_\lambda(X_\lambda(t)) - \frac{\gamma}{t^p}(X_\lambda(t) - X_0) \tag{19}$$

*with initial value condition $X_\lambda(0) = X_0 \in dom(\mathbb{A})$. The solution uniquely exists by Theorem B.2, denote the solution as $X_\lambda$. Now for $T > 0$ and a positive sequence $\{\lambda_n\}_{n \in \mathbb{N}}$ such that $\lim_{n \to \infty} \lambda_n = 0$, define a sequence of solutions as $\mathcal{F}_T = \{X_{\lambda_n} : [0, T] \to \mathbb{R}^n \mid m \in \mathbb{N}\}$. Then there is a subsequence $\{\lambda_{n_k}\}_{k \in \mathbb{N}}$ such that $X_{\lambda_{n_k}}$ converges to the solution of (6) uniformly on $[0, T]$.*

From Corollary B.8, Corollary B.9 and Lemma B.10 (iii), following lemma is immediate.

**Lemma B.12.** *Let $X_\lambda$ be the solution of (19). Then following is true for $t \in [0, T]$ for all $T > 0$.*

$$\left\| \dot{X}_\lambda(t) \right\| \leq M_{dot}(T) = \begin{cases} \|m(\mathbb{A}(X_0))\| & 0 < p < 1 \\ \frac{1}{\sqrt{\gamma + 1}} \|m(\mathbb{A}(X_0))\| & p = 1 \\ \sqrt{\frac{p}{\gamma} T^{p-1}} \|m(\mathbb{A}(X_0))\| & p > 1 \end{cases}$$

*and*

$$\|\mathbb{A}_\lambda(X_\lambda(t))\| \leq M_{\mathbb{A}}(T) = \begin{cases} \sqrt{(\gamma + 1) \left( 1 + \frac{T^{1-p}}{p} \right)} \|m(\mathbb{A}(X_0))\| & 0 < p < 1 \\ \|m(\mathbb{A}(X_0))\| & p = 1 \\ \sqrt{\frac{p(\gamma + 1)}{\gamma} \left( T^{p-1} + \frac{1}{p} \right)} \|m(\mathbb{A}(X_0))\| & p > 1. \end{cases}$$

*Proof.* Replace $\tilde{A}$ with $\mathbb{A}_\lambda$ and $\tilde{X}$ with $X_\lambda$ in Corollary B.8 and Corollary B.9. Applying $\|\mathbb{A}_\lambda(X_0)\| \leq \|m(\mathbb{A}(X_0))\|$ from Lemma B.10 (iii), we're done. □

While we're concluding the existence of converging sequence, we will exploit following lemma from [11, Chapter 0.3, Theorem 4]. For convenience, we restate the lemma here.

**Lemma B.13.** *Let us consider a sequence of absolutely continuous functions $x_k(\cdot)$ from an interval $I$ of $\mathbb{R}$ to a Banach space $X$ satisfying*

   *(i) $\forall t \in I$, $\{x_k(t)\}_{k \in \mathbb{N}}$ is a relatively compact subset of $X$*

   *(ii) there exists a positive function $c(\cdot) \in L^1(I, [0, \infty))$ such that, for almost all $t \in I$, $\|\dot{x}_k(t)\| \le c(t)$*

*Then there exists a subsequence (again denoted by) $x_k(\cdot)$ converging to an absolutely continuous function $x(\cdot)$ from $I$ to $X$ in the sense that*

   *(i) $x_k(\cdot)$ converges to $x(\cdot)$ over compact subsets of $I$*

   *(ii) $\dot{x}_k(\cdot)$ converges weakly to $\dot{x}(\cdot)$ in $L^1(I, X)$*

*Proof.* See [11, Chapter 0.3, Theorem 4]. $\qquad\square$

From Lemma B.12 we can immediately check norm of all derivatives $\dot{X}_{\lambda_m}$ are bounded by $M_{dot}(T)$. So condition (ii) holds with $M_{dot}(T)$. For condition (i), we prove $\mathcal{F}_T$ is convergent in $C([0,T], \mathbb{R}^n)$.

**Lemma B.14.** $\mathcal{F}_T$ *is convergent sequence in $C([0,T], \mathbb{R}^n)$. In other words, $\forall \epsilon > 0$, there is $N > 0$ such that $n, m > N$ implies $\sup_{t \in [0,T]} \|X_{\lambda_n}(t) - X_{\lambda_m}(t)\| < \epsilon$*

*Proof.* We will show $X_{\lambda_n}$ is Cauchy sequence in $C([0,T], \mathbb{R}^n)$. Let $\nu, \lambda > 0$. From (19), we see for $t \in (0, T]$

$$\frac{d}{dt}\frac{1}{2}\|X_\nu(t) - X_\lambda(t)\|^2 = \left\langle \dot{X}_\nu(t) - \dot{X}_\lambda(t), X_\nu(t) - X_\lambda(t) \right\rangle$$
$$= -\langle \mathbb{A}_\nu(X_\nu(t)) - \mathbb{A}_\lambda(X_\lambda(t)), X_\nu(t) - X_\lambda(t) \rangle - \frac{\gamma}{t^p}\|X_\nu(t) - X_\lambda(t)\|^2$$
$$\le -\langle \mathbb{A}_\nu(X_\nu(t)) - \mathbb{A}_\lambda(X_\lambda(t)), X_\nu(t) - X_\lambda(t) \rangle.$$

From definition of resolvent we know $\mathbb{I} - \mathbb{J}_{\lambda\mathbb{A}} = \lambda\mathbb{A}_\lambda$. And from Lemma B.10 (iv) we know $\mathbb{A}_\lambda(X_\lambda(t)) \in \mathbb{A}(\mathbb{J}_{\lambda\mathbb{A}}(X_\lambda(t)))$. Thus from monotone inequality we see

$$-\langle \mathbb{A}_\nu(X_\nu(t)) - \mathbb{A}_\lambda(X_\lambda(t)), X_\nu(t) - X_\lambda(t) \rangle$$
$$= -\langle \mathbb{A}_\nu(X_\nu(t)) - \mathbb{A}_\lambda(X_\lambda(t)), \nu\mathbb{A}_\nu(X_\nu(t)) - \lambda\mathbb{A}_\lambda(X_\lambda(t)) \rangle$$
$$\quad - \langle \mathbb{A}_\nu(X_\nu(t)) - \mathbb{A}_\lambda(X_\lambda(t)), \mathbb{J}_{\nu\mathbb{A}}(X_\nu(t)) - \mathbb{J}_{\lambda\mathbb{A}}(X_\lambda(t)) \rangle$$
$$\le -\langle \mathbb{A}_\nu(X_\nu(t)) - \mathbb{A}_\lambda(X_\lambda(t)), \nu\mathbb{A}_\nu X_\nu(t) - \lambda\mathbb{A}_\lambda(X_\lambda(t)) \rangle$$
$$= (\nu + \lambda)\langle \mathbb{A}_\nu(X_\nu(t)), \mathbb{A}_\lambda(X_\lambda(t)) \rangle - \left( \nu\|\mathbb{A}_\nu X_\nu(t)\|^2 + \lambda\|\mathbb{A}_\lambda X_\lambda(t)\|^2 \right).$$

By Young's inequality

$$(\nu + \lambda)\langle \mathbb{A}_\nu(X_\nu(t)), \mathbb{A}_\lambda(X_\lambda(t)) \rangle - \left( \nu\|\mathbb{A}_\nu X_\nu(t)\|^2 + \lambda\|\mathbb{A}_\lambda X_\lambda(t)\|^2 \right)$$
$$\le \nu\left( \|\mathbb{A}_\nu(X_\nu(t))\|^2 + \frac{1}{4}\|\mathbb{A}_\lambda(X_\lambda(t))\|^2 \right) + \lambda\left( \|\mathbb{A}_\lambda(X_\lambda(t))\|^2 + \frac{1}{4}\|\mathbb{A}_\nu(X_\nu(t))\|^2 \right)$$
$$\quad - \left( \nu\|\mathbb{A}_\nu X_\nu(t)\|^2 + \lambda\|\mathbb{A}_\lambda X_\lambda(t)\|^2 \right)$$
$$= \frac{1}{4}\left( \nu\|\mathbb{A}_\lambda(X_\lambda(t))\|^2 + \lambda\|\mathbb{A}_\nu(X_\nu(t))\|^2 \right).$$

Now applying Lemma B.12 we have

$$\frac{d}{dt}\frac{1}{2}\|X_\nu(t) - X_\lambda(t)\|^2 \le \frac{1}{4}(\nu + \lambda)M_{\mathbb{A}}(T)^2.$$

Then integrating both sides from $0$ to $t$ we have

$$\|X_\nu(t) - X_\lambda(t)\|^2 \le \frac{t}{2}(\nu + \lambda)M_{\mathbb{A}}(T)^2. \tag{20}$$

Now take $\epsilon > 0$. Then there is $N > 0$ such that for $n > N$, $\lambda_n < \frac{\epsilon}{TM_{\mathbb{A}}(T)^2}$ holds. Then for $t \in [0, T]$, $n, m > N$, we have

$$\|X_{\lambda_n}(t) - X_{\lambda_m}(t)\|^2 \leq \frac{t}{2}(\lambda_n + \lambda_m)M_{\mathbb{A}}(T)^2 < \epsilon.$$

Therefore we get the desired result. $\qquad\square$

Finally, we are ready to prove Proposition B.11, which implies the main theorem.

*Proof of Proposition B.11.* Take $T > 0$. We know $X_{\lambda_n}$ uniformly converges on $[0, T]$ by Lemma B.14. Name the limit as $X$, i.e. define $X \colon [0, T] \to \mathbb{R}^n$ as $X(t) = \lim_{n \to \infty} X_{\lambda_n}(t)$. Then as $X_{\lambda_n}(0) = X_0$ for all $n \in \mathbb{N}$, we see $X$ satisfies the initial condition. It remains to show $X$ satisfies (6) almost everywhere.

Recall $\{\dot{X}_{\lambda_n}\}$ is bounded in $L^\infty([0, T], \mathbb{R}^n)$ by Lemma B.12. Thus we can apply Lemma B.13, there is a subsequence $\{\dot{X}_{\lambda_{n_k}}\}$ converges weakly to $\dot{X}$ in $L^1([0, T], \mathbb{R}^n)$. Furtheremore we have $\dot{X} \in L^\infty([0, T], \mathbb{R}^n)$ and so $\{\dot{X}_{\lambda_{n_k}}\}$ also converges weakly to $\dot{X}$ in $L^2([0, T], \mathbb{R}^n)$ as well.

For $\lambda > 0$, define $f_\lambda \colon [0, T] \to \mathbb{R}^n$ as

$$f_\lambda(t) = \begin{cases} \frac{\gamma}{t^p}(X_\lambda(t) - X_0) & \text{if } t > 0 \\ 0 & \text{if } t = 0. \end{cases}$$

Then for $f \colon [0, T] \to \mathbb{R}^n$ defined as $f(t) = \frac{\gamma}{t^p}(X(t) - X_0)$ for $t > 0$ and $f(0) = 0$, we have $\lim_{k \to \infty} f_{\lambda_{n_k}}(t) = f(t)$. As $\|f_\lambda(t)\| = \left\|\dot{X}_\lambda(t) + \mathbb{A}_{\lambda_n}(X_\lambda)(t)\right\| \leq M_{dot}(T) + M_{\mathbb{A}}(T)$ for $t \in (0, T]$ by Lemma B.12, we have

$$\|f_\lambda(t) - f(t)\|^2 \leq (\|f_\lambda(t)\| + \|f(t)\|)^2 \leq 4(M_{dot}(T) + M_{\mathbb{A}}(T))^2.$$

Therefore by dominated convergence theorem we have

$$\lim_{k \to \infty} \int_0^T \left\|f_{\lambda_{n_k}}(t) - f(t)\right\|^2 dt = \int_0^T \lim_{k \to \infty} \left\|f_{\lambda_{n_k}}(t) - f(t)\right\|^2 dt = 0,$$

we conclude $f_{\lambda_{n_k}}$ strongly converges to $f$ in $L^2([0, T], \mathbb{R}^n)$.

Now consider $F_\lambda \colon [0, T] \to \mathbb{R}^n$ defined as

$$F_\lambda(t) = \begin{cases} \dot{X}_\lambda(t) + \frac{\gamma}{t^p}(X_\lambda(t) - X_0) & \text{if } t > 0 \\ -\mathbb{A}_\lambda(X_0) & \text{if } t = 0 \end{cases}$$

Note since $X_\lambda$ are solution to ODE (19), we have

$$F_\lambda(t) = -\mathbb{A}_\lambda(X_\lambda(t)).$$

Then for $F \colon [0, T] \to \mathbb{R}^n$ defined as

$$F(t) = \begin{cases} \dot{X}(t) + \frac{\gamma}{t^p}(X(t) - X_0) & \text{if } t > 0 \\ -m(\mathbb{A}(X_0)) & \text{if } t = 0, \end{cases}$$

we see $\{F_{\lambda_{n_k}}\}_{k \in \mathbb{N}}$ converges weakly to $F$ in $L^2([0, T], \mathbb{R}^n)$.

On the other hand, by Lemma B.10 (iv) and the fact $-F_{\lambda_{n_k}}(t) = \mathbb{A}_{\lambda_{n_k}}(X_{\lambda_{n_k}}(t))$, we see

$$-F_{\lambda_{n_k}}(t) \in \mathbb{A}(\mathbb{J}_{\lambda_{n_k}\mathbb{A}}(X_{\lambda_{n_k}}(t))).$$

Observe, from the definition of $\mathbb{A}_{\lambda_n}$ and Lemma B.12, we see

$$\|X_{\lambda_n}(t) - \mathbb{J}_{\lambda_n\mathbb{A}}(X_{\lambda_n}(t))\| = \lambda_n \|\mathbb{A}_{\lambda_n}(X_{\lambda_n}(t))\| \leq \lambda_n M_{\mathbb{A}}(T).$$

Since $X_{\lambda_n}$ converges to $X$ in $\mathcal{C}([0, T], \mathbb{R}^n)$, by taking $n \to \infty$ above inequality we see $\mathbb{J}_{\lambda_n\mathbb{A}}(X_{\lambda_n})$ also converges to $X$ in $\mathcal{C}([0, T], \mathbb{R}^n)$.

Now for $\mathcal{A} \colon L^2([0, T], \mathbb{R}^n) \to L^2([0, T], \mathbb{R}^n)$ defined as $(\mathcal{A}(x))(t) = \mathbb{A}(x(t))$ almost everywhere, by [11, Chapter 3.1, Proposition 4], $\mathcal{A}$ is maximal monotone since $\mathbb{A}$ is maximal monotone. Since $-F_{\lambda_k}$ weakly converges to $-F$ in $L^2([0, T], \mathbb{R}^n)$

and $\mathbb{J}_{\lambda_k \mathbb{A}}(X_{\lambda_k})$ strongly converges to $X$ in $L^2([0,T], \mathbb{R}^n)$, by [11, Chapter 3.1, Proposition 2] we have $-F \in \mathcal{A}(X)$ in $L^2([0,T], \mathbb{R}^n)$. Therefore for almost all $t \in (0,T]$ we have

$$-\left(\dot{X}(t) + \frac{\gamma}{t^p}(X(t) - X_0)\right) \in \mathbb{A}(X(t)).$$

Reorganizing the result with respect to $\dot{X}$, we have following is true for almost all $t \in (0,T]$

$$\dot{X}(t) \in -\mathbb{A}(X(t)) - \frac{\gamma}{t^p}(X(t) - X_0).$$

Since $T > 0$ was arbitrary, we conclude $X$ satisfies above inclusion for almost all $t \in (0, \infty)$. $\qquad \square$

## C   Proof of existence and uniquecess of the solution of (10)

As uniqueness comes from Theorem B.1, we only need to show the existence. What we need for the existence proof are

(i) Nonincreasing function $\tilde{U}(t)$ which contains $\left\|\dot{\tilde{X}}\right\|^2$ as in Lemma B.5.

(ii) Uniform boundedness of $\dot{\tilde{X}}_\delta(t)$ for $t \in [0,T]$ as shown in Lemma B.6.

(iii) $\dot{\tilde{X}}(0) = \lim_{t \to 0+} \frac{\tilde{X}(t) - X_0}{t} = \lim_{t \to 0+} \dot{\tilde{X}}(t)$ as shown in the existence proof of Lipschitz case.

(iv) Uniform boundedness of $\left\|\dot{X}_\lambda(t)\right\|$ and $\|\mathbb{A}_\lambda(X_\lambda(t))\|$ for $t \in [0,T]$ as shown in Lemma B.12

We now show these steps can be also done to the $\beta(t) = \frac{2\mu}{e^{2\mu t} - 1}$.

### (i) Nonincreasing function $\tilde{U}(t)$ which contains $\left\|\dot{\tilde{X}}\right\|^2$

From

$$\dot{\tilde{X}}(t) = -\tilde{A}(\tilde{X}(t)) - \frac{2\mu}{e^{2\mu t} - 1}(\tilde{X}(t) - X_0)$$

for almost all $t$ we have

$$\ddot{\tilde{X}}(t) = -\frac{d}{dt}\tilde{A}(\tilde{X}(t)) + \left(\frac{2\mu}{e^{2\mu t} - 1}\right)^2 e^{2\mu t}(\tilde{X}(t) - X_0) - \frac{2\mu}{e^{2\mu t} - 1}\dot{\tilde{X}}(t)$$

Define

$$\tilde{U}(t) = e^{-2\mu t}\left\|\dot{\tilde{X}}(t)\right\|^2 + \left(\frac{2\mu}{e^{2\mu t} - 1}\right)^2 \left\|\tilde{X}(t) - X_0\right\|^2$$

Therefore for almost all $t > 0$,

$$
\begin{aligned}
\dot{\tilde{U}}(t) &= 2e^{-2\mu t}\left\langle \dot{\tilde{X}}(t), \ddot{\tilde{X}}(t)\right\rangle - 2\mu e^{-2\mu t}\left\|\dot{\tilde{X}}(t)\right\|^2 \\
&\quad + 2\left(\frac{2\mu}{e^{2\mu t} - 1}\right)^2\left\langle \dot{\tilde{X}}(t), \tilde{X}(t) - X_0\right\rangle - \left(\frac{2\mu}{e^{2\mu t} - 1}\right)^3 2e^{2\mu t}\left\|\tilde{X}(t) - X_0\right\|^2 \\
&= -2e^{-2\mu t}\left\langle \dot{\tilde{X}}(t), \frac{d}{dt}\tilde{A}(\tilde{X}(t))\right\rangle + 2\left(\frac{2\mu}{e^{2\mu t} - 1}\right)^2\left\langle \dot{\tilde{X}}(t), \tilde{X}(t) - X_0\right\rangle - 2e^{-2\mu t}\frac{2\mu}{e^{2\mu t} - 1}\left\|\dot{\tilde{X}}(t)\right\|^2 \\
&\quad - 2\mu e^{-2\mu t}\left\|\dot{\tilde{X}}(t)\right\|^2 + 2\left(\frac{2\mu}{e^{2\mu t} - 1}\right)^2\left\langle \dot{\tilde{X}}(t), \tilde{X}(t) - X_0\right\rangle - \left(\frac{2\mu}{e^{2\mu t} - 1}\right)^3 2e^{2\mu t}\left\|\tilde{X}(t) - X_0\right\|^2 \\
&= -2\mu e^{-2\mu t}\left\|\dot{\tilde{X}}(t)\right\|^2 - 2e^{-2\mu t}\left\langle \dot{\tilde{X}}(t), \frac{d}{dt}\tilde{A}(\tilde{X}(t))\right\rangle - \frac{2\mu e^{-2\mu t}}{e^{2\mu t} - 1}\left\|\dot{\tilde{X}}(t) - \frac{2\mu e^{2\mu t}}{e^{2\mu t} - 1}\left(\tilde{X}(t) - X_0\right)\right\|^2 \\
&\leq 0.
\end{aligned}
$$

**(ii) Uniform boundedness of $\dot{\tilde{X}}_\delta(t)$**

As (13), we define $\dot{\tilde{X}}_\delta(t)$ as the solution of

$$\dot{\tilde{X}}_\delta(t) = \begin{cases} -\tilde{A}(\tilde{X}_\delta(t)) - \frac{2\mu}{e^{2\mu\delta}-1}(\tilde{X}_\delta(t) - X_0) & 0 \le t \le \delta \\ -\tilde{A}(\tilde{X}_\delta(t)) - \frac{2\mu}{e^{2\mu t}-1}(\tilde{X}_\delta(t) - X_0) & t \ge \delta \end{cases}$$

Again with same arguments of Lemma B.6 we have $\left\| \tilde{X}_\delta(\delta) - X_0 \right\| \le \delta \left\| \tilde{A}(X_0) \right\|$ and $\left\| \dot{\tilde{X}}_\delta(\delta) \right\| \le \left\| \tilde{A}(X_0) \right\|$. Now for $t \in [0, T]$

$$\left\| \dot{\tilde{X}}_\delta(t) \right\| \le \sqrt{e^{2\mu t}\tilde{U}(t)} \le \sqrt{e^{2\mu t}\tilde{U}(\delta)} = e^{\mu t}\sqrt{e^{-2\mu\delta}\left\| \dot{\tilde{X}}_\delta(\delta) \right\|^2 + \left( \frac{2\mu}{e^{2\mu\delta}-1} \right)^2 \left\| \tilde{X}_\delta(\delta) - X_0 \right\|^2}$$

$$\le e^{\mu t}\sqrt{e^{-2\mu\delta}\left\| \tilde{A}(X_0) \right\|^2 + \left( \frac{2\mu\delta}{e^{2\mu\delta}-1} \right)^2 \left\| \tilde{A}(X_0) \right\|^2}$$

$$\le e^{\mu T}\sqrt{2}\left\| \tilde{A}(X_0) \right\|.$$

**(iii) $\dot{\tilde{X}}(0) = \lim_{t\to 0+} \frac{\tilde{X}(t)-X_0}{t} = \lim_{t\to 0+} \dot{\tilde{X}}(t)$**

Define $C(t) := 1 - e^{-2\mu t}$. Then $\dot{C}(t) = C(t)\beta(t)$ for $\beta(t) = \frac{2\mu}{e^{2\mu t}-1}$. And since

$$\lim_{t\to 0+} \frac{C(tv)}{C(t)} = \lim_{t\to 0+} \frac{1 - e^{-2\mu tv}}{1 - e^{-2\mu t}} = v,$$

with same argument done to arrive (15), we have

$$\dot{\tilde{X}}(0) = \lim_{t\to 0+} \frac{\tilde{X}(t)-X_0}{t} = -\int_0^1 \lim_{t\to 0+} \left( \frac{C(tv)}{C(t)}\tilde{A}(\tilde{X}(tv)) \right) dv = -\frac{1}{2}\tilde{A}(X_0)$$

Now from ODE $\dot{\tilde{X}}(t) = -\tilde{A}(\tilde{X}(t)) - \frac{2\mu}{e^{2\mu t}-1}(\tilde{X}(t) - X_0)$, by taking limit both sides by $t \to 0+$ we have

$$\lim_{t\to 0+} \dot{\tilde{X}}(t) = -\tilde{A}(\tilde{X}(0)) - \lim_{t\to 0+} \frac{2\mu t}{e^{2\mu t}-1}\frac{\tilde{X}(t)-X_0}{t} = -\frac{1}{2}\tilde{A}(X_0)$$

Therefore, $\lim_{t\to 0+} \frac{\tilde{X}(t)-X_0}{t} = \lim_{t\to 0+} \dot{\tilde{X}}(t)$.

**(iv) Uniform boundedness of $\left\| \dot{X}_\lambda(t) \right\|$ and $\|\mathbb{A}_\lambda(t)\|$ for $t \in [0, T]$**

Recall $U(t) = e^{-2\mu t}\left\| \dot{X}_\lambda(t) \right\|^2 + \left( \frac{2\mu}{e^{2\mu t}-1} \right)^2 \|X_\lambda(t) - X_0\|^2$ is nonincreasing. So from (iii), we have

$$e^{-2\mu t}\left\| \dot{X}_\lambda(t) \right\|^2 + \left( \frac{2\mu}{e^{2\mu t}-1} \right)^2 \|X_\lambda(t) - X_0\|^2 \le \lim_{t\to 0+} \left( e^{-2\mu t}\left\| \dot{X}_\lambda(t) \right\|^2 + \left( \frac{2\mu}{e^{2\mu t}-1} \right)^2 \|X_\lambda(t) - X_0\|^2 \right)$$

$$= \frac{1}{4}\|\mathbb{A}_\lambda(X_0)\|^2 + \frac{1}{4}\|\mathbb{A}_\lambda(X_0)\|^2 = \frac{1}{2}\|\mathbb{A}_\lambda(X_0)\|^2$$

Therefore we have from Lemma B.10 (iii)

$$e^{-2\mu t}\left\| \dot{X}_\lambda(t) \right\|^2 \le \frac{1}{2}\|\mathbb{A}_\lambda(X_0)\|^2 \le \frac{1}{2}\|m(\mathbb{A}(X_0))\|^2 \implies \left\| \dot{X}_\lambda(t) \right\| \le \frac{e^{\mu T}}{\sqrt{2}}\|m(\mathbb{A}(X_0))\|, \tag{21}$$

and by Young's inequality

$$\|\mathbb{A}_\lambda(X_\lambda(t))\|^2 \leq \left\| \dot{X}_\lambda(t) + \frac{2\mu}{e^{2\mu t} - 1}(X_\lambda(t) - X_0) \right\|^2$$

$$= 2\left( \left\| \dot{X}_\lambda(t) \right\|^2 + \left( \frac{2\mu}{e^{2\mu t} - 1} \right)^2 \|X_\lambda(t) - X_0\|^2 \right)$$

$$\leq 2\left( \frac{e^{2\mu T}}{2} \|\mathbb{A}_\lambda(X_0)\|^2 + \frac{1}{2} \|\mathbb{A}_\lambda(X_0)\|^2 \right) = (e^{2\mu T} + 1)\|\mathbb{A}_\lambda(X_0)\|^2 \leq (e^{2\mu T} + 1)\|m(\mathbb{A}(X_0))\|^2 .$$

Therefore

$$\|\mathbb{A}_\lambda(X_\lambda(t))\| \leq \sqrt{e^{2\mu T} + 1}\,\|m(\mathbb{A}(X_0))\| . \tag{22}$$

# D  Omitted proofs for derivation of anchor ODE (4)

## D.1  Preparation for the proof of Theorem 2.1

We first provide the boundedness of trajectories as a lemma. As mentioned in the discussion after Lemma 2.3, boundedness of trajectories is an immediate corollary of Lemma 2.3. However, to address cases that are slightly more generalized, we present a proof using an argument similar to the one used in the proof of Lemma 2.3. Note the proof argument of following lemma is valid for the solution of differential equation (11) with satisfying the assumptions in Theorem 7.1 as well.

**Lemma D.1.** *(Boundedness of solutions) Suppose $X(\cdot)$ is the solution of the differential inclusion (3). Then for all $X_\star \in \mathrm{Zer}\mathbb{A}$, $t \in [0, \infty)$, following holds.*

$$\|X(t) - X_\star\| \leq \|X_0 - X_\star\| .$$

*And so, $\|X(t) - X_0\| \leq 2\|X_0 - X_\star\|$.*

*Proof.* It is trivial for $t = 0$, so we may assume $t > 0$.
Take $X_\star \in \mathrm{Zer}\mathbb{A}$. By monotonicity of $\mathbb{A}$ and Young's inequality, we get the following inequality.

$$\frac{d}{dt}\|X(t) - X_\star\|^2 = 2\left\langle \dot{X}(t), X(t) - X_\star \right\rangle = -2\left\langle \tilde{\mathbb{A}}(X(t)) + \beta(t)(X(t) - X_0), X(t) - X_\star \right\rangle$$

$$= -2\left\langle \tilde{\mathbb{A}}(X(t)) + \beta(t)(X(t) - X_\star) - \beta(t)(X_0 - X_\star), X(t) - X_\star \right\rangle$$

$$\leq -2\beta(t)\|X(t) - X_\star\|^2 + 2\beta(t)\langle X_0 - X_\star, X(t) - X_\star \rangle$$

$$\leq -2\beta(t)\|X(t) - X_\star\|^2 + \beta(t)\left( \|X_0 - X_\star\|^2 + \|X(t) - X_\star\|^2 \right)$$

$$= -\beta(t)\|X(t) - X_\star\|^2 + \beta(t)\|X_0 - X_\star\|^2 .$$

Now again define $C(t) = e^{\int_v^t \beta(s)ds}$ for some $v > 0$, then we see $\dot{C}(t) = C(t)\beta(t)$. Moving $-\beta(t)\|X(t) - X_\star\|^2$ to the left hand side and multiplying both sides by $C(t)$, we have

$$\frac{d}{dt}\left( C(t)\|X(t) - X_\star\|^2 \right) = C(t)\frac{d}{dt}\|X(t) - X_\star\|^2 + C(t)\beta(t)\|X(t) - X_\star\|^2$$

$$\leq C(t)\beta(t)\|X_0 - X_\star\|^2 = \frac{d}{dt}C(t)\|X_0 - X_\star\|^2 .$$

Integrating both sides from $\epsilon > 0$ to $t$ we have

$$C(t)\|X(t) - X_\star\|^2 - C(\epsilon)\|X(\epsilon) - X_\star\|^2 \leq C(t)\|X_0 - X_\star\|^2 - C(\epsilon)\|X_0 - X_\star\|^2 .$$

As $\beta > 0$, we have $C$ is nonnegative and nondecreasing, $\lim_{\epsilon \to 0+} C(\epsilon)$ exists. Taking limit $\epsilon \to 0+$ both sides we have and dividing both sides by $C(t)$ we conclude

$$\|X(t) - X_\star\|^2 \leq \|X_0 - X_\star\|^2 .$$

The latter statement holds directly from triangular inequality,

$$\|X(t) - X_0\| \leq \|X(t) - X_\star\| + \|X_\star - X_0\| \leq 2\|X_0 - X_\star\| .$$

$\square$

Following lemma shows APPM is an instance of Halpern method. It is immediate from induction, but we state it as a lemma due as its importance.

**Lemma D.2.** *Consider a method defined as*

$$x^{k+1} = \mathbb{J}_\mathbb{A} y^k$$
$$y^{k+1} = (1 - \beta_k)\left(2x^{k+1} - y^k\right) + \beta_k x^0,$$

*for $k = 0, 1, \ldots$, with initial condition $y^0 = x^0$. Then for reflected resolvent $\mathbb{R}_\mathbb{A}$ defined as $\mathbb{R}_\mathbb{A} = 2\mathbb{J}_\mathbb{A} - \mathbb{I}$, above method is equivalent to*

$$\tilde{y}^{k+1} = \beta_k \tilde{y}^0 + (1 - \beta_k)\mathbb{T}\tilde{y}^k$$

*when $\mathbb{T} = \mathbb{R}_\mathbb{A}$, $\tilde{y}^0 = y^0$. Here equivalence means $\tilde{y}^k = y^k$ holds for $k = 0, 1, \ldots$.*

*Proof.* Proof by induction. As $\tilde{y}^0 = y^0$ by assumption, the statement is true for $k = 0$. Now suppose $y^k = \tilde{y}^k$ holds for $k \in \mathbb{N}$, then

$$\tilde{y}^{k+1} = (1 - \beta_k)\mathbb{R}_\mathbb{A}\tilde{y}^k + \beta_0 \tilde{y}^0 = (1 - \beta_k)\left(2\mathbb{J}_\mathbb{A} - \mathbb{I}\right)\tilde{y}^k + \beta_0 \tilde{y}^0$$
$$= (1 - \beta_k)\left(2\mathbb{J}_\mathbb{A} - \mathbb{I}\right)y^k + \beta_0 y^0 = (1 - \beta_k)\left(2x^{k+1} - y^k\right) + \beta_0 y^0 = y^{k+1}.$$

Thus $\tilde{y}^{k+1} = y^{k+1}$, the statement is true for $k + 1$. By induction, we get the desired result. $\qquad\square$

### D.2 Proof of Theorem 2.1

Let $\mathbb{A} \colon \mathbb{R}^n \to \mathbb{R}^n$ be a maximal monotone operator, and $h, \lambda, \delta > 0$. Again, denote $\mathbb{A}_\lambda = \frac{1}{\lambda}(\mathbb{I} - \mathbb{A}_{\lambda\mathbb{A}}) = \frac{1}{\lambda}(\mathbb{I} - (\mathbb{I} + \lambda\mathbb{A})^{-1})$. Since various kind of terms appear in the proof, we first organize the terms and notations.

- $X$ : Solution of differential inclusion,

$$\dot{X} \in -\mathbb{A}(X) - \frac{1}{t}(X - X_0).$$

- $X_\lambda$ : Solution of differential equation,

$$\dot{X}_\lambda = -\mathbb{A}_\lambda(X_\lambda) - \frac{1}{t}(X_\lambda - X_0).$$

- $X_{\lambda,\delta}$ : Solution of approximated differential equation,

$$\dot{X}_{\lambda,\delta}(t) = \begin{cases} -\mathbb{A}_\lambda(X_{\lambda,\delta})(t) - \frac{1}{\delta}(X_{\lambda,\delta}(t) - X_0) & 0 \le t < \delta \\ -\mathbb{A}_\lambda(X_{\lambda,\delta})(t) - \frac{1}{t}(X_{\lambda,\delta}(t) - X_0) & t \ge \delta. \end{cases} \tag{23}$$

- $X_{\lambda,\delta}^k$ : Sequence obtained by taking Euler discretization of ODE (23),

$$X_{\lambda,\delta}^{k+1} = \begin{cases} X_\delta^k - \left(2h\mathbb{A}_\lambda(X_{\lambda,\delta}^k) + \frac{2h}{\delta}(X_{\lambda,\delta}^k - X_0)\right) & 0 \le k < \frac{\delta}{2h} \\ X_\delta^k - \left(2h\mathbb{A}_\lambda(X_{\lambda,\delta}^k) + \frac{1}{k}(X_{\lambda,\delta}^k - X_0)\right) & k \ge \frac{\delta}{2h}. \end{cases}$$

- $x_{h,\lambda}^k$ : Sequence obtained from APPM with operator $h\mathbb{A}_\lambda$, i.e.

$$x_{h,\lambda}^k = \mathbb{J}_{h\mathbb{A}_\lambda} y_{h,\lambda}^{k-1}$$
$$y_{h,\lambda}^k = \frac{k}{k+1}(2x_{h,\lambda}^k - y_{h,\lambda}^{k-1}) + \frac{1}{k+1}X_0.$$

- $x_h^k$ : Sequence obtained from APPM with operator $h\mathbb{A}$, i.e.

$$x_h^k = \mathbb{J}_{h\mathbb{A}} y_h^{k-1}$$
$$y_h^k = \frac{k}{k+1}(2x_h^k - y_h^{k-1}) + \frac{1}{k+1}X_0.$$

We want to show for fixed $T > 0$,

$$\lim_{h \to 0+} \sup_{0 \le k < \frac{T}{2h}} \left\| x_h^k - X(2hk) \right\| = 0.$$

Equivalently we may show for fixed $T > 0$, for every $\{h_n\}_{n \in \mathbb{N}}$ such that $h_n > 0$ and converges to 0,

$$\lim_{n \to \infty} \sup_{0 \le k < \frac{T}{2h_n}} \left\| x_{h_n}^k - X(2h_n k) \right\| = 0.$$

We will show this by considering inequality

$$\left\| x_h^k - X(2hk) \right\| \le \| X(2hk) - X_\lambda(2hk) \| + \| X_\lambda(2hk) - X_{\lambda,\delta}(2hk) \| \tag{24}$$
$$+ \left\| X_{\lambda,\delta}(2hk) - X_{\lambda,\delta}^k \right\| + \left\| X_{\lambda,\delta}^k - x_\lambda^k \right\| + \left\| x_{h,\lambda}^k - x_h^k \right\| =: S\left( h, \lambda, \delta, k \right).$$

Our goal is to show, for every $\{h_n\}_{n \in \mathbb{N}}$ such that $h_n > 0$ and converges to 0, there is a sequence $\{(\delta_n, \lambda_n)\}_{n \in \mathbb{N}}$ such that $\lim_{n \to \infty} \sup_{0 \le k < \frac{T}{2h_n}} S\left( h_n, \lambda_n, \delta_n, k \right) = 0$, and thus

$$\lim_{n \to \infty} \sup_{0 \le k < \frac{T}{2h_n}} \left\| x_{h_n}^k - X(2h_n k) \right\| \le \lim_{n \to \infty} \sup_{0 \le k < \frac{T}{2h_n}} S\left( h_n, \lambda_n, \delta_n, k \right) = 0.$$

To clarify our goal, we need to find proper $\{(\delta_n, \lambda_n)\}_{n \in \mathbb{N}}$ in terms of $\{h_n\}_{n \in \mathbb{N}}$. As $\{h_n\}_{n \in \mathbb{N}}$ is determined, $\left\| x_{h_n}^k - X(2h_n k) \right\|$ is fixed and doesn't change by the choice of $\{(\delta_n, \lambda_n)\}_{n \in \mathbb{N}}$. But if we find $\{(\delta_n, \lambda_n)\}_{n \in \mathbb{N}}$ that makes $S\left( h_n, \lambda_n, \delta_n, k \right)$ small, since (24) holds for any choice of $\delta, \lambda$, right choice of $\{(\delta_n, \lambda_n)\}_{n \in \mathbb{N}}$ can gaurantee $\left\| x_{h_n}^k - X(2h_n k) \right\|$ is small. Thus to find such $\{(\delta_n, \lambda_n)\}_{n \in \mathbb{N}}$, we will observe each terms in $S$ to find the required conditions.

**Lemma D.3.** *For $h, \lambda, \delta > 0$ following is true.*

*(i)* $\| X(2hk) - X_\lambda(2hk) \| = \mathcal{O}\left( \sqrt{\lambda} \right)$

*(ii)* $\| X_\lambda(2hk) - X_{\lambda,\delta}(2hk) \| = \mathcal{O}\left( \delta \right)$

*(iii)* $\left\| x_{h,\lambda}^k - x_h^k \right\| = O(\lambda).$

*For $L_{\lambda,\delta} = \max \left\{ \frac{1}{\lambda}, \frac{\sqrt{2}}{\lambda} \| m(\mathbb{A}(X_0)) \|, \frac{1}{\delta}, \frac{4\sqrt{2}}{\delta} \| m(\mathbb{A}(X_0)) \| \right\}$,*

*(iv)* $\left\| X_{\lambda,\delta}(2hk) - X_{\lambda,\delta}^k \right\| = \mathcal{O}\left( h e^{2L_{\lambda,\delta}T} \right).$

*Further more if $0 < \frac{h}{\lambda} < \frac{1}{2}$,*

*(v)* $\left\| X_{\lambda,\delta}^k - x_\lambda^k \right\| = \mathcal{O}(h) + \mathcal{O}\left( h^2 L_{\lambda,\delta} e^{2L_{\lambda,\delta}T} \right)$
$$+ 3^{\frac{T}{\lambda}} \left( \mathcal{O}\left( \frac{h}{\lambda} \right) + \mathcal{O}\left( \frac{h}{\lambda} e^{2L_{\lambda,\delta}T} \right) + \mathcal{O}(h) + \mathcal{O}\left( \frac{h^2}{\lambda} e^{2L_{\lambda,\delta}T} \right) + \mathcal{O}\left( \frac{e^{2\delta L_{\lambda,\delta}}}{L_{\lambda,\delta}} \right) + \mathcal{O}(\delta) \right).$$

We prove this lemma in next subsection, here we assume the lemma is true and prove Theorem 2.1. Suppose above lemma is true. The calculations are messy but the strategy is simple; balancing the speed of the terms $h, \delta, \lambda$ going zero to make above terms reach to zero. Above lemma motivate to take sequences as

$$\delta_n = \min \left\{ \frac{h_n}{2}, \frac{8MT}{\log_3\left( \frac{1}{h_n} \right)} \right\}$$

$$\lambda_n = \max \left\{ \delta_n, \frac{\sqrt{2} \| m(\mathbb{A}(X_0)) \| \delta_n}{M}, \frac{2T}{\log_3\left( \frac{1}{\delta_n} \right)} \right\}.$$

where $M = \max \left\{ 1, 4\sqrt{2} \| m(\mathbb{A}(X_0)) \| \right\}$. When $\lim_{n \to \infty} h_n \to 0$ with $h_n > 0$, we can easily check $\lim_{n \to \infty} \delta_n = 0$ and $\lim_{n \to \infty} \lambda_n = 0$. So the cases (i), (ii), (iii) go to zero.

Now observe from the definition of $\lambda_n$ we have,

$$\lambda_n \geq \max\left\{\frac{\sqrt{2}\,\|m(\mathbb{A}(X_0))\|\,\delta_n}{M}, \delta_n\right\} \quad\implies\quad \frac{M}{\delta_n} \geq \max\left\{\frac{\sqrt{2}\,\|m(\mathbb{A}(X_0))\|}{\lambda_n}, \frac{1}{\lambda_n}\right\} \quad\implies\quad L_{\lambda_n,\delta_n} = \frac{M}{\delta_n}.$$

Thus

$$e^{2L_{\lambda,\delta}T} \leq e^{\frac{2MT}{\delta_n}} \leq 3^{\frac{2MT}{\delta_n}} \leq 3^{\frac{1}{4}\log_3\left(\frac{1}{h_n}\right)} = \frac{1}{h_n^{1/4}}$$

$$3^{\frac{T}{\lambda_n}} \leq 3^{\frac{MT}{\delta_n}} \leq 3^{MT\frac{\log_3\left(\frac{1}{h_n}\right)}{8MT}} = \frac{1}{h_n^{1/8}}$$

$$\frac{e^{2\delta_n L_{\lambda_n,\delta_n}}}{L_{\lambda_n,\delta_n}} = e^{2M}\frac{\delta_n}{M} = \mathcal{O}\left(\delta_n\right).$$

Therefore when $\lim_{n\to\infty} h_n \to 0$,

$$h_n e^{2L_{\lambda_n,\delta_n}T} \leq h_n^{3/4} \to 0$$

$$h_n^2 L_{\lambda_n,\delta_n} e^{2L_{\lambda_n,\delta_n}T} = \frac{h_n^2}{2T}(2L_{\lambda_n,\delta_n}T)e^{2L_{\lambda_n,\delta_n}T} \leq \frac{h_n^2}{2T}e^{4L_{\lambda_n,\delta_n}T} \leq \frac{h_n^{3/2}}{2T} \to 0$$

$$3^{\frac{T}{\lambda_n}}h_n = h_n^{7/8} \to 0$$

$$3^{\frac{T}{\lambda_n}}\frac{h_n}{\lambda_n} = 3^{\frac{T}{\lambda_n}}\frac{T}{\lambda_n}\frac{h_n}{T} \leq 3^{\frac{2T}{\lambda_n}}\frac{h_n}{T} \leq \frac{h_n^{3/4}}{T} \to 0$$

$$3^{\frac{T}{\lambda_n}}\frac{h_n}{\lambda_n}e^{2L_{\lambda_n,\delta_n}T} \leq \frac{h_n^{3/4}}{T}e^{2L_{\lambda_n,\delta_n}T} \leq \frac{h_n^{1/2}}{T} \to 0$$

$$3^{\frac{T}{\lambda_n}}\frac{h_n^2}{\lambda_n}e^{2L_{\lambda_n,\delta_n}T} \leq \frac{h_n^{3/2}}{T} \to 0$$

$$3^{\frac{T}{\lambda_n}}\delta_n \leq 3^{T\frac{\log_3(\delta_n)}{2T}}\delta_n = \delta_n^{1/2} \to 0.$$

As $\lim_{n\to\infty} h_n = 0$, without loss of generality we may assume $h_n < 1$. Since $h_n > 0$ we have $\lambda_n$ is well-defined and satisfies the condition for Lemma D.3. Thus terms for the case (iv) and (v) go to zero as well. Therefore we have $\lim_{n\to\infty}\sup_{0\leq k<\frac{T}{2h_n}} S\left(h_n, \lambda_n, \delta_n, k\right) = 0$, as $\{h_n\}_{n\in\mathbb{N}}$ is arbitrary, we get the desired result.

### D.2.1  Proof for case (i), (ii), (iii), (iv) of Lemma D.3

As case proof for (v) need lot of work, we provide it in a different subsection and here we provide the proofs for the cases from (i) to (iv).

(i) $\|X(2hk) - X_\lambda(2hk)\| = \mathcal{O}\left(\sqrt{\lambda}\right)$
This is result of Lemma B.14. Considering (20) with $p = 1$, taking limit $\nu \to 0$ we know

$$\sup_{t\in[0,T]}\|X(t) - X_\lambda(t)\| \leq \sqrt{\frac{\lambda T}{2}}\,\|m(\mathbb{A}(X_0))\| = \mathcal{O}\left(\sqrt{\lambda}\right).$$

(ii) $\|X_\lambda(2hk) - X_{\lambda,\delta}(2hk)\| = \mathcal{O}\left(\delta\right)$
This is result of Proposition B.7. Consider (14) with $\gamma = p = 1$ for Lemma B.6 and taking limit $\nu \to 0$. Then applying Lemma B.10 (iii) we have

$$\sup_{t\in[0,T]}\|X_\lambda(t) - X_{\lambda,\delta}(t)\| \leq 2\sqrt{2}\delta\,\|\mathbb{A}_\lambda(X_0)\| \leq 2\sqrt{2}\delta\,\|m(\mathbb{A}(X_0))\| = \mathcal{O}\left(\delta\right).$$

(iii) $\left\|x_{h,\lambda}^k - x_h^k\right\| = O(\lambda)$

We first show show a general fact about Yosida approximation and resolvent. From [15, Proposition 23.7 (iv)] we have

$$\|\mathbf{J}_{h\mathbb{A}}(x) - \mathbf{J}_{h\mathbb{A}_\lambda}(x)\| = \left\|\mathbf{J}_{h\mathbb{A}}(x) - \left(\mathbb{I} + \frac{1}{1+\lambda}\left(\mathbf{J}_{(1+\lambda)h\mathbb{A}} - \mathbb{I}\right)\right)(x)\right\|$$

$$= \left\|\mathbf{J}_{h\mathbb{A}}(x) - \left(\frac{\lambda}{1+\lambda}x + \frac{1}{1+\lambda}\mathbf{J}_{(1+\lambda)h\mathbb{A}}(x)\right)\right\|$$

$$\leq \frac{1}{1+\lambda}\left\|\mathbf{J}_{h\mathbb{A}}(x) - \mathbf{J}_{(1+\lambda)h\mathbb{A}}(x)\right\| + \frac{\lambda}{1+\lambda}\left\|x - \mathbf{J}_{h\mathbb{A}}(x)\right\|$$

From [15, Proposition 23.31 (iii)], we have

$$\left\|\mathbf{J}_{h\mathbb{A}}(x) - \mathbf{J}_{(1+\lambda)h\mathbb{A}}(x)\right\| \leq \lambda \left\|\mathbf{J}_{h\mathbb{A}}(x) - x\right\|.$$

Combining two facts we get

$$\|\mathbf{J}_{h\mathbb{A}}(x) - \mathbf{J}_{h\mathbb{A}_\lambda}(x)\| \leq \frac{2\lambda}{1+\lambda}\left\|x - \mathbf{J}_{h\mathbb{A}}(x)\right\|.$$

From Lemma D.2, we know the iteration of $y^k$ sequence in (5) is equivalent to below sequence

$$y^{k+1} = \frac{1}{k+1}X_0 + \frac{k}{k+1}\left(2\mathbf{J}_{h\mathbb{A}} - \mathbb{I}\right)(y^k).$$

Using this alternating form we have

$$\left\|y_h^{k+1} - y_{h,\lambda}^{k+1}\right\| = \frac{k}{k+1}\left\|(2\mathbf{J}_{h\mathbb{A}} - \mathbb{I})(y_h^k) - (2\mathbf{J}_{h\mathbb{A}_\lambda} - \mathbb{I})(y_{h,\lambda}^k)\right\|$$

$$\leq \frac{k}{k+1}\left(\left\|(2\mathbf{J}_{h\mathbb{A}} - \mathbb{I})(y_h^k) - (2\mathbf{J}_{h\mathbb{A}_\lambda} - \mathbb{I})(y_h^k)\right\| + \left\|\mathbf{R}_{h\mathbb{A}_\lambda}(y_h^k) - \mathbf{R}_{h\mathbb{A}_\lambda}(y_{h,\lambda}^k)\right\|\right)$$

$$\leq \frac{k}{k+1}\left(2\left\|\mathbf{J}_{h\mathbb{A}}(y_h^k) - \mathbf{J}_{h\mathbb{A}_\lambda}(y_h^k)\right\| + \left\|y_h^k - y_{h,\lambda}^k\right\|\right)$$

$$\leq \frac{k}{k+1}\left(\frac{4\lambda}{1+\lambda}\left\|y_h^k - \mathbf{J}_{h\mathbb{A}}(y_h^k)\right\| + \left\|y_h^k - y_{h,\lambda}^k\right\|\right)$$

$$\leq \frac{k}{k+1}\left(\frac{4\lambda}{1+\lambda}\frac{\|X_0 - X_\star\|}{k} + \left\|y_h^k - y_{h,\lambda}^k\right\|\right).$$

The first inequality comes from triangular inequality. The second inequality is from nonexpansiveness of reflected resolvent $\mathbb{R}_{\mathbb{A}} = 2\mathbf{J}_{\mathbb{A}} - \mathbb{I}$, [15, Corollary 23.11]. The third inequality is from the inequality shown previously. The last inequality comes from the convergence rate of APPM[35, Theorem 4.1], $\left\|y_h^k - \mathbf{J}_{h\mathbb{A}}(y_h^k)\right\| \leq \frac{\|X_0 - X_\star\|}{k}$.

Now multiplying both sides by $k+1$ and summing up from 0 to $k$ we get

$$(k+1)\left\|y_h^{k+1} - y_{h,\lambda}^{k+1}\right\| \leq \sum_{i=0}^{k}\frac{4\lambda}{1+\lambda}\|X_0 - X_\star\| = (k+1)\frac{4\lambda}{1+\lambda}\|X_0 - X_\star\|.$$

Finally, from the relation between $x^k$ and $y^k$ in APPM we have

$$\left\|x_h^{k+1} - x_{h,\lambda}^{k+1}\right\| = \left\|\mathbf{J}_{h\mathbb{A}}(y_h^{k+1}) - \mathbf{J}_{h\mathbb{A}_\lambda}(y_{h,\lambda}^{k+1})\right\|$$

$$\leq \left\|\mathbf{J}_{h\mathbb{A}_\lambda}(y_h^{k+1}) - \mathbf{J}_{h\mathbb{A}_\lambda}(y_{h,\lambda}^{k+1})\right\| + \left\|\mathbf{J}_{h\mathbb{A}}(y_h^{k+1}) - \mathbf{J}_{h\mathbb{A}_\lambda}(y_h^{k+1})\right\|$$

$$\leq \left\|y_h^{k+1} - y_{h,\lambda}^{k+1}\right\| + \frac{2\lambda}{1+\lambda}\left\|y_h^{k+1} - \mathbf{J}_{h\mathbb{A}}(y_h^{k+1})\right\|$$

$$\leq \frac{4\lambda}{1+\lambda}\|X_0 - X_\star\| + \frac{2\lambda}{1+\lambda}\frac{\|X_0 - X_\star\|}{k+1}$$

$$= \left(2 + \frac{1}{k+1}\right)\frac{2\lambda}{1+\lambda}\|X_0 - X_\star\| = O(\lambda).$$

(iv) $\left\| X_{\lambda,\delta}\left(2hk\right) - X_{\lambda,\delta}^k \right\| = \mathcal{O}\left(he^{2L_{\lambda,\delta}T}\right)$

From (23), we can consider $\dot{X}_{\lambda,\delta}$ as of function $F\colon \mathbb{R}^n \times [0,\infty) \to \mathbb{R}^n$ defined as below

$$F(X,t) = \begin{cases} -\mathbb{A}_\lambda(X) - \frac{1}{\delta}(X - X_0) & 0 \leq t < \delta \\ -\mathbb{A}_\lambda(X) - \frac{1}{t}(X - X_0) & t > \delta. \end{cases} \tag{25}$$

Note $F$ is $2\max\left\{\frac{1}{\lambda}, \frac{1}{\delta}\right\}$-Lipschitz with respect to $X$.

For convenience name $\alpha = 2h$. Define $\epsilon_k := X_{\lambda,\delta}(\alpha k) - X_{\lambda,\delta}^k$. By definition of Euler discretization and from fundamental theorem of calculus, we have the following

$$X_{\lambda,\delta}^{k+1} = X_{\lambda,\delta}^k + \alpha F(X_{\lambda,\delta}^k, \alpha k)$$

$$X_{\lambda,\delta}(\alpha(k+1)) = X_{\lambda,\delta}(\alpha k) + \int_{\alpha k}^{\alpha(k+1)} \dot{X}_{\lambda,\delta}(t)dt$$

$$= X_{\lambda,\delta}(\alpha k) + \int_{\alpha k}^{\alpha(k+1)} \left(\dot{X}_{\lambda,\delta}(\alpha k)) + \int_{\alpha k}^{t} \ddot{X}_{\lambda,\delta}(s)ds\right) dt$$

$$= X_{\lambda,\delta}(\alpha k) + \alpha F(X_{\lambda,\delta}(\alpha k), \alpha k) + \int_{\alpha k}^{\alpha(k+1)} \int_{\alpha k}^{t} \ddot{X}_{\lambda,\delta}(s)\, ds\, dt.$$

From Lemma B.4 we have $F(X_{\lambda,\delta}, t) = \dot{X}_{\lambda,\delta}(t)$ is Lipschitz continuous respect to $t$, so $\ddot{X}_{\lambda,\delta}$ is defined almost everywhere and fundamental theorem of calculus is valid. Therefore we have

$$\epsilon_{k+1} = X_{\lambda,\delta}(\alpha(k+1)) - X_{\lambda,\delta}^{k+1}$$

$$= X_{\lambda,\delta}(\alpha k) - X_{\lambda,\delta}^k + \alpha\left(F(X_{\lambda,\delta}(\alpha k), \alpha k) - F(X_{\lambda,\delta}^k, \alpha k)\right) + \int_{\alpha k}^{\alpha(k+1)} \int_{\alpha k}^{t} \ddot{X}_{\lambda,\delta}(s)\, ds\, dt$$

As $F$ is $2\max\left\{\frac{1}{\lambda}, \frac{1}{\delta}\right\}$-Lipschitz with respect to first variable, we have

$$\|\epsilon_{k+1}\| \leq \left\| X_{\lambda,\delta}(\alpha k) - X_{\lambda,\delta}^k \right\| + \alpha\left\| F(X_{\lambda,\delta}(\alpha k), \alpha k) - F(X_{\lambda,\delta}^k, \alpha k) \right\| + \int_{\alpha k}^{\alpha(k+1)} \int_{\alpha k}^{t} \left\| \ddot{X}_{\lambda,\delta}(s) \right\|\, ds\, dt$$

$$\leq \left(1 + 2\alpha\max\left\{\frac{1}{\lambda}, \frac{1}{\delta}\right\}\right)\|\epsilon_k\| + \int_{\alpha k}^{\alpha(k+1)} \int_{\alpha k}^{\alpha(k+1)} \left\| \ddot{X}_{\lambda,\delta}(s) \right\|\, ds\, dt.$$

Now we observe $\left\| \ddot{X}_{\lambda,\delta} \right\|$ is bounded. By differentiating $\dot{X}_{\lambda,\delta}$, as

$$\ddot{X}_{\lambda,\delta} = -\frac{d}{dt}\mathbb{A}_\lambda(X_{\lambda,\delta}) + \frac{1}{t^2}\left(\dot{X}_{\lambda,\delta} - X_0\right) - \frac{1}{t}\dot{X}_{\lambda,\delta} = -\frac{d}{dt}\mathbb{A}_\lambda(X_{\lambda,\delta}) + \frac{1}{t}\left(-\mathbb{A}_\lambda(X_{\lambda,\delta}) - \dot{X}_{\lambda,\delta}\right) - \frac{1}{t}\dot{X}_{\lambda,\delta}$$

for $t > \delta$, we have for almost every $t$

$$\ddot{X}_{\lambda,\delta} = \begin{cases} -\frac{d}{dt}\mathbb{A}_\lambda(X_{\lambda,\delta}) + \frac{1}{\delta}\dot{X}_{\lambda,\delta} & 0 \leq t < \delta \\ -\frac{d}{dt}\mathbb{A}_\lambda(X_{\lambda,\delta}) - \frac{1}{t}\mathbb{A}_\lambda(X_{\lambda,\delta}) - \frac{2}{t}\dot{X}_{\lambda,\delta} & t \geq \delta \end{cases}.$$

Considering $\gamma = 1$, $p = 1$ to Lemma B.6 and from Lemma B.10 (iv) we have

$$\left\| \dot{X}_{\lambda,\delta}(t) \right\| \leq \sqrt{2}\left\| \mathbb{A}_\lambda(X_0) \right\| \leq \sqrt{2}\left\| m(\mathbb{A}(X_0)) \right\|,$$

and thus

$$\left\| \mathbb{A}_\lambda(X_{\lambda,\delta}(t)) \right\| = \left\| \dot{X}_{\lambda,\delta}(t) + \frac{1}{\max\left\{\delta, t\right\}}(X_{\lambda,\delta}(t) - X_0) \right\|$$

$$\leq \left\| \dot{X}_{\lambda,\delta}(t) \right\| + \frac{1}{\max\left\{\delta, t\right\}}\left\| X_{\lambda,\delta}(t) - X_0 \right\|$$

$$\leq \left\| \dot{X}_{\lambda,\delta}(t) \right\| + \frac{1}{\max\left\{\delta, t\right\}}\int_0^t \left\| \dot{X}_{\lambda,\delta}(s) \right\|\, ds \leq 2\sqrt{2}\left\| m(\mathbb{A}(X_0)) \right\|.$$

And since $\mathbb{A}_\lambda$ is $\frac{1}{\lambda}$-Lipschitz, we know $\mathbb{A}_\lambda \circ X_{\lambda,\delta}$ is $\frac{1}{\lambda}\sqrt{2}\,\|m(\mathbb{A}(X_0))\|$-Lispchitz, thus we have for almost all $t$,

$$\left\|\frac{d}{dt}\mathbb{A}_\lambda(X_{\lambda,\delta}(t))\right\| \leq \frac{\sqrt{2}\,\|m(\mathbb{A}(X_0))\|}{\lambda}.$$

Applying these facts we have

$$\left\|\ddot{X}_{\lambda,\delta}(t)\right\| \leq \left\|\frac{d}{dt}\mathbb{A}_\lambda(X_{\lambda,\delta}(t))\right\| + \frac{1}{\delta}\left\|\mathbb{A}_\lambda(X_{\lambda,\delta}(t))\right\| + \frac{2}{\delta}\left\|\dot{X}_{\lambda,\delta}(t)\right\|$$

$$\leq \frac{\sqrt{2}\,\|m(\mathbb{A}(X_0))\|}{\lambda} + \frac{4\sqrt{2}}{\delta}\,\|m(\mathbb{A}(X_0))\| \leq 2\max\left\{\frac{\sqrt{2}\,\|m(\mathbb{A}(X_0))\|}{\lambda}, \frac{4\sqrt{2}}{\delta}\,\|m(\mathbb{A}(X_0))\|\right\}.$$

Therefore

$$\|\epsilon_{k+1}\| \leq \left(1 + 2\alpha\max\left\{\frac{1}{\lambda}, \frac{1}{\delta}\right\}\right)\|\epsilon_k\| + 2\max\left\{\frac{\sqrt{2}\,\|m(\mathbb{A}(X_0))\|}{\lambda}, \frac{4\sqrt{2}}{\delta}\,\|m(\mathbb{A}(X_0))\|\right\}\alpha^2 \qquad (26)$$

Now for $L_{\delta,\lambda} = \max\left\{\frac{1}{\lambda}, \frac{\sqrt{2}\|m(\mathbb{A}(X_0))\|}{\lambda}, \frac{1}{\delta}, \frac{4\sqrt{2}}{\delta}\,\|m(\mathbb{A}(X_0))\|\right\}$, we show

$$\|\epsilon_k\| \leq he^{2L_{\lambda,\delta}T}.$$

Multiplying $(1 + 2\alpha L_{\lambda,\delta})^{-(k+1)}$ to (26) we have

$$(1 + 2\alpha L_{\lambda,\delta})^{-(k+1)}\|\epsilon_{k+1}\| \leq (1 + 2\alpha L_{\lambda,\delta})^{-k}\|\epsilon_k\| + (1 + 2\alpha L_{\lambda,\delta})^{-(k+1)}L_{\lambda,\delta}\alpha^2$$

As $\|\epsilon_0\| = \|X_0 - X(0)\| = 0$, summing up from $0$ to $k-1$ we have

$$(1 + 2\alpha L_{\lambda,\delta})^{-k}\|\epsilon_k\| \leq \sum_{i=1}^{k}(1 + 2\alpha L_{\lambda,\delta})^{-i}L_{\lambda,\delta}\alpha^2$$

$$= \frac{(1 + 2\alpha L_{\lambda,\delta})^{-1}\left(1 - (1 + 2\alpha L_{\lambda,\delta})^{-k}\right)}{1 - (1 + 2\alpha L_{\lambda,\delta})^{-1}}L_{\lambda,\delta}\alpha^2$$

$$= \frac{1}{2}\left(1 - (1 + 2\alpha L_{\lambda,\delta})^{-k}\right)\alpha.$$

Multiplying $(1 + 2\alpha L_{\lambda,\delta})^k$ to both sides and applying $\alpha = 2h$ we have

$$\|\epsilon_k\| \leq \frac{\alpha}{2}\left((1 + 2\alpha L_{\lambda,\delta})^k - 1\right) = h\left((1 + 4hL_{\lambda,\delta})^k - 1\right). \qquad (27)$$

Now from

$$(1 + 4hL_{\lambda,\delta})^k \leq \left((1 + 4hL_{\lambda,\delta})^{\frac{1}{4hL_{\lambda,\delta}}}\right)^{4hL_{\lambda,\delta}k} \leq e^{4hL_{\lambda,\delta}k},$$

applying $k \leq \frac{T}{2h}$

$$\|\epsilon_k\| \leq h\left(e^{4hL_{\lambda,\delta}k} - 1\right) \leq he^{2L_{\lambda,\delta}T}.$$

Therefore

$$\left\|X_{\lambda,\delta}(2hk) - X_{\lambda,\delta}^k\right\| \leq he^{2L_{\lambda,\delta}T} = \mathcal{O}\left(he^{2L_{\lambda,\delta}T}\right). \qquad (28)$$

### D.3 Proof for case (v) of Lemma D.3

As APPM has coefficient $\frac{1}{k+1}$, we consider $X_{\lambda,\delta}^{k+1}$ instead of $X_{\lambda,\delta}^k$ due to calculation simplicity. From triangular inequality, we have

$$\left\|X_{\lambda,\delta}^k - x_\lambda^k\right\| \leq \left\|X_{\lambda,\delta}^k - X_{\lambda,\delta}^{k+1}\right\| + \left\|X_{\lambda,\delta}^{k+1} - x_\lambda^k\right\|.$$

We will show

$$\left\| X_{\lambda,\delta}^k - X_{\lambda,\delta}^{k+1} \right\| = \mathcal{O}(h) + \mathcal{O}\left(h^2 L_{\lambda,\delta} e^{2L_{\lambda,\delta}T}\right)$$

$$\left\| X_{\lambda,\delta}^{k+1} - x_\lambda^k \right\| = 3^{\frac{T}{\lambda}}\left(\mathcal{O}\left(\frac{h}{\lambda}\right) + \mathcal{O}\left(\frac{h}{\lambda}e^{2L_{\lambda,\delta}T}\right) + \mathcal{O}(h) + \mathcal{O}\left(\frac{h^2}{\lambda}e^{2L_{\lambda,\delta}T}\right) + \mathcal{O}\left(\frac{e^{2\delta L_{\lambda,\delta}}}{L_{\lambda,\delta}}\right) + \mathcal{O}(\delta)\right).$$

First one is simple. Since $X_{\lambda,\delta}^{k+1} = X_{\lambda,\delta}^k + 2hF(X_{\lambda,\delta}, 2hk)$ and $F$ is $2\max\left\{\frac{1}{\lambda}, \frac{1}{\delta}\right\}$-Lipschitz with respect to the first variable, we have

$$\left\| X_{\lambda,\delta}^k - X_{\lambda,\delta}^{k+1} \right\| = 2h\left\| F(X_{\lambda,\delta}^k, 2hk) \right\|$$

$$\leq 2h\left\| F(X_{\lambda,\delta}(2hk), 2hk) \right\| + 2h\left\| F(X_{\lambda,\delta}(2hk), 2hk) - F(X_{\lambda,\delta}^k, 2hk) \right\|$$

$$\leq 2h\left\| \dot{X}_{\lambda,\delta}(2hk) \right\| + 4h\max\left\{\frac{1}{\lambda}, \frac{1}{\delta}\right\}\left\| X_{\lambda,\delta}(2hk) - X_{\lambda,\delta}^k \right\|$$

$$\leq 2\sqrt{2}h\left\| m(\mathbb{A}_\lambda(X_0)) \right\| + 4h^2 L_{\lambda,\delta} e^{2L_{\lambda,\delta}T}$$

$$= \mathcal{O}(h) + \mathcal{O}\left(h^2 L_{\lambda,\delta} e^{2L_{\lambda,\delta}T}\right).$$

Second one is complicated, we present our proof with dividing steps to subsections.

### D.3.1 Recursive inequality for $\epsilon_k = \left\| X_{\lambda,\delta}^{k+1} - x_{h,\lambda}^k \right\|$

Define $\epsilon_k = \left\| X_{\lambda,\delta}^{k+1} - x_{h,\lambda}^k \right\|$. Recall, $X_{\lambda,\delta}$ was solution of approximated ODE

$$\dot{X}_{\lambda,\delta}(t) = F(X_{\lambda,\delta}, t) = \begin{cases} -\mathbb{A}_\lambda(X_{\lambda,\delta})(t) - \frac{1}{\delta}(X(t) - X_0) & 0 \leq t < \delta \\ -\mathbb{A}_\lambda(X_{\lambda,\delta})(t) - \frac{1}{t}(X(t) - X_0) & t \geq \delta \end{cases}.$$

We now wish to write $\epsilon_{k+1}$ in terms of $\epsilon_k$. As $\epsilon_{k+1}$ involves $X_{\lambda,\delta}^{k+2}$, we first write it explicitly.

$$X_{\lambda,\delta}^{k+2} = X_{\lambda,\delta}^{k+1} + 2hF\left(X_{\lambda,\delta}^{k+1}, 2h(k+1)\right)$$

$$= \begin{cases} X_{\lambda,\delta}^{k+1} - \left(2h\mathbb{A}_\lambda(X_{\lambda,\delta}^{k+1}) + \frac{2h}{\delta}(X_{\lambda,\delta}^{k+1} - X_0)\right) & 0 \leq k+1 < \frac{\delta}{2h} \\ X_{\lambda,\delta}^{k+1} - \left(2h\mathbb{A}_\lambda(X_{\lambda,\delta}^{k+1}) + \frac{1}{k+1}(X_{\lambda,\delta}^{k+1} - X_0)\right) & k+1 \geq \frac{\delta}{2h}. \end{cases}$$

Now we find recursive inequality considering two cases.

(i) $k+1 \geq \frac{\delta}{2h}$
Recall APPM (5) was defined as

$$x_{h,\lambda}^k = \mathbb{J}_{h\mathbb{A}_\lambda} y_{h,\lambda}^{k-1}$$

$$y_{h,\lambda}^k = \frac{k}{k+1}(2x_{h,\lambda}^k - y_{h,\lambda}^{k-1}) + \frac{1}{k+1}X_0,$$

and substituting $y_{h,\lambda}^{k-1} = x_{h,\lambda}^k + h\mathbb{A}_\lambda(x_{h,\lambda}^k)$, we get a one line expression

$$x_{h,\lambda}^{k+1} = \frac{k}{k+1}x_{h,\lambda}^k - h\left(\mathbb{A}_\lambda(x_{h,\lambda}^{k+1}) + \frac{k}{k+1}\mathbb{A}_\lambda(x_{h,\lambda}^k)\right) + \frac{1}{k+1}X_0.$$

Rewriting $X_{\lambda,\delta}^{k+2}$ to make easier to compare with above,

$$X_{\lambda,\delta}^{k+2} = \frac{k}{k+1}X_{\lambda,\delta}^{k+1} - h\left(\mathbb{A}_\lambda(X_{\lambda,\delta}^{k+2}) + \frac{k}{k+1}\mathbb{A}_\lambda(X_{\lambda,\delta}^{k+1})\right) + \frac{1}{k+1}X_0$$

$$+ h\left(\mathbb{A}_\lambda(X_{\lambda,\delta}^{k+2}) - \mathbb{A}_\lambda(X_{\lambda,\delta}^{k+1})\right) - \frac{h}{k+1}\mathbb{A}_\lambda(X_{\lambda,\delta}^{k+1}).$$

As $\mathbb{A}_\lambda$ is $\frac{1}{\lambda}$-Lipschitz

$$\epsilon_{k+1} \leq \frac{k}{k+1}\epsilon_k + \frac{h}{\lambda}\left(\epsilon_{k+1} + \frac{k}{k+1}\epsilon_k\right) + \frac{h}{\lambda}\left\|X_{\lambda,\delta}^{k+2} - X_{\lambda,\delta}^{k+1}\right\| + \frac{h}{k+1}\left\|\mathbb{A}_\lambda(X_{\lambda,\delta}^{k+1})\right\|.$$

Organizing,

$$\epsilon_{k+1} \leq \frac{1}{1-\frac{h}{\lambda}}\left(\frac{k}{k+1}\left(1+\frac{h}{\lambda}\right)\epsilon_k + \underbrace{h\left(\frac{1}{\lambda}\left\|X_{\lambda,\delta}^{k+2} - X_{\lambda,\delta}^{k+1}\right\| + \frac{1}{k+1}\left\|\mathbb{A}_\lambda(X_{\lambda,\delta}^{k+1})\right\|\right)}_{:=e_k}\right).$$

(ii) $k+1 < \frac{\delta}{2h}$

Rewriting $X_{\lambda,\delta}^{k+2}$ we have

$$X_{\lambda,\delta}^{k+2} = \frac{k}{k+1}X_{\lambda,\delta}^{k+1} - h\left(\mathbb{A}_\lambda(X_{\lambda,\delta}^{k+2}) + \frac{k}{k+1}\mathbb{A}_\lambda(X_{\lambda,\delta}^{k+1})\right) + \frac{1}{k+1}X_0$$
$$+ h\left(\mathbb{A}_\lambda(X_{\lambda,\delta}^{k+2}) - \mathbb{A}_\lambda(X_{\lambda,\delta}^{k+1})\right) + \frac{h}{k+1}\mathbb{A}_\lambda(X_{\lambda,\delta}^{k+1}) + \left(\frac{1}{k+1} - \frac{2h}{\delta}\right)(X_{\lambda,\delta}^{k+1} - X_0).$$

The only difference between the case (i) is the last term. Therefore with same calculation, we get below inequality.

$$\epsilon_{k+1} \leq \frac{1}{1-\frac{h}{\lambda}}\left(\frac{k}{k+1}\left(1+\frac{h}{\lambda}\right)\epsilon_k + \underbrace{h\left(\frac{1}{\lambda}\left\|X_{\lambda,\delta}^{k+2} - X_{\lambda,\delta}^{k+1}\right\| + \frac{1}{k+1}\left\|\mathbb{A}_\lambda(X_{\lambda,\delta}^{k+1})\right\|\right) + \left|\frac{1}{k+1} - \frac{2h}{\delta}\right|\left\|X_{\lambda,\delta}^{k+1} - X_0\right\|}_{:=e_k}\right).$$

From case (i) and (ii), by defining

$$e_k = \begin{cases} h\left(\frac{1}{\lambda}\left\|X_{\lambda,\delta}^{k+2} - X_{\lambda,\delta}^{k+1}\right\| + \frac{1}{k+1}\left\|\mathbb{A}_\lambda(X_{\lambda,\delta}^{k+1})\right\|\right) + \left|\frac{1}{k+1} - \frac{2h}{\delta}\right|\left\|X_{\lambda,\delta}^{k+1} - X_0\right\| & 0 \leq k+1 < \frac{\delta}{2h} \\ h\left(\frac{1}{\lambda}\left\|X_{\lambda,\delta}^{k+2} - X_{\lambda,\delta}^{k+1}\right\| + \frac{1}{k+1}\left\|\mathbb{A}_\lambda(X_{\lambda,\delta}^{k+1})\right\|\right) & k+1 \geq \frac{\delta}{2h}, \end{cases}$$

we can write a recursive inequality for all $k$ as below

$$\epsilon_{k+1} \leq \frac{1}{1-\frac{h}{\lambda}}\left(\frac{k}{k+1}\left(1+\frac{h}{\lambda}\right)\epsilon_k + e_k\right).$$

**D.3.2** $\sup_{0 \leq i \leq \frac{T}{2h}} \epsilon_i = 3^{\frac{T}{\lambda}}\left(\mathcal{O}\left(\frac{h}{\lambda}\right) + \mathcal{O}\left(\frac{h}{\lambda}e^{2L_{\lambda,\delta}T}\right) + \mathcal{O}(h) + \mathcal{O}\left(\frac{h^2}{\lambda}e^{2L_{\lambda,\delta}T}\right) + \mathcal{O}\left(\frac{e^{2\delta L_{\lambda,\delta}}}{L_{\lambda,\delta}}\right) + \mathcal{O}(\delta)\right)$

We will now sum up above inequality. Multiplying $(k+1)\left(\frac{1-\frac{h}{\lambda}}{1+\frac{h}{\lambda}}\right)^{k+1}$ both sides we have

$$\left(\frac{1-\frac{h}{\lambda}}{1+\frac{h}{\lambda}}\right)^{k+1}(k+1)\epsilon_{k+1} \leq \left(\frac{1-\frac{h}{\lambda}}{1+\frac{h}{\lambda}}\right)^k k\epsilon_k + \frac{\left(1-\frac{h}{\lambda}\right)^{k+1}}{\left(1+\frac{h}{\lambda}\right)^k}(k+1)e_k.$$

By summing up above inequality from 0 to $k-1$, we have

$$\left(\frac{1-\frac{h}{\lambda}}{1+\frac{h}{\lambda}}\right)^k k\epsilon_k \leq \sum_{i=0}^{k-1}\frac{\left(1-\frac{h}{\lambda}\right)^{i+1}}{\left(1+\frac{h}{\lambda}\right)^i}(i+1)e_i.$$

Note $\epsilon_0$ vanished as it is multiplied with 0. Reorganizing with respect to $\epsilon_k$ we have

$$\epsilon_k \leq \left(\frac{1+\frac{h}{\lambda}}{1-\frac{h}{\lambda}}\right)^k \frac{1}{k}\sum_{i=0}^{k-1}\frac{\left(1-\frac{h}{\lambda}\right)^{i+1}}{\left(1+\frac{h}{\lambda}\right)^i}(i+1)e_i$$

$$\leq \left(\frac{1+\frac{h}{\lambda}}{1-\frac{h}{\lambda}}\right)^{\frac{T}{2h}}\frac{1}{k}\sum_{i=0}^{k-1}(i+1)e_i = \left(\left(\frac{1+\frac{h}{\lambda}}{1-\frac{h}{\lambda}}\right)^{\frac{\lambda}{h}}\right)^{\frac{T}{2\lambda}}\frac{1}{k}\sum_{i=0}^{k-1}(i+1)e_i$$

For second inequality follows from the fact $0 < \frac{h}{\lambda} < \frac{1}{2}$, which implies $0 < \frac{\left(1-\frac{h}{\lambda}\right)^{i+1}}{\left(1+\frac{h}{\lambda}\right)^i} \leq 1$ and $1 < \frac{1+\frac{h}{\lambda}}{1-\frac{h}{\lambda}}$. Observe $f(x) = \left(\frac{1+x}{1-x}\right)^{\frac{1}{x}}$ is nondecreasing in $x \in (0,1)$ since

$$f'(x) = -\frac{\left(\frac{1+x}{1-x}\right)^{\frac{1}{x}}\left((x^2-1)\log\left(\frac{1+x}{1-x}\right) + 2x\right)}{x^2(x^2-1)}.$$

Therefore, from $f\left(\frac{1}{2}\right) = 9$ we have

$$\epsilon_k \leq f\left(\frac{1}{2}\right)^{\frac{T}{2\lambda}} \frac{1}{k}\sum_{i=0}^{k-1}(i+1)e_i = 3^{\frac{T}{\lambda}}\frac{1}{k}\sum_{i=0}^{k-1}(i+1)e_i.$$

Now we show

$$\frac{1}{k}\sum_{i=0}^{k-1}(i+1)e_i = \mathcal{O}\left(\frac{h}{\lambda}\right) + \mathcal{O}\left(\frac{h}{\lambda}e^{2L_{\lambda,\delta}T}\right) + \mathcal{O}(h) + \mathcal{O}\left(\frac{h^2}{\lambda}e^{2L_{\lambda,\delta}T}\right) + \mathcal{O}(\delta) + \mathcal{O}\left(\frac{e^{2\delta L_{\lambda,\delta}}}{L_{\lambda,\delta}}\right)$$

for $0 < T < \infty$, $0 \leq k < \frac{T}{2h}$. Name $N_k = \min\left\{k, \left\lfloor\frac{\delta}{2h}\right\rfloor\right\}$. From the definition of $e_i$ we have

$$\frac{1}{k}\sum_{i=0}^{k-1}(i+1)e_i$$

$$= \frac{1}{k}\sum_{i=0}^{k-1}h(i+1)\left(\frac{1}{\lambda}\left\|X_{\lambda,\delta}^{i+2} - X_{\lambda,\delta}^{i+1}\right\| + \frac{1}{i+1}\left\|\mathbb{A}_\lambda(X_{\lambda,\delta}^{i+1})\right\|\right) + \frac{1}{k}\sum_{i=0}^{N_k-1}(i+1)\left|\frac{1}{i+1} - \frac{2h}{\delta}\right|\left\|X_{\lambda,\delta}^{i+1} - X_0\right\|$$

$$\leq \sum_{i=0}^{k-1}h\left(\frac{1}{\lambda}\left\|X_{\lambda,\delta}^{i+2} - X_{\lambda,\delta}^{i+1}\right\| + \frac{1}{k}\left\|\mathbb{A}_\lambda(X_{\lambda,\delta}^{i+1})\right\|\right) + \frac{1}{k}\sum_{i=0}^{N_k-1}\left\|X_{\lambda,\delta}^{i+1} - X_0\right\|,$$

where inequality follows from the fact $\frac{i+1}{k} \leq 1$ for $0 \leq i \leq k-1$ and $\frac{1}{i+1} \geq \frac{2h}{\delta}$ for $0 \leq i \leq N_k - 1$.

Now let's observe each term. From Lemma B.12 and Lemma D.1 we know

$$\left\|\dot{X}_{\lambda,\delta}(t)\right\| \leq \sqrt{2}\left\|\mathbb{A}_\lambda(X_0)\right\| \leq \sqrt{2}\left\|m(\mathbb{A}(X_0))\right\|$$
$$\left\|\mathbb{A}_\lambda(X(t))\right\| \leq \left\|\mathbb{A}_\lambda(X_0)\right\| \leq \left\|m(\mathbb{A}(X_0))\right\|$$
$$\left\|X_{\lambda,\delta}(t) - X_0\right\| \leq 2\left\|X_0 - X_\star\right\|.$$

Name $M = \max\left\{\sqrt{2}\left\|m(\mathbb{A}(X_0))\right\|, 2\left\|X_0 - X_\star\right\|\right\}$.

(i) $\frac{h}{\lambda}\left\|X_{\lambda,\delta}^{i+2} - X_{\lambda,\delta}^{i+1}\right\|$

First observe

$$h\sum_{i=0}^{k-1}\frac{1}{\lambda}\left\|X_{\lambda,\delta}(2h(i+2)) - X_{\lambda,\delta}(2h(i+1))\right\|$$

$$= \frac{h}{\lambda}\sum_{i=0}^{k-1}\left\|\int_{2h(i+1)}^{2h(i+2)}\dot{X}_{\lambda,\delta}(t)dt\right\| \leq \frac{h}{\lambda}\sum_{i=0}^{k-1}\int_{2h(i+1)}^{2h(i+2)}\left\|\dot{X}_{\lambda,\delta}(t)\right\|dt \leq \frac{h}{\lambda}k(2h)M \leq \frac{h}{\lambda}TM.$$

Thus,

$$h \sum_{i=0}^{k-1} \frac{1}{\lambda} \left\| X_{\lambda,\delta}^{i+2} - X_{\lambda,\delta}^{i+1} \right\|$$

$$\leq \frac{h}{\lambda} \sum_{i=0}^{k-1} \left( \left\| X_{\lambda,\delta}^{i+2} - X_{\lambda,\delta}(2h(i+2)) \right\| + \| X_{\lambda,\delta}(2h(i+2)) - X_{\lambda,\delta}(2h(i+1)) \| + \left\| X_{\lambda,\delta}(2h(i+1)) - X_{\lambda,\delta}^{i+1} \right\| \right)$$

$$\leq \frac{h}{\lambda} \left( TM + 2 \sum_{i=0}^{k-1} h e^{2L_{\lambda,\delta}T} \right)$$

$$\leq \frac{h}{\lambda} \left( TM + T e^{2L_{\lambda,\delta}T} \right).$$

Therefore

$$\frac{h}{\lambda} \sum_{i=0}^{k-1} \left\| X_{\lambda,\delta}^{i+2} - X_{\lambda,\delta}^{i+1} \right\| = \mathcal{O}\left( \frac{h}{\lambda} \right) + \mathcal{O}\left( \frac{h}{\lambda} e^{2L_{\lambda,\delta}T} \right).$$

(ii) $\frac{h}{k} \left\| \mathbb{A}_\lambda(X_{\lambda,\delta}^{i+1}) \right\|$

Since $\mathbb{A}_\lambda$ is $\frac{1}{\lambda}$-Lipschitz continuous and from (28)

$$\sum_{i=0}^{k-1} \frac{h}{k} \left\| \mathbb{A}_\lambda(X_{\lambda,\delta}^{i+1}) \right\| \leq \frac{h}{k} \sum_{i=0}^{k-1} \left( \left\| \mathbb{A}_\lambda(X_{\lambda,\delta}^i) - \mathbb{A}_\lambda(X_{\lambda,\delta}(2hi)) \right\| + \| \mathbb{A}_\lambda(X_{\lambda,\delta}(2hi)) \| \right)$$

$$\leq \frac{h}{k} \sum_{i=0}^{k-1} \left( \frac{h}{\lambda} e^{2L_{\lambda,\delta}T} + M \right) = h \left( \frac{h}{\lambda} e^{2L_{\lambda,\delta}T} + M \right).$$

Therefore

$$\sum_{i=0}^{k-1} \frac{h}{k} \left\| \mathbb{A}_\lambda(X_{\lambda,\delta}^{i+1}) \right\| = \mathcal{O}(h) + \mathcal{O}\left( \frac{h^2}{\lambda} e^{2L_{\lambda,\delta}T} \right).$$

(iii) $\frac{1}{k} \sum_{i=0}^{N_k-1} \left\| X_{\lambda,\delta}^{i+1} - X_0 \right\|$ for $N_k = \min\left\{ k, \left\lfloor \frac{\delta}{2h} \right\rfloor \right\}$.

First, observe

$$\| X_0 - X_{\lambda,\delta}(t) \| = \left\| \int_0^t \dot{X}_{\lambda,\delta}(s) ds \right\| \leq \int_0^t \left\| \dot{X}_{\lambda,\delta}(s) \right\| ds \leq tM.$$

From (27)

$$\left\| X_{\lambda,\delta}^{i+1} - X_0 \right\| \leq \left\| X_{\lambda,\delta}^{i+1} - X_{\lambda,\delta}(2h(i+1)) \right\| + \| X_{\lambda,\delta}(2h(i+1)) - X_0 \|$$

$$\leq h \left( 1 + 4hL_{\lambda,\delta} \right)^{i+1} + 2h(i+1)M.$$

Now as $i + 1 \leq N_k = \min\left\{ k, \left\lfloor \frac{\delta}{2h} \right\rfloor \right\}$,

$$\frac{1}{k} \sum_{i=0}^{N_k-1} \left\| X_{\lambda,\delta}^{i+1} - X_0 \right\| \leq \frac{1}{k} \sum_{i=0}^{N_k-1} h \left( (1 + 4hL_{\lambda,\delta})^{i+1} + 2(i+1)M \right)$$

$$\leq \frac{h}{k} \frac{(1 + 4hL_{\lambda,\delta})^{N_k} - 1}{4hL_{\lambda,\delta}} + 2hM \sum_{i=0}^{N_k-1} \frac{i+1}{k}$$

$$\leq \frac{e^{4hL_{\lambda,\delta}N_k}}{kL_{\lambda,\delta}} + 2hN_kM$$

$$\leq \frac{e^{2\delta L_{\lambda,\delta}}}{L_{\lambda,\delta}} + 2\delta M = \mathcal{O}\left( \frac{e^{2\delta L_{\lambda,\delta}}}{L_{\lambda,\delta}} \right) + \mathcal{O}(\delta).$$

From (i), (ii), (iii) we have

$$\frac{1}{k}\sum_{i=0}^{k-1}(i+1)e_i = \mathcal{O}\left(\frac{h}{\lambda}\right) + \mathcal{O}\left(\frac{h}{\lambda}e^{2L_{\lambda,\delta}T}\right) + \mathcal{O}(h) + \mathcal{O}\left(\frac{h^2}{\lambda}e^{2L_{\lambda,\delta}T}\right) + \mathcal{O}\left(\frac{e^{2\delta L_{\lambda,\delta}}}{L_{\lambda,\delta}}\right) + \mathcal{O}(\delta).$$

Therefore,

$$\epsilon_k = 3^{\frac{T}{\lambda}}\left(\mathcal{O}\left(\frac{h}{\lambda}\right) + \mathcal{O}\left(\frac{h}{\lambda}e^{2L_{\lambda,\delta}T}\right) + \mathcal{O}(h) + \mathcal{O}\left(\frac{h^2}{\lambda}e^{2L_{\lambda,\delta}T}\right) + \mathcal{O}\left(\frac{e^{2\delta L_{\lambda,\delta}}}{L_{\lambda,\delta}}\right) + \mathcal{O}(\delta)\right).$$

## D.4  Derivation of ODE (4) from EAG and FEG

### D.4.1  Derivation from EAG

For $L$-Lipschitz continuous monotone operator $\mathbb{A}$ and stepsize $h > 0$, EAG-C [75] is defined as

$$z^{k+\frac{1}{2}} = z^k - \frac{1}{k+2}\left(z^k - z^0\right) - h\mathbb{A}(z^k)$$

$$z^{k+1} = z^k - \frac{1}{k+2}\left(z^k - z^0\right) - h\mathbb{A}(z^{k+\frac{1}{2}}).$$

Dividing the second line by $h$ and reorganizing we have

$$\frac{z^{k+1} - z^k}{h} = -\mathbb{A}(z^{k+\frac{1}{2}}) - \frac{1}{h(k+2)}\left(z^k - z^0\right)$$

$$= -\mathbb{A}(z^k) - \frac{1}{h(k+2)}\left(z^k - z^0\right) - \left(\mathbb{A}(z^{k+\frac{1}{2}}) - \mathbb{A}(z^k)\right).$$

Identify $hk = t$, $z^k = X(t)$, $z^0 = X_0$. As $z^k$ is a converging sequence [76], it is bounded. Thus we see

$$\left\|\mathbb{A}(z^{k+\frac{1}{2}}) - \mathbb{A}(z^k)\right\| \leq L\left\|z^{k+\frac{1}{2}} - z^k\right\|$$

$$= L\left\|\frac{h}{h(k+2)}(z^k - z^0) + h\mathbb{A}(z^k)\right\|$$

$$= Lh\left\|\frac{1}{t+2h}(X(t) - X_0) + \mathbb{A}(X(t))\right\| = \mathcal{O}(h).$$

Therefore taking limit $h \to 0+$ we have

$$\dot{X}(t) = -\mathbb{A}(X(t)) - \frac{1}{t}(X(t) - X_0).$$

### D.4.2  Derivation from FEG

For $L$-Lipschitz continuous monotone operator $\mathbb{A}$ and stepsize $h > 0$, FEG [42] is defined as

$$z^{k+\frac{1}{2}} = z^k - \frac{1}{k+1}\left(z^k - z^0\right) - \frac{k}{k+1}h\mathbb{A}(z^k)$$

$$z^{k+1} = z^k - \frac{1}{k+1}\left(z^k - z^0\right) - h\mathbb{A}(z^{k+\frac{1}{2}}).$$

Dividing the second line by $h$ and reorganizing we have

$$\frac{z^{k+1} - z^k}{h} = -\mathbb{A}(z^{k+\frac{1}{2}}) - \frac{1}{h(k+1)}\left(z^k - z^0\right)$$

$$= -\mathbb{A}(z^k) - \frac{1}{h(k+1)}\left(z^k - z^0\right) - \left(\mathbb{A}(z^{k+\frac{1}{2}}) - \mathbb{A}(z^k)\right).$$

Identify $hk = t$, $z^k = X(t)$, $z^0 = X_0$. As $z^k$ is a converging sequence [76], it is bounded. Thus we see

$$\left\| \mathbb{A}(z^{k+\frac{1}{2}}) - \mathbb{A}(z^k) \right\| \leq L \left\| z^{k+\frac{1}{2}} - z^k \right\|$$

$$= L \left\| \frac{h}{h(k+1)}(z^k - z^0) + h\frac{hk}{h(k+1)}\mathbb{A}(z^k) \right\|$$

$$= Lh \left\| \frac{1}{t+h}(X(t) - X_0) + \frac{t}{t+h}\mathbb{A}(X(t)) \right\| = \mathcal{O}(h).$$

Therefore taking limit $h \to 0+$ we have

$$\dot{X}(t) = -\mathbb{A}(X(t)) - \frac{1}{t}(X(t) - X_0).$$

# E    Proof of convergence analysis for monotone $\mathbb{A}$

## E.1    Extending $\tilde{\mathbb{A}}$ to $[0, \infty)$

**Lemma E.1.** *Suppose $\mathbb{A}$ is a maximal monotone operator. Let $X$ be the solution for* (3). *Define $S$ as*

$$S = \left\{ t \in [0, \infty) \mid \dot{X}(t) \in -\mathbb{A}(X(t)) - \beta(t)(X(t) - X_0) \text{ is true} \right\}.$$

*Then as $X$ is the solution for* (3), *we know $[0, \infty) \backslash S$ is of measure zero.*

*Define $\tilde{\mathbb{A}}(X) \colon S \to \mathbb{R}^n$ as*

$$\tilde{\mathbb{A}}(X)(t) = -\dot{X}(t) - \beta(t)(X(t) - X_0) \tag{29}$$

*and denote $\tilde{\mathbb{A}}(X(t)) = \tilde{\mathbb{A}}(X)(t)$. Assume for every $T > 0$, there is $M > 0$ such that for all $t \in S \cap [0, T]$*

$$\left\| \tilde{\mathbb{A}}(X(t)) \right\| \leq M.$$

*Then $\tilde{\mathbb{A}}(X)$ can be extended to $t \in [0, \infty)$ with satisfying following properties.*

  *(i)  $\tilde{\mathbb{A}}(X(t)) \in \mathbb{A}(X(t))$ for all $t \in [0, \infty)$.*

  *(ii)  For $t \in [0, \infty) \backslash S$, there is a sequence $\{t_k\}_{k \in \mathbb{N}}$ such that $t_k \in S$, $\lim_{k \to \infty} t_k = t$ and $\tilde{\mathbb{A}}(X(t_k))$ converges to $\tilde{\mathbb{A}}(X(t))$.*

*Proof.* Take $t \in [0, \infty)$, and take a sequence $\{t_n\}_{n \in \mathbb{N}}$ such that $t_n \in S$ and $\lim_{n \to \infty} t_n = t$. As a converging sequence $t_n$ is bounded, there is $T > 0$ such that $t_n \in [0, T]$. For that $T$, we have $\left\| \tilde{\mathbb{A}}(X(t_n)) \right\| \leq M$ by the assumption. As $n \mapsto \tilde{\mathbb{A}}(X(t_n))$ is a bounded sequence, there is a subsequence $\{t_{n_k}\}_{k \in \mathbb{N}}$ such that $k \mapsto \tilde{\mathbb{A}}(X(t_{n_k}))$ converges. Name the limit as $u = \lim_{k \to \infty} \tilde{\mathbb{A}}(X(t_{n_k}))$. On the other hand, as $X$ is a continuous curve we have $\lim_{k \to \infty} X(t_{n_k}) = X(t)$. Then since $\mathbb{A}$ is maximally monotone, from [15, Proposition 20.38] we have $(X(t), u) \in \mathrm{Gra}\,\mathbb{A}$. Defining $\tilde{\mathbb{A}}(X(t))$ as $u$, we get the desired result. $\qquad\square$

**Corollary E.2.** *Let $X$ be the solution for* (6). *Then $\tilde{\mathbb{A}}(X)$ defined as* (29) *has extension with properties stated in Lemma E.1.*

*Proof.* First consider the case $\beta(t) = \frac{\gamma}{t^p}$, $p > 0$, $\gamma > 0$. Take $T > 0$. It is enough to show there is $M > 0$ such that for $t \in S \cap [0, T]$

$$\left\| \tilde{\mathbb{A}}(X(t)) \right\| \leq M.$$

Recall from Proposition B.11 for Yosida approximation $\mathbb{A}_\lambda$, we denoted $X_\lambda$ as the solution of

$$\dot{X}_\lambda = -\mathbb{A}_\lambda(X_\lambda) - \frac{\gamma}{t^p}(X_\lambda - X_0),$$

and we have shown there is a sequence $\{\lambda_n\}_{n \in \mathbb{N}}$ such that $X_{\lambda_n}$ uniformly converges to $X$ on $[0, T]$.

From Lemma B.12, we see for $h \neq 0$

$$\left\| \frac{X_\lambda(t+h) - X_\lambda(t)}{h} \right\| \leq \frac{\int_t^{t+h} \left\| \dot{X}_\lambda(s) \right\| ds}{h} \leq \frac{\int_t^{t+h} M_{dot}(T) ds}{h} = M_{dot}(T),$$

thus

$$\left\| \frac{X(t+h) - X(t)}{h} \right\| = \lim_{\lambda \to 0+} \left\| \frac{X_\lambda(t+h) - X_\lambda(t)}{h} \right\| \leq M_{dot}(T).$$

Therefore for $t \in S \cap [0,T]$, $\dot{X}(t) = \lim_{h \to 0} \frac{X(t+h) - X(t)}{h}$ holds, we conclude

$$\left\| \dot{X}(t) \right\| = \lim_{h \to 0} \left\| \frac{X(t+h) - X(t)}{h} \right\| \leq M_{dot}(T).$$

And also from Lemma B.12, for $t \in [0,T]$ we have

$$\left\| \frac{\gamma}{t^p} (X_\lambda(t) - X_0) \right\| = \left\| \dot{X}_\lambda(t) + \mathbb{A}_\lambda(t) \right\| \leq \left\| \dot{X}_\lambda(t) \right\| + \| \mathbb{A}_\lambda(t) \| \leq M_{dot}(T) + M_{\mathbb{A}}(T),$$

therefore

$$\left\| \frac{\gamma}{t^p} (X(t) - X_0) \right\| = \lim_{\lambda \to 0+} \left\| \frac{\gamma}{t^p} (X_\lambda(t) - X_0) \right\| \leq M_{dot}(T) + M_{\mathbb{A}}(T).$$

Gathering the result, for $t \in S \cap [0,T]$ we have

$$\left\| \tilde{\mathbb{A}}(X(t)) \right\| = \left\| \dot{X}(t) + \frac{\gamma}{t^p} (X(t) - X_0) \right\| \leq \left\| \dot{X}(t) \right\| + \left\| \frac{\gamma}{t^p} (X(t) - X_0) \right\| \leq 2 M_{dot}(T) + M_{\mathbb{A}}(T).$$

$\square$

## E.2 Proof of Proposition 3.2

We prove this theorem by deriving the energy with dilated coordinate $W(t) = C(t)(X(t) - X_0)$ and conservation law from [67]. Think of dilated coordinate $W(t) = C(t)(X(t) - X_0)$. As we're considering the case $\tilde{\mathbb{A}}$ is Lipschitz continuous, the differential inclusion (3) becomes ODE

$$\dot{X}(t) = -\tilde{\mathbb{A}}(X(t)) - \beta(t)(X(t) - X_0).$$

Rewriting the ODE in terms of $W$, we have

$$\frac{1}{C(t)} \left( \dot{W}(t) - \beta(t) W(t) \right) = -\tilde{\mathbb{A}}(X(W(t), t)) - \frac{\beta(t)}{C(t)} W(t)$$

where $X(W(t), t) = X(t) = \frac{W(t)}{C(t)} + X_0$. Organizing, we have

$$0 = \dot{W}(t) + C(t) \tilde{\mathbb{A}}(X(W(t), t)).$$

From Lemma B.4 we know $\tilde{\mathbb{A}}(X(W, t))$ is differentiable almost everywhere, by differentiating we obtain second order ODE which holds almost everywhere

$$0 = \ddot{W} + C(t) \beta(t) \tilde{\mathbb{A}}(X(W(t), t)) + C(t) \frac{d}{dt} \tilde{\mathbb{A}}(X(W(t), t))$$

$$= \ddot{W} - \beta(t) \dot{W} + C(t) \frac{d}{dt} \tilde{\mathbb{A}}(X(W(t), t)). \tag{30}$$

Now by taking inner product with $\dot{W}$ and integrating, we get equality which was refered as conservation law in [67]

$$E_1 \equiv \frac{1}{2} \left\| \dot{W}(t) \right\|^2 - \int_{t_0}^t \beta(s) \left\| \dot{W}(s) \right\|^2 ds + \int_{t_0}^t C(s) \left\langle \frac{d}{ds} \tilde{\mathbb{A}}(X(s)), \dot{W}(s) \right\rangle ds.$$

Since $\dot{W}(t) = C(t)\left(\dot{X}(t) + \beta(t)(X(t) - X_0)\right)$, we can rewrite the integrand in the last term as

$$\int_{t_0}^t C(s) \left\langle \frac{d}{ds}\tilde{\mathbf{A}}(X(s)), \dot{W}(s) \right\rangle ds = \int_{t_0}^t C(s)^2 \left\langle \frac{d}{ds}\tilde{\mathbf{A}}(X(s)), \dot{X}(s) \right\rangle ds + \int_{t_0}^t \left\langle \frac{d}{ds}\tilde{\mathbf{A}}(X(s)), C(s)\beta(s)W(s) \right\rangle ds.$$

Note the purpose was to obtain the first term, which is nonnegative due to monotonicity of $\tilde{\mathbf{A}}$. Now taking integration by parts to the second term we have

$$\int_{t_0}^t \left\langle \frac{d}{ds}\tilde{\mathbf{A}}(X(s)), C(s)\beta(s)W(s) \right\rangle ds$$

$$= \left[\langle \tilde{\mathbf{A}}(X(W,s)), C(s)\beta(s)W(s) \rangle\right]_{t_0}^t - \int_{t_0}^t \left\langle \tilde{\mathbf{A}}(X(W,s)), \left(C(s)\beta(s)^2 + C(s)\dot{\beta}(s)\right)W(s) + C(s)\beta(s)\dot{W}(s) \right\rangle ds$$

$$= \left[\langle \tilde{\mathbf{A}}(X(W,s)), C(s)\beta(s)W(s) \rangle\right]_{t_0}^t + \int_{t_0}^t \beta(s)\left\|\dot{W}(s)\right\|^2 ds + \int_{t_0}^t \left(\beta(s)^2 + \dot{\beta}(s)\right)\left\langle \dot{W}(s), W(s) \right\rangle ds$$

$$= \left[\langle \tilde{\mathbf{A}}(X(W,s)), C(s)\beta(s)W(s) \rangle\right]_{t_0}^t + \int_{t_0}^t \beta(s)\left\|\dot{W}(s)\right\|^2 ds$$

$$+ \left[\left(\beta(s)^2 + \dot{\beta}(s)\right)\frac{1}{2}\|W(s)\|^2\right]_{t_0}^t - \frac{1}{2}\int_{t_0}^t \left(2\beta(s)\dot{\beta}(s) + \ddot{\beta}(s)\right)\|W(s)\|^2 ds$$

On the second equality, we used the fact $\dot{W}(t) = -C(t)\tilde{\mathbf{A}}(X(W(t), t))$. Note the fundamental theorem of calculus for $\langle \tilde{\mathbf{A}}(X(W,s)), C(s)\beta(s)W(s) \rangle$ is valid since $\tilde{\mathbf{A}}(X(W,s))$ is Lipschitz continuous and $C(s)\beta(s)W(s)$ is continuously differentiable in $[t_0, t]$, so their inner product is absolutely continuous in $[t_0, t]$.

Observe the integrand in the last term can be rewritten as

$$\left(2\beta(s)\dot{\beta}(s) + \ddot{\beta}(s)\right)\|W(s)\|^2 = C(s)^2\left(2\beta(s)\dot{\beta}(s) + \ddot{\beta}(s)\right)\|X(s) - X_0\|^2 = \frac{d}{ds}\left(C(s)^2\dot{\beta}(s)\right)\|X(s) - X_0\|^2.$$

Now gathering the results, we conclude

$$E_1 \equiv \frac{1}{2}\left\|\dot{W}(t)\right\|^2 - \int_{t_0}^t \beta(s)\left\|\dot{W}(s)\right\|^2 ds + \int_{t_0}^t C(s) \left\langle \frac{d}{ds}\tilde{\mathbf{A}}(X(s)), \dot{W}(s) \right\rangle ds$$

$$= \frac{C(t)^2}{2}\left\|\tilde{\mathbf{A}}(X(t))\right\|^2 + \left[\langle \tilde{\mathbf{A}}(X(W(s),s)), C(s)\beta(s)W(s) \rangle\right]_{t_0}^t + \left[\left(\beta(s)^2 + \dot{\beta}(s)\right)\frac{1}{2}\|W(s)\|^2\right]_{t_0}^t$$

$$+ \int_{t_0}^t C(s)^2 \left\langle \frac{d}{ds}\tilde{\mathbf{A}}(X(s)), \dot{X}(s) \right\rangle ds - \frac{1}{2}\int_{t_0}^t \frac{d}{ds}\left(C(s)^2\dot{\beta}(s)\right)\|X(s) - X_0\|^2 ds$$

$$= \frac{C(t)^2}{2}\left(\|\tilde{\mathbf{A}}(X(t))\|^2 + 2\beta(t)\langle \tilde{\mathbf{A}}(X(t)), X(t) - X_0 \rangle + \left(\beta(t)^2 + \dot{\beta}(t)\right)\|X(t) - X_0\|^2\right)$$

$$+ \int_{t_0}^t C(s)^2 \left\langle \frac{d}{ds}\tilde{\mathbf{A}}(X(s)), \dot{X}(s) \right\rangle ds - \frac{1}{2}\int_{t_0}^t \frac{d}{ds}\left(C(s)^2\dot{\beta}(s)\right)\|X(s) - X_0\|^2 ds$$

$$\underbrace{- \frac{C(t_0)^2}{2}\left(2\beta(t_0)\langle \tilde{\mathbf{A}}(X(t_0)), X(t_0) - X_0 \rangle + \left(\beta(t_0)^2 + \dot{\beta}(t_0)\right)\|X(t_0) - X_0\|^2\right)}_{=\text{constant}}$$

Moving the constant terms to left hand side and naming $E = E_1 - \text{constant}$, we get the desired result.

### E.3 Proof of Corollary 3.3

### E.3.1 $V$ is nonincreasing when $\tilde{\mathbf{A}}$ is Lipshitz continuous monotone

We first check it is true for the case $\tilde{\mathbf{A}}$ is Lipschitz continuous. Then from Proposition 3.2 we can write $V(t)$ as

$$V(t) = E - 2\int_{t_0}^t C(s)^2 \left\langle \frac{d}{ds}\tilde{\mathbf{A}}(X(s)), \dot{X}(s) \right\rangle ds$$

with $E$ in Proposition 3.2. As $\tilde{\mathbb{A}}$ is monotone, from (1) we know $C(s)^2 \left\langle \frac{d}{ds}\tilde{\mathbb{A}}(X(s)), \dot{X}(s) \right\rangle \geq 0$ holds almost everywhere. Therefore for $h > 0$

$$V(t+h) - V(t) = -2\int_t^{t+h} C(s)^2 \left\langle \frac{d}{ds}\tilde{\mathbb{A}}(X(s)), \dot{X}(s) \right\rangle ds \leq 0,$$

we see $V$ is a nonincreasing function. Therefore for all $t > 0$, $V(t) \leq \lim_{\epsilon \to 0+} V(\epsilon)$. It remains to show the limit $\lim_{\epsilon \to 0+} V(\epsilon)$ exists.

### E.3.2 Calculation of $V(0) = \lim_{t \to 0+} V(t)$ for Lipshitz continuous monotone $\tilde{\mathbb{A}}$

In this section, we calculate $\lim_{t \to 0+} V(t)$ when $\tilde{\mathbb{A}}$ is Lipshitz continuous. Recall $V$ was defined as

$$V(t) = \frac{C(t)^2}{2}\left( \left\|\tilde{\mathbb{A}}(X(t))\right\|^2 + 2\beta(t)\left\langle \tilde{\mathbb{A}}(X(t)), X(t) - X_0 \right\rangle + \left(\beta(t)^2 + \dot{\beta}(t)\right)\|X(t) - X_0\|^2\right)$$
$$- \int_{t_0}^t \frac{d}{ds}\left(\frac{C(s)^2\dot{\beta}(s)}{2}\right)\|X(s) - X_0\|^2 ds.$$

From now we denote $V$ for the case $t_0 = 0$ as $V^0$.

We first check

$$C(t) = \begin{cases} t^\gamma & p = 1 \\ e^{\frac{\gamma}{1-p}t^{1-p}} & p > 0, p \neq 1. \end{cases} \tag{31}$$

Observing

$$\int_1^t \frac{\gamma}{s}ds = \gamma \log t$$

$$\int_0^t \frac{\gamma}{s^p}ds = \frac{\gamma}{1-p}t^{1-p} \qquad \text{for } 0 < p < 1$$

$$\int_\infty^t \frac{\gamma}{s^p}ds = \frac{\gamma}{1-p}t^{1-p} \qquad \text{for } p > 1,$$

we see $C(t)$ defined above agrees with the definition of $C(t)$ for each case. Note

$$\lim_{t \to 0+} C(t) = \begin{cases} 0 & p \geq 1 \\ 1 & 0 < p < 1. \end{cases}$$

Now we will show

$$\lim_{t \to 0+} V^0(t) = \lim_{t \to 0+} \frac{C(t)^2}{2}\left\|\tilde{\mathbb{A}}(X(t))\right\|^2 = \begin{cases} 0 & \text{if } p \geq 1 \\ \frac{\|\tilde{\mathbb{A}}(X_0)\|^2}{2} & \text{if } 0 < p < 1. \end{cases} \tag{32}$$

To do so, we first show for $t > 0$

$$\lim_{\epsilon \to 0+} \int_\epsilon^t \frac{d}{ds}\left(\frac{C(s)^2\dot{\beta}(s)}{2}\right)\|X(s) - X_0\|^2 ds < \infty.$$

We first provide an elementary fact as a lemma.

**Lemma E.3.** *Let $f\colon (0, \infty) \to \mathbb{R}$ is a continuous function. Suppose there is $q < 1$, $0 < l < \infty$ such that*

$$\limsup_{t \to 0+} |f(t)t^q| < l.$$

*Then for $t > 0$, $f \in L^1([0, t], \mathbb{R})$.*

*Proof.* Since $\limsup_{s\to 0+} f(s)s^q = l$, there is $\epsilon \in (0, t)$ such that

$$0 < s < \epsilon \implies |f(s)s^q| < 2|l|.$$

Then since $q < 1$

$$\int_0^\epsilon |f(s)|\, ds = \int_0^\epsilon |f(s)s^q|\frac{1}{s^q}ds \le \int_0^\epsilon \frac{2|l|}{s^q}ds = \left[\frac{2|l|}{1-q}s^{1-q}\right]_0^\epsilon = \frac{2|l|}{1-q}\epsilon^{1-q} < \infty.$$

By the way as $f(t)$ is continuous on $[\epsilon, t]$, $M = \max_{s\in[\epsilon,t]} |f(s)|$ exists. Therefore,

$$\int_0^t |f(s)|ds = \int_0^\epsilon |f(s)|ds + \int_\epsilon^t |f(s)|ds \le \frac{2|l|}{1-q}\epsilon^{1-q} + M(t - \epsilon) < \infty.$$

$\square$

Applying Lemma E.3, we will show for $t > 0$

$$\left|s^2 \frac{d}{ds}\left(C(s)^2 \dot{\beta}(s)\right)\right| \in L^1([0,t], \mathbb{R}). \tag{33}$$

Observe

$$\frac{d}{ds}\left(C(s)^2 \dot{\beta}(s)\right) = 2C(s)^2\beta(s)\dot{\beta}(s) + C(s)^2\ddot{\beta}(s) = C(s)^2\left(-\frac{2p\gamma^2}{s^{2p+1}} + \frac{p(1+p)\gamma}{s^{p+2}}\right). \tag{34}$$

(i) $p > 1$

We first show $\lim_{s\to 0+} C(s)^2 \frac{1}{s^n} = 0$ for all $n > 0$. Take $n > 0$. Then there is some $k \in \mathbb{N}$ such that $k(p-1) > n$. With change of variable $u = \frac{1}{s}$ and L'Hóptial's rule we see

$$\lim_{s\to 0+} C(s)^2 \frac{1}{s^n} = \lim_{u\to\infty} \frac{u^n}{e^{\frac{2\gamma}{p-1}u^{p-1}}}$$

$$= \lim_{u\to\infty} \frac{nu^{n-1}}{2\gamma u^{p-2}e^{\frac{2\gamma}{p-1}u^{p-1}}} = \frac{n}{2\gamma}\lim_{u\to\infty} \frac{u^{n+1-p}}{e^{\frac{2\gamma}{p-1}u^{p-1}}} = \cdots = \frac{\prod_{m=0}^{k-1}(n - m(p-1))}{(2\gamma)^k}\lim_{u\to\infty} \frac{u^{n-k(p-1)}}{e^{\frac{2\gamma}{p-1}u^{p-1}}} = 0. \tag{35}$$

And thus

$$\lim_{s\to 0+}\left|s^2 \frac{d}{ds}\left(\frac{C(s)^2 \dot{\beta}(s)}{2}\right)\right| = \lim_{s\to 0+}\left|C(s)^2\left(-\frac{2p\gamma^2}{s^{2p-1}} + \frac{p(1+p)\gamma}{s^p}\right)\right| = 0.$$

By Lemma E.3, we conclude $\int_0^t \left|s^2 \frac{d}{ds}\left(\frac{C(s)^2 \dot{\beta}(s)}{2}\right)\right| ds < \infty$.

(ii) $p = 1$

Since $C(s) = s^\gamma$, we see

$$\lim_{s\to 0+}\left|s^2 \frac{d}{ds}\left(\frac{C(s)^2 \dot{\beta}(s)}{2}\right)\right| \cdot s^{1-\gamma} = \lim_{s\to 0+}\left|s^2 \cdot \gamma(\gamma - 1)s^{2\gamma-3} \cdot s^{1-\gamma}\right| = \gamma\,|\gamma - 1| \lim_{s\to 0+} s^\gamma = 0.$$

Since $1 - \gamma < 1$, by Lemma E.3 we conclude $\int_0^t \left|s^2 \frac{d}{ds}\left(\frac{C(s)^2 \dot{\beta}(s)}{2}\right)\right| ds < \infty$.

(iii) $0 < p < 1$

Since $\lim_{s\to 0+} C(s) = \lim_{s\to 0+} e^{\frac{\gamma}{1-p}s^{1-p}} = 1$, we see

$$\limsup_{s\to 0+}\left|s^2 \frac{d}{ds}\left(\frac{C(s)^2 \dot{\beta}(s)}{2}\right)\right| \cdot s^p = \limsup_{s\to 0+} C(s)^2 \left|-2p\gamma^2 s^{1-p} + p(1+p)\gamma\right| = p(1+p)\gamma.$$

Since $p < 1$, by Lemma E.3 we conclude $\int_0^t \left|s^2 \frac{d}{ds}\left(\frac{C(s)^2 \dot{\beta}(s)}{2}\right)\right| ds < \infty$.

Naming the bound in Corollary B.8 as $M(T)$, we know for $0 < s < T$

$$\left\| \frac{X(s) - X_0}{s} \right\| \leq \frac{\int_0^s \left\| \dot{X}(u) \right\| du}{s} \leq \frac{\int_0^s M(T) du}{s} = M(T).$$

Therefore applying (33), for $0 < t < T$ we have

$$\int_0^t \left| \frac{d}{ds} \left( \frac{C(s)^2 \dot{\beta}(s)}{2} \right) \right| \|X(s) - X_0\|^2 \, ds \leq M(T)^2 \int_0^t \left| s^2 \frac{d}{ds} \left( \frac{C(s)^2 \dot{\beta}(s)}{2} \right) \right| ds < \infty.$$

Hence we know $V^0(t)$ is well defined and since $\lim_{t \to 0+} \int_0^t \frac{d}{ds} \left( \frac{C(s)^2 \dot{\beta}(s)}{2} \right) \|X(s) - X_0\|^2 \, ds = 0$, we have

$$\lim_{t \to 0+} V^0(t) = \lim_{t \to 0+} \frac{C(t)^2}{2} \left( \left\| \tilde{\mathbb{A}}(X(t)) \right\|^2 + 2\beta(t) \left\langle \tilde{\mathbb{A}}(X(t)), X(t) - X_0 \right\rangle + \left( \beta(t)^2 + \dot{\beta}(t) \right) \|X(t) - X_0\|^2 \right).$$

Now we are ready to show the desired result.

(i) $p > 1$

As we know $X(t) - X_0$ and $\tilde{\mathbb{A}}(X(t))$ are bounded from Lemma D.1 and Corollary B.9. Therefore from (35) we have

$$0 = \lim_{t \to 0+} C(t)^2 = \lim_{t \to 0+} C(t)^2 \beta(t) = \lim_{t \to 0+} C(t)^2 \left( \beta(t)^2 + \dot{\beta}(t) \right),$$

therefore $\lim_{t \to 0+} V^0(t) = 0$.

(ii) $0 < p \leq 1$

As $\left\| \frac{X(t) - X_0}{t} \right\| \leq M(T) < \infty$ for $0 < t < T$, we see

$$\limsup_{t \to 0+} C(t)^2 \beta(t) \left\langle \tilde{\mathbb{A}}(X(t)), X(t) - X_0 \right\rangle \leq \gamma \limsup_{t \to 0+} C(t)^2 t^{1-p} \left\| \tilde{\mathbb{A}}(X(t)) \right\| M(T) = 0$$

$$\limsup_{t \to 0+} C(t)^2 \left( \beta(t)^2 + \dot{\beta}(t) \right) \|X(t) - X_0\|^2 \leq \gamma \limsup_{t \to 0+} C(t)^2 \left( \gamma t^{2-2p} - t^{1-p} \right) M(T)^2 = 0.$$

Therefore

$$\lim_{t \to 0+} V^0(t) = \lim_{t \to 0+} \frac{C(t)^2}{2} \left\| \tilde{\mathbb{A}}(X(t)) \right\|^2 = \begin{cases} 0 & \text{if } p = 1 \\ \frac{\left\| \tilde{\mathbb{A}}(X_0) \right\|^2}{2} & \text{if } 0 < p < 1. \end{cases}$$

From (i) and (ii) we get the desired conclusion

$$V^0(0) = \lim_{t \to 0+} V^0(t) = \begin{cases} 0 & \text{if } p \geq 1 \\ \frac{\left\| \tilde{\mathbb{A}}(X_0) \right\|^2}{2} & \text{if } 0 < p < 1, \end{cases}$$

and therefore for general $t_0 \geq 0$,

$$V(0) = \lim_{t \to 0+} V(t) = \lim_{t \to 0+} V^0(t) - \int_0^{t_0} \frac{d}{ds} \left( \frac{C(s)^2 \dot{\beta}(s)}{2} \right) \|X(s) - X_0\|^2 \, ds$$

is well-defined.

### E.3.3  $V(t) \leq \lim_{n \to \infty} V_{\lambda_n}(0)$ **holds for** $t \in S$ **and general maximal monotone** $\mathbb{A}$

Define $S$ as defined in Lemma E.1. Take $t \in S$, let $T > t$. Let $\{\lambda_n\}_{n \in \mathbb{N}}$ be a positive sequence $\lambda_n$ that $\lim_{n \to \infty} \lambda_n = 0$, $X_{\lambda_n}$ converges to $X$ uniformly on $[0, T]$ and $\dot{X}_{\lambda_n}$ converges weakly to $\dot{X}$ in $L^2([0, T], \mathbb{R}^n)$. Recall existence of such sequence was gauranteed by Proposition B.11.

Recall we denoted $X_\lambda$ as the solution of the ODE (19). Denote $V_\lambda$ as $V$ for the case $\mathbb{A} = \mathbb{A}_\lambda$, i.e.

$$V_\lambda(t) = \frac{C(t)^2}{2} \left( \|\mathbb{A}_\lambda(X_\lambda(t))\|^2 + 2\beta(t) \langle \mathbb{A}_\lambda(X_\lambda(t)), X_\lambda(t) - X_0 \rangle + \left( \beta(t)^2 + \dot\beta(t) \right) \|X_\lambda(t) - X_0\|^2 \right)$$

$$- \int_{t_0}^t \frac{d}{ds} \left( \frac{C(s)^2 \dot\beta(s)}{2} \right) \|X_\lambda(t) - X_0\|^2 \, ds.$$

Note equality $\dot X(t) = -\tilde{\mathbb{A}}(X(t)) - \beta(t)(X(t) - X_0)$ holds since $t \in S$, therefore we have

$$V(t) = \frac{C(t)^2}{2} \left( \left\|\dot X(t)\right\|^2 + \dot\beta(t) \|X(t) - X_0\|^2 \right) - \int_{t_0}^t \frac{d}{ds} \left( \frac{C(s)^2 \dot\beta(s)}{2} \right) \|X(s) - X_0\|^2 \, ds.$$

The goal of this section is to show

$$V(t) \leq \limsup_{n\to\infty} V_{\lambda_n}(t) \leq \limsup_{n\to\infty} V_{\lambda_n}(0) = \lim_{n\to\infty} V_{\lambda_n}(0).$$

(1) $V(t) \leq \limsup_{n\to\infty} V_{\lambda_n}(t)$

First observe, from Lemma B.12 we know

$$\frac{\|X_\lambda(s) - X_0\|}{s} \leq \frac{\int_0^s \left\|\dot X_\lambda(s)\right\| ds}{s} \leq \frac{\int_0^s M_{dot}(T) ds}{s} = M_{dot}(T)$$

holds for $s \leq T$. Thus from (33) we have

$$\left| \frac{d}{ds} \left( \frac{C(s)^2 \dot\beta(s)}{2} \right) \|X_\lambda(s) - X_0\|^2 \right| \leq \left| s^2 \frac{d}{ds} \left( \frac{C(s)^2 \dot\beta(s)}{2} \right) \right| M_{dot}(T)^2 \in L^1([0,T], \mathbb{R}).$$

Therefore applying dominated convergence theorem, we have for $t_0 \in [0, T]$

$$\lim_{n\to\infty} \int_{t_0}^t \frac{d}{ds} \left( \frac{C(s)^2 \dot\beta(s)}{2} \right) \|X_{\lambda_n}(s) - X_0\|^2 \, ds = \int_{t_0}^t \frac{d}{ds} \left( \frac{C(s)^2 \dot\beta(s)}{2} \right) \|X(s) - X_0\|^2 \, ds. \tag{36}$$

From elementary analysis, we can easily check $\limsup_{n\to\infty}(a_n + b_n) = \limsup_{n\to\infty} a_n + \lim_{n\to\infty} b_n$ holds when $\lim_{n\to\infty} b_n$ exists. Thus we have

$$\limsup_{n\to\infty} V_{\lambda_n}(t) - V(t) = \frac{C(t)^2}{2} \left( \limsup_{n\to\infty} \left\|\dot X_{\lambda_n}(t)\right\|^2 - \left\|\dot X(t)\right\|^2 \right).$$

Therefore it is suffices to show following lemma.

**Lemma E.4.** *Suppose for $T > 0$ and sequence $\{\lambda_n\}_{n\in\mathbb{N}}$, $X_{\lambda_n}$ converges to $X$ uniformly on $[0, T]$ and $\dot X_{\lambda_n}$ converges weakly to $\dot X$ in $L^2([0, T], \mathbb{R}^n)$. Let $t \in S \cap [0, T]$. Then following inequality is true*

$$\left\|\dot X(t)\right\| \leq \limsup_{n\to\infty} \left\|\dot X_{\lambda_n}(t)\right\|.$$

*Proof.* (i) $\left\|\dot X(s)\right\| \leq \limsup_{n\to\infty} \left\|\dot X_{\lambda_n}(s)\right\|$ holds for almost every $s \in [0, T]$.

Let $D$ be a measurable subset $D \subset [0, T]$. Since $\dot X_{\lambda_n} \rightharpoonup \dot X$ in $L^2([0,T], \mathbb{R}^n)$ and $\chi_D \dot X \in L^2([0, T], \mathbb{R}^n)$, we have

$$\int_D \left\|\dot X(s)\right\|^2 ds = \int_0^T \left\langle \dot X(s), \chi_D(s)\dot X(s) \right\rangle ds = \lim_{n\to\infty} \int_0^T \left\langle \dot X_{\lambda_n}(s), \chi_D(s)\dot X(s) \right\rangle ds$$

$$= \limsup_{n\to\infty} \int_D \left\langle \dot X_{\lambda_n}(s), \dot X(s) \right\rangle ds \leq \limsup_{n\to\infty} \int_D \left\|\dot X_{\lambda_n}(s)\right\| \left\|\dot X(s)\right\| ds.$$

The inequality comes from the Cauchy–Schwarz inequality. Now from Reverse Fatou's Lemma we have

$$\limsup_{n\to\infty} \int_D \left\|\dot X_{\lambda_n}(s)\right\| \left\|\dot X(s)\right\| ds \leq \int_D \limsup_{n\to\infty} \left\|\dot X_{\lambda_n}(s)\right\| \left\|\dot X(s)\right\| ds.$$

Therefore combining two inequalities we have

$$\int_D \left( \limsup_{n\to\infty} \left\| \dot{X}_{\lambda_n}(s) \right\| - \left\| \dot{X}(s) \right\| \right) \left\| \dot{X}(s) \right\| ds \geq 0.$$

As $D$ was arbitrary measurable subset of $[0, T]$, we conclude for almost every $s \in [0, T]$

$$\left( \limsup_{n\to\infty} \left\| \dot{X}_{\lambda_n}(s) \right\| - \left\| \dot{X}(s) \right\| \right) \left\| \dot{X}(s) \right\| \geq 0 \quad \implies \quad \left\| \dot{X}(s) \right\| \leq \limsup_{n\to\infty} \left\| \dot{X}_{\lambda_n}(s) \right\|.$$

(ii) $\left\| \dot{X}(t) \right\| \leq \limsup_{n\to\infty} \left\| \dot{X}_{\lambda_n}(t) \right\|$ for $t \in S \cap [0, T]$.

Let $t \in S \cap [0, T]$. Then for $h > 0$ such that $t + h < T$, since $S$ is measure zero, from (i) we have

$$\int_t^{t+h} \left\| \dot{X}(s) \right\| ds \leq \int_t^{t+h} \limsup_{n\to\infty} \left\| \dot{X}_{\lambda_n}(s) \right\| ds.$$

Now, for some $a > 0$ consider

$$U_{\lambda_n}(s) = \left\| \dot{X}_{\lambda_n}(s) \right\|^2 + \underbrace{\frac{\gamma p}{s^{p+1}} \|X_{\lambda_n}(s) - X_0\|^2 + \int_a^s \frac{\gamma p(1-p)}{u^{p+2}} \|X_{\lambda_n}(u) - X_0\|^2 \, du}_{=f_n(s)}.$$

Then from the proof Lemma B.5, we know

$$\dot{U}_{\lambda_n}(s) = -2 \left\langle \dot{X}_{\lambda_n}(s), \frac{d}{ds} \mathbb{A}_{\lambda_n}(X(s)) \right\rangle - \frac{2\gamma}{s^p} \left\| \dot{X}_{\lambda_n}(s) - \frac{p}{s}(X_{\lambda_n}(s) - X_0) \right\|^2 \leq 0$$

holds for almost every $s$, thus $U_{\lambda_n}$ is nonincreasing.

By the way, as $X_{\lambda_n}$ converges to $X$ uniformly on $[0, T]$, using dominated convergence theorem we have

$$\lim_{n\to\infty} f_n(s) = \frac{\gamma p}{s^{p+1}} \|X(s) - X_0\|^2 + \int_a^s \frac{\gamma p(1-p)}{u^{p+2}} \|X(u) - X_0\|^2 \, du.$$

Denote $f(s) = \lim_{n\to\infty} f_n(s)$.

Using above facts, for $s \in [t, t+h]$ we have

$$\limsup_{n\to\infty} \left\| \dot{X}_{\lambda_n}(s) \right\| = \limsup_{n\to\infty} \sqrt{U_{\lambda_n}(s) - f_n(s)}$$

$$\leq \limsup_{n\to\infty} \sqrt{U_{\lambda_n}(t) - f_n(s)} = \sqrt{\limsup_{n\to\infty} U_{\lambda_n}(t) - f(s)}. \tag{37}$$

Therefore,

$$\|X(t+h) - X(t)\| \leq \int_t^{t+h} \left\| \dot{X}(s) \right\| ds \leq \int_t^{t+h} \limsup_{n\to\infty} \left\| \dot{X}_{\lambda_n}(s) \right\| ds \leq \int_t^{t+h} \sqrt{\limsup_{n\to\infty} U_{\lambda_n}(t) - f(s)} ds.$$

Note $\sqrt{\limsup_{n\to\infty} U_{\lambda_n}(t) - f(s)}$ is a continuous function with respect to $s$. Thus we have

$$\lim_{h\to 0} \frac{1}{h} \int_t^{t+h} \sqrt{\limsup_{n\to\infty} U_{\lambda_n}(t) - f(s)} ds = \sqrt{\limsup_{n\to\infty} U_{\lambda_n}(t) - f(t)}$$

$$= \sqrt{\limsup_{n\to\infty} \left\| \dot{X}_{\lambda_n}(t) \right\|^2 + \lim_{n\to\infty} f_n(t) - f(t)} = \limsup_{n\to\infty} \left\| \dot{X}_{\lambda_n}(t) \right\|.$$

Finally as $t \in S$, $\dot{X}(t) = \lim_{h\to 0} \frac{X(t+h) - X(t)}{h}$ exists. Therefore

$$\left\| \dot{X}(t) \right\| = \lim_{h\to 0+} \frac{\|X(t+h) - X(t)\|}{h}$$

$$\leq \lim_{h\to 0+} \frac{1}{h} \int_t^{t+h} \sqrt{\limsup_{n\to\infty} U_{\lambda_n}(t) - f(s)} ds = \limsup_{n\to\infty} \left\| \dot{X}_{\lambda_n}(t) \right\|,$$

we conclude the desired result.

$\square$

(2) $\limsup_{n\to\infty} V_{\lambda_n}(t) \le \limsup_{n\to\infty} V_{\lambda_n}(0)$

As $\mathbb{A}_{\lambda_n}$ is Lipschitz continuous, from Appendix E.3.1 we know $V_{\lambda_n}$ is nonincreasing, and thus

$$V_{\lambda_n}(t) \le \lim_{\epsilon \to 0+} V_{\lambda_n}(\epsilon) = V_{\lambda_n}(0).$$

Taking limsup both sides we get the desired result.

(3) $\limsup_{n\to\infty} V_{\lambda_n}(0) = \lim_{n\to\infty} V_{\lambda_n}(0)$

From (32) and (ii) of Lemma B.10, we know

$$\lim_{n\to\infty} V^0_{\lambda_n}(0) = \begin{cases} 0 & \text{if } p \ge 1 \\ \frac{\|m(\mathbb{A}(X_0))\|^2}{2} & \text{if } 0 < p < 1. \end{cases}$$

And applying (36) we have

$$\lim_{n\to\infty} V_{\lambda_n}(0) = \lim_{n\to\infty} V^0_{\lambda_n}(0) - \int_0^{t_0} \frac{d}{ds}\left(\frac{C(s)^2 \dot\beta(s)}{2}\right) \|X(s) - X_0\|^2 \, ds.$$

As the limit $\lim_{n\to\infty} V_{\lambda_n}(0)$ exists, the limsup concides with the limit.

### E.3.4 Proof for general maximal monotone $\mathbb{A}$

First, we show $V(t) \le \lim_{n\to\infty} V_{\lambda_n}(0)$ holds for $t \in [0, \infty)$. Next, we show $\lim_{n\to\infty} V_{\lambda_n}(0) = \lim_{t\to 0+} V(t)$. Then we have

$$V(t) \le \lim_{n\to\infty} V_{\lambda_n}(0) = \lim_{t\to 0+} V(t) = V(0),$$

which is our desired result.

(i) $V(t) \le \lim_{n\to\infty} V_{\lambda_n}(0)$ holds for $t \in [0, \infty)$

Take $t \in [0, \infty)$. We know the inequality is true when $t \in S$, thus assume $t \notin S$. Then from Lemma E.1, we know there is a sequence $\{t_k\}_{k\in\mathbb{N}}$ such that $t_k \in S$, $\lim_{k\to\infty} t_k = t$ and $\tilde{\mathbb{A}}(X(t_k))$ converges to $\tilde{\mathbb{A}}(X(t))$. As $t_k \in S$, $V(t_k) \le \lim_{n\to\infty} V_{\lambda_n}(0)$ holds. Since $\lim_{k\to\infty} V(t_k) = V(t)$, taking limit $k \to \infty$ to the inequality we get the desired result.

(ii) $\lim_{n\to\infty} V_{\lambda_n}(0) = \lim_{t\to 0+} V(t)$

Take a sequence $\{t_k\}_{k\in\mathbb{N}}$ such that $t_k > 0$ and $\lim_{k\to\infty} t_k = 0$. We wish to show $\lim_{k\to\infty} V(t_k) = \lim_{n\to\infty} V_{\lambda_n}(0)$. Note the arguments in Appendix E.3.2 are valid when $\left\|\tilde{\mathbb{A}}(X(t_k))\right\|$ and $\frac{\|X(t_k)-X_0\|}{t_k}$ are bounded for all $k \in \mathbb{N}$. The boundedness of $\left\|\tilde{\mathbb{A}}(X(t_k))\right\|$ comes from Corollary E.2. And from Lemma B.12 we have

$$\frac{\|X(t_k) - X_0\|}{t_k} = \lim_{\lambda \to 0+} \frac{\|X_\lambda(t_k) - X_0\|}{t_k} \le \lim_{\lambda \to 0+} \frac{\int_0^{t_k} \left\|\dot{X}_\lambda(s)\right\| ds}{t_k} \le M_{dot}(t_k) \le \sup_{k\in\mathbb{N}} M_{dot}(t_k).$$

Therefore applying the arguments in Appendix E.3.2 we have

$$\lim_{k\to\infty} V^0(t_k) = \lim_{k\to\infty} \frac{C(t_k)^2}{2}\left\|\tilde{\mathbb{A}}(X(t_k))\right\|^2.$$

Thus it remains to show the limit on the right hand side exists and is equal to $\lim_{n\to\infty} V^0_{\lambda_n}(0)$.

For $p \ge 1$, we know $\lim_{k\to\infty} C(t_k)^2 = 0$, thus $\lim_{k\to\infty} \frac{C(t_k)^2}{2}\left\|\tilde{\mathbb{A}}(X(t_k))\right\|^2 = 0$ since $\left\|\tilde{\mathbb{A}}(X(t_k))\right\|$ is bounded. As $\lim_{n\to\infty} V^0_{\lambda_n}(0) = 0$ from (32), we're done.

Now consider the case $0 < p < 1$. Suppose $\tilde{\mathbb{A}}(X(t_{k_l}))$ is a convergent subsequence of $\tilde{\mathbb{A}}(X(t_k))$. First observe from (i) we have

$$V^0(t_{k_l}) \le \lim_{n\to\infty} V^0_{\lambda_n}(0) = \frac{\|m(\mathbb{A}(X_0))\|^2}{2}.$$

From above inequality, recalling $\lim_{l\to\infty} C(t_{k_l})^2 = 1$ we have

$$\frac{\left\|\lim_{l\to\infty} \tilde{\mathbb{A}}(X(t_{k_l}))\right\|^2}{2} = \lim_{l\to\infty} V^0(t_{k_l}) \le \frac{\|m(\mathbb{A}(X_0))\|^2}{2}.$$

By the way as $\lim_{l\to\infty} X(t_{k_l}) = X_0$ and $\tilde{\mathbb{A}}(X(t_{k_l})) \in \mathbb{A}(X(t_{k_l}))$, by closed graph theorem we have $\lim_{l\to\infty} \tilde{\mathbb{A}}(X(t_{k_l})) \in \mathbb{A}(X_0)$. As of $m(\mathbb{A}(X_0))$ is the element in $\mathbb{A}(X_0)$ with smallest norm, we have $\lim_{l\to\infty} \tilde{\mathbb{A}}(X(t_{k_l})) = m(\mathbb{A}(X_0))$. As all convergent subsequence converges to the same limit $m(\mathbb{A}(X_0))$, we conclude $\lim_{k\to\infty} \tilde{\mathbb{A}}(X(t_k)) = m(\mathbb{A}(X_0))$.

Therefore

$$\lim_{k\to\infty} V^0(t_k) = \lim_{k\to\infty} \frac{C(t_k)^2}{2} \left\| \tilde{\mathbb{A}}(X(t_k)) \right\|^2 = \frac{\| m(\mathbb{A}(X_0)) \|^2}{2} = \lim_{n\to\infty} V_{\lambda_n}^0(0).$$

As $t_k$ was arbitrary positive sequence converges to 0, we conclude $\lim_{n\to\infty} V_{\lambda_n}^0(0) = \lim_{t\to 0+} V^0(t) = V^0(0)$. Therefore

$$\lim_{n\to\infty} V_{\lambda_n}(0) = V^0(0) - \int_0^{t_0} \frac{d}{ds} \left( \frac{C(s)^2 \dot{\beta}(s)}{2} \right) \|X(s) - X_0\|^2 \, ds = \lim_{t\to 0+} V(t).$$

### E.4  Proof of Lemma 3.4

Most of the proof is done in the main text. Recall $\Phi(t)$ is defined as $\Phi(t) = \left\| \tilde{\mathbb{A}}(X(t)) \right\|^2 + 2\beta(t) \left\langle \tilde{\mathbb{A}}(X(t)), X(t) - X_0 \right\rangle$, and from (8) we have

$$\Phi(t) \geq \frac{1}{2} \left\| \tilde{\mathbb{A}}(X(t)) \right\|^2 - 2\beta(t)^2 \|X_0 - X_\star\|^2.$$

On the other hand,

$$\frac{2V(t)}{C(t)^2} - \Phi(t) = \left( \beta(t)^2 + \dot{\beta}(t) \right) \|X(t) - X_0\|^2 - \frac{2}{C(t)^2} \int_{t_0}^t \frac{d}{ds} \left( \frac{C(s)^2 \dot{\beta}(s)}{2} \right) \|X(s) - X_0\|^2 \, ds.$$

Note $C(t) \neq 0$ if $t > 0$, we can divide with $C(t)^2$. Now from Corollary 3.3 we have we have $V(t) \geq V(0)$ for $t > 0$, and so $\frac{V(t)}{C(t)^2} - \frac{V(0)}{C(t)^2} \geq 0$. Thus for $t > \epsilon$

$$\frac{2V(0)}{C(t)^2} - \left( \beta(t)^2 + \dot{\beta}(t) \right) \|X(t) - X_0\|^2 + \frac{2}{C(t)^2} \int_{t_0}^t \frac{d}{ds} \left( \frac{C(s)^2 \dot{\beta}(s)}{2} \right) \|X(s) - X_0\|^2 \, ds$$

$$= \frac{2V(0)}{C(t)^2} - \left( \frac{2V(t)}{C(t)^2} - \Phi(t) \right)$$

$$= \left( \frac{2V(0)}{C(t)^2} - \frac{2V(t)}{C(t)^2} \right) + \Phi(t) \geq \Phi(t) \geq \frac{1}{2} \left\| \tilde{\mathbb{A}}(X(t)) \right\|^2 - 2\beta(t)^2 \|X_0 - X_\star\|^2.$$

Moving $2\beta(t)^2 \|X_0 - X_\star\|^2$ to the left hand side and multiplying with 2 we get the desired result

$$\left\| \tilde{\mathbb{A}}(X(t)) \right\|^2 \leq 4\beta(t)^2 \|X_0 - X_\star\|^2 + \frac{4V(0)}{C(t)^2} - 2 \left( \beta(t)^2 + \dot{\beta}(t) \right) \|X(t) - X_0\|^2$$

$$+ \frac{2}{C(t)^2} \int_{t_0}^t \frac{d}{ds} \left( C(s)^2 \dot{\beta}(s) \right) \|X(s) - X_0\|^2 \, ds.$$

### E.5  Proof of Theorem 3.1

Restating Lemma 3.4, we know for $t > 0$,

$$\left\| \tilde{\mathbb{A}}(X(t)) \right\|^2 \leq 4\beta(t)^2 \|X_0 - X_\star\|^2 + \frac{2V(0)}{C(t)^2} - 2 \left( \beta(t)^2 + \dot{\beta}(t) \right) \|X(t) - X_0\|^2 \qquad (7)$$

$$+ \frac{2}{C(t)^2} \int_{t_0}^t \frac{d}{ds} \left( C(s)^2 \dot{\beta}(s) \right) \|X(s) - X_0\|^2 \, ds.$$

As we know $\|X(t) - X_0\| \le 2\|X_0 - X_\star\|$ from Lemma D.1, it is clear that

$$4\beta(t)^2 \|X_0 - X_\star\|^2 = \mathcal{O}\left(\beta(t)^2\right)$$

$$\frac{2V(0)}{C(t)^2} = \mathcal{O}\left(\frac{1}{C(t)^2}\right) \tag{38}$$

$$2\left(\beta(t)^2 + \dot\beta(t)\right)\|X(t) - X_0\|^2 = \mathcal{O}\left(\beta(t)^2\right) + \mathcal{O}\left(\dot\beta(t)\right).$$

Therefore it remains to show

$$\frac{2}{C(t)^2}\int_{t_0}^t \frac{d}{ds}\left(C(s)^2\dot\beta(s)\right)\|X(s) - X_0\|^2\, ds = \mathcal{O}\left(\beta(t)^2\right) + \mathcal{O}\left(\frac{1}{C(t)^2}\right) + \mathcal{O}\left(\dot\beta(t)\right).$$

We can check there is $T > 0$ such that for $s > T$ the sign of $\frac{d}{ds}\left(C(s)^2\dot\beta(s)\right)$ does not change, for $\beta(t) = \frac{\gamma}{t^p}$ with $\gamma > 0$, $p > 0$. We first proceed our proof with assuming this condition. The point for this condition is that following equality holds for $t > T$.

$$\left|\int_T^t \frac{d}{ds}\left(C(s)^2\dot\beta(s)\right)ds\right| = \int_T^t \left|\frac{d}{ds}\left(C(s)^2\dot\beta(s)\right)\right|ds.$$

Applying above equality and using $\|X(t) - X_0\| \le 2\|X_0 - X_\star\|$, we see

$$\left|\frac{2}{C(t)^2}\int_{t_0}^t \frac{d}{ds}\left(C(s)^2\dot\beta(s)\right)\|X(s) - X_0\|^2\, ds\right|$$

$$\le \frac{2}{C(t)^2}\underbrace{\left|\int_{t_0}^T \frac{d}{ds}\left(C(s)^2\dot\beta(s)\right)\|X(s) - X_0\|^2\, ds\right|}_{=M} + \frac{2}{C(t)^2}\int_T^t \left|\frac{d}{ds}\left(C(s)^2\dot\beta(s)\right)\|X(s) - X_0\|^2\right|ds$$

$$\le \frac{2M}{C(t)^2} + \frac{2}{C(t)^2}\int_T^t \left|\frac{d}{ds}\left(C(s)^2\dot\beta(s)\right)4\|X_0 - X_\star\|^2\right|ds$$

$$= \frac{2M}{C(t)^2} + \frac{8\|X_0 - X_\star\|^2}{C(t)^2}\left|C(t)^2\dot\beta(t) - C(T)^2\dot\beta(T)\right|$$

$$\le \frac{2}{C(t)^2}\left(M + 4\|X_0 - X_\star\|^2 C(T)^2\left|\dot\beta(T)\right|\right) + 8\|X_0 - X_\star\|^2\left|\dot\beta(t)\right| = \mathcal{O}\left(\frac{1}{C(t)^2}\right) + \mathcal{O}\left(\dot\beta(t)\right).$$

Therefore from (7) and (38), we conclude

$$\left\|\tilde{\mathbf{A}}(X(t))\right\|^2 = \mathcal{O}\left(\beta(t)^2\right) + \mathcal{O}\left(\frac{1}{C(t)^2}\right) + \mathcal{O}\left(\dot\beta(t)\right).$$

It remains to show there is $T > 0$ such that for $s > T$ the sign of $\frac{d}{ds}\left(C(s)^2\dot\beta(s)\right)$ does not change. It can be shown by easy, but a little complicated calculations. Since $\dot C(t) = C(t)\beta(t)$, we see

$$\frac{d}{ds}\left(C(s)^2\dot\beta(s)\right) = 2C(s)^2\beta(s)\dot\beta(s) + C(s)^2\ddot\beta(s) = C(s)^2\left(2\beta(s)\dot\beta(s) + \ddot\beta(s)\right).$$

Therefore it is enough to check the sign of $2\beta(s)\dot\beta(s) + \ddot\beta(s)$. Observe

$$2\beta(s)\dot\beta(s) + \ddot\beta(s) = -\frac{2p\gamma^2}{s^{2p+1}} + \frac{p(1+p)\gamma}{s^{p+2}} = \begin{cases} \frac{2}{s^3}\gamma(1-\gamma) & \text{if } p = 1 \\ \frac{p(p+1)\gamma}{s^{2p+1}}\left(s^{p-1} - \frac{2\gamma}{p+1}\right) & \text{if } p \ne 1. \end{cases}$$

Thus when $p = 1$, for all $s > 0$ it is nonpositive if $\gamma \ge 1$ and nonnegative if $0 < \gamma < 1$.

When $p \ne 1$, we see

$$\lim_{s \to 0+} s^{p-1} = \begin{cases} \infty & \text{if } 0 < p < 1 \\ 0 & \text{if } p > 1 \end{cases}, \qquad \lim_{s \to \infty} s^{p-1} = \begin{cases} 0 & \text{if } 0 < p < 1 \\ \infty & \text{if } p > 1 \end{cases}$$

Therefore by intermediate value theorem there is $T > 0$ such that $T^{p-1} - \frac{2\gamma}{p+1} = 0$, and for that $T$ we have $s^{p-1} - \frac{2\gamma}{p+1}$ is nonpositive for $s > T$ if $0 < p < 1$ and nonnegative for $s > T$ if $p > 1$. This concludes the proof.

### E.5.1 Proof for the convergence rate in Table 1

Recall, from (31) we have

$$C(t) = \begin{cases} t^\gamma & p = 1 \\ e^{\frac{\gamma}{1-p} t^{1-p}} & p > 0, \ p \neq 1. \end{cases}$$

We now observe $\dot\beta(t)$ does not effect to convergence rate. In other words, we show $\dot\beta(t)$ is not the slowest one that goes to zero compared to $\beta(t)^2$ and $\frac{1}{C(t)^2}$ for every case.

(i) $0 < p \leq 1$
  Comparing $\dot\beta(t) = -\frac{p\gamma}{t^{p+1}}$ with $\beta(t)^2 = \frac{\gamma^2}{t^{2p}}$, we see $\dot\beta(t) = \mathcal{O}\left(\beta(t)^2\right)$ when $0 < p \leq 1$.

(ii) $p > 1$
  For $p > 1$, we see $\lim_{t\to\infty} \frac{1}{C(t)^2} = 1 \neq 0$. As $\lim_{t\to\infty} \dot\beta(t) = -\lim_{t\to\infty} \frac{p\gamma}{t^{p+1}} = 0$, we have $\dot\beta(t) = \mathcal{O}\left(\frac{1}{C(t)^2}\right)$.

From (i) and (ii), we conclude

$$\left\|\tilde{\mathbf{A}}(X(t))\right\|^2 = \mathcal{O}\left(\beta(t)^2\right) + \mathcal{O}\left(\frac{1}{C(t)^2}\right).$$

Now the results in Table 1 is straightfoward.

- $p = 1$
  When $p = 1$, we have $C(t) = t^\gamma$. Comparing $\frac{1}{C(t)^2} = \frac{1}{t^{2\gamma}}$ and $\beta(t)^2 = \frac{\gamma^2}{t^2}$,

  (1) $\left\|\tilde{\mathbf{A}}(X(t))\right\|^2 = \mathcal{O}\left(\beta(t)^2\right) = \mathcal{O}\left(\frac{1}{t^2}\right)$ if $\gamma \geq 1$.
  (2) $\left\|\tilde{\mathbf{A}}(X(t))\right\|^2 = \mathcal{O}\left(\frac{1}{C(t)^2}\right) = \mathcal{O}\left(\frac{1}{t^{2\gamma}}\right)$ if $0 < \gamma < 1$.

- $0 < p < 1$
  When $0 < p < 1$, we have

$$\lim_{t\to\infty} \frac{1}{\beta(t)^2} \cdot \frac{1}{C(t)^2} = \frac{1}{\gamma^2} \lim_{t\to\infty} \frac{t^2}{e^{\frac{2\gamma}{1-p} t^{1-p}}} = 0.$$

  Therefore, $\left\|\tilde{\mathbf{A}}(X(t))\right\|^2 = \mathcal{O}\left(\beta(t)^2\right) = \mathcal{O}\left(\frac{1}{t^{2p}}\right)$.

- $p > 1$
  We observed previously that $\lim_{t\to\infty} \frac{1}{C(t)^2} = 1 \neq 0$ when $p > 1$. Therefore $\left\|\tilde{\mathbf{A}}(X(t))\right\|^2 = \mathcal{O}\left(\frac{1}{C(t)^2}\right) = \mathcal{O}(1)$.

### E.6 Proof of Theorem 3.5

Name

$$\bar{X}_\star = \Pi_{\mathrm{Zer}\,\mathbf{A}}(X_0) = \operatorname*{argmin}_{z \in \mathrm{Zer}\mathbf{A}} \|z - X_0\|.$$

We first show

$$\limsup_{t\to\infty} \left\langle X(t) - \bar{X}_\star, X_0 - \bar{X}_\star \right\rangle \leq 0.$$

Proof by contradiction. Suppose not.
Then there is $\epsilon > 0$ such that for every $k \in \mathbb{N}$, there is $n_k \in [0, \infty)$ such that

$$\left\langle X(n_k) - \bar{X}_\star, X_0 - \bar{X}_\star \right\rangle > \epsilon$$

By the way, from Lemma D.1 we know $X(n_k) \in \bar{B}_{\|X_0 - \bar{X}_\star\|}(\bar{X}_\star)$. Thus by Bolzano-Weierstrass theorem, there is a converging subseqeunce $\{X\left(n_{k(l)}\right)\}_{l\in(N)}$. Name the limit as $X_\infty$. Since $\lim_{t\to\infty} \left\|\tilde{\mathbf{A}}(X(t))\right\| = 0$, $\{\tilde{\mathbf{A}}(X(n_{k(l)})\}_{l\in(N)}$ converges to 0. Then from [15, Proposition 20.38], $(X_\infty, 0) \in \mathrm{Gra}\,\mathbf{A}$, i.e. $X_\infty \in \mathrm{Zer}\mathbf{A}$.

On the other hand, as $\mathbb{A}$ is maximal monotone, by [15, Proposition 23.39] $\text{Zer}\,\mathbb{A}$ is closed and convex. Since $\bar{X}_\star = \Pi_{\text{Zer}\,\mathbb{A}}(X_0)$, by [15, Theorem 3.16] we have $\langle X_\infty - \bar{X}_\star, X_0 - \bar{X}_\star \rangle \leq 0$. Thus

$$0 < \epsilon \leq \lim_{l \to \infty} \langle X\left(n_{k(l)}\right) - \bar{X}_\star, X_0 - \bar{X}_\star \rangle = \langle X_\infty - \bar{X}_\star, X_0 - \bar{X}_\star \rangle \leq 0,$$

this is a contradiction, therefore we get the desired result.

Now we show $\lim_{t \to \infty} \left\| X(t) - \bar{X}_\star \right\| = 0$. Recall, for $t > 0$

$$\dot{X} \in -\mathbb{A}(X(t)) - \beta(t)(X - X_0).$$

From $\dot{C}(t) = C(t)\beta(t)$, and monotonicity of $\mathbb{A}$, we observe

$$\begin{aligned}
\frac{d}{dt}\left(C(t)^2 \left\| X(t) - \bar{X}_\star \right\|^2\right) &= 2C(t)^2 \left\langle \dot{X}(t), X(t) - \bar{X}_\star \right\rangle + 2C(t)^2 \beta(t) \left\| X(t) - \bar{X}_\star \right\|^2 \\
&= 2C(t)^2 \left\langle -\tilde{\mathbb{A}}(X(t)) - \beta(t)(X(t) - X_0), X(t) - \bar{X}_\star \right\rangle + 2C(t)^2 \beta(t) \left\| X(t) - \bar{X}_\star \right\|^2 \\
&\leq -2C(t)^2 \beta(t) \left\langle X(t) - X_0, X(t) - \bar{X}_\star \right\rangle + 2C(t)^2 \beta(t) \left\| X(t) - \bar{X}_\star \right\|^2 \\
&= 2C(t)^2 \beta(t) \left\langle X_0 - \bar{X}_\star, X(t) - \bar{X}_\star \right\rangle.
\end{aligned}$$

Now take $\epsilon$. From $\limsup_{t \to \infty} \left\langle X(t) - \bar{X}_\star, X_0 - \bar{X}_\star \right\rangle \leq 0$, there is $M > 0$ such that

$$t > M \quad \Longrightarrow \quad \left\langle X_0 - \bar{X}_\star, X(t) - \bar{X}_\star \right\rangle < \epsilon.$$

For that $M$ and $t > M$, integrating from $M$ to $t$ we get

$$\begin{aligned}
\left[C(s)^2 \left\| X(s) - \bar{X}_\star \right\|^2\right]_M^t &= C(t)^2 \left\| X(t) - \bar{X}_\star \right\|^2 - C(M)^2 \left\| X(M) - \bar{X}_\star \right\|^2 \\
&\leq \int_M^t 2C(s)^2 \beta(s) \left\langle X_0 - \bar{X}_\star, X(s) - \bar{X}_\star \right\rangle ds \\
&\leq \int_M^t \left| 2C(s)^2 \beta(s) \left\langle X_0 - \bar{X}_\star, X(s) - \bar{X}_\star \right\rangle \right| ds \\
&\leq \int_M^t 2C(s)^2 \beta(s)\epsilon \, ds = \epsilon \left[C(s)^2\right]_M^t = \epsilon \left(C(t)^2 - C(M)^2\right).
\end{aligned}$$

By dividing $C(t)^2$ and organizing,

$$\left\| X(t) - \bar{X}_\star \right\|^2 \leq \epsilon \left(1 - \left(\frac{C(M)}{C(t)}\right)^2\right) + \left(\frac{C(M)}{C(t)}\right)^2 \left\| X(M) - \bar{X}_\star \right\|^2.$$

By the way from assumption, we have $\lim_{t \to \infty} \frac{1}{C(t)} = 0$. Therefore we conclude

$$\lim_{t \to \infty} \left\| X(t) - \bar{X}_\star \right\|^2 \leq \epsilon.$$

Since $\epsilon > 0$ was arbitrary, we get the desired result.

## F    Proof for worst case examples

The explicit solution for linear $A$ is crucially used in worst case examples. Therefore, we first provide proof for it.

### F.1    Proof for explicit solution for linear $A$

#### F.1.1    Proof of Lemma 4.1

Observing

$$X(t) = \sum_{n=0}^\infty \frac{(-tA)^n}{\Gamma(n + \gamma + 1)}\Gamma(\gamma + 1)X_0 = X_0 + \sum_{n=1}^\infty \frac{(-tA)^n}{\Gamma(n + \gamma + 1)}\Gamma(\gamma + 1)X_0,$$

we can check $X(0) = X_0$. Now by using the property of Gamma function $\Gamma(x+1) = x\Gamma(x)$, and paying attention to the lower bound of the summation index we have

$$\dot{X}(t) = \sum_{n=1}^{\infty} \frac{(-nA)(-tA)^{n-1}}{\Gamma(n+\gamma+1)} \Gamma(\gamma+1)X_0 = A \sum_{n=1}^{\infty} \frac{(-n-\gamma+\gamma)(-tA)^{n-1}}{\Gamma(n+\gamma+1)} \Gamma(\gamma+1)X_0$$

$$= -A \sum_{n=1}^{\infty} \frac{(-tA)^{n-1}}{\Gamma(n+\gamma)} \Gamma(\gamma+1)X_0 + \gamma \sum_{n=1}^{\infty} \frac{(-t)^{n-1}A^n}{\Gamma(n+\gamma+1)} \Gamma(\gamma+1)X_0$$

$$= -A \sum_{n=0}^{\infty} \frac{(-tA)^{n}}{\Gamma(n+\gamma+1)} \Gamma(\gamma+1)X_0 + \frac{\gamma}{(-t)} \sum_{n=1}^{\infty} \frac{(-t)^{n}A^n}{\Gamma(n+\gamma+1)} \Gamma(\gamma+1)X_0 = -AX(t) - \frac{\gamma}{t}\left(X(t) - X_0\right).$$

The solution can be written in another form

$$X(t) = \gamma e^{-tA} t^{-\gamma} \int_0^t u^{\gamma-1} e^{uA} \, du \, X_0. \tag{39}$$

As this is a special case of (9), here we just briefly check this satisfies the ODE and check other details in the proof of Lemma 4.2. By product rule of differentiation,

$$\dot{X} = \gamma \left(\frac{d}{dt} e^{-tA}\right) t^{-\gamma} \int_0^t u^{\gamma-1} e^{uA} \, du \, X_0 + \gamma e^{-tA} \left(\frac{d}{dt} t^{-\gamma}\right) \int_0^t u^{\gamma-1} e^{uA} \, du \, X_0 + \gamma e^{-tA} t^{-\gamma} \frac{d}{dt}\left(\int_0^t u^{\gamma-1} e^{uA} \, du \, X_0\right)$$

$$= -A(X(t)) - \frac{\gamma}{t} X(t) + \gamma e^{-tA} t^{-\gamma} \left(t^{\gamma-1} e^{tA}\right) X_0$$

$$= -A(X(t)) - \frac{\gamma}{t}\left(X(t) - X_0\right).$$

### F.1.2  Proof of Lemma 4.2

Define

$$X_*(t) = \left(I - \frac{Ae^{-tA}}{C(t)}\left(\int_0^t e^{sA} C(s)ds\right)\right) X_0.$$

We show $X_*$ is a well-defined solution for (3) with linear $\mathbb{A}$, then show $X_*$ is equal to $X(t)$ defined in (9).

We first check well-definedness. By definition, $C(t) = e^{\int_{t_0}^t \beta(s)ds}$ is nondecreasing. Also, $\|e^{tA}\|$ is also nondecreasing since from monotonicity of $A$ we have for all $x \in \mathbb{R}^n$

$$\frac{d}{dt} \|e^{tA}x\|^2 = 2\left\langle A(e^{tA}x), e^{tA}x\right\rangle \geq 0.$$

Now we see $X(t)$ is well-defined since

$$\left\| e^{-tA} \frac{1}{C(t)}\left(\int_0^t e^{sA} C(s)ds\right)\right\| = \left\|\int_0^t e^{(s-t)A} \frac{C(s)}{C(t)} ds\right\|$$

$$\leq \int_0^t \left\|e^{(s-t)A}\right\| \left|\frac{C(s)}{C(t)}\right| ds \leq \int_0^t (1 \cdot 1)ds = t < \infty.$$

Also above inequality implies second term reaches to zero as $t \to 0+$, we have $\lim_{t\to 0+} X_*(t) = X_0$. Thus defining $X_*(0) = X_0$ if necessary, we see $X_*$ satisfies the initial condition with $X_* \in \mathcal{C}([0,\infty), \mathbb{R}^n)$.

We then check $X_*(t)$ becomes the solution. This is immediate from product rule for differentiation. Since $\frac{d}{dt}\left(\frac{1}{C(t)}\right) = -\frac{1}{C(t)^2} C(t)\beta(t) = -\frac{1}{C(t)}\beta(t)$, we have

$$\dot{X}_*(t) = -\left(\frac{d}{dt} e^{-tA}\right) \frac{1}{C(t)} \left(\int_0^t e^{sA} C(s)ds\right) X_0$$

$$- Ae^{-tA} \left(\frac{d}{dt} \frac{1}{C(t)}\right) \left(\int_0^t e^{sA} C(s)ds\right) X_0$$

$$- Ae^{-tA} \frac{1}{C(t)} \frac{d}{dt}\left(\int_0^t e^{sA} C(s)ds\right) X_0$$

$$= -A(X_*(t) - X_0) - \beta(t)(X_*(t) - X_0) - A(X_0) = -A(X(t)) - \beta(t)(X_*(t) - X_0).$$

Finally we now show $X(t) = X_*(t)$, where $X(t)$ is defined as (9). From integral by parts we have

$$
X(t) = e^{-tA} \frac{1}{C(t)} \left( \int_0^t e^{sA} C(s) \beta(s) ds + C(0)I \right) X_0
$$

$$
= e^{-tA} \frac{1}{C(t)} \left( \left[ e^{sA} C(s) \right]_0^t - \int_0^t A e^{sA} C(s) ds + C(0)I \right) X_0
$$

$$
= \left( I - A e^{-tA} \frac{1}{C(t)} \left( \int_0^t e^{sA} C(s) ds \right) \right) X_0 = X_*(t).
$$

## F.2   Proof of Theorem 4.3

If $\beta(t) \equiv 0$, for $A := \begin{pmatrix} 0 & 1 \\ -1 & 0 \end{pmatrix}$, we have $X(t) = e^{-tA}$, so

$$
\lim_{t \to \infty} \|A(X(t))\| = 1 \neq 0.
$$

So let's consider the case $\beta$ is not $\beta(t) \equiv 0$ and $\beta(t) \geq 0$. For

$$
A_\xi = 2\pi\xi \begin{pmatrix} 0 & 1 \\ -1 & 0 \end{pmatrix}
$$

name the solution for ODE $\dot{X}_\xi = -A_\xi(X_\xi) - \beta(t)(X_\xi - X_0)$ as $X_\xi$. By Lemma 4.2 we have

$$
X_\xi(t) = \frac{e^{-tA_\xi}}{C(t)} \left( \int_0^t e^{sA_\xi} C(s) \beta(s) ds + C(0)I \right) X_0.
$$

We want to show,

$$
\exists \xi \in \mathbb{R}, \qquad \lim_{t \to \infty} \|A_\xi(X_\xi(t))\| \neq 0
$$

From $\lim_{t \to \infty} \frac{1}{C(t)} \neq 0$, we see $\lim_{t \to \infty} \left\| \frac{e^{-tA_\xi}}{C(t)} \right\| = 1 \cdot \frac{1}{C(t)} \neq 0$. And since $A_\xi$ is invertible except the case $\xi = 0$, we have

$$
\lim_{t \to \infty} \|A_\xi(X_\xi(t))\| = 0 \iff \lim_{t \to \infty} \|X_\xi(t)\| = 0
$$

$$
\iff \left\| \lim_{t \to \infty} \int_0^t e^{sA_\xi} C(s) \beta(s) ds + C(0)I \right\| = 0.
$$

Now let assume $\forall \xi \in \mathbb{R}$, $\lim_{t \to \infty} \|A_\xi(X_\xi(t))\| = 0$ and lead to contradiction. Define $f : \mathbb{R} \to \mathbb{R}$ as

$$
f(s) = \begin{cases} C(s)\beta(s) & s > 0 \\ 0 & s \leq 0. \end{cases}
$$

Then $\int_{-\infty}^\infty f(s) ds = [C(s)]_0^\infty = \lim_{t \to \infty} C(t) - C(0) < \infty$ we have $f \in L^1(\mathbb{R}, \mathbb{R})$. Note $\lim_{t \to \infty} C(t)$ exists, since $C$ is nondecreasing and by the assumption is bounded.

Now setting $X_0 = (1,0)^T$ and from $e^{sA_\xi} = \begin{pmatrix} \cos 2\pi\xi s & \sin 2\pi\xi s \\ -\sin 2\pi\xi s & \cos 2\pi\xi s \end{pmatrix}$, we see

$$
\int_0^\infty e^{sA_\xi} C(s) \beta(s) ds \, X_0 = \left( \int_0^\infty \cos(2\pi s\xi) f(s) ds \, , \, \int_0^\infty \sin(-2\pi s\xi) f(s) ds \right)^T
$$

$$
= \left( Re(\hat{f}(\xi)) \, , \, Im(\hat{f}(\xi)) \right)^T
$$

where $\hat{f}(\xi)$ is Fourier transform of $f$.

From assumption we have $\forall \xi \in \mathbb{R}$, $\left\| \int_0^\infty e^{sA_\xi} C(s) \beta(s) ds X_0 + C(0)X_0 \right\| = 0$, we have

$$
\left( Re(\hat{f}(\xi)) \, , \, Im(\hat{f}(\xi)) \right)^T \equiv -C(0)X_0
$$

By the way, from Fourier theory we know $\hat{f}$ vanishes at infinity , thus

$$\|C(0)X_0\| = \lim_{\xi \to \infty} \left|\hat{f}(\xi)\right| = 0.$$

Therefore we have $\hat{f}(\xi) = 0$ for all $\xi \in \mathbb{R}$.

Now $\hat{f} \equiv 0$ is clearly $L^1(\mathbb{R}, \mathbb{R})$, we can apply *Fourier inversion formula* and conclude $f \equiv 0$ almost everywhere.

However, $f(s) = C(s)\beta(s)$ is not zero almost everywhere since $\beta(s)$ is not constantly zero. Thus a contradiction, we get the desired result.

### F.3   Proof of Theorem 4.4

#### F.3.1   Proof for the case $p = 1, \gamma \geq 1$

Recall from (39), we know

$$X(t) = \gamma e^{-tA} t^{-\gamma} \int_0^t e^{sA} s^{\gamma-1} ds \, X_0.$$

Without loss of generality, we consider the case $X_0 = (1,0)^T$.

(i) $\gamma = 1$
Plugging $\gamma = 1$ gives

$$X(t) = \frac{e^{-tA}}{t} A^{-1} \left[e^{sA}\right]_0^t X_0 = \frac{1}{t} A^{-1} \left(I - e^{-tA}\right)(1,0)^T = \frac{1}{t} A^{-1} (1 - \cos t, \sin t)^T$$

Therefore

$$\lim_{t \to \infty} \|tA(X(t))\| = \lim_{t \to \infty} \left\|(1 - \cos t, \sin t)^T\right\| \neq 0.$$

(ii) $\gamma > 1$
We will show $\lim_{t \to \infty} \|tA(X(t))\| = \gamma$. With change of variable $s = tv$ and integration by parts we have

$$\left(\frac{1}{\gamma}\right) tA(X(t)) = e^{-tA} At^{-(\gamma-1)} \int_0^t e^{sA} s^{\gamma-1} ds \, X_0$$

$$= e^{-tA} tA \int_0^1 e^{tvA} v^{\gamma-1} dv \, X_0$$

$$= e^{-tA} \left[e^{tvA} v^{\gamma-1}\right]_0^1 X_0 - (\gamma - 1) e^{-tA} \int_0^1 e^{tvA} v^{\gamma-2} dv \, X_0$$

$$= X_0 - (\gamma - 1) e^{-tA} \left(\int_0^1 \cos(tv) v^{\gamma-2} dv \, , \, \int_0^1 \sin(tv) v^{\gamma-2} dv\right)^T$$

By the way, from $\gamma > 1$ we have $v^{\gamma-2} \in L^1[0,1]$, by Riemann-Lebesgue lemma we have

$$\lim_{t \to \infty} \int_0^1 \cos(tv) v^{\gamma-2} dv = \lim_{t \to \infty} \int_0^1 \sin(tv) v^{\gamma-2} dv = 0.$$

Observe as $e^{-tA}$ is a rotation, we know

$$\left\|e^{-tA}\left(\int_0^1 \cos(tv) v^{\gamma-2} dv \, , \, \int_0^1 \sin(tv) v^{\gamma-2} dv\right)\right\| = \left\|\left(\int_0^1 \cos(tv) v^{\gamma-2} dv \, , \, \int_0^1 \sin(tv) v^{\gamma-2} dv\right)\right\|.$$

Taking limit $t \to \infty$ we know right hand side converges to zero, we conclude

$$\lim_{t \to \infty} e^{-tA}\left(\int_0^1 \cos(tv) v^{\gamma-2} dv \, , \, \int_0^1 \sin(tv) v^{\gamma-2} dv\right) = 0.$$

Therefore,

$$\lim_{t \to \infty} \|tA(X(t))\| = \left\|\gamma X_0 - \gamma(\gamma - 1) \lim_{t \to \infty} e^{-tA}\left(\int_0^1 \cos(tv) v^{\gamma-2} dv \, , \, \int_0^1 \sin(tv) v^{\gamma-2} dv\right)^T\right\|$$

$$= \gamma \|X_0\| = \gamma$$

**F.3.2   Proof for the case $p = 1, \gamma < 1$**

Since $A, e^{tA}$ are invertible, we see

$$\lim_{t\to\infty} t^{2\gamma} \|A(X(t))\|^2 \neq 0 \iff \lim_{t\to\infty} t^\gamma \|A(X(t))\| \neq 0 \iff \lim_{t\to\infty} t^\gamma \left\|\frac{1}{\gamma} e^{tA} X(t)\right\| \neq 0.$$

Therefore it is enough to observe $\frac{1}{\gamma} e^{tA} t^\gamma X(t)$. Again for $X_0 = (1, 0)^T$

$$\frac{1}{\gamma} e^{tA} t^\gamma X(t) = \int_0^t e^{sA} s^{\gamma-1} ds \ X_0$$

$$= \left(\int_0^t \cos(s) s^{\gamma-1} ds \ , \ \int_0^t \sin(s) s^{\gamma-1} ds\right)^T.$$

Since $s^{\gamma-1}$ decrease monotonically to zero, we may apply similar argument with alternative series test.

Define $a_n = \int_{(n-1)\pi}^{n\pi} \sin(s) s^{\gamma-1} ds$ and name $S_n = \sum_{k=1}^n a_n$. Then for $m \in \mathbb{N}, t > 2m\pi$ we see

$$S_{2m} \leq \int_0^t \sin(s) s^{\gamma-1} ds \leq S_{2m-1}.$$

By the way $S = \lim_{n\to\infty} S_n$ exists by alternative series test, thus we have

$$\lim_{t\to\infty} \int_0^t \sin(s) s^{\gamma-1} ds = S$$

by squeeze theorem. Note $S \geq S_2 > 0$. With similar argument, we can conclude $\lim_{t\to\infty} \int_0^t \cos(s) s^{\gamma-1} ds$ also exists.

Therefore we conclude $\lim_{t\to\infty} \|t^\gamma X(t)\| \neq 0$ since

$$\lim_{t\to\infty} \|t^\gamma X(t)\| = \lim_{t\to\infty} \gamma \left\|\left(\int_0^t \cos(s) s^{\gamma-1} ds \ , \ \int_0^t \sin(s) s^{\gamma-1} ds\right)^T\right\| \geq \gamma S > 0.$$

**F.3.3   Proof for the case $0 < p < 1$**

Recall from (9), we have

$$X(t) = \frac{e^{-tA}}{C(t)} \left(\int_0^t e^{sA} C(s)\beta(s) ds + C(0)I\right) X_0.$$

Our goal is to show

$$\lim_{t\to\infty} t^{2p} \|A(X(t))\|^2 \neq 0.$$

We first observe, it is enough to show

$$\lim_{t\to\infty} \frac{1}{\beta(t)C(t)} \int_0^t C(s)\beta(s) \sin s \, ds \neq 0.$$

This follows from below facts.

(1) Since $\beta(t) = \frac{\gamma}{t^p}$ and $A$ is linear and invertible,

$$\lim_{t\to\infty} t^{2p} \|A(X(t))\|^2 = \lim_{t\to\infty} \frac{\gamma^2}{\beta(t)^2} \|A(X(t))\|^2 \neq 0 \iff \lim_{t\to\infty} \left\|\frac{1}{\beta(t)} X(t)\right\| \neq 0.$$

(2) Recall from (31), we have $C(t) = e^{\frac{\gamma}{1-p} t^{1-p}}$ for $\beta(t) = \frac{\gamma}{t^p}$, so $C(0) = \lim_{t\to 0+} C(t) = 1$ and $\lim_{t\to\infty} \frac{1}{\beta(t)C(t)} = 0$.
Thus $\lim_{t\to\infty} \frac{1}{\beta(t)C(t)} \|C(0)I\| = 0$, second term is ignorable. Therefore the problem reduces to

$$\lim_{t\to\infty} \left\|\frac{e^{-tA}}{\beta(t)C(t)} \left(\int_0^t C(s)\beta(s) e^{-sA} ds\right) X_0\right\| \neq 0.$$

(3) Since $e^{tA}$ is linear and invertible, problem reduces to

$$\lim_{t\to\infty}\left\|\frac{1}{\beta(t)C(t)}\left(\int_0^t C(s)\beta(s)e^{-sA}ds\right)X_0\right\|\neq 0.$$

(4) Again without loss of generality, let $X_0=(1,0)^T$. Recalling $e^{-sA}=\begin{pmatrix}\cos s & \sin s\\ -\sin s & \cos s\end{pmatrix}$, focusing on one component we see it is sufficient to show

$$\lim_{t\to\infty}\frac{1}{\beta(t)C(t)}\int_0^t C(s)\beta(s)\sin s\,ds\neq 0.$$

Now we prove the statement. For convenience, name $D(t)=\beta(t)C(t)$. We want to show

$$\lim_{t\to\infty}\underbrace{\int_0^t\frac{D(s)}{D(t)}\sin s\,ds}_{=S(t)}\neq 0.$$

We first observe

$$\lim_{t\to\infty}\sup_{\delta\in[0,\delta]}\left|\frac{D(t+\delta)}{D(t+\pi)}-1\right|=0.$$

To do so, we first show $\lim_{t\to\infty}\frac{D(t+\delta)}{D(t+\pi)}=1$ for $\delta\in[0,\pi]$. Take $\delta\in[0,\pi]$. Observe

$$\frac{D(t+\delta)}{D(t+\pi)}=\frac{\beta(t+\delta)C(t+\delta)}{\beta(t+\pi)C(t+\pi)}=\left(\frac{t+\pi}{t+\delta}\right)^p\frac{e^{\frac{\gamma}{1-p}(t+\delta)^{1-p}}}{e^{\frac{\gamma}{1-p}(t+\pi)^{1-p}}}=\left(\frac{t+\pi}{t+\delta}\right)^p e^{\frac{\gamma}{1-p}(t+\delta)^{1-p}\left(1-\left(\frac{t+\pi}{t+\delta}\right)^{1-p}\right)}.$$

Considering L'ôspital's rule for the exponent, we see

$$\lim_{t\to\infty}(t+\delta)^{1-p}\left(1-\left(\frac{t+\pi}{t+\delta}\right)^{1-p}\right)=\lim_{t\to\infty}\frac{1-\left(\frac{t+\pi}{t+\delta}\right)^{1-p}}{(t+\delta)^{p-1}}=\lim_{t\to\infty}\frac{\frac{-\pi(p-1)\left(\frac{t+\pi}{t+\delta}\right)^{-p}}{(t+\delta)^2}}{(p-1)(t+\delta)^{p-2}}=\lim_{t\to\infty}\frac{-\pi}{(t+\pi)^p}=0.$$

As the exponent reaches to zero, we conclude $\lim_{t\to\infty}\frac{D(t+\delta)}{D(t+\pi)}=1$.

Now from

$$\dot D(t)=C(t)\left(\beta(t)^2+\dot\beta(t)\right)=e^{\frac{\gamma}{1-p}t^{1-p}}\gamma t^{-2p-1}\left(\gamma t-pt^p\right),$$

we see $D(t)$ is nondecreasing for $t\geq\left(\frac{\gamma}{p}\right)^{\frac{1}{p-1}}$ since $0<p<1$. Therefore for $t\geq\left(\frac{\gamma}{p}\right)^{\frac{1}{p-1}}$ we have $\sup_{\delta\in[0,\delta]}\left|\frac{D(t+\delta)}{D(t+\pi)}-1\right|=\max\left\{\left|\frac{D(t)}{D(t+\pi)}-1\right|,\left|\frac{D(t+\delta)}{D(t+\pi)}-1\right|\right\}$, so it reaches to zero as $t\to\infty$.

Now we prove desired statement. Proof by contradiction. Suppose $\lim_{t\to\infty}S(t)=0$. Then for $0<\epsilon<\frac{1}{2}$, there is $T_1>0$ such that $t>T_1$ implies $|S(t)|<\epsilon$. Now for $t>T_1$, observe

$$2\epsilon>|S(t+\pi)-S(t)|=\left|\int_0^{t+\pi}\frac{D(s)}{D(t+\pi)}\sin s\,ds-\int_0^t\frac{D(s)}{D(t)}\sin s\,ds\right|$$

$$=\left|\frac{1}{D(t+\pi)}\left(\int_0^{t+\pi}D(s)\sin s\,ds-\int_0^t D(s)\sin s\,ds\right)+\left(\frac{1}{D(t+\pi)}-\frac{1}{D(t)}\right)\int_0^t D(s)\sin s\,ds\right|$$

$$\geq\left|\int_t^{t+\pi}\frac{D(s)}{D(t+\pi)}\sin s\,ds\right|-\left|\left(\frac{D(t)}{D(t+\pi)}-1\right)\int_0^t\frac{D(s)}{D(t)}\sin s\,ds\right|.$$

On the other hand, there is $T_2$ such that $t>T_2$ implies $\sup_{\delta\in[0,\pi]}\left|\frac{D(t+\delta)}{D(t+\pi)}-1\right|<\frac{\epsilon}{2}$. Now for $t=2n\pi>\max\{T_1,T_2\}$,

$$2\epsilon>\left|\int_{2n\pi}^{(2n+1)\pi}\frac{D(s)}{D(t+\pi)}\sin s\,ds\right|-\left|\frac{D(t)}{D(t+\pi)}-1\right|\times|S(t)|>\int_{2n\pi}^{(2n+1)\pi}(1-\epsilon)|\sin s|\,ds-\epsilon=2-2\epsilon.$$

This contradicts the fact $\epsilon<\frac{1}{2}$, we prove the assumption $\lim_{t\to\infty}\int_0^t S(t)=0$ is not true. Therefore $\lim_{t\to\infty}\int_0^t S(t)\neq 0$.

# G  Proof of convergence analysis for discrete counterpart

## G.1  Correspondence between discrete method in Theorem 5.1 and continuous model (6)

To check the correspondence of the method with (6), we provide a informal derivation. Assume operator $\mathbb{A}$ is continuous. Then we have $y^k = x^{k+1} + \mathbb{A}x^{k+1}$, by substituting $y^k$ and $y^{k-1}$, the method can be equivalently written as

$$x^{k+1} + \mathbb{A}x^{k+1} = \frac{k^p}{k^p + \gamma}(x^k - \mathbb{A}x^k) + \frac{\gamma}{k^p + \gamma}x^0. \tag{40}$$

This can be considered as a special case of below method, with $h = 1$.

$$x^{k+1} + h\mathbb{A}x^{k+1} = \frac{k^p}{k^p + h^{1-p}\gamma}(x^k - h\mathbb{A}x^k) + \frac{h^{1-p}\gamma}{k^p + h^{1-p}\gamma}x^0.$$

Dividing by $h$ both sides and reorganizing, we have

$$\frac{x^{k+1} - x^k}{h} = -\mathbb{A}x^{k+1} - \frac{h^p k^p}{h^p k^p + h\gamma}\mathbb{A}x^k - \frac{\gamma}{h^p k^p + h\gamma}\left(x^k - x^0\right).$$

Identifying $hk = t$, $x^k = X(t)$ and taking $h \to 0$, we have

$$\dot{X}(t) = -2\mathbb{A}(X(t)) - \frac{\gamma}{t^p}\left(X - X_0\right).$$

As monotonicity is preserved for scalar multiple, rescaling $2\mathbb{A}$ to $\mathbb{A}$ does not change the class of operators that the ODE covers. For notation simplicity, replacing $2\mathbb{A}$ to $\mathbb{A}$ we have

$$\dot{X}(t) = -\mathbb{A}(X(t)) - \frac{\gamma}{t^p}\left(X - X_0\right).$$

## G.2  Proof of boundedness of $\left\|x^k\right\|$

While proving Theorem 5.1, we need a upper bound for $\left\|x^k\right\|$. Therefore we first prove following lemma.

**Lemma G.1.** *Let $\mathbb{A}$ be a maximal monotone operator. Consider a method*

$$x^k = \mathbb{J}_{\mathbb{A}}y^{k-1}$$
$$y^k = (1 - \beta_k)\left(2x^k - y^{k-1}\right) + \beta_k x^0$$

*for $k = 1, 2, \ldots$, with sequence $\{\beta_k\}_{k \in \mathbb{N}}$, $0 \leq \beta_k \leq 1$ and initial condition $y^0 = x^0 \in \mathbb{R}^n$. Then following holds*

$$\left\|x^k - x^\star\right\| \leq \left\|x^0 - x^\star\right\|.$$

*for $x^\star \in \mathrm{Zer}\mathbb{A}$. And so, $\left\|x^k - x^0\right\| \leq 2\left\|x^0 - x^\star\right\|$.*

*Proof.* Recall from Lemma D.2, we know for $\mathbb{T} = \mathbb{R}_{\mathbb{A}} = 2\mathbb{J}_{\mathbb{A}} - \mathbb{I}$ given method is equivalent to below method

$$y^{k+1} = \beta_k y^0 + (1 - \beta_k)\mathbb{T}y^k.$$

Note that $x^\star \in \mathrm{Zer}\,\mathbb{A} \Leftrightarrow 0 \in \mathbb{A}x^\star \Leftrightarrow x^\star = \mathbb{J}_{\mathbb{A}}x^\star \Leftrightarrow x^\star = \mathbb{T}x^\star \Leftrightarrow x^\star \in \mathrm{Fix}\,\mathbb{T}$. We first prove $\left\|y^k - x^\star\right\| \leq \left\|y^0 - x^\star\right\|$ by induction. The statement is trivially true when $k = 0$. Now suppose the statement is true for $k \in \mathbb{N}$. Then

$$\begin{aligned}
\left\|y^{k+1} - x^\star\right\| &= \left\|\beta_k\left(y^0 - x^\star\right) + (1 - \beta_k)\left(\mathbb{T}y^k - x^\star\right)\right\| \\
&\leq \beta_k\left\|y^0 - x^\star\right\| + (1 - \beta_k)\left\|\mathbb{T}y^k - x^\star\right\| \\
&\leq \beta_k\left\|y^0 - x^\star\right\| + (1 - \beta_k)\left\|y^k - x^\star\right\| \\
&\leq \beta_k\left\|y^0 - x^\star\right\| + (1 - \beta_k)\left\|y^0 - x^\star\right\| = \left\|y^0 - x^\star\right\|.
\end{aligned}$$

The first inequality is just triangular inequality, the second inequality comes from the fact $x^\star \in \mathrm{Fix}\,\mathbb{T}$ and $\mathbb{T}$ is nonexpansive, and the last inequality is from induction hypothesis. By induction, we have $\left\|y^k - x^\star\right\| \leq \left\|y^0 - x^\star\right\|$ for $k = 0, 1, \ldots$.

Define $\tilde{\mathbf{A}}x^k = y^{k-1} - x^k$, then $\tilde{\mathbf{A}}x^k \in \mathbb{A}x^k$ since $\mathbb{J}_{\mathbb{A}}y^{k-1} = x^k$. Observe for $k = 1, 2, \ldots$, from monotone inequality we have

$$\left\|y^{k-1} - x^\star\right\|^2 = \left\|\tilde{\mathbf{A}}x^k + (x^k - x^\star)\right\|^2 = \left\|\tilde{\mathbf{A}}x^k\right\|^2 + 2\left\langle \tilde{\mathbf{A}}x^k, x^k - x^\star\right\rangle + \left\|x^k - x^\star\right\|^2 \geq \left\|x^k - x^\star\right\|^2.$$

Therefore we conclude

$$\left\|x^k - x^\star\right\| \leq \left\|y^{k-1} - x^\star\right\| \leq \left\|y^0 - x^\star\right\| = \left\|x^0 - x^\star\right\|.$$

The latter statement holds directly from triangular inequality,

$$\left\|x^k - x^0\right\| \leq \left\|x^k - x^\star\right\| + \left\|x^\star - x^0\right\| \leq 2\left\|x^0 - x^\star\right\|.$$

$\square$

### G.3 Proof of Theorem 5.1

The outline of the proofs originate from continuous proofs. To simplify calculations, instead of directly deriving discrete counterpart of Proposition 3.2, we consider rescaled conservation law for each cases. By considering dilated coordinate $W_1 = X - X_0$ for $p > 1$, $W_2 = t^p(X - X_0)$ for $0 < p < 1$, $W_3 = t(X - X_0)$ for $p = 1$, $\gamma \geq 1$ and $W_4 = t^\gamma(X - X_0)$ for $p = 1$, $0 < \gamma < 1$, we obtain below conservation laws.

$$E_1 \equiv \left\|\tilde{\mathbf{A}}(X(t))\right\|^2 + \frac{2\gamma}{t^p}\left\langle \tilde{\mathbf{A}}(X(t)), X(t) - X_0\right\rangle + \left(\frac{\gamma^2}{t^{2p}} + \frac{\gamma p}{t^{p+1}}\right)\|X(t) - X_0\|^2$$
$$+ \int_0^t \left\langle \frac{d}{ds}\tilde{\mathbf{A}}(X(s)), \dot{X}(s)\right\rangle ds + \int_0^t \frac{\gamma}{s^p}\left\|\dot{X}(s) + \frac{p}{s}(X(s) - X_0)\right\|^2 ds + \frac{1}{2}\int_0^t \frac{\gamma p(p-1)}{s^{p+2}}\|X(s) - X_0\|^2 ds$$

$$E_2 \equiv t^{2p}\left\|\tilde{\mathbf{A}}(X(t))\right\|^2 + 2\gamma t^p\left\langle \tilde{\mathbf{A}}(X(t)), X(t) - X_0\right\rangle + \left(\gamma^2 - \gamma p t^{p-1}\right)\|X(t) - X_0\|^2$$
$$+ \int_0^t 2s^{2p}\left\langle \frac{d}{ds}\tilde{\mathbf{A}}(X(s)), \dot{X}(s)\right\rangle ds + \int_0^t 2s^p\left(\gamma - ps^{p-1}\right)\left\|\dot{X}(s)\right\|^2 ds + \int_0^t \gamma p(p-1)s^{p-2}\|X(s) - X_0\|^2 ds$$

$$E_3 \equiv t^2\left\|\tilde{\mathbf{A}}(X(t))\right\|^2 + 2\gamma t\left\langle \tilde{\mathbf{A}}(X(t)), X(t) - X_0\right\rangle + \gamma(\gamma - 1)\|X(t) - X_0\|^2$$
$$+ \int_{t_0}^t 2s^2\left\langle \frac{d}{ds}\tilde{\mathbf{A}}(X(s)), \dot{X}(s)\right\rangle ds + \int_{t_0}^t 2s(\gamma - 1)\left\|\dot{X}(s)\right\|^2 ds$$

$$E_4 \equiv t^{2\gamma}\left\|\tilde{\mathbf{A}}(X(t))\right\|^2 + 2\gamma t^{2\gamma-1}\left\langle \tilde{\mathbf{A}}(X(t)), X(t) - X_0\right\rangle + \gamma(\gamma - 1)t^{2\gamma-2}\|X(t) - X_0\|^2$$
$$+ \int_0^t 2s^{2\gamma}\left\langle \frac{d}{ds}\tilde{\mathbf{A}}(X(s)), \dot{X}(s)\right\rangle ds + \int_0^t 2\gamma(\gamma - 1)s^{2\gamma-3}\|X(s) - X_0\|^2 ds.$$

Lyapunov style proof can be obtained by considering below functions

$$U_1(t) = \left\|\tilde{\mathbf{A}}(X(t))\right\|^2 + \frac{2\gamma}{t^p}\left\langle \tilde{\mathbf{A}}(X(t)), X(t) - X_0\right\rangle + \left(\frac{\gamma^2}{t^{2p}} + \frac{\gamma p}{t^{p+1}}\right)\|X(t) - X_0\|^2$$

$$U_2(t) = t^{2p}\left\|\tilde{\mathbf{A}}(X(t))\right\|^2 + 2\gamma t^p\left\langle \tilde{\mathbf{A}}(X(t)), X(t) - X_0\right\rangle + \left(\gamma^2 - \gamma p t^{p-1}\right)\|X(t) - X_0\|^2$$

$$U_3(t) = t^2\left\|\tilde{\mathbf{A}}(X(t))\right\|^2 + 2\gamma t\left\langle \tilde{\mathbf{A}}(X(t)), X(t) - X_0\right\rangle + \gamma(\gamma - 1)\|X(t) - X_0\|^2$$

$$U_4(t) = t^{2\gamma}\left\|\tilde{\mathbf{A}}(X(t))\right\|^2 + \gamma t^{2\gamma-1}\left\langle \tilde{\mathbf{A}}(X(t)), X(t) - X_0\right\rangle + \gamma(\gamma - 1)t^{2\gamma-2}\|X(t) - X_0\|^2.$$

The main blocks of the calculations corresponds to continuous cases, but there are some 'errors' in terms of $\tilde{\mathbf{A}}x^k$ and $x^k - x^0$ occur due to discretization. The proofs are done by showing these 'errors' don't effect to the conclusions.

(0) Preparation.

For all cases, we will consider functions of the form

$$U^k = a_k\left\|\tilde{\mathbf{A}}x^k\right\|^2 + b_k\left\langle \tilde{\mathbf{A}}x^k, x^k - x^0\right\rangle + c_k\left\|x^k - x^0\right\|^2, \tag{41}$$

and consider $U^{k+1} - U^k$. As similar calculations will be repeated, we first organize the repeating calculations. That is, we prove following equality is true.

$$U^{k+1} - U^k + \lambda_k \left\langle \tilde{\mathbf{A}} x^{k+1} - \tilde{\mathbf{A}} x^k, x^{k+1} - x^k \right\rangle - \tau_k \left\langle \tilde{\mathbf{A}} x^{k+1} - \tilde{\mathbf{A}} x^k, x^{k+1} - x^k \right\rangle \tag{42}$$

$$= \left( a_{k+1} - \lambda_k \frac{k^p + \gamma}{k^p} \right) \|\tilde{\mathbf{A}} x^{k+1}\|^2 + \left( \lambda_k \frac{k^p}{k^p + \gamma} - a_k \right) \|\tilde{\mathbf{A}} x^k\|^2$$

$$+ \left( b_{k+1} - \lambda_k \frac{\gamma}{k^p} - \tau_k - c_{k+1} \frac{k^p + \gamma}{k^p} \right) \left\langle \tilde{\mathbf{A}} x^{k+1}, x^{k+1} - x^0 \right\rangle + (\tau_k - c_{k+1}) \left\langle \tilde{\mathbf{A}} x^{k+1}, x^k - x^0 \right\rangle$$

$$+ (\tau_k - c_{k+1}) \left\langle \tilde{\mathbf{A}} x^k, x^{k+1} - x^0 \right\rangle + \left( \lambda_k \frac{\gamma}{k^p + \gamma} - b_k - \tau_k - c_{k+1} \frac{k^p}{k^p + \gamma} \right) \left\langle \tilde{\mathbf{A}} x^k, x^k - x^0 \right\rangle$$

$$- c_{k+1} \frac{\gamma}{k^p} \left\| x^{k+1} - x^0 \right\|^2 - c_{k+1} \frac{\gamma}{k^p + \gamma} \left\| x^k - x^0 \right\|^2 + (c_{k+1} - c_k) \left\| x^k - x^0 \right\|^2.$$

Observe, this can be shown by checking below equalities are true.

$$\left\langle \tilde{\mathbf{A}} x^{k+1} - \tilde{\mathbf{A}} x^k, x^{k+1} - x^k \right\rangle \tag{43}$$

$$= -\frac{k^p + \gamma}{k^p} \|\tilde{\mathbf{A}} x^{k+1}\|^2 - \frac{\gamma}{k^p} \langle \tilde{\mathbf{A}} x^{k+1}, x^{k+1} - x^0 \rangle + \frac{k^p}{k^p + \gamma} \|\tilde{\mathbf{A}} x^k\|^2 + \frac{\gamma}{k^p + \gamma} \langle \tilde{\mathbf{A}} x^k, x^k - x^0 \rangle$$

$$\left\langle \tilde{\mathbf{A}} x^{k+1} - \tilde{\mathbf{A}} x^k, x^{k+1} - x^k \right\rangle \tag{44}$$

$$= \left\langle \tilde{\mathbf{A}} x^{k+1}, x^{k+1} - x^0 \right\rangle + \left\langle \tilde{\mathbf{A}} x^k, x^k - x^0 \right\rangle - \left( \left\langle \tilde{\mathbf{A}} x^{k+1}, x^k - x^0 \right\rangle + \left\langle \tilde{\mathbf{A}} x^k, x^{k+1} - x^0 \right\rangle \right)$$

$$c_{k+1} \left\| x^{k+1} - x^0 \right\|^2 - c_k \left\| x^k - x^0 \right\|^2 \tag{45}$$

$$= -c_{k+1} \frac{k^p + \gamma}{k^p} \left\langle \tilde{\mathbf{A}} x^{k+1}, x^{k+1} - x^0 \right\rangle - c_{k+1} \left\langle \tilde{\mathbf{A}} x^{k+1}, x^k - x^0 \right\rangle$$

$$- c_{k+1} \left\langle \tilde{\mathbf{A}} x^k, x^{k+1} - x^0 \right\rangle - c_{k+1} \frac{k^p}{k^p + \gamma} \left\langle \tilde{\mathbf{A}} x^k, x^k - x^0 \right\rangle$$

$$- c_{k+1} \frac{\gamma}{k^p} \left\| x^{k+1} - x^0 \right\|^2 - c_{k+1} \frac{\gamma}{k^p + \gamma} \left\| x^k - x^0 \right\|^2 + (c_{k+1} - c_k) \left\| x^k - x^0 \right\|^2.$$

– Proof for (43)

Recall, the method was defined as

$$y^k = \frac{k^p}{k^p + \gamma} (2x^k - y^{k-1}) + \frac{\gamma}{k^p + \gamma} x^0$$

$$x^{k+1} = \mathbf{J}_\mathbb{A} y^k.$$

By considering $x^{k+1} + \tilde{\mathbf{A}} x^{k+1} = y^k$, substituting $y^k$ and $y^{k-1}$ we can rewrite the method as

$$x^{k+1} + \tilde{\mathbf{A}} x^{k+1} = \left( 1 - \frac{\gamma}{k^p + \gamma} \right) (x^k - \tilde{\mathbf{A}} x^k) + \frac{\gamma}{k^p + \gamma} x^0. \tag{46}$$

By multiplying $k^p + \gamma$ to both sides and reorganizing we have,

$$(k^p + \gamma) \{ (x^{k+1} - x^0) + \tilde{\mathbf{A}} x^{k+1} \} = k^p \{ (x^k - x^0) - \tilde{\mathbf{A}} x^k \}.$$

By subtracting both sides by $k^p(x^{k+1} - x^0)$ and $(k^p + \gamma)(x^k - x^0)$ respectively, above equation can be rewritten as

$$-k^p (\tilde{\mathbf{A}} x^k + (x^{k+1} - x^k)) = (k^p + \gamma) \tilde{\mathbf{A}} x^{k+1} + \gamma(x^{k+1} - x^0)$$

$$-(k^p + \gamma)(\tilde{\mathbf{A}} x^{k+1} + (x^{k+1} - x^k)) = k^p \tilde{\mathbf{A}} x^k + \gamma(x^k - x^0).$$

From above, we get the following

$$(k^p + \gamma) k^p \left\langle \tilde{\mathbf{A}} x^{k+1} - \tilde{\mathbf{A}} x^k, x^{k+1} - x^k \right\rangle$$

$$= (k^p + \gamma) k^p \left\langle \tilde{\mathbf{A}} x^{k+1}, x^{k+1} - x^k \right\rangle + (k^p + \gamma) k^p \left\langle \tilde{\mathbf{A}} x^{k+1}, \tilde{\mathbf{A}} x^k \right\rangle$$

$$- (k^p + \gamma) k^p \left\langle \tilde{\mathbf{A}} x^k, x^{k+1} - x^k \right\rangle - (k^p + \gamma) k^p \left\langle \tilde{\mathbf{A}} x^{k+1}, \tilde{\mathbf{A}} x^k \right\rangle$$

$$= -(k^p + \gamma) \langle \tilde{\mathbf{A}} x^{k+1}, -k^p(\tilde{\mathbf{A}} x^k + (x^{k+1} - x^k)) \rangle + k^p \langle \tilde{\mathbf{A}} x^k, -(k^p + \gamma)(\tilde{\mathbf{A}} x^{k+1} + (x^{k+1} - x^k)) \rangle$$

$$= -(k^p + \gamma) \langle \tilde{\mathbf{A}} x^{k+1}, (k^p + \gamma) \tilde{\mathbf{A}} x^{k+1} + \gamma(x^{k+1} - x^0) \rangle + k^p \langle \tilde{\mathbf{A}} x^k, k^p \tilde{\mathbf{A}} x^k + \gamma(x^k - x^0) \rangle$$

$$= -(k^p + \gamma) \left( (k^p + \gamma) \|\tilde{\mathbf{A}} x^{k+1}\|^2 + \gamma \langle \tilde{\mathbf{A}} x^{k+1}, x^{k+1} - x^0 \rangle \right) + k^p \left( \|\tilde{\mathbf{A}} x^k\|^2 + \gamma \langle \tilde{\mathbf{A}} x^k, x^k - x^0 \rangle \right).$$

Now dividing both sides by $k^p(k^p + \gamma)$, we get the desired result.

– Proof for (44)

This can be checked by just expanding the inner product of left hand side.

– Proof for (45)

First, observe

$$
c_{k+1} \left\|x^{k+1} - x^0\right\|^2 - c_k \left\|x^k - x^0\right\|^2
$$
$$
= c_{k+1} \left( \left\|x^{k+1} - x^0\right\|^2 - \left\|x^k - x^0\right\|^2 \right) + (c_{k+1} - c_k) \left\|x^k - x^0\right\|^2
$$
$$
= c_{k+1} \left\langle x^{k+1} - x^k, \left(x^{k+1} - x^0\right) + \left(x^k - x^0\right) \right\rangle + (c_{k+1} - c_k) \left\|x^k - x^0\right\|^2
$$
$$
= c_{k+1} \left( \left\langle x^{k+1} - x^k, x^{k+1} - x^0 \right\rangle + \left\langle x^{k+1} - x^k, x^k - x^0 \right\rangle \right) + (c_{k+1} - c_k) \left\|x^k - x^0\right\|^2.
$$

Reorganizing (40), we get two different expressions for $x^{k+1} - x^k$.

$$
x^{k+1} - x^k = - \left( \tilde{\mathbf{A}}x^{k+1} + \frac{k^p}{k^p + \gamma} \tilde{\mathbf{A}}x^k \right) - \frac{\gamma}{k^p + \gamma} \left(x^k - x^0\right)
$$
$$
x^{k+1} - x^k = - \left( \frac{k^p + \gamma}{k^p} \tilde{\mathbf{A}}x^{k+1} + \tilde{\mathbf{A}}x^k \right) - \frac{\gamma}{k^p} \left(x^{k+1} - x^0\right).
$$

Now plugging these to previous equality, be get the desired result.

$$
c_{k+1} \left\|x^{k+1} - x^0\right\|^2 - c_k \left\|x^k - x^0\right\|^2
$$
$$
= -c_{k+1} \left( \left\langle \frac{k^p + \gamma}{k^p} \tilde{\mathbf{A}}x^{k+1} + \tilde{\mathbf{A}}x^k + \frac{\gamma}{k^p} \left(x^{k+1} - x^0\right), x^{k+1} - x^0 \right\rangle \right.
$$
$$
\left. + \left\langle \tilde{\mathbf{A}}x^{k+1} + \frac{k^p}{k^p + \gamma} \tilde{\mathbf{A}}x^k + \frac{\gamma}{k^p + \gamma} \left(x^k - x^0\right), x^k - x^0 \right\rangle \right) + (c_{k+1} - c_k) \left\|x^k - x^0\right\|^2
$$
$$
= -c_{k+1} \frac{k^p + \gamma}{k^p} \left\langle \tilde{\mathbf{A}}x^{k+1}, x^{k+1} - x^0 \right\rangle - c_{k+1} \left\langle \tilde{\mathbf{A}}x^{k+1}, x^k - x^0 \right\rangle
$$
$$
- c_{k+1} \left\langle \tilde{\mathbf{A}}x^k, x^{k+1} - x^0 \right\rangle - c_{k+1} \frac{k^p}{k^p + \gamma} \left\langle \tilde{\mathbf{A}}x^k, x^k - x^0 \right\rangle
$$
$$
- c_{k+1} \frac{\gamma}{k^p} \left\|x^{k+1} - x^0\right\|^2 - c_{k+1} \frac{\gamma}{k^p + \gamma} \left\|x^k - x^0\right\|^2 + (c_{k+1} - c_k) \left\|x^k - x^0\right\|^2
$$

(i) $\left\|\tilde{\mathbf{A}}(x^k)\right\|^2 = \mathcal{O}(1)$ for $p > 0, \gamma > 0$.

Plugging

$$
a_k = 1 + \frac{\gamma}{2k^p}, \qquad b_k = \frac{\gamma}{k^p}, \qquad c_{k+1} = \frac{\gamma k^p}{4\left(k^p + \frac{\gamma}{2}\right)} \left( \frac{\gamma}{k^{2p}} - \left( \frac{1}{k^p} - \frac{1}{(k+1)^p} \right) \right)
$$
$$
\lambda_k = 1 + \frac{\gamma}{2k^p}, \qquad \tau_k = c_{k+1}
$$

to (41) and (42), we obtain

$$
U^{k+1} - U^k + \left( 1 + \frac{\gamma}{2k^p} - \frac{\gamma k^p}{4\left(k^p + \frac{\gamma}{2}\right)} \left( \frac{\gamma}{k^{2p}} - \left( \frac{1}{k^p} - \frac{1}{(k+1)^p} \right) \right) \right) \left\langle \tilde{\mathbf{A}}x^{k+1} - \tilde{\mathbf{A}}x^k, x^{k+1} - x^k \right\rangle
$$
$$
= - \left( \frac{\gamma \left(2k^p + \gamma\right)}{2k^{2p}} + \frac{\gamma}{2} \left( \frac{1}{k^p} - \frac{1}{(k+1)^p} \right) \right) \|\tilde{\mathbf{A}}x^{k+1}\|^2 - \frac{\gamma \left(2k^p + \gamma\right)}{2k^p \left(k^p + \gamma\right)} \|\tilde{\mathbf{A}}x^k\|^2
$$
$$
- \left( \frac{\gamma^2}{k^{2p}} + \frac{\gamma}{2} \left( \frac{1}{k^p} - \frac{1}{(k+1)^p} \right) \right) \left\langle \tilde{\mathbf{A}}x^{k+1}, x^{k+1} - x^0 \right\rangle
$$
$$
- \left( \frac{\gamma^2}{k^p \left(k^p + \gamma\right)} + \frac{\gamma k^p}{2(k^p + \gamma)} \left( \frac{1}{k^p} - \frac{1}{(k+1)^p} \right) \right) \left\langle \tilde{\mathbf{A}}x^k, x^k - x^0 \right\rangle
$$
$$
- \frac{\gamma^2}{2\left(2k^p + \gamma\right)} \left( \frac{\gamma}{k^{2p}} - \left( \frac{1}{k^p} - \frac{1}{(k+1)^p} \right) \right) \left\|x^{k+1} - x^0\right\|^2
$$

$$-\frac{\gamma^2 k^p}{2\left(k^p+\gamma\right)\left(2k^p+\gamma\right)}\left(\frac{\gamma}{k^{2p}}-\left(\frac{1}{k^p}-\frac{1}{(k+1)^p}\right)\right)\left\|x^k-x^0\right\|^2+\left(c_{k+1}-c_k\right)\left\|x^k-x^0\right\|^2$$

$$=-\frac{\gamma\left(2k^p+\gamma\right)}{2k^{2p}}\left\|\tilde{\mathbf{A}}x^{k+1}+\frac{1}{2k^p+\gamma}\left(\gamma+\frac{k^{2p}}{2}\left(\frac{1}{k^p}-\frac{1}{(k+1)^p}\right)\right)\left(x^{k+1}-x^0\right)\right\|^2$$

$$-\frac{\gamma\left(2k^p+\gamma\right)}{2k^p\left(k^p+\gamma\right)}\left\|\tilde{\mathbf{A}}x^k+\frac{1}{2k^p+\gamma}\left(\gamma+\frac{k^{2p}}{2}\left(\frac{1}{k^p}-\frac{1}{(k+1)^p}\right)\right)\left(x^k-x^0\right)\right\|^2$$

$$-\frac{1}{2}\left(\frac{\gamma}{k^p}-\frac{\gamma}{(k+1)^p}\right)\left\|\tilde{\mathbf{A}}x^{k+1}\right\|^2$$

$$-\underbrace{\frac{\gamma}{2\left(2k^p+\gamma\right)}\left(2\gamma\left(\frac{1}{(k+1)^p}-\frac{1}{k^p}\right)+\frac{k^{2p}}{4}\left(\frac{1}{k^p}-\frac{1}{(k+1)^p}\right)^2\right)}_{s_{k,1}}\left\|x^{k+1}-x^0\right\|^2$$

$$-\underbrace{\frac{\gamma k^p}{2\left(k^p+\gamma\right)\left(2k^p+\gamma\right)}\left(2\gamma\left(\frac{1}{(k+1)^p}-\frac{1}{k^p}\right)+\frac{k^{2p}}{4}\left(\frac{1}{k^p}\right)^2-\frac{1}{(k+1)^p}\right)}_{s_{k,0}}\left\|x^k-x^0\right\|^2$$

$$+\left(c_{k+1}-c_k\right)\left\|x^k-x^0\right\|^2.$$

The continuous counterpart of above equality is

$$\dot{U}_1(t)+\left\langle\frac{d}{dt}\tilde{\mathbf{A}}(X(t)),\dot{X}(t)\right\rangle$$

$$=-\frac{\gamma}{t^p}\left\|\tilde{\mathbf{A}}(X(t))+\left(\frac{\gamma}{t^p}+\frac{p}{t}\right)(X(t)-X_0)\right\|^2-\frac{1}{2}\frac{\gamma p(p-1)}{t^{p+2}}\left\|X(t)-X_0\right\|^2,$$

which can be obtained by differentiating and reorganizing the conservation law for $E_1$. Note, as $\frac{k^{2p}}{2}\left(\frac{1}{k^p}-\frac{1}{(k+1)^p}\right)=$ $\mathcal{O}\left(k^{2p}\right)\mathcal{O}\left(\frac{1}{k^{p+1}}\right)=\mathcal{O}\left(\frac{1}{k^{1-p}}\right)$, the order of the term $\frac{1}{2k^p+\gamma}\left(\gamma+\frac{k^{2p}}{2}\left(\frac{1}{k^p}-\frac{1}{(k+1)^p}\right)\right)$ corresponds to $\left(\frac{\gamma}{t^p}+\frac{p}{t}\right)$. Thus we can see the sum of first two terms on the right hand side of the equality for discrete setting corresponds to the first term of the right hand side of the equality for continuous setting.

We can observe that $s_{k,1},s_{k,0}=\mathcal{O}\left(\frac{1}{k^{2p+1}}\right)+\mathcal{O}\left(\frac{1}{k^{p+2}}\right)$. And since $c_k=\mathcal{O}\left(\frac{1}{k^{2p}}\right)+\mathcal{O}\left(\frac{1}{k^{p+1}}\right)$, we have $c_{k+1}-c_k=$ $\mathcal{O}\left(\frac{1}{k^{2p+1}}\right)+\mathcal{O}\left(\frac{1}{k^{p+2}}\right)$ as well. Therefore the coefficients of $\left\|x^{k+1}-x^0\right\|^2$ and $\left\|x^k-x^0\right\|^2$ are summable for $p>0$. From Lemma G.1, we know $\left\|x^{k+1}-x^0\right\|^2$ and $\left\|x^k-x^0\right\|^2$ are bounded by $4\left\|x^0-x^\star\right\|^2$.

The term $\left\langle\tilde{\mathbf{A}}x^{k+1}-\tilde{\mathbf{A}}x^k,x^{k+1}-x^k\right\rangle$ on the left hand side is nonnegative from monotonicity of $\tilde{\mathbf{A}}$, and the coefficient is nonnegative as well. The coefficient of $\left\|\tilde{\mathbf{A}}x^{k+1}\right\|^2$ in the right hand side, $-\frac{1}{2}\left(\frac{\gamma}{k^p}-\frac{\gamma}{(k+1)^p}\right)$, is nonpositive. As a result, we get below inequality.

$$U^{k+1}\leq U^k+\frac{1}{2}\left(\frac{\gamma}{k^p}-\frac{\gamma}{(k+1)^p}\right)\left\|x^{k+1}-x^0\right\|^2+\left(c_{k+1}-c_k\right)\left\|x^k-x^0\right\|^2$$

$$\leq U^1+2\left\|x^0-x^\star\right\|^2\sum_{m=1}^{\infty}\left(\left(\frac{\gamma}{k^p}-\frac{\gamma}{(k+1)^p}\right)+2\left(c_{k+1}-c_k\right)\right)=M.$$

As done in (8), by monotonicity of $\mathbb{A}$ and Young's inequality

$$M\geq U^k=a_k\left\|\tilde{\mathbf{A}}x^k\right\|^2+b_k\left\langle\tilde{\mathbf{A}}x^k,x^k-x^0\right\rangle+c_k\left\|x^k-x^0\right\|^2$$

$$\geq a_k\left\|\tilde{\mathbf{A}}x^k\right\|^2+b_k\left\langle\tilde{\mathbf{A}}x^k,x^\star-x^0\right\rangle$$

$$\geq a_k\left\|\tilde{\mathbf{A}}x^k\right\|^2-\frac{1}{2}\left(\left\|\tilde{\mathbf{A}}x^k\right\|^2+b_k^2\left\|x^\star-x^0\right\|^2\right)=\frac{1}{2}\left(1+\frac{\gamma}{k^p}\right)\left\|\tilde{\mathbf{A}}x^k\right\|^2+\frac{\gamma^2}{2k^{2p}}\left\|x^\star-x^0\right\|^2.$$

Reorganizing, we get the desired result

$$\left\|\tilde{\mathbf{A}}x^k\right\|^2 \le 2\left(1 - \frac{\gamma}{k^p}\right)^{-1}\left(M + \frac{\gamma^2}{2k^{2p}}\left\|x^\star - x^0\right\|^2\right)$$

$$= 2\left(1 + \frac{\gamma}{k^p - \gamma}\right)\left(M + \frac{\gamma^2}{2k^{2p}}\left\|x^\star - x^0\right\|^2\right) = \mathcal{O}\left(1\right).$$

(ii) $\left\|\tilde{\mathbf{A}}(x^k)\right\|^2 = \mathcal{O}\left(\frac{1}{k^{2p}}\right)$ for $0 < p < 1, \gamma > 0$.

Plugging

$$a_{k+1} = k^p\left(k^p + \frac{\gamma}{2}\right), \qquad\qquad b_{k+1} = \gamma k^p, \qquad\qquad c_{k+1} = \frac{k^p\gamma^2}{4\left(k^p + \frac{\gamma}{2}\right)}$$

$$\lambda_k = a_{k+1}, \qquad\qquad \tau_k = c_{k+1}$$

to (41) and (42) with $p = 1$, we obtain

$$U^{k+1} - U^k + \left(k^p\left(k^p + \frac{\gamma}{2}\right) - \frac{k^p\gamma^2}{4\left(k^p + \frac{\gamma}{2}\right)}\right)\left\langle\tilde{\mathbf{A}}x^{k+1} - \tilde{\mathbf{A}}x^k, x^{k+1} - x^k\right\rangle$$

$$= -\gamma\left(k^p + \frac{\gamma}{2}\right)\|\tilde{\mathbf{A}}x^{k+1}\|^2 - \gamma^2\langle\tilde{\mathbf{A}}x^{k+1}, x^{k+1} - x^0\rangle - \frac{\gamma^3}{4\left(k^p + \frac{\gamma}{2}\right)}\left\|x^{k+1} - x^0\right\|^2$$

$$+ \underbrace{\left(\frac{k^{2p}\left(k^p + \frac{\gamma}{2}\right)}{k^p + \gamma} - (k-1)^p\left((k-1)^p + \frac{\gamma}{2}\right)\right)}_{=q_k}\|\tilde{\mathbf{A}}x^k\|^2 + \underbrace{\frac{\gamma\left(-(k-1)^pk^p + k^{2p} - \gamma(k-1)^p\right)}{k^p + \gamma}}_{=r_k}\langle\tilde{\mathbf{A}}x^k, x^k - x^0\rangle$$

$$- \frac{\gamma^3k^p}{4\left(k^p + \gamma\right)\left(k^p + \frac{\gamma}{2}\right)}\left\|x^k - x^0\right\|^2 + \frac{\gamma^3\left(k^p - (k-1)^p\right)}{8\left((k-1)^p + \frac{\gamma}{2}\right)\left(k^p + \frac{\gamma}{2}\right)}\left\|x^k - x^0\right\|^2$$

$$= -\gamma\left(k^p + \frac{\gamma}{2}\right)\left\|\tilde{\mathbf{A}}x^{k+1} + \frac{\gamma}{2k^p + \gamma}\left(x^{k+1} - x^0\right)\right\|^2 + q_k\left\|\tilde{\mathbf{A}}x^k + \frac{r_k}{2q_k}\left(x^k - x^0\right)\right\|^2$$

$$- \underbrace{\left(\frac{r_k^2}{4q_k} + \frac{\gamma^3k^p}{4\left(k^p + \gamma\right)\left(k^p + \frac{\gamma}{2}\right)} + \frac{\gamma^3\left(k^p - (k-1)^p\right)}{8\left((k-1)^p + \frac{\gamma}{2}\right)\left(k^p + \frac{\gamma}{2}\right)}\right)}_{s_k}\left\|x^k - x^0\right\|^2.$$

The continuous counterpart of this equality is

$$\dot{U}_2(t) + 2t^{2p}\left\langle\frac{d}{dt}\tilde{\mathbf{A}}(X(t)), \dot{X}(t)\right\rangle$$

$$= -2t^p\left(\gamma - pt^{p-1}\right)\left\|\tilde{\mathbf{A}}(X(t)) + \frac{\gamma}{t^p}\left(X(t) - X_0\right)\right\|^2 - \gamma p(p-1)t^{p-2}\left\|X(t) - X_0\right\|^2$$

which can be obtained by differentiating the conservation law for $E_2$. The first term on the right hand side correspond to the sum of first two terms in the right hand side of discrete equality. Thus we may expect the order of the coefficients for the terms would match, and we will check the expectation is indeed true.

With some calculation, we can observe

$$s_k = \frac{2\gamma^2(k-1)^pk^{2p}\left((k-1)^p - k^p\right)^2}{4\left((k-1)^p + \frac{\gamma}{2}\right)\left(k^p + \frac{\gamma}{2}\right)d_k},$$

where

$$d_k = 2k^p(k-1)^p\left((k-1)^p + \frac{\gamma}{2}\right) - 2k^{3p} - \gamma k^{2p} + 2\gamma(k-1)^p\left((k-1)^p + \frac{\gamma}{2}\right).$$

By considering Newton expansion and from $0 < p < 1$, we see

$$d_k = 2k^{3p} + \gamma k^{2p} + \mathcal{O}\left(k^{3p-1}\right) - 2k^{3p} - \gamma k^{2p} + 2\gamma k^{2p} + \mathcal{O}\left(k^p\right)$$

$$= 2\gamma k^{2p} + \mathcal{O}\left(k^{3p-1}\right) + \mathcal{O}\left(k^p\right) = \mathcal{O}\left(k^{2p}\right).$$

Thus we can check the leading order of numerator is $p + 2p + (2p - 2) = 5p - 2$, and leading order of the denominator is $p + p + 2p = 4p$. As $5p - 2 - 4p = p - 2$, we have $s_k \in \mathcal{O}\left(k^{p-2}\right)$, which matches with the continuous counterpart. Therefore $\sum_{k=1}^{\infty} s_k < \infty$.

On the other hand, we see

$$
\begin{aligned}
q_k &= \frac{k^{2p}\left(k^p + \frac{\gamma}{2}\right)}{k^p + \gamma} - (k-1)^p\left((k-1)^p + \frac{\gamma}{2}\right) \\
&\leq k^{2p} - \left(k^p - pk^{p-1} + \mathcal{O}\left(k^{p-2}\right)\right)\left(\left(k^p - pk^{p-1} + \mathcal{O}\left(k^{p-2}\right)\right) + \frac{\gamma}{2}\right) \\
&= -\frac{\gamma}{2}k^p + 2pk^{2p-1} + \mathcal{O}\left(k^{2p-2}\right).
\end{aligned}
$$

Since $p > 2p - 1$, we have $\lim_{k \to \infty} q_k = -\infty$. Note this matches with the continuous counterpart as well.

Therefore there is $N > 0$ such that for $k > N$, $q_k < 0$. Now for $k > N$ we have

$$
U_{k+1} \leq U_k + s_k \left\|x^k - x^0\right\|^2 \leq U_N + 4\left(\sum_{m=N}^{\infty} s_m\right)\left\|x^0 - x^\star\right\|^2 = M.
$$

Thus for $k > N$, by monotonicity of $\mathbb{A}$ and Young's inequality

$$
\begin{aligned}
M \geq U^{k+1} &= k^p\left(k^p + \frac{\gamma}{2}\right)\left\|\tilde{\mathbb{A}}x^{k+1}\right\|^2 + \gamma k^p\left\langle\tilde{\mathbb{A}}x^{k+1}, x^{k+1} - x^0\right\rangle + \frac{k^p\gamma^2}{4\left(k^p + \frac{\gamma}{2}\right)}\left\|x^{k+1} - x^0\right\|^2 \\
&\geq k^p\left(k^p + \frac{\gamma}{2}\right)\left\|\tilde{\mathbb{A}}x^{k+1}\right\|^2 + \gamma k^p\left\langle\tilde{\mathbb{A}}x^{k+1}, x^\star - x^0\right\rangle \\
&\geq k^p\left(k^p + \frac{\gamma}{2}\right)\left\|\tilde{\mathbb{A}}x^{k+1}\right\|^2 - \frac{k^p}{2}\left(\left(k^p + \frac{\gamma}{2}\right)\left\|\tilde{\mathbb{A}}x^{k+1}\right\|^2 + \frac{\gamma^2}{k^p + \frac{\gamma}{2}}\left\|x^\star - x^0\right\|^2\right) \\
&= \frac{k^p\left(k^p + \frac{\gamma}{2}\right)}{2}\left\|\tilde{\mathbb{A}}x^{k+1}\right\|^2 - \frac{k^p\gamma^2}{2k^p + \gamma}\left\|x^\star - x^0\right\|^2.
\end{aligned}
$$

Reorganizing, we get the desired result.

$$
\left\|\tilde{\mathbb{A}}x^{k+1}\right\|^2 \leq \frac{2}{k^p\left(k^p + \frac{\gamma}{2}\right)}\left(M + \frac{\gamma^2}{2}\left\|x^0 - x^\star\right\|^2\right) = \mathcal{O}\left(\frac{1}{k^{2p}}\right).
$$

(iii) $\left\|\tilde{\mathbb{A}}(x^k)\right\|^2 = \mathcal{O}\left(\frac{1}{k^2}\right)$ for $p = 1, \gamma \geq 1$.
Plugging

$$
\begin{aligned}
a_{k+1} &= k^2, & b_{k+1} &= \gamma\left(k - \frac{1}{2}(\gamma - 1)\right), & c_{k+1} &= \frac{1}{4}\gamma(\gamma - 1) \\
\lambda_k &= k(k+1), & \tau_k &= c_{k+1}
\end{aligned}
$$

to (41) and (42), we obtain

$$U^{k+1} - U^k + \left( k(k+1) - \frac{1}{4}\gamma(\gamma-1) \right) \langle \tilde{\mathbb{A}}x^{k+1} - \tilde{\mathbb{A}}x^k, x^{k+1} - x^k \rangle$$

$$= -(\gamma-1)(k+1)\|\tilde{\mathbb{A}}x^{k+1}\|^2 - \frac{k^2(\gamma-1)}{k+\gamma}\|\tilde{\mathbb{A}}x^k\|^2$$

$$- \frac{\gamma(\gamma-1)\left(k+\frac{\gamma}{4}\right)}{k}\langle \tilde{\mathbb{A}}x^{k+1}, x^{k+1} - x^0 \rangle - \frac{\gamma(\gamma-1)\left(k-\frac{\gamma}{4}\right)}{k+\gamma}\langle \tilde{\mathbb{A}}x^k, x^k - x^0 \rangle$$

$$- \frac{\gamma^2(\gamma-1)}{4k}\|x^{k+1} - x^0\|^2 - \frac{\gamma^2(\gamma-1)}{4(k+\gamma)}\|x^k - x^0\|^2$$

$$= -(\gamma-1)(k+1)\left\| \tilde{\mathbb{A}}x^{k+1} + \frac{\gamma}{2(k+1)}\left(1+\frac{\gamma}{4k}\right)(x^{k+1} - x^0) \right\|^2$$

$$- \frac{k^2(\gamma-1)}{k+\gamma}\left\| \tilde{\mathbb{A}}x^k + \frac{\gamma}{2k}\left(1-\frac{\gamma}{4k}\right)(x^k - x^0) \right\|^2$$

$$- \underbrace{\frac{\gamma^2(\gamma-1)}{64k(k+1)}\left(8(\gamma-2) - \frac{\gamma^2}{k}\right)}_{=s_{k,1}=\mathcal{O}\left(\frac{1}{k^2}\right)}\|x^{k+1} - x^0\|^2 - \underbrace{\frac{\gamma^3(\gamma-1)}{64k(k+\gamma)}\left(8 - \frac{\gamma}{k}\right)}_{=s_{k,0}=\mathcal{O}\left(\frac{1}{k^2}\right)}\|x^k - x^0\|^2 .$$

The continuous counterpart of above equality is

$$\dot{U}_3(t) + 2t^2 \left\langle \frac{d}{dt}\tilde{\mathbb{A}}(X(t)), \dot{X}(t) \right\rangle = -2t(\gamma-1)\left\| \tilde{\mathbb{A}}(X(t)) + \frac{\gamma}{t}(X(t) - X_0) \right\|^2$$

which can be obtained by differentiating the conservation law for $E_3$. Note the terms match with same order of coefficients, except two $\|x^{k+1} - x^0\|^2$ and $\|x^k - x^0\|^2$, while these terms are summable as $s_{k,1}, s_{k,0} = \mathcal{O}\left(\frac{1}{k^2}\right)$. Therefore we have

$$U^{k+1} \leq U^k - s_{k,1}\|x^{k+1} - x^0\|^2 - s_{k,0}\|x^k - x^0\|^2$$

$$\leq U^1 + 4\|x^0 - x^\star\|^2 \sum_{m=1}^{\infty}(s_{m,1} + s_{m,0}) = M.$$

Thus by monotonicity of $\mathbb{A}$ and Young's inequality

$$M \geq U^k = k^2\|\tilde{\mathbb{A}}x^{k+1}\|^2 + \gamma\left(k - \frac{1}{2}(\gamma-1)\right)\langle \tilde{\mathbb{A}}x^k, x^k - x^0 \rangle + \frac{1}{4}\gamma(\gamma-1)\|x^k - x^0\|^2$$

$$= k^2\|\tilde{\mathbb{A}}x^{k+1}\|^2 + \gamma\left(k - \frac{1}{2}(\gamma-1)\right)\langle \tilde{\mathbb{A}}x^k, x^\star - x^0 \rangle$$

$$\geq k^2\|\tilde{\mathbb{A}}x^k\|^2 - \frac{1}{2}\left( \left(k - \frac{1}{2}(\gamma-1)\right)^2\|\tilde{\mathbb{A}}x^k\|^2 + \gamma^2\|x^\star - x^0\|^2 \right)$$

$$\geq \frac{k^2}{2}\|\tilde{\mathbb{A}}x^k\|^2 - \frac{\gamma^2}{2}\|x^\star - x^0\|^2 .$$

Reorganizing, we get the desired result.

$$\|\tilde{\mathbb{A}}x^k\|^2 \leq \frac{2}{k^2}\left( M + \frac{\gamma^2}{2}\|x^0 - x^\star\|^2 \right) = \mathcal{O}\left(\frac{1}{k^2}\right).$$

(iv) $\left\|\tilde{\mathbb{A}}(x^k)\right\|^2 = \mathcal{O}\left(\frac{1}{k^{2\gamma}}\right)$ for $p = 1, 0 < \gamma < 1$.
Plugging

$$a_k = k^{2\gamma}, \qquad\qquad b_k = \gamma k\left(k - \frac{1}{4}\right)^{2\gamma-2}, \qquad\qquad c_{k+1} = \frac{1}{4}\gamma(\gamma-1)k^{2\gamma-2}$$

$$\lambda_k = k^{2\gamma-1}(k+\gamma), \qquad \tau_k = c_{k+1}$$

to (41) and (42) with $p = 1$, we obtain

$$U^{k+1} - U^k + \left( k^{2\gamma-1}(k+\gamma) + \frac{1}{4}\gamma(1-\gamma)k^{2\gamma-2} \right) \left\langle \tilde{\mathbf{A}}x^{k+1} - \tilde{\mathbf{A}}x^k, x^{k+1} - x^k \right\rangle$$

$$= \underbrace{\left( (k+1)^{2\gamma} - k^{2\gamma-2}(k+\gamma)^2 \right)}_{=q_k} \left\| \tilde{\mathbf{A}}x^{k+1} \right\|^2$$

$$+ \underbrace{\left( \gamma(k+1)\left(k+\frac{3}{4}\right)^{2\gamma-2} - \gamma k^{2\gamma-2}(k+\gamma) - \left(1 + \frac{k+\gamma}{k}\right)\frac{1}{4}\gamma(\gamma-1)k^{2\gamma-2} \right)}_{=s_{k,1}} \left\langle \tilde{\mathbf{A}}x^{k+1}, x^{k+1} - x^0 \right\rangle$$

$$+ \underbrace{\left( \gamma k^{2\gamma-1} - \gamma k\left(k-\frac{1}{4}\right)^{2\gamma-2} - \left(1 + \frac{k}{k+\gamma}\right)\frac{1}{4}\gamma(\gamma-1)k^{2\gamma-2} \right)}_{=s_{k,0}} \left\langle \tilde{\mathbf{A}}x^k, x^k - x^0 \right\rangle$$

$$- \frac{1}{4}\gamma^2(\gamma-1)k^{2\gamma-3} \left\| x^{k+1} - x^0 \right\|^2 - \frac{1}{4}\gamma^2(\gamma-1)k^{2\gamma-3}\frac{1}{1+\frac{\gamma}{k}} \left\| x^k - x^0 \right\|^2$$

$$+ \frac{1}{4}\gamma(\gamma-1)\left( k^{2\gamma-2} - (k-1)^{2\gamma-2} \right) \left\| x^k - x^0 \right\|^2$$

The continuous counterpart of this equality is

$$\dot{U}_4(t) + 2t^{2\gamma} \left\langle \frac{d}{dt}\tilde{\mathbf{A}}(X(t)), \dot{X}(t) \right\rangle ds = -2\gamma(\gamma-1)t^{2\gamma-3} \left\| X(t) - X_0 \right\|^2$$

which can be obtained by differentiating the conservation law for $E_4$. Thus we may expect the order of the matching terms are equal, and the terms do not occur in the continuous version do not bother our desired conclusion. We check our expectation is true.

Terms $\left\| x^{k+1} - x^0 \right\|^2$ and $\left\| x^k - x^0 \right\|^2$ correspond to $\left\| X - X_0 \right\|^2$. The coefficients for $\left\| x^{k+1} - x^0 \right\|^2$ and $\left\| x^k - x^0 \right\|^2$ are clearly $\mathcal{O}\left(k^{2\gamma-3}\right)$, which equals the order of continuous counterpart. Since $\gamma < 1$, we know these terms are summable.

Next we observe $q_k \leq 0$. Observe

$$q_k \leq 0 \iff \frac{(k+1)^{2\gamma}}{k^{2\gamma}} \leq \frac{(k+\gamma)^2}{k^2}$$

$$\iff \left(1 + \frac{1}{k}\right)^{2\gamma} \leq \left(1 + \frac{\gamma}{k}\right)^2 \iff \left(1 + \frac{1}{k}\right)^\gamma \leq 1 + \frac{\gamma}{k}.$$

To check the last inequality is true, consider $f(x) = x^\gamma$. Since this function is concave for $0 < \gamma < 1$, we see

$$\left(1 + \frac{1}{k}\right)^\gamma = f\left(1 + \frac{1}{k}\right) \leq f(1) + \frac{1}{k}f'(1) = 1 + \frac{\gamma}{k}.$$

Finally we focus on $s_{k,0}$ and $s_{k,1}$. As cross terms don't appear in continuous version, we may expect these terms are 'small', or in mathematical words, summable. From Cauchy-Schwarz inequality we know

$$\left\langle \tilde{\mathbf{A}}x^{k+1}, x^{k+1} - x^0 \right\rangle \leq \left\| \tilde{\mathbf{A}}x^{k+1} \right\| \left\| x^{k+1} - x^0 \right\|$$

$$\left\langle \tilde{\mathbf{A}}x^k, x^k - x^0 \right\rangle \leq \left\| \tilde{\mathbf{A}}x^k \right\| \left\| x^k - x^0 \right\|.$$

Since $\left\| \tilde{\mathbf{A}}x^{k+1} \right\|$ and $\left\| \tilde{\mathbf{A}}x^k \right\|$ are bounded from the proof for case (i), we know two innerproduct terms are bounded. Thus if we show $\sum_{k=1}^\infty |s_{k,0}|, \sum_{k=1}^\infty |s_{k,1}| < \infty$, we can conclude $U^k$ is bounded.
Considering Newton expansion, we see

$$\gamma(k+1)\left(k + \frac{3}{4}\right)^{2\gamma-2} - \gamma k^{2\gamma-2}(k+\gamma)$$

$$= \gamma k^{2\gamma-1} + \gamma\left(1 + \frac{3}{2}(\gamma-1)\right)k^{2\gamma-2} + \mathcal{O}\left(k^{2\gamma-3}\right) - \gamma k^{2\gamma-1} - \gamma^2 k^{2\gamma-2}$$

$$= \frac{1}{2}\gamma(\gamma-1)k^{2\gamma-2} + \mathcal{O}\left(k^{2\gamma-3}\right)$$

Therefore

$$s_{k,1} = \frac{1}{2}\gamma(\gamma-1)k^{2\gamma-2} + \mathcal{O}\left(k^{2\gamma-3}\right) - \left(2+\frac{\gamma}{k}\right)\frac{1}{4}\gamma(\gamma-1)k^{2\gamma-2} = \mathcal{O}\left(k^{2\gamma-3}\right).$$

With similar argument

$$
\begin{aligned}
s_{k,0} &= \gamma k^{2\gamma-1} - \gamma k\left(k-\frac{1}{4}\right)^{2\gamma-2} - \left(1+\frac{k}{k+\gamma}\right)\frac{1}{4}\gamma(\gamma-1)k^{2\gamma-2} \\
&= \gamma k^{2\gamma-1} - \gamma k\left(k^{2\gamma-2} - (2\gamma-2)\frac{1}{4}k^{2\gamma-3} + \mathcal{O}\left(k^{2\gamma-4}\right)\right) - \left(2-\frac{\gamma}{k+\gamma}\right)\frac{1}{4}\gamma(\gamma-1)k^{2\gamma-2} \\
&= \frac{1}{2}\gamma(\gamma-1)k^{2\gamma-2} + \mathcal{O}\left(k^{2\gamma-3}\right) - \left(\frac{1}{2}\gamma(\gamma-1)k^{2\gamma-2} + \mathcal{O}\left(k^{2\gamma-3}\right)\right) = \mathcal{O}\left(k^{2\gamma-3}\right)
\end{aligned}
$$

Therefore, we have

$$
\begin{aligned}
U^{k+1} &\le U^k + s_{k,1}\left\|\tilde{\mathbf{A}}x^{k+1}\right\|\left\|x^{k+1}-x^0\right\| + s_{k,0}\left\|\tilde{\mathbf{A}}x^k\right\|\left\|x^k-x^0\right\| \\
&\quad - c_{k+1}\frac{\gamma}{k}\left\|x^{k+1}-x^0\right\|^2 - c_{k+1}\frac{\gamma}{k+\gamma}\left\|x^k-x^0\right\|^2 + (c_{k+1}-c_k)\left\|x^k-x^0\right\|^2 \\
&\le \sum_{m=1}^{\infty}\left(|s_{m,1}|\left\|\tilde{\mathbf{A}}x^{k+1}\right\|\left\|x^{k+1}-x^0\right\| + |s_{m,0}|\left\|\tilde{\mathbf{A}}x^{k+1}\right\|\left\|x^{k+1}-x^0\right\|\right) \\
&\quad + \sum_{m=1}^{\infty}\left(\left|c_{m+1}\frac{\gamma}{m}\right|\left\|x^{k+1}-x^0\right\|^2 + \left(\left|c_{m+1}\frac{\gamma}{m+\gamma}\right| + |c_{k+1}-c_k|\right)\left\|x^k-x^0\right\|^2\right) = M_1.
\end{aligned}
$$

By Young's inequality

$$
\begin{aligned}
M_1 + \frac{1}{4}\gamma(1-\gamma)k^{2\gamma-2}\left\|x^k-x^0\right\|^2 &\ge k^{2\gamma}\|\tilde{\mathbf{A}}x^k\|^2 + \gamma k\left(k-\frac{1}{4}\right)^{2\gamma-2}\langle\tilde{\mathbf{A}}x^k, x^k-x^0\rangle \\
&\ge k^{2\gamma}\|\tilde{\mathbf{A}}x^k\|^2 + \gamma k\left(k-\frac{1}{4}\right)^{2\gamma-2}\langle\tilde{\mathbf{A}}x^k, x^\star-x^0\rangle \\
&= k^{2\gamma}\|\tilde{\mathbf{A}}x^k\|^2 + \left\langle k^\gamma\tilde{\mathbf{A}}x^k, \gamma k^{1-\gamma}\left(k-\frac{1}{4}\right)^{2\gamma-2}(x^\star-x^0)\right\rangle \\
&\ge k^{2\gamma}\|\tilde{\mathbf{A}}x^k\|^2 - \frac{1}{2}k^{2\gamma}\|\tilde{\mathbf{A}}x^k\|^2 - \frac{\gamma^2}{2}\left(k^{1-\gamma}\left(k-\frac{1}{4}\right)^{2\gamma-2}\right)^2\left\|x^\star-x^0\right\|^2.
\end{aligned}
$$

Since $k^{1-\gamma}\left(k-\frac{1}{4}\right)^{2\gamma-2} = \mathcal{O}\left(k^{\gamma-1}\right)$ and $\gamma < 1$, this terms goes to zero as $k\to\infty$ thus there is some $M_2 > 0$ such that $\frac{1}{2}\left(\gamma k^{1-\gamma}\left(k-\frac{1}{4}\right)^{2\gamma-2}\right)^2\left\|x^k-x^\star\right\|^2 \le M_2$ for all $k\ge 1$. Reorganizng terms, we obtain the desired result

$$\|\tilde{\mathbf{A}}x^k\|^2 \le \frac{2}{k^{2\gamma}}\left(M_1 + M_2 + \gamma(1-\gamma)k^{2\gamma-2}\left\|x^\star-x^0\right\|^2\right) = \mathcal{O}\left(\frac{1}{k^{2\gamma}}\right).$$

# H   Proof of convergence analysis for strongly monotone $\mathbb{A}$

## H.1   Proof of Proposition 6.1

We take dilated coordinate $W(t) = C(t)(X(t) - X_0)$ as did in the proof of Proposition 3.2. Recall from (30), the second order version of the ODE was written as $0 = \ddot{W} - \beta(t)\dot{W} + C(t)\frac{d}{dt}\tilde{\mathbb{A}}(X(t))$. Now for we multiply $R(t)^2$ in the ODE and obtain

$$0 = R(t)^2\left(\ddot{W} - \beta(t)\dot{W} + C(t)\frac{d}{dt}\tilde{\mathbb{A}}(X)\right).$$

Now taking inner product with $\dot{W}$ and integrating we have

$$E_1 \equiv \frac{R(t)^2}{2} \left\| \dot{W}(t) \right\|^2 - \underbrace{\int_{t_0}^{t} R(s)^2 \left( \frac{\dot{R}(s)}{R(s)} \left\| \dot{W}(s) \right\|^2 \right) ds}_{*} - \int_{t_0}^{t} \beta(s) R(s)^2 \left\| \dot{W}(s) \right\|^2 ds$$

$$+ \int_{t_0}^{t} C(s) R(s)^2 \left\langle \frac{d}{ds} \tilde{\mathbb{A}}(X(s)), \dot{W}(s) \right\rangle ds.$$

Note the second term which is obtained from integration by parts, would not appear if $R(t) = 1$ as in Proposition 3.2. This is key term to exploit the condition $\mathbb{A}$ is strongly monotone.

Now again with $\dot{W}(t) = C(t)\left( \dot{X}(t) + \beta(s)(X(t) - X_0) \right)$, we rewrite the last term as

$$\int_{t_0}^{t} C(s) R(s)^2 \left\langle \frac{d}{ds} \tilde{\mathbb{A}}(X(s)), \dot{W}(s) \right\rangle ds$$

$$= \int_{t_0}^{t} C(s)^2 R(s)^2 \left\langle \frac{d}{ds} \tilde{\mathbb{A}}(X(s)), \dot{X}(s) \right\rangle ds + \int_{t_0}^{t} \left\langle \frac{d}{ds} \tilde{\mathbb{A}}(X(s)), C(s) R(s)^2 \beta(s) W(s) \right\rangle ds.$$

Taking integration by parts to the second term we have

$$\int_{t_0}^{t} \left\langle \frac{d}{ds} \tilde{\mathbb{A}}(X(s)), C(s) R(s)^2 \beta(s) W(s) \right\rangle ds - \left[ \left\langle \tilde{\mathbb{A}}(X(W(s), s)), C(s) R(s)^2 \beta(s) W(s) \right\rangle \right]_{t_0}^{t}$$

$$= -\int_{t_0}^{t} \left\langle \tilde{\mathbb{A}}(X(W, t)), \left( \beta(s)^2 + 2 \frac{\dot{R}(s)}{R(s)} \beta(s) + \dot{\beta}(s) \right) C(s) R(s)^2 W(s) + C(s) R(s)^2 \beta(s) \dot{W}(s) \right\rangle ds$$

$$= \int_{t_0}^{t} \beta(s) R(s)^2 \left\| \dot{W}(s) \right\|^2 ds + \int_{t_0}^{t} \left( \beta(s)^2 + \underbrace{2 \frac{\dot{R}(s)}{R(s)} \beta(s)}_{*} + \dot{\beta}(s) \right) R(s)^2 \left\langle \dot{W}(s), W(s) \right\rangle ds.$$

The fact $C(t)\tilde{\mathbb{A}}(X(W, t)) = -\dot{W}(t)$ is applied to the second equality. Note the fundamental theorem of calculus is valid since $C(s) R(s)^2 \beta(s) W(s)$ is differentiable, $\tilde{\mathbb{A}}(X(W(s), t))$ is Lipschitz continuous in $[t_0, t]$ by Lemma B.4, and so their inner product is absolutely continuous in $[t_0, t]$.

Now consider the second integrand except the term marked with *. From integration by parts we have

$$\int_{t_0}^{t} \left( \beta(s)^2 + \dot{\beta}(s) \right) R(s)^2 \left\langle \dot{W}(s), W(s) \right\rangle ds - \left[ \left( \beta(s)^2 + \dot{\beta}(s) \right) R(s)^2 \frac{1}{2} \| W(s) \|^2 \right]_{t_0}^{t}$$

$$= -\frac{1}{2} \int_{t_0}^{t} \left( \left( 2\beta(s)\dot{\beta}(s) + \ddot{\beta}(s) \right) R(s)^2 + \left( \underbrace{\beta(s)^2}_{*} + \dot{\beta}(s) \right) 2 R(s) \dot{R}(s) \right) \| W(s) \|^2 ds.$$

The integrand except * marked term can be rewritten as

$$\frac{1}{2} \int_{t_0}^{t} \left( 2\beta(s)\dot{\beta}(s) R(s) + \ddot{\beta}(s) R(s) + 2\dot{\beta}(s) \dot{R}(s) \right) R(s) \| W(s) \|^2 ds$$

$$= \frac{1}{2} \int_{t_0}^{t} C(s)^2 \left( 2\beta(s)\dot{\beta}(s) R(s) + \ddot{\beta}(s) R(s) + 2\dot{\beta}(s) \dot{R}(s) \right) R(s) \| X(s) - X_0 \|^2 ds$$

$$= \frac{1}{2} \int_{t_0}^{t} \frac{d}{ds} \left( C(s)^2 R(s)^2 \dot{\beta}(s) \right) \| X(s) - X_0 \|^2 ds.$$

Now collecting the terms marked with *, we have

$$-\int_{t_0}^{t} R(s)^2 \left( \frac{\dot{R}(s)}{R(s)} \left\| \dot{W}(s) \right\|^2 \right) ds + \int_{t_0}^{t} 2 \frac{\dot{R}(s)}{R(s)} \beta(s) R(s)^2 \left\langle \dot{W}(s), W(s) \right\rangle ds - \int_{t_0}^{t} \beta(s)^2 R(s) \dot{R}(s) \| W(s) \|^2 ds$$

$$= -\int_{t_0}^{t} R(s)^2 \frac{\dot{R}(s)}{R(s)} \left\| \dot{W}(s) - \beta(s) W(s) \right\|^2 ds = -\int_{t_0}^{t} R(s)^2 C(s)^2 \frac{\dot{R}(s)}{R(s)} \left\| \dot{X}(s) \right\|^2 ds.$$

Collecting all results we have

$$
E_1 \equiv \frac{R(t)^2}{2} \left\| \dot{W}(t) \right\|^2 + \left[ \langle \tilde{\mathbf{A}}(X(W,s)), C(s)R(s)^2\beta(s)W(s) \rangle \right]_{t_0}^t + \left[ \left( \beta(s)^2 + \dot{\beta}(s) \right) R(s)^2 \frac{1}{2} \| W(s) \|^2 \right]_{t_0}^t
$$

$$
+ \int_{t_0}^t C(s)^2 R(s)^2 \left\langle \frac{d}{ds} \tilde{\mathbf{A}}(X(s)), \dot{X}(s) \right\rangle ds - \int_{t_0}^t R(s)^2 C(s)^2 \frac{\dot{R}(s)}{R(s)} \left\| \dot{X}(s) \right\|^2 ds
$$

$$
+ \frac{1}{2} \int_{t_0}^t \frac{d}{ds} \left( C(s)^2 R(s)^2 \dot{\beta}(s) \right) \| X(s) - X_0 \|^2 ds
$$

$$
= \frac{C(t)^2 R(t)^2}{2} \left( \| \mathbf{A}(X(t)) \|^2 + 2\beta(t) \langle \mathbf{A}(X(t)), X(t) - X_0 \rangle + \left( \beta(t)^2 + \dot{\beta}(t) \right) \| X(t) - X_0 \|^2 \right)
$$

$$
+ \int_{t_0}^t C(s)^2 R(s)^2 \left( \left\langle \frac{d}{ds} \tilde{\mathbf{A}}(X(s)), \dot{X}(s) \right\rangle - \frac{\dot{R}(s)}{R(s)} \left\| \dot{X}(s) \right\|^2 \right) ds - \int_{t_0}^t \frac{d}{ds} \left( \frac{C(s)^2 R(s)^2 \dot{\beta}(s)}{2} \right) \| X(s) - X_0 \|^2 ds
$$

$$
\underbrace{- \frac{C(t_0)^2 R(t_0)^2}{2} \left( 2\beta(t_0) \langle \mathbf{A}(X(t_0)), X(t_0) - X_0 \rangle + \left( \beta(t_0)^2 + \dot{\beta}(t_0) \right) \| X(t_0) - X_0 \|^2 \right)}_{\text{constant}}.
$$

Renaming $E = E_1 - \text{constant}$, we get the desired result.

## H.2 Proof of Theorem 6.2

### H.2.1 Proof of the inequality $V(t) \leq V(0)$

The basic structure of the proof is same as Appendix E.3. We do not repeat the whole proof here, instead we check the steps done in Appendix E.3 are also valid for the setup in Theorem 6.2.

(i) $V$ is nonincreasing for Lipschitz continuous $\mu$-strongly monotone $\tilde{\mathbf{A}}$.

Recall $V$ in Theorem 6.2 was defined as

$$
V(t) = \frac{(e^{\mu t} - e^{-\mu t})^2}{2} \left\| \tilde{\mathbf{A}}(X(t)) \right\|^2 + 2\mu(1 - e^{-2\mu t}) \left\langle \tilde{\mathbf{A}}(X(t)), X(t) - X_0 \right\rangle - 2\mu^2 \left( 1 - e^{-2\mu t} \right) \| X(t) - X_0 \|^2. \tag{47}
$$

We first check following equality is true.

$$
V(t) = \frac{C(t)^2 R(t)^2}{2} \left( \left\| \tilde{\mathbf{A}}(X(t)) \right\|^2 + 2\beta(t) \left\langle \tilde{\mathbf{A}}(X(t)), X(t) - X_0 \right\rangle + \left( \beta(t)^2 + \dot{\beta}(t) \right) \| X(t) - X_0 \|^2 \right). \tag{48}
$$

Recall we're considering (10), $\beta(t) = \frac{2\mu}{e^{2\mu t} - 1}$. As $\frac{d}{dt} \log \left( 1 - e^{-2\mu t} \right) = \frac{2\mu e^{-2\mu t}}{1 - e^{-2\mu t}} = \frac{2\mu}{e^{2\mu t} - 1} = \beta(t)$, we have

$$
C(t) = e^{\int_\infty^t \frac{2\mu}{e^{2\mu s} - 1} ds} = e^{\log\left(1 - e^{-2\mu t}\right)} = 1 - e^{-2\mu t}.
$$

As $\dot{\beta}(t) = -\frac{4\mu^2 e^{2\mu t}}{(e^{2\mu t} - 1)^2}$ and $R(t) = e^{\mu t}$,

$$
\frac{C(t)^2 R(t)^2}{2} = \frac{1}{2} \left( 1 - e^{-2\mu t} \right)^2 e^{2\mu t} = \frac{(e^{\mu t} - e^{-\mu t})^2}{2}
$$

$$
C(t)^2 R(t)^2 \beta(t) = e^{-2\mu t} \left( e^{2\mu t} - 1 \right)^2 \frac{2\mu}{e^{2\mu t} - 1} = 2\mu \left( 1 - e^{-2\mu t} \right)
$$

$$
\frac{C(t)^2 R(t)^2}{2} \left( \beta(t)^2 + \dot{\beta}(t) \right) = \frac{(e^{\mu t} - e^{-\mu t})^2}{2} \left( \left( \frac{2\mu}{e^{2\mu t} - 1} \right)^2 - \frac{4\mu^2 e^{2\mu t}}{(e^{2\mu t} - 1)^2} \right) = 2\mu^2 \left( e^{-2\mu t} - 1 \right).
$$

This proves the desired equality. Now we show

$$
V(t) = E - \int_{t_0}^t C(s)^2 R(s)^2 \left( \left\langle \frac{d}{ds} \tilde{\mathbf{A}}(X(s)), \dot{X}(s) \right\rangle - \frac{\dot{R}(s)}{R(s)} \left\| \dot{X}(s) \right\|^2 \right) ds, \tag{49}
$$

where $E$ is from Proposition 6.1 which was defined as

$$E = \frac{C(t)^2 R(t)^2}{2} \left( \left\| \tilde{\mathbb{A}}(X(t)) \right\|^2 + 2\beta(t) \left\langle \tilde{\mathbb{A}}(X(t)), X(t) - X_0 \right\rangle + \left( \beta(t)^2 + \dot{\beta}(t) \right) \|X(t) - X_0\|^2 \right)$$

$$+ \int_{t_0}^t C(s)^2 R(s)^2 \left( \left\langle \frac{d}{ds} \tilde{\mathbb{A}}(X(s)), \dot{X}(s) \right\rangle - \frac{\dot{R}(s)}{R(s)} \left\| \dot{X}(s) \right\|^2 \right) ds - \int_{t_0}^t \frac{d}{ds} \left( \frac{C(s)^2 R(s)^2 \dot{\beta}(s)}{2} \right) \|X(t) - X_0\|^2 \, ds.$$

From (48) and the definition of $E$, it is enough to show $\frac{d}{ds} \left( \frac{C(s)^2 R(s)^2 \dot{\beta}(s)}{2} \right) = 0$. Since

$$\frac{C^2(t) R^2(t) \dot{\beta}(t)}{2} = \left( 1 - e^{-2\mu t} \right)^2 e^{2\mu t} \left( -\frac{4\mu^2 e^{2\mu t}}{(e^{2\mu t} - 1)^2} \right) = -4\mu^2,$$

we see $\frac{d}{ds} \left( \frac{C(s)^2 R(s)^2 \dot{\beta}(s)}{2} \right) = 0$.

Now since $\tilde{\mathbb{A}}$ is Lipschitz continuous, we know $E$ is constant from Proposition 6.1. Therefore from (49), for $t > 0$, $|h| < t$ we have

$$V(t + h) - V(t) = \int_t^{t+h} C(s)^2 R(s)^2 \left( \left\langle \frac{d}{ds} \tilde{\mathbb{A}}(X(s)), \dot{X}(s) \right\rangle - \frac{\dot{R}(s)}{R(s)} \left\| \dot{X}(s) \right\|^2 \right) ds.$$

As $\frac{\dot{R}(s)}{R(s)} = \mu$ and $\tilde{\mathbb{A}}$ is $\mu$-strongly monotone, from (2) we see

$$\left\langle \frac{d}{ds} \tilde{\mathbb{A}}(X(s)), \dot{X}(s) \right\rangle - \frac{\dot{R}(s)}{R(s)} \left\| \dot{X}(s) \right\|^2 \geq 0.$$

Therefore $V(t + h) - V(t) \geq 0$ for $h > 0$, we get the desired result.

(ii) Calculation of $V(0)$ for Lipshitz continuous monotone $\tilde{\mathbb{A}}$
Plugging $t = 0$ to (47) we immediately obtain $V(0) = 0$.

(iii) $V(t) \leq 0$ holds for all $t \in [0, \infty)$ and general maximal $\mu$-strongly monotone $\mathbb{A}$
We check the arguments in Appendix E.3.3 and Appendix E.3.4 are also valid here.
Define $S$ as defined in Lemma E.1. Take $t \in S$, let $T > t$. As checked in Appendix C, the arguments used in the proof Proposition B.11 is also valid for the case $\beta(t) = \frac{2\mu}{e^{2\mu t} - 1}$. This fact provides the required sequence $\{X_{\lambda_n}\}_{n \in \mathbb{N}}$, where $X_{\lambda_n}$ converges to $X$ uniformly on $[0, T]$, and $\dot{X}_{\lambda_n}$ converges weakly to $\dot{X}$ in $L^2([0, T], \mathbb{R}^n)$. As in Appendix E.3.3, denote $V_\lambda$ as $V$ for the solution with $\mathbb{A}_\lambda$. Then from (i) we know $V_{\lambda_n}$ is nonincreasing for all $n \in \mathbb{N}$, we have $\limsup_{n \to \infty} V_{\lambda_n}(t) \leq \limsup_{n \to \infty} V_{\lambda_n}(0) = 0$.

Moreover, we can check the extension of $\tilde{\mathbb{A}}$ defined in Lemma E.1 is also valid. From (21) and (22), we know $\left\| \dot{X}_\lambda(t) \right\| \leq \frac{e^{\mu T}}{\sqrt{2}} \|m(\mathbb{A}(X_0))\|$ and $\|\mathbb{A}_\lambda(X_\lambda(t))\| \leq \sqrt{e^{2\mu T} + 1} \|m(\mathbb{A}(X_0))\|$ for all $\lambda > 0, t \in [0, T]$. Therefore we can prove Corollary E.2 for the case $\beta(t) = \frac{2\mu}{e^{2\mu t} - 1}$ with the same proof, replacing $M_{\mathbb{A}}(T)$ by $\sqrt{e^{2\mu T} + 1} \|m(\mathbb{A}(X_0))\|$ and $M_{dot}(T)$ by $\frac{e^{\mu T}}{\sqrt{2}} \|m(\mathbb{A}(X_0))\|$. Thus $\tilde{\mathbb{A}}(X(t))$ is well-defined for $t = 0$, plugging $t = 0$ to (47) we obtain $V(0) = 0$.
Therefore we have

$$\limsup_{n \to \infty} V_{\lambda_n}(t) \leq \limsup_{n \to \infty} V_{\lambda_n}(0) = 0 = V(0),$$

it remains to show $V(t) \leq \limsup_{n \to \infty} V_{\lambda_n}(t)$. Observe, as equality $\dot{X}(t) = -\tilde{\mathbb{A}}(X(t)) - \beta(t)(X(t) - X_0)$ holds since $t \in S$, from (48) we have

$$V(t) = \frac{C(t)^2 R(t)^2}{2} \left( \left\| \dot{X}(t) \right\|^2 + \dot{\beta}(t) \|X(t) - X_0\|^2 \right).$$

Therefore if is suffices to check Lemma E.4 is valid here with some $U_{\lambda_n}$. For some $a > 0$, Define $U_{\lambda_n} : [0, \infty) \to \mathbb{R}$ as

$$U_{\lambda_n}(t) = \underbrace{\left\| \dot{X}_{\lambda_n}(t) \right\|^2 - \dot{\beta}(t) \|X_{\lambda_n}(t) - X_0\|^2 + \int_a^t \left( \ddot{\beta}(s) - \frac{2\dot{\beta}(s)^2}{\beta(s)} \right) \|X_{\lambda_n}(s) - X_0\|^2 \, ds}_{f_n(t)}.$$

We proceed similar argument with Lemma B.5. Differentiating $\dot{X}_{\lambda_n}(t) = -\mathbb{A}_{\lambda_n}(X(t)) - \beta(t)(X_{\lambda_n}(t) - X_0)$, we have for almost all $t > 0$

$$\ddot{X}_{\lambda_n}(t) = -\frac{d}{dt}\mathbb{A}_{\lambda_n}(X(t)) - \dot{\beta}(t)(X_{\lambda_n}(t) - X_0) - \beta(t)\dot{X}_{\lambda_n}(t).$$

Therefore for almost all $t > 0$,

$$\dot{U}_{\lambda_n}(t)$$

$$= 2\left\langle \dot{X}_{\lambda_n}(t), \ddot{X}_{\lambda_n}(t) \right\rangle - \ddot{\beta}(t)\left\| X_{\lambda_n}(t) - X_0 \right\|^2 - 2\dot{\beta}(t)\left\langle \dot{X}_{\lambda_n}(t), X_{\lambda_n}(t) - X_0 \right\rangle + \left( \ddot{\beta}(t) - \frac{2\dot{\beta}(t)^2}{\beta(t)} \right)\left\| X_{\lambda_n}(t) - X_0 \right\|^2$$

$$= 2\left\langle \dot{X}_{\lambda_n}(t), -\frac{d}{dt}\mathbb{A}_{\lambda_n}(X(t)) - \dot{\beta}(t)(X_{\lambda_n}(t) - X_0) - \beta(t)\dot{X}(t) \right\rangle$$

$$\quad - 2\dot{\beta}(t)\left\langle \dot{X}_{\lambda_n}(t), X_{\lambda_n}(t) - X_0 \right\rangle - \frac{2\dot{\beta}(t)^2}{\beta(t)}\left\| X_{\lambda_n}(t) - X_0 \right\|^2$$

$$= -2\left\langle \dot{X}_{\lambda_n}(t), \frac{d}{dt}\mathbb{A}_{\lambda_n}(X(t)) \right\rangle - 2\beta(t)\left\| \dot{X}_{\lambda_n}(t) + \frac{\dot{\beta}(t)}{\beta(t)}(X_{\lambda_n}(t) - X_0) \right\|^2 \le 0.$$

Therefore $U_{\lambda_n}$ is nonincreasing, we can prove $\left\| \dot{X}(t) \right\| \le \limsup_{n\to\infty}\left\| \dot{X}_{\lambda_n}(t) \right\|^2$ with same argument in Lemma E.4.

Therefore we have $V(t) \le V(0) = 0$ for $t \in S$. Extending the result to $t \in [0, \infty)$ can be done with the same argument done in Appendix E.3.4.

### H.2.2 Proof for convergence rate

Recall

$$V(t) = \frac{(e^{\mu t} - e^{-\mu t})^2}{2}\left\| \tilde{\mathbb{A}}(X(t)) \right\|^2 + 2\mu(1 - e^{-2\mu t})\left\langle \tilde{\mathbb{A}}(X(t)), X(t) - X_0 \right\rangle - 2\mu^2\left(1 - e^{-2\mu t}\right)\left\| X(t) - X_0 \right\|^2.$$

Observe

$$\frac{2V(t)}{1 - e^{-2\mu t}} = (e^{2\mu t} - 1)\|\tilde{\mathbb{A}}(X(t))\|^2 + 4\mu\left\langle \tilde{\mathbb{A}}(X(t)), X(t) - X_0 \right\rangle - 4\mu^2\left\| X(t) - X_0 \right\|^2$$

$$= (e^{\mu t} - 1)\underbrace{\left( \|\tilde{\mathbb{A}}(X(t))\|^2 - 4\mu\left\langle \tilde{\mathbb{A}}(X(t)), X(t) - X_0 \right\rangle + 4\mu^2\left\| X(t) - X_0 \right\|^2 \right)}_{= \left\| \tilde{\mathbb{A}}(X(t)) - 2\mu(X(t) - X_0) \right\|^2}$$

$$\quad + e^{\mu t}(e^{\mu t} - 1)\|\tilde{\mathbb{A}}(X(t))\|^2 + \underbrace{e^{\mu t}\left( 4\mu\left\langle \tilde{\mathbb{A}}(X(t)), X(t) - X_0 \right\rangle - 4\mu^2\left\| X(t) - X_0 \right\|^2 \right)}_{=p(t)}.$$

From the law of cosines, we have $\|X(t) - X_0\|^2 = \|X(t) - X_\star\|^2 - 2\left\langle X(t) - X_0, X_0 - X_\star \right\rangle - \|X_\star - X_0\|^2$. Applying this to $p(t)$ we have

$$p(t) = 4\mu e^{\mu t}\left( \left\langle \tilde{\mathbb{A}}(X(t)), X(t) - X_\star \right\rangle - \left\langle \tilde{\mathbb{A}}(X(t)), X_0 - X_\star \right\rangle - \mu\|X(t) - X_0\|^2 \right)$$

$$= 4\mu e^{\mu t}\left( \left\langle \tilde{\mathbb{A}}(X(t)) + \mu(X(t) - X_\star), X(t) - X_\star \right\rangle - \left\langle \tilde{\mathbb{A}}(X(t)) - 2\mu(X(t) - X_0), X_0 - X_\star \right\rangle - \mu\|X_0 - X_\star\|^2. \right)$$

Thus

$$\frac{2V(t)}{1 - e^{-2\mu t}} = (e^{\mu t} - 1)\left\| \tilde{\mathbb{A}}(X(t)) - 2\mu(X(t) - X_0) \right\|^2 + e^{\mu t}(e^{\mu t} - 1)\|\tilde{\mathbb{A}}(X(t))\|^2 + p(t)$$

$$= (e^{\mu t} - 1)\|\tilde{\mathbb{A}}(X(t)) - 2\mu(X(t) - X_0)\|^2 - 4\mu e^{\mu t}\langle \tilde{\mathbb{A}}(X(t)) - 2\mu(X(t) - X_0), X_0 - X_\star\rangle$$

$$\quad + e^{\mu t}(e^{\mu t} - 1)\|\tilde{\mathbb{A}}(X(t))\|^2 + 4\mu e^{\mu t}\langle \tilde{\mathbb{A}}(X(t)) - \mu(X(t) - X_\star), X(t) - X_\star\rangle + 4\mu^2 e^{\mu t}\|X_0 - X_\star\|^2$$

$$= (e^{\mu t} - 1)\left\| \tilde{\mathbb{A}}(X(t)) - 2\mu(X(t) - X_0) - \frac{2\mu e^{\mu t}}{e^{\mu t} - 1}(X_0 - X_\star) \right\|^2 - \frac{4\mu^2 e^{2\mu t}}{e^{\mu t} - 1}\|X_0 - X_\star\|^2$$

$$\quad + e^{\mu t}(e^{\mu t} - 1)\|\tilde{\mathbb{A}}(X(t))\|^2 + 4\mu e^{\mu t}\left( \left\langle \tilde{\mathbb{A}}(X(t)), X(t) - X_\star \right\rangle - \mu\left\| X(t) - X_\star \right\|^2 \right) + 4\mu^2 e^{\mu t}\|X_0 - X_\star\|^2.$$

Now since $\tilde{\mathbb{A}}$ is $\mu$-strongly monotone, $\langle \tilde{\mathbb{A}}(X(t)), X(t) - X_\star \rangle - \mu \|X(t) - X_\star\|^2 \geq 0$. From previous section we know $0 = V(0) \geq V(t)$ for all $t > 0$. Therefore for all $t > 0$

$$0 \geq \frac{2V(t)}{1 - e^{-2\mu t}} \geq e^{\mu t}(e^{\mu t} - 1)\|\tilde{\mathbb{A}}(X(t))\|^2 + \left(4\mu^2 e^{\mu t} - \frac{4\mu^2 e^{2\mu t}}{e^{\mu t} - 1}\right)\|X_0 - X_\star\|^2$$

$$= e^{\mu t}(e^{\mu t} - 1)\|\tilde{\mathbb{A}}(X(t))\|^2 - \frac{4\mu^2 e^{\mu t}}{e^{\mu t} - 1}\|X_0 - X_\star\|^2.$$

Organizing, we conclude

$$\|\tilde{\mathbb{A}}(X(t))\|^2 \leq 4\left(\frac{\mu}{e^{\mu t} - 1}\right)^2 \|X_0 - X_\star\|^2.$$

### H.2.3 Informal derivation of ODE from the method

Assume $\mathbb{A}\colon \mathbb{R}^n \to \mathbb{R}^n$ be a continuous monotone operator. In [54], the method OS-PPM is presented as

$$x^k = \mathbb{J}_{h\mathbb{A}} y^{k-1}$$
$$y^k = \left(1 - \frac{1}{s_k}\right)\left\{x^k - \frac{1}{\nu}(y^{k-1} - x^k)\right\} + \frac{1}{s_k} y^0$$

where $y^0 = x^0$, $\nu = 1 + 2h\mu$ and $s_k = 1 + \nu^2 + \cdots + \nu^{2k} = \frac{\nu^{2k+2} - 1}{\nu^2 - 1}$. Using $y^{k-1} = x^k + h\mathbb{A}x^k$, substituting $y^k$ and $y^{k-1}$ this method can be expressed in a single line,

$$x^{k+1} + h\mathbb{A}x^{k+1} = \left(1 - \frac{1}{s_k}\right)\left(x^k - \frac{h}{\nu}\mathbb{A}x^k\right) + \frac{1}{s_k}x^0.$$

Reorganizing and dividing both sides by $h$, we have

$$\frac{x^{k+1} - x^k}{h} = -\mathbb{A}x^{k+1} - \left(\frac{1}{\nu} - \frac{1}{\nu s_k}\right)\mathbb{A}x^k - \frac{1}{h s_k}(x^k - x^0).$$

Identifying $x^0 = X_0$, $2hk = t$, $x^k = X(t)$,

$$\frac{X(t + 2h) - X(t)}{h} = -\mathbb{A}(X(t + 2h)) - \left(\frac{1}{\nu} - \frac{1}{\nu s_k}\right)\mathbb{A}(X(t)) - \frac{1}{h s_k}(X(t) - X_0).$$

Now observe

$$\lim_{h \to 0^+} h s_k = \lim_{h \to 0^+} h\frac{\nu^{2k+2} - 1}{\nu^2 - 1} = \lim_{h \to 0^+} h\frac{(1 + 2h\mu)^{2k+2} - 1}{(1 + 2h\mu)^2 - 1}$$
$$= \lim_{h \to 0^+} \frac{(1 + 2h\mu)^2(1 + 2h\mu)^{t/h} - 1}{4\mu(1 + h\mu)} = \frac{e^{2\mu t} - 1}{4\mu}$$
$$\lim_{h \to 0^+} \frac{1}{\nu} = \lim_{h \to 0^+} \frac{1}{1 + 2h\mu} = 1$$
$$\lim_{h \to 0^+} \frac{1}{\nu s_k} = \lim_{h \to 0^+} \frac{1}{\nu}\frac{1}{h s_k}h = 1 \times \frac{4\mu}{e^{2\mu t} - 1} \times 0 = 0.$$

Taking limit $h \to 0^+$ and organizing,

$$2\dot{X}(t) = -\mathbb{A}(X(t)) - (1 + 0)\mathbb{A}(X(t)) - \frac{4\mu}{e^{2\mu t} - 1}(X(t) - X_0).$$

By diving both sides by 2, we get the desired anchor ODE

$$\dot{X}(t) = -\mathbb{A}(X(t)) - \frac{2\mu}{e^{2\mu t} - 1}(X(t) - X_0).$$

# I  Proof omitted in Section 7

## I.1  Proof for Theorem 7.1

We first check

$$\left\|\tilde{\mathbf{A}}(X(t))\right\|^2 \leq 4\beta(t)^2 \left\|X_0 - X_\star\right\|^2 = \mathcal{O}\left(\beta(t)^2\right),$$

where

$$\dot{X}(t) = -\tilde{\mathbf{A}}(X(t)) - \beta(t)(X(t) - X_0)$$

$$\beta(t) = -\frac{\left\|\tilde{\mathbf{A}}(X(t))\right\|^2}{2\left\langle \tilde{\mathbf{A}}(X(t)), X(t) - X_0 \right\rangle}.$$

By definition of $\beta(t)$, $\Phi(t)$ defined in (8) becomes zero, i.e.

$$0 \equiv \Phi(t) = \left\|\tilde{\mathbf{A}}(X(t))\right\|^2 + 2\beta(t)\left\langle \tilde{\mathbf{A}}(X(t)), X - X_0 \right\rangle.$$

Plugging $\Phi(t) = 0$ to inequality (8) we have

$$0 \geq \frac{1}{2}\left\|\tilde{\mathbf{A}}(X(t))\right\|^2 - 2\beta(t)^2 \left\|X_0 - X_\star\right\|^2.$$

Reorganizing, we get the desired result. Other results need some works, we provide the proof with steps into subsections.

### I.1.1  $\beta(t) > 0$ for $t > 0$

Taking inner product with $\tilde{\mathbf{A}}(X)$ to the ODE and applying $\frac{1}{2}\Phi(t) = 0$ we have

$$\left\langle \dot{X}(t), \tilde{\mathbf{A}}(X(t)) \right\rangle = -\left\|\tilde{\mathbf{A}}(X(t))\right\|^2 - \beta(t)\left\langle \tilde{\mathbf{A}}(X(t)), X(t) - X_0 \right\rangle = -\frac{1}{2}\left\|\tilde{\mathbf{A}}(X(t))\right\|^2.$$

As $\mathbb{A}$ is assumed to be continuous, taking limit $t \to 0+$ we have

$$\lim_{t \to 0+}\left\langle \dot{X}(t), \tilde{\mathbf{A}}(X(t)) \right\rangle = \left\langle \lim_{t \to 0+} \dot{X}(t), \tilde{\mathbf{A}}(X_0) \right\rangle = -\frac{1}{2}\lim_{t \to 0+}\left\|\tilde{\mathbf{A}}(X(t))\right\|^2 = -\frac{1}{2}\left\|\tilde{\mathbf{A}}(X_0)\right\|^2.$$

On the other hand, by assumption we have $\lim_{t \to 0+} \dot{X}(t) = \lim_{t \to 0+} \frac{X(t) - X_0}{t}$ and $\left\|\tilde{\mathbf{A}}(X_0)\right\| \neq 0$, thus

$$\lim_{t \to 0+} \frac{1}{t}\left\langle \tilde{\mathbf{A}}(X(t)), X(t) - X_0 \right\rangle = \left\langle \tilde{\mathbf{A}}(X_0), \lim_{t \to 0+} \dot{X}(t) \right\rangle = -\frac{1}{2}\left\|\tilde{\mathbf{A}}(X_0)\right\|^2 < 0. \tag{50}$$

Therefore there is $\epsilon > 0$ such that $\left\langle \tilde{\mathbf{A}}(X(t)), X(t) - X_0 \right\rangle < 0$ for $t \in (0, \epsilon)$, thus for $t \in (0, \epsilon)$ we have $\left\langle \tilde{\mathbf{A}}(X(t)), X(t) - X_0 \right\rangle \neq 0$ so $\beta(t)$ is well-defined and satisfies

$$\beta(t) = -\frac{\left\|\tilde{\mathbf{A}}(X(t))\right\|^2}{2\left\langle \tilde{\mathbf{A}}(X(t)), X(t) - X_0 \right\rangle} > 0.$$

Observe the denominator of $\beta(t)$ is zero when $\left\|\tilde{\mathbf{A}}(X(t))\right\| = 0$. As $\beta$ is assumed to be well-defined, we have $\left\|\tilde{\mathbf{A}}(X(t))\right\| \neq 0$ for all $t > 0$. Since $\beta$ is continuous as $\mathbb{A}$ and $X$ are continuous, by intermediate value theorem we have $\left\|\tilde{\mathbf{A}}(X(t))\right\| > 0$ for all $t > 0$.

### I.1.2  Proof for the main statements

We first show following lemma.

**Lemma I.1.** *Following equality holds for almost every $t > 0$.*

$$\left(\dot{\beta}(t) + \beta(t)^2\right)\left\langle \tilde{\mathbf{A}}(X(t)), X(t) - X_0 \right\rangle = \left\langle \frac{d}{dt}\tilde{\mathbf{A}}(X(t)), \dot{X}(t) \right\rangle. \tag{51}$$

*Proof.* Since $\tilde{\mathbf{A}}$ is Lipschitz continuous by assumption, by Lemma B.4 we know $\tilde{\mathbf{A}}(X(t))$ is differentiable almost everywhere. Differentiating $\Phi(t)$ we have

$$
\begin{aligned}
0 &= \dot{\Phi}(t) \\
&= 2\left\langle \frac{d}{dt}\tilde{\mathbf{A}}(X(t)), \tilde{\mathbf{A}}(X(t))\right\rangle + 2\dot{\beta}(t)\left\langle \tilde{\mathbf{A}}(X(t)), X(t) - X_0\right\rangle + 2\beta(t)\left\langle \frac{d}{dt}\tilde{\mathbf{A}}(X(t)), X(t) - X_0\right\rangle + 2\beta(t)\left\langle \tilde{\mathbf{A}}(X(t)), \dot{X}(t)\right\rangle \\
&= 2\left\langle \frac{d}{dt}\tilde{\mathbf{A}}(X(t)), \tilde{\mathbf{A}}(X(t)) + \beta(t)(X(t) - X_0)\right\rangle + 2\dot{\beta}(t)\left\langle \tilde{\mathbf{A}}(X(t)), X(t) - X_0\right\rangle \\
&\quad + 2\beta(t)\left\langle \tilde{\mathbf{A}}(X(t)), -\tilde{\mathbf{A}}(X(t)) - \beta(t)(X(t) - X_0)\right\rangle \\
&= -2\left\langle \frac{d}{dt}\tilde{\mathbf{A}}(X(t)), \dot{X}(t)\right\rangle - 2\beta(t)\left\|\tilde{\mathbf{A}}(X(t))\right\|^2 + 2\left(\dot{\beta}(t) - \beta(t)^2\right)\left\langle \tilde{\mathbf{A}}(X(t)), X(t) - X_0\right\rangle \\
&= -2\left\langle \frac{d}{dt}\tilde{\mathbf{A}}(X(t)), \dot{X}(t)\right\rangle + 4\beta(t)^2\left\langle \tilde{\mathbf{A}}(X(t)), X(t) - X_0\right\rangle + 2\left(\dot{\beta}(t) - \beta(t)^2\right)\left\langle \tilde{\mathbf{A}}(X(t)), X(t) - X_0\right\rangle \\
&= -2\left\langle \frac{d}{dt}\tilde{\mathbf{A}}(X(t)), \dot{X}(t)\right\rangle + 2\left(\dot{\beta}(t) + \beta(t)^2\right)\left\langle \tilde{\mathbf{A}}(X(t)), X(t) - X_0\right\rangle.
\end{aligned}
$$

for $t \in (0, \infty)$ almost everywhere. The equalities come from the ODE $\dot{X}(t) = -\tilde{\mathbf{A}}(X(t)) - \beta(t)(X(t) - X_0)$ and the fact $\Phi(t) = 0$. Reorganizing, we have the desired equation (51). $\qquad\square$

Now we show the upper bounds of $\beta(t)$ for each monotone and strongly monotone case. Observe from (51) and the definition of $\beta(t)$, we have for almost all $t \in (0, \infty)$

$$
-\frac{\dot{\beta}(t) + \beta(t)^2}{2\beta(t)}\left\|\tilde{\mathbf{A}}(X(t))\right\|^2 = \left\langle \frac{d}{dt}\tilde{\mathbf{A}}(X), \dot{X}\right\rangle. \tag{52}
$$

(i) When $\tilde{\mathbf{A}}$ is monotone.

From (52) and (1) we have for almost all $t \in (0, \infty)$

$$
-\frac{\dot{\beta}(t) + \beta(t)^2}{2\beta(t)}\left\|\tilde{\mathbf{A}}(X(t))\right\|^2 = \left\langle \frac{d}{dt}\tilde{\mathbf{A}}(X), \dot{X}\right\rangle \geq 0.
$$

Since $\beta(t) > 0$ we have

$$
\dot{\beta}(t) + \beta(t)^2 \leq 0.
$$

almost everywhere. Now dividing both sides by $\beta(t)^2$ we have

$$
1 \leq -\frac{\dot{\beta}(t)}{\beta(t)^2}
$$

holds almost everywhere. Since $-\frac{\dot{\beta}(t)}{\beta(t)^2} = \frac{d}{dt}\left(\frac{1}{\beta(t)}\right)$, integrating above inequality both side from $\delta$ to $t$ we have

$$
t - \delta \leq \frac{1}{\beta(t)} - \frac{1}{\beta(\delta)} \implies \beta(t) \leq \frac{1}{t - \delta + \frac{1}{\beta(\delta)}}.
$$

By the way, from (50) we have

$$
\lim_{t\to 0+} t\beta(t) = -\lim_{t\to 0+}\frac{\left\|\tilde{\mathbf{A}}(X(t))\right\|^2}{2\left\langle \tilde{\mathbf{A}}(X(t)), \frac{X(t) - X_0}{t}\right\rangle} = 1,
$$

thus $\lim_{\delta\to 0+}\beta(\delta) = \infty$, so $\lim_{\delta\to 0+}\frac{1}{\beta(\delta)} = 0$. Therefore taking limit $\delta \to 0+$ we have

$$
\beta(t) \leq \lim_{\delta\to 0+}\frac{1}{t - \delta + \frac{1}{\beta(\delta)}} = \frac{1}{t}.
$$

(ii) When $\tilde{\mathbf{A}}$ is $\mu$-strongly monotone.

From (52) and (2) for almost all $t \in (0, \infty)$ we have

$$-\frac{\dot{\beta}(t) + \beta(t)^2}{2\beta(t)} \left\| \tilde{\mathbf{A}}(X(t)) \right\|^2 = \left\langle \frac{d}{dt}\tilde{\mathbf{A}}(X), \dot{X} \right\rangle \geq \mu \left\| \dot{X}(t) \right\|^2. \tag{53}$$

On the other hand, observe

$$\left\| \dot{X}(t) \right\|^2 = \left\| \tilde{\mathbf{A}}(X(t)) + \beta(t)(X(t) - X_0) \right\|^2$$

$$= \underbrace{\left\| \tilde{\mathbf{A}}(X(t)) \right\|^2 + 2\beta(t) \left\langle \tilde{\mathbf{A}}(X(t)), X(t) - X_0 \right\rangle}_{=\Phi(t)=0} + \beta(t)^2 \left\| X(t) - X_0 \right\|^2 = \beta(t)^2 \left\| X(t) - X_0 \right\|^2.$$

From Cauchy-Schwarz inequality we see

$$\left\| \tilde{\mathbf{A}}(X(t)) \right\|^2 = -2\beta(t) \left\langle \tilde{\mathbf{A}}(X(t)), X(t) - X_0 \right\rangle \leq 2\beta(t) \left\| \tilde{\mathbf{A}}(X(t)) \right\| \left\| X(t) - X_0 \right\|,$$

therefore $\left\| \tilde{\mathbf{A}}(X(t)) \right\| \leq 2\beta(t) \left\| X(t) - X_0 \right\|$. Combining above observations, we have for almost all $t > 0$

$$\frac{4 \left\| \dot{X}(t) \right\|^2}{\left\| \tilde{\mathbf{A}}(X(t)) \right\|^2} = \frac{4\beta(t)^2 \left\| X(t) - X_0 \right\|^2}{\left\| \tilde{\mathbf{A}}(X(t)) \right\|^2} \geq 1. \tag{54}$$

From (53) and (54), we get an inequality for $\beta(t)$ and $\dot{\beta}(t)$

$$-\frac{\dot{\beta}(t) + \beta(t)^2}{2\beta(t)} = \frac{\dot{\beta}(t)}{2\beta(t)} + \frac{\beta(t)}{2} \geq \frac{\mu}{4}.$$

Moving $\frac{\beta(t)}{2}$ to the right-hand side and reorganizing, we have for almost all $t > 0$

$$1 \leq -\frac{\dot{\beta}(t)}{\beta(t)(\frac{\mu}{2} + \beta(t))} = -\frac{\dot{\beta}(t)}{\frac{\mu}{2}} \left( \frac{1}{\beta(t)} - \frac{1}{\frac{\mu}{2} + \beta(t)} \right) \quad \Longrightarrow \quad \frac{\mu}{2} \leq -\frac{\dot{\beta}(t)}{\beta(t)} + \frac{\dot{\beta}(t)}{\frac{\mu}{2} + \beta(t)}.$$

Integrating above inequality both sides from $\delta$ to $t$ we have

$$\frac{\mu}{2}(t - \delta) \leq \left[ -\log \beta(t) + \log \left( \frac{\mu}{2} + \beta(t) \right) \right]_\delta^t = \log \frac{\frac{\mu}{2} + \beta(t)}{\beta(t)} - \log \frac{\frac{\mu}{2} + \beta(\delta)}{\beta(\delta)}.$$

As observed in case (i) we know $\lim_{\delta \to 0+} \beta(\delta) = \infty$, taking limit $\delta \to 0+$ both sides we have

$$\frac{\mu t}{2} \leq \log \frac{\frac{\mu}{2} + \beta(t)}{\beta(t)} = \log \left( 1 + \frac{\mu/2}{\beta(t)} \right) \Longrightarrow e^{\mu t/2} \leq 1 + \frac{\mu/2}{\beta(t)}.$$

Organizing, we get the desired result.

$$\beta(t) \leq \frac{\mu/2}{e^{\mu t/2} - 1}.$$

## I.2 Correspondence between the ODE (11) and the discrete method in Theorem 7.2

Observe, the case $h = 1$ for below method corresponds to the method provided in Theorem 7.2.

$$x^k = \mathbb{J}_{h\mathbb{A}} y^{k-1}$$

$$\beta_k = \begin{cases} \dfrac{\|h\tilde{\mathbf{A}}x^k\|^2}{-\langle h\tilde{\mathbf{A}}x^k, \, x^k - x^0 \rangle + \|h\tilde{\mathbf{A}}x^k\|^2} & \text{if } \|\tilde{\mathbf{A}}x^k\|^2 \neq 0 \\ 0 & \text{if } \|\tilde{\mathbf{A}}x^k\|^2 = 0 \end{cases}$$

$$y^k = (1 - \beta_k)(2x^k - y^{k-1}) + \beta_k x^0.$$

We now show when $\mathbb{A}$ is continuous and $\|\tilde{\mathbb{A}}x^k\|^2 \neq 0$ for all $k \geq 0$, we obtain the ODE (11) when we take limit $h \to 0+$. Note when $\mathbb{A}$ is continuous $\mathbb{A}$ equals to $\tilde{\mathbb{A}}$.

From $h\tilde{\mathbb{A}}x^{k+1} + x^{k+1} = y^k$, substituting $y^k$ and $y^{k-1}$ we get a single line expression

$$h\tilde{\mathbb{A}}x^{k+1} + x^{k+1} = (1 - \beta_k)(x^k - h\tilde{\mathbb{A}}x^k) + \beta_k x^0.$$

Reorganizing and dividing both sides by $h$, we have

$$\frac{x^{k+1} - x^k}{h} = -\tilde{\mathbb{A}}x^{k+1} - (1 - \beta_k)\tilde{\mathbb{A}}x^k - \frac{\beta_k}{h}(x^k - x^0).$$

Identify $x^0 = X_0$, $2hk = t$, and $x^k = X(t)$. Then we see

$$\frac{\beta_k}{h} = \frac{h^2\|\tilde{\mathbb{A}}x^k\|^2}{-h^2\langle \tilde{\mathbb{A}}x^k, \, x^k - x^0\rangle + h^3\|\tilde{\mathbb{A}}x^k\|^2} = \frac{\|\tilde{\mathbb{A}}(X(t))\|^2}{-\langle \tilde{\mathbb{A}}(X(t)), \, X(t) - X_0\rangle + h\|\tilde{\mathbb{A}}(X(t))\|^2}.$$

Thus $\beta_k = \mathcal{O}(h)$, we have $\lim_{h\to 0+}\beta_k = 0$. Now taking limit $h \to 0+$ we have

$$2\dot{X}(t) = -2\tilde{\mathbb{A}}(X(t)) - \frac{\left\|\tilde{\mathbb{A}}(X(t))\right\|^2}{\left\langle \tilde{\mathbb{A}}(X(t)), X(t) - X_0\right\rangle}(X(t) - X_0).$$

Dividing both sides by 2, we obtain (11).

### I.2.1 Correspondence between convergence rates in Theorem 7.1 and Theorem 7.2

From identification above, we see $\beta(t)$ corresponds to $\frac{\beta_k}{2h}$. Therefore we see with identification $x^0 = X_0$, $x^\star = X_\star$, $2hk = t$, and $x^k = X(t)$ we have

$$\left\|h\tilde{\mathbb{A}}(x^{k+1})\right\|^2 \leq \beta_k^2 \left\|x^0 - x^\star\right\|^2 = 4h^2\frac{\beta_k^2}{(2h)^2}\left\|x^0 - x^\star\right\|^2 \quad \xrightarrow{\text{divide by } h^2, \ h\to 0+} \quad \left\|\tilde{\mathbb{A}}(X(t))\right\|^2 \leq 4\beta(t)^2 \left\|X_0 - X_\star\right\|^2.$$

And we can also check that the bound $\beta_k \leq \frac{1}{k+1}$ for the monotone case, is equivalent to $\frac{\beta_k}{2h} \leq \frac{1}{2hk + \mathcal{O}(h)}$ and corresponds to $\beta(t) \leq \frac{1}{t}$ as well.

Now suppose $\mathbb{A}$ is $\mu$-strongly monotone and $L$-Lipschitz continuous. Then $h\mathbb{A}$ is $h\mu$-strongly monotone and $hL$-Lipschitz continuous, we have the following inequality from Theorem 7.2

$$\frac{\beta_k}{2h} \leq \frac{\frac{\mu}{2(1+(hL)^2)}}{\left(1 + \frac{h\mu}{1+(hL)^2}\right)^k - 1 + \frac{h\mu}{1+(hL)^2}}. \tag{55}$$

Recalling the identification $2hk = t$, we see

$$\left(1 + \frac{h\mu}{1+(hL)^2}\right)^k = \left(1 + \frac{h\mu}{1+(hL)^2}\right)^{\frac{t}{2h}} = \left(\left(1 + \frac{h\mu}{1+(hL)^2}\right)^{\frac{1+(hL)^2}{h\mu}}\right)^{\frac{\mu t}{2(1+(hL)^2)}} \quad \xrightarrow{h\to 0+} \quad e^{\mu t/2}.$$

Therefore identifying $\beta(t) = \frac{\beta_k}{2h}$ and taking limit $h \to 0+$ to the inequality (55), we have

$$\beta(t) \leq \frac{\mu/2}{e^{\mu t/2} - 1}.$$

### I.3 Proof of Theorem 7.2

Recall, the method was defined as

$$x^k = \mathbb{J}_{\mathbb{A}}y^{k-1}$$

$$\beta_k = \begin{cases} \dfrac{\|\tilde{\mathbb{A}}x^k\|^2}{-\langle \tilde{\mathbb{A}}x^k, \, x^k - x^0\rangle + \|\tilde{\mathbb{A}}x^k\|^2} & \text{if } \|\tilde{\mathbb{A}}x^k\|^2 \neq 0 \\ 0 & \text{if } \|\tilde{\mathbb{A}}x^k\|^2 = 0 \end{cases}$$

$$y^k = (1 - \beta_k)(2x^k - y^{k-1}) + \beta_k x^0.$$

First, we assume $\tilde{\mathbf{A}}x^k \neq 0$ for all $k \geq 0$. Define

$$\tilde{\Phi}^k = (1 - \beta_k)\|\tilde{\mathbf{A}}x^k\|^2 + \beta_k\langle\tilde{\mathbf{A}}x^k,\, x^k - x^0\rangle$$

for $k = 1, 2, \ldots$, and

$$\Phi^k = \|\tilde{\mathbf{A}}x^k\|^2 + \beta_{k-1}\langle\tilde{\mathbf{A}}x^k,\, x^k - x^0\rangle$$

for $k = 2, 3, \ldots$. Note by definition of $\beta_k$, we have $\tilde{\Phi}^k = 0$. Our goal is to prove

$$\Phi^{k+1} \leq 0.$$

Then with the same argument with (8) we can conclude $\|\tilde{\mathbf{A}}x^{k+1}\|^2 \leq \beta_k^2\|x^0 - x^\star\|^2$. To do so, we first show following lemma.

**Lemma I.2.** *For $k \geq 1$, following is true.*

$$(1 - \beta_k)\tilde{\Phi}^k - \Phi^{k+1} = (1 - \beta_k)\langle\tilde{\mathbf{A}}x^{k+1} - \tilde{\mathbf{A}}x^k,\, x^{k+1} - x^k\rangle. \tag{56}$$

*Proof.* Recall $y^k = x^{k+1} + \tilde{\mathbf{A}}x^{k+1}$. Substituting $y^k$ and $y^{k-1}$, the method is equivalent to

$$x^{k+1} + \tilde{\mathbf{A}}x^{k+1} = (1 - \beta_k)(x^k - \tilde{\mathbf{A}}x^k) + \beta_k x^0.$$

From above we can get two different expression of $(1 - \beta_k)(x^{k+1} - x^k)$.

$$\begin{aligned}
(1 - \beta_k)(x^{k+1} - x^k) &= -(1 - \beta_k)(\tilde{\mathbf{A}}x^{k+1} + \tilde{\mathbf{A}}x^k) - \beta_k\left(\tilde{\mathbf{A}}x^{k+1} + (x^{k+1} - x^0)\right) \\
&= -(1 - \beta_k)\left[(\tilde{\mathbf{A}}x^{k+1} + \tilde{\mathbf{A}}x^k) - \beta_k\left(\tilde{\mathbf{A}}x^k - (x^k - x^0)\right)\right].
\end{aligned}$$

With reorganizing, first equality can be obtained by subtracting both sides by $\beta_k(x^{k+1} - x^0)$ and the second equality can be obtained by multiplying both sides by $(1 - \beta_k)$.

From this we have

$$\begin{aligned}
&(1 - \beta_k)\langle\tilde{\mathbf{A}}x^{k+1} - \tilde{\mathbf{A}}x^k,\, x^{k+1} - x^k\rangle \\
&= \langle\tilde{\mathbf{A}}x^{k+1},\, (1 - \beta_k)(x^{k+1} - x^k)\rangle - \langle\tilde{\mathbf{A}}x^k,\, (1 - \beta_k)(x^{k+1} - x^k)\rangle \\
&= \langle\tilde{\mathbf{A}}x^{k+1},\, -(1 - \beta_k)(\tilde{\mathbf{A}}x^{k+1} + \tilde{\mathbf{A}}x^k) - \beta_k\{\tilde{\mathbf{A}}x^{k+1} + (x^{k+1} - x^0)\}\rangle \\
&\quad - (1 - \beta_k)\langle\tilde{\mathbf{A}}x^k,\, -(\tilde{\mathbf{A}}x^{k+1} + \tilde{\mathbf{A}}x^k) + \beta_k\{\tilde{\mathbf{A}}x^k - (x^k - x^0)\}\rangle \\
&= -(1 - \beta_k)\|\tilde{\mathbf{A}}x^{k+1}\|^2 - \beta_k\langle\tilde{\mathbf{A}}x^{k+1},\, \tilde{\mathbf{A}}x^{k+1} + (x^{k+1} - x^0)\rangle \\
&\quad + (1 - \beta_k)\left(\|\tilde{\mathbf{A}}x^k\|^2 - \beta_k\langle\tilde{\mathbf{A}}x^k,\, \tilde{\mathbf{A}}x^k - (x^k - x^0)\rangle\right) \\
&= (1 - \beta_k)\left((1 - \beta_k)\|\tilde{\mathbf{A}}x^k\|^2 + \beta_k\langle\tilde{\mathbf{A}}x^k,\, x^k - x^0\rangle\right) - \|\tilde{\mathbf{A}}x^{k+1}\|^2 - \beta_k\langle\tilde{\mathbf{A}}x^{k+1},\, x^{k+1} - x^0\rangle \\
&= (1 - \beta_k)\tilde{\Phi}^k - \Phi^{k+1}.
\end{aligned}$$

$\qquad\square$

Since $\mathbf{A}$ is monotone we have $\langle\tilde{\mathbf{A}}x^{k+1} - \tilde{\mathbf{A}}x^k,\, x^{k+1} - x^k\rangle \geq 0$, we see that the right-hand side of (56) is greater or equal to 0 if $1 - \beta_k \geq 0$. Thus it remains to show $1 - \beta_k \geq 0$. As the index is quite confusing, we provide it as a lemma to avoid confusion while proceeding the proof.

**Lemma I.3.** *If $\tilde{\mathbf{A}}x^k \neq 0$ for all $k \geq 1$, then*

$$\langle\tilde{\mathbf{A}}x^k, x^k - x^0\rangle < 0, \quad \beta_k \in (0, 1), \quad \Phi^{k+1} \leq 0$$

*for all $k \geq 1$.*

*Proof.* Proof by induction. From $x^1 = \mathbb{J}_A y^0 = \mathbb{J}_A x^0$, we have $x^1 + \tilde{\mathbf{A}}x^1 = x^0$ and so $x^1 - x^0 = -\tilde{\mathbf{A}}x^1$. Applying these facts we have

$$\langle\tilde{\mathbf{A}}x^1,\, x^1 - x^0\rangle = -\|\tilde{\mathbf{A}}x^1\|^2 < 0,$$

$$\beta_1 = \frac{\|\tilde{\mathbf{A}}x^1\|^2}{-\langle\tilde{\mathbf{A}}x^1,\, x^1 - x^0\rangle + \|\tilde{\mathbf{A}}x^1\|^2} = \frac{1}{2} \in (0, 1).$$

As $1 - \beta_1 > 0$, from (56) we have

$$(1 - \beta_1)\tilde{\Phi}^1 - \Phi^2 = (1 - \beta_1)\langle \tilde{\mathbb{A}}x^2 - \tilde{\mathbb{A}}x^1,\ x^2 - x^1 \rangle \geq 0.$$

As $\tilde{\Phi}^1 = 0$ by definition, we have $\Phi^2 \leq 0$. Therefore the statement is true for $k = 1$.

Now suppose the statements are true for $k$. By induction hypothesis, we know

$$0 \geq \Phi^{k+1} = \|\tilde{\mathbb{A}}x^{k+1}\|^2 + \beta_k \langle \tilde{\mathbb{A}}x^{k+1},\ x^{k+1} - x^0 \rangle.$$

As $\beta_k > 0$ from induction hypothesis and $\tilde{\mathbb{A}}x^{k+1} \neq 0$ by assumption, reorganizing $\Phi^k \leq 0$ we have

$$\langle \tilde{\mathbb{A}}x^{k+1},\ x^{k+1} - x^0 \rangle \leq -\frac{\|\tilde{\mathbb{A}}x^{k+1}\|^2}{\beta_k} < 0.$$

And therefore

$$\beta_{k+1} = \frac{\|\tilde{\mathbb{A}}x^{k+1}\|^2}{\underbrace{-\langle \tilde{\mathbb{A}}x^{k+1},\ x^{k+1} - x^0 \rangle}_{>0} + \|\tilde{\mathbb{A}}x^{k+1}\|^2} \in (0, 1).$$

Since $1 - \beta_{k+1} > 0$, by (56) we have

$$(1 - \beta_{k+1})\tilde{\Phi}^{k+1} - \Phi^{k+2} = (1 - \beta_{k+1})\langle \tilde{\mathbb{A}}x^{k+2} - \tilde{\mathbb{A}}x^{k+1},\ x^{k+2} - x^{k+1} \rangle \geq 0.$$

Since $\tilde{\Phi}^{k+1} = 0$ by definition, we have $\Phi^{k+2} \leq 0$. Therefore the statements are true for $k + 1$, by induction, we get the desired result. $\qquad\square$

Suppose $\|\tilde{\mathbb{A}}x^k\|^2 \neq 0$ for all $k \geq 1$. From the lemma we know $\Phi^{k+1} \leq 0$ for $k \geq 1$, so with the same argument of (8) we have for $x^\star \in \mathrm{Zer}\mathbb{A}$

$$\begin{aligned}
0 \geq \Phi^{k+1} &= \|\tilde{\mathbb{A}}x^{k+1}\|^2 + \beta_k \langle \tilde{\mathbb{A}}x^{k+1},\ x^{k+1} - x^\star \rangle - \beta_k \langle \tilde{\mathbb{A}}x^{k+1},\ x^0 - x^\star \rangle \\
&\geq \|\tilde{\mathbb{A}}x^{k+1}\|^2 - \beta_k \langle \tilde{\mathbb{A}}x^{k+1},\ x^0 - x^\star \rangle \\
&\geq |\tilde{\mathbb{A}}x^{k+1}|^2 - \left( \frac{1}{2}\|\tilde{\mathbb{A}}x^{k+1}\|^2 + \frac{\beta_k^2}{2}\|x^0 - x^\star\|^2 \right) = \frac{1}{2}\|\tilde{\mathbb{A}}x^{k+1}\|^2 - \frac{\beta_k^2}{2}\|x^0 - x^\star\|^2.
\end{aligned}$$

Organizing, we get

$$\|\tilde{\mathbb{A}}x^{k+1}\|^2 \leq \beta_k^2 \|x^0 - x^\star\|^2.$$

Now we show the upper bound of $\beta_k$. Observe from (56) and the fact $\tilde{\Phi} = 0$, we have

$$\begin{aligned}
(1 - \beta_k)\left\langle \tilde{\mathbb{A}}x^{k+1} - \tilde{\mathbb{A}}x^k,\ x^{k+1} - x^k \right\rangle &= (1 - \beta_k)\tilde{\Phi}^k - \Phi^{k+1} \\
&= \frac{\beta_k}{\beta_{k+1}}\tilde{\Phi}^{k+1} - \Phi^{k+1} = \left( \frac{\beta_k}{\beta_{k+1}} - \beta_k - 1 \right)\left\|\tilde{\mathbb{A}}x^{k+1}\right\|^2.
\end{aligned}$$

As $1 - \beta_k \neq 0$ for $k \geq 1$ by Lemma I.3, dividing both sides by $1 - \beta_k$, we get the discrete counterpart of (52),

$$\left\langle \tilde{\mathbb{A}}x^{k+1} - \tilde{\mathbb{A}}x^k,\ x^{k+1} - x^k \right\rangle = \frac{\frac{\beta_k}{\beta_{k+1}} - \beta_k - 1}{1 - \beta_k}\left\|\tilde{\mathbb{A}}x^{k+1}\right\|^2. \tag{57}$$

   (i) When $\mathbb{A}$ is monotone.

From (57) and monotonicity of $\mathbb{A}$ we have

$$0 \leq \left\langle \tilde{\mathbb{A}}x^{k+1} - \tilde{\mathbb{A}}x^k,\ x^{k+1} - x^k \right\rangle = \frac{\frac{\beta_k}{\beta_{k+1}} - \beta_k - 1}{1 - \beta_k}\left\|\tilde{\mathbb{A}}x^{k+1}\right\|^2.$$

From Lemma I.3 we have $1 - \beta_k > 0$, therefore

$$\frac{\beta_k}{\beta_{k+1}} - \beta_k - 1 \geq 0 \quad \implies \quad \frac{1}{\beta_k} + 1 \leq \frac{1}{\beta_{k+1}}.$$

Summing up, as $\beta_1 = \frac{1}{2}$ and $\frac{1}{\beta_k} > 0$ we have

$$\frac{1}{\beta_1} + (k - 1) = k + 1 \leq \frac{1}{\beta_k} \quad \implies \quad \beta_k \leq \frac{1}{k + 1}.$$

(ii) When $\mathbb{A}$ is $\mu$-strongly monotone.

Since $\mathbb{A}$ is $\mu$-strongly monotone, from (57) we have

$$\mu \left\| x^{k+1} - x^k \right\|^2 \leq \left\langle \tilde{\mathbb{A}} x^{k+1} - \tilde{\mathbb{A}} x^k, x^{k+1} - x^k \right\rangle = \frac{\frac{\beta_k}{\beta_{k+1}} - \beta_k - 1}{1 - \beta_k} \left\| \tilde{\mathbb{A}} x^{k+1} \right\|^2 .$$

Define $r_k = \frac{\left\| x^{k+1} - x^k \right\|^2}{\left\| \tilde{\mathbb{A}} x^{k+1} \right\|^2}$ for $k = 0, 1, \ldots$. Dividing both sides by $\left\| \tilde{\mathbb{A}} x^{k+1} \right\|^2$ and organizing, we have

$$r_k \mu \leq \frac{\frac{\beta_k}{\beta_{k+1}} - \beta_k - 1}{1 - \beta_k} \quad \Longrightarrow \quad r_k \mu (1 - \beta_k) \leq \frac{\beta_k}{\beta_{k+1}} - \beta_k - 1 = \beta_k \left( \frac{1}{\beta_{k+1}} - 1 \right) - \beta_k - (1 - \beta_k).$$

Dividing both sides by $\beta_k$ and reorganizing, we get a recursive inequality for $\frac{1}{\beta_k} - 1$

$$(1 + r_k \mu) \left( \frac{1}{\beta_k} - 1 \right) + 1 \leq \frac{1}{\beta_{k+1}} - 1. \tag{58}$$

We now prove an upper bound of $\beta_k$ from above inequality.

**Lemma I.4.** *Suppose $\mathbb{A}$ be a $\mu$-strongly monotone operator. Let $\beta_k$ be a sequence defined as Theorem 7.2 and let $r_k = \frac{\left\| x^{k+1} - x^k \right\|^2}{\left\| \tilde{\mathbb{A}} x^{k+1} \right\|^2}$ for $k = 0, 1, \ldots$. Then following holds for $k = 1, 2, \ldots$.*

$$\beta_k \leq \frac{1}{\sum_{j=1}^{k-1} \prod_{i=j}^{k-1} (1 + r_i \mu) + 2} .$$

*Proof.* First observe, the statement is equivalent to

$$\frac{1}{\beta_k} - 1 \geq \sum_{j=1}^{k-1} \prod_{i=j}^{k-1} (1 + r_i \mu) + 1.$$

The proof can be done by induction with (58).

When $k = 1$, recalling $\beta_1 = \frac{1}{2}$ from the proof of Lemma I.3, we can check the inequality is true.

Now suppose the inequality is true for $k = m$. Then from (58) we have

$$\frac{1}{\beta_{m+1}} - 1 \geq (1 + r_m \mu) \left( \frac{1}{\beta_m} - 1 \right) + 1$$

$$\geq (1 + r_m \mu) \left( \sum_{j=1}^{m-1} \prod_{i=j}^{m-1} (1 + r_i \mu) + 1 \right) + 1$$

$$= \left( \sum_{j=1}^{m-1} \prod_{i=j}^{m} (1 + r_i \mu) + (1 + r_m \mu) \right) + 1 = \sum_{j=1}^{m} \prod_{i=j}^{m} (1 + r_i \mu) + 1.$$

Therefore, we get the desired result. $\qquad\square$

Under the identification considered in Appendix I.2, the continuous counterpart of $r_k$ is

$$r_k = \frac{\left\| x^{k+1} - x^k \right\|^2}{\left\| h \tilde{\mathbb{A}} x^{k+1} \right\|^2} = 4 \frac{\left\| x^{k+1} - x^k \right\|^2}{(2h)^2} \frac{1}{\left\| \tilde{\mathbb{A}} x^{k+1} \right\|^2} \xrightarrow{h \to 0+} \frac{4 \left\| \dot{X}(t) \right\|^2}{\left\| \tilde{\mathbb{A}}(X(t)) \right\|^2},$$

and is greater or equal to 1 by (54). We obtained an exponential convergence rate in continuous setup from this fact. In the same spirit, we can get an exponential convergence rate for discrete setup if there is a positive lower bound for $r_k$, we provide it as a corollary of Lemma I.4.

**Corollary I.5.** *Consider the setup of Lemma I.4. Suppose there is $r \geq 0$ such that $r_k \geq r$ for $k = 0, 1, \ldots$. Then following is true for $k = 1, 2, \ldots$.*

$$\beta_k \leq \frac{r\mu}{(1 + r\mu)^k - 1 + r\mu}.$$

*Proof.* From $r_k \geq r$ we have

$$\sum_{j=1}^{k-1} \prod_{i=j}^{k-1} (1 + r_i \mu) + 2 \geq \sum_{j=1}^{k-1} \prod_{i=j}^{k-1} (1 + r\mu) + 2$$

$$= \sum_{j=1}^{k-1} (1 + r\mu)^{k-j} + 2 = \sum_{l=0}^{k-1} (1 + r\mu)^l + 1 = \frac{(1 + r\mu)^k - 1 + r\mu}{r\mu}.$$

Applying Lemma I.4, we get the desired result. $\qquad\square$

We now show $r_k \geq \frac{1}{1+L^2}$ holds when $\mathbb{A}$ is furthermore $L$-Lipschitz continuous.

(iii) When $\mathbb{A}$ is $\mu$-strongly monotone and $L$-Lipschitz continuous.
Recall from the proof of Lemma I.2, we know

$$x^{k+1} - x^k = -\mathbb{A}x^{k+1} - (1 - \beta_k)\mathbb{A}x^k - \beta_k(x^k - x^0).$$

Taking inner product with $\tilde{\mathbb{A}}x^k$ both sides we have

$$\left\langle \tilde{\mathbb{A}}x^k, x^{k+1} - x^k \right\rangle = -\left\langle \mathbb{A}x^{k+1}, \tilde{\mathbb{A}}x^k \right\rangle - \tilde{\Phi}^k = -\left\langle \mathbb{A}x^{k+1}, \tilde{\mathbb{A}}x^k \right\rangle.$$

From above equality we can check

$$\left\| x^{k+1} - x^k \right\|^2 + \left\| \mathbb{A}x^{k+1} - \tilde{\mathbb{A}}x^k \right\|^2 = \left\| \mathbb{A}x^{k+1} \right\|^2 + \left\| x^{k+1} - x^k + \tilde{\mathbb{A}}x^k \right\|^2 \geq \left\| \mathbb{A}x^{k+1} \right\|^2.$$

As $\mathbb{A}$ is $L$-Lipschitz continuous, we have

$$\left(1 + L^2\right) \left\| x^{k+1} - x^k \right\|^2 \geq \left\| x^{k+1} - x^k \right\|^2 + \left\| \mathbb{A}x^{k+1} - \tilde{\mathbb{A}}x^k \right\|^2 \geq \left\| \mathbb{A}x^{k+1} \right\|^2.$$

Dividing both sides by $\left(1 + L^2\right) \left\| \mathbb{A}x^{k+1} \right\|^2$ we get a lowerbound for $r_k$

$$r_k = \frac{\left\| x^{k+1} - x^k \right\|^2}{\left\| \mathbb{A}x^{k+1} \right\|^2} \geq \frac{1}{1 + L^2}.$$

Applying Corollary I.5 we get the desired result

$$\beta_k \leq \frac{\mu/(1 + L^2)}{\left(1 + \mu/(1 + L^2)\right)^k - 1 + \mu/(1 + L^2)}.$$

Note if we take limit $\mu \to 0^+$, with substitution $\alpha = \frac{\mu}{1+L^2}$ we have

$$\lim_{\mu \to 0^+} \frac{\mu/(1 + L^2)}{\left(1 + \mu/(1 + L^2)\right)^k - 1 + \mu/(1 + L^2)} = \frac{1}{\lim_{\alpha \to 0^+} \frac{1}{\alpha} \left((1 + \alpha)^k - 1\right) + 1} = \frac{1}{k + 1},$$

which is the bound for the monotone case.

### I.3.1 If there is $k \geq 0$ such that $\tilde{\mathbb{A}}x^k = 0$

Suppose $\tilde{\mathbb{A}}x^k = 0$ for some $k$. Let $x^N$ be the very first iterate such that $\tilde{\mathbb{A}}x^N = 0$. Then from previous argument we know Theorem 7.2 is true for $k < N$. Thus it remains to show the statements are true for $k \geq N$.

From $\tilde{\mathbb{A}}x^N = 0$ we know $y^{N-1} = x^N + \tilde{\mathbb{A}}x^N = x^N$. And since $\tilde{\mathbb{A}}x^N = 0$ implies $\beta_N = 0$, from the definition of the method we have $y^N = 2x^N - y^{N-1} = x^N$. Therefore

$$x^{N+1} = \mathbb{J}_{\mathbb{A}} y^N = \mathbb{J}_{\mathbb{A}} x^N = x^N,$$

we conclude $x^k = x^N \in \mathrm{Zer}\mathbb{A}$ for all $k \geq N$. Thus $\beta_k = 0$, $\left\| \tilde{\mathbb{A}}x^k \right\| = 0$ for all $k \geq N$, the Theorem 7.2 is trivially true for $k \geq N$.

## J   Details of experiment in Section 7.1

We solve a compressed sensing problem of Shi et al. [63] which is formulated as an $\ell_1$-regularized least-squared problem

$$\underset{x\in\mathbb{R}^d}{\text{minimize}} \quad \frac{1}{n}\sum_{i=1}^{n}\left\{\frac{1}{2}\|A_{(i)}x - b_i\|^2 + \rho\|x\|_1\right\}.$$

We solve this problem in decentralized manner due to the problem setup where the network of local agents are as Figure 2 and each agents communicate only with their neighbors, the nodes connected to each agents by edge. We use Metropolis-Hastings matrix as our mixing matrix $W \in \mathbb{R}^{n\times n}$ and apply PG-EXTRA. Let $W_{i,j}$ denote $(i,j)$-th entry of $W$ and $N_i \subseteq \{1,2,\ldots,n\}$ denote the index of the agents in the neighborhood of agent $i$. Consider

$$\mathbf{x}_i^{k+1} = \text{Prox}_{\alpha\rho\|\cdot\|_1}\left(\sum_{j\in N_i}W_{i,j}\mathbf{x}_j^k - \alpha A_{(i)}^\mathsf{T}\left(A_{(i)}\mathbf{x}_i^k - b_{(i)}\right) - \mathbf{w}_i^k\right)$$

$$\mathbf{w}_i^{k+1} = \mathbf{w}_i^k + \frac{1}{2}\left(x_i^k - \sum_{j\in N_i}W_{i,j}\mathbf{x}_j^k\right), \quad k = 0,1,\ldots \qquad\text{(PG-EXTRA)}$$

to the problem above. Under suitable choice of parameters, PG-EXTRA can be seen as a fixed-point iteration of an averaged operator with respect to $\|\cdot\|_M$ [74, Theorem 2], where the metric matrix $M$ is defined as

$$M = \begin{bmatrix}(1/\alpha)I & U^\mathsf{T} \\ U & \alpha I\end{bmatrix},$$

and $U$ is a symmetric definite matrix with $U^2 = \frac{1}{2}(I - W)$. That is, denoting $\mathbf{x}^k, \mathbf{w}^k \in \mathbb{R}^{d\times n}$ as the vertical stack of $\mathbf{x}_i^k$'s and $\mathbf{w}_i^k$'s respectively [58, Chapter 11.3], PG-EXTRA can be rewritten as $(\mathbf{x}^{k+1}, \mathbf{w}^{k+1}) = \mathbb{T}(\mathbf{x}^k, \mathbf{w}^k)$. Using this $\mathbb{T}$, we proceed the experiment with the Halpern method

$$(\mathbf{x}^{k+1}, \mathbf{w}^{k+1}) = \beta_k(\mathbf{x}^0, \mathbf{w}^0) + (1 - \beta_k)\mathbb{T}(\mathbf{x}^k, \mathbf{w}^k).$$

When $\mathbb{T}$ is an averaged operator, it is nonexpansive, we know $\mathbb{T} = \mathbb{R}_\mathbb{A} = 2\mathbb{J}_\mathbb{A} - \mathbb{I}$ for some maximal monotone operator $\mathbb{A}$ [54, Lemma 2.1]. Considering the equivalence discussed in Lemma D.2, we see above Halpern method is equivalent to our presented algorithms of the form

$$x^{k+1} = \mathbb{J}_\mathbb{A}y^k$$
$$y^{k+1} = (1 - \beta_k)\left(2x^{k+1} - y^k\right) + \beta_k x^0,$$

by corresponding $y^k = (\mathbf{x}^{k+1}, \mathbf{w}^{k+1})$. Note the operator norm $\left\|\tilde{\mathbb{A}}x^k\right\|_M^2$ can be calculated by considering below equation

$$\frac{1}{2}\left(\mathbb{T}y^{k-1} - y^{k-1}\right) = \mathbb{J}_\mathbb{A}y^{k-1} - y^{k-1} = x^k - \left(\tilde{\mathbb{A}}x^k - x^k\right) = -\tilde{\mathbb{A}}x^k.$$

We use the anchor coefficients $\beta_k = \frac{1}{k+1}$, $\beta_k = \frac{\gamma}{k^p + \gamma}$ with $p = 1.5$, $\gamma = 2.0$ and the adaptive choice of $\beta_k$ in Theorem 7.2 with $M$-norm. Note, in the experiment the adaptive coefficient is calculated by considering below equation

$$\frac{1}{2}\frac{\left\|\mathbb{T}y^{k-1} - y^{k-1}\right\|_M^2}{\left\|\mathbb{T}y^{k-1} - y^{k-1}\right\|_M^2 + \left\langle\mathbb{T}y^{k-1} - y^{k-1}, y^{k-1} - x^0\right\rangle_M} = \frac{1}{2}\frac{4\left\|\tilde{\mathbb{A}}x^k\right\|_M^2}{4\left\|\tilde{\mathbb{A}}x^k\right\|_M^2 + \left\langle-2\tilde{\mathbb{A}}x^k, \tilde{\mathbb{A}}x^k + x^k - x^0\right\rangle_M}$$

$$= \frac{\left\|\tilde{\mathbb{A}}x^k\right\|_M^2}{-\left\langle\tilde{\mathbb{A}}x^k, x^k - x^0\right\rangle_M + \left\|\tilde{\mathbb{A}}x^k\right\|_M^2}.$$

We choose the dimension of signal $d = 100$, the number of agents $n = 20$, the number of measurement for each agent $m_i = 4$, $\ell_1$-regularization parameter $\rho = 0.01$, and algorithm parameter $\alpha = 0.01$.

## K   Broader Impacts

Our work focuses on the theoretical aspects of convex optimization algorithms. There are no negative social impacts that we anticipate from our theoretical results.

## L   Limitations

Our analysis concerns convex optimization. Although this assumption is standard in optimization theory, many functions that arise in machine learning practice are not convex.

