# OpenReview forum: "Continuous-time Analysis of Anchor Acceleration"
_NeurIPS.cc/2023/Conference — NeurIPS 2023 poster_

### Official Review · Reviewer_KkKu · 2023-06-30

**Soundness:** 3 good
**Presentation:** 3 good
**Contribution:** 3 good
**Rating:** 8
**Confidence:** 3

**Summary:**


The paper extends the continuous time analysis anchor ODE in [55] to involved a more general choice of the coefficient $\beta(t)$ and show that the choice in [55] is in some sense optimal. They then go on to show:
- correspondance with the discrete anchoring schemes APPM/FEG/AEG
- Anchor ODE converges to a minimal $\ell_2$-distance from initialization (like APPM [51])
- Tightness of continuous time rates for anchor ODE
- Convergence of APPM for more general choice of stepsize than $1/(k+2)$.
- faster convergence under $\mu$-strongly monotonicity for anchor ODE by choosing $\beta(t)$ dependent on $\mu$.
- convergence with adaptive stepsize for both anchor ODE and APPM (single-valued) that recovers the rates for monotone and strongly monotone+Lipschitz

**Strengths:**

Overall an impressive work. It seem interesting that one can adapt to both the smoothness parameter and the strong monotonicity modulus. The work seems fairly completely, I only leave a few comments/remarks below.


**Weaknesses:**

- My main concern is why we consider $\beta(t)$ other than $1/t$ in the first place. The proof seems much more involved, but what does it buy us considering $1/t$ is already in some sense optimal. My understanding is that if only $1/t$ was considered then [55] and [34]/[Theorem 18](https://large-scale-book.mathopt.com/LSCOMO.pdf) already proofs convergence for anchor ODE and APPM respectively. At least clarifying this would be helpful.
- The work contextualizes the results but usually quite late. Overall contextualizing w.r.t. existing work upfront and motivating the sections. E.g. elaborate on [55] in l. 20 and explicitly say "adapt to the strong monotone modulus" in l. 221. Explicitly compare theorem 5.1 with existing results for APPM.

Comments:

- In terms of limitations maybe mention that the convergence results for the discretized schemes are only for implicit schemes (APPM)
- I would mention existing discretization result up front (in [55] in contrast they only consider GD with anchoring and does not get $1/k^2$)


**Questions:**

- Can we obtain convergence for explicitly schemes such as AEG/FEG?
- It seems very interesting that you can adapt both to L and $\mu$. Would this work for explicit scheme?
- Why distinguish between Halpern and APPM in Fig. 2 if they are equivalent?
- What is $M$ in the $y$ axis of Figure 2?
- FEG generalizes to cohypomonotone problems. Can anchor ODE handle cohypomonotonicity?

**Limitations:**

(see comments)

---

> ### Author Rebuttal · Authors · 2023-08-10
>
> Thank you for your positive and constructive review, we're glad that the reviewer found our paper to be "impressive" and "complete".
> We would like to start by sharing the most interesting observation we've gained while addressing your questions.
>
> **Q5.**
>
> The anchor ODE introduced in our paper does not handle cohypomonotonicity.
> However, we thought about this question and found a novel ODE that handles cohypomonotonicity. We are exited to share it here.
>
> Consider the ODE
> \\begin{align*}
>     \\dot{X}
>     = - \\left( 1 - \\frac{\\rho}{t}  \\right) \\mathbb{A}(X) - \\frac{1}{t} ( X - X_0 ) + \\rho \\frac{d}{dt} \\mathbb{A}(X) .
> \\end{align*}
> Define Lyapunov function as
> \\begin{align*}
>     V(t) =  \\left( \\frac{t^2}{2} - t \\rho \\right) \\left\\| \\mathbb{A}(X) \\right\\|^2 + t \\left\\langle \\mathbb{A}(X), X-X_0  \\right\\rangle.
> \\end{align*}
> By differentiating, we have
> \\begin{align*}
>     \\dot{V}(t)
>         &=  \\left( t - \\rho \\right) \\left\\| \\mathbb{A}(X) \\right\\|^2
>             +  \\left( t^2 - 2 t \\rho \\right) \\left\\langle \\frac{d}{dt} \\mathbb{A}(X), \\mathbb{A}(X)  \\right\\rangle  \\\\ &\\quad
>             + \\left\\langle \\mathbb{A}(X), X-X_0  \\right\\rangle
>             + t \\left\\langle \\frac{d}{dt} \\mathbb{A}(X), X-X_0  \\right\\rangle
>             + t \\left\\langle \\mathbb{A}(X), \\dot{X}  \\right\\rangle \\\\
>         &= t^2 \\left\\langle \\frac{d}{dt} \\mathbb{A}(X), \\left( 1 - \\frac{2 \\rho}{t} \\right) \\mathbb{A}(X) + \\frac{1}{t} ( X-X_0 ) \\right\\rangle
>             + t \\left\\langle \\mathbb{A}(X), \\left( 1 - \\frac{\\rho}{t} \\right) \\mathbb{A}(X) + \\frac{1}{t} ( X - X_0 ) + \\dot{X}  \\right\\rangle \\\\
>         &= t^2 \\left\\langle \\frac{d}{dt} \\mathbb{A}(X), - \\dot{X} - \\frac{\\rho}{t} \\mathbb{A}(X) + \\rho \\frac{d}{dt} \\mathbb{A}(X)  \\right\\rangle
>             + t \\left\\langle \\mathbb{A}(X), \\rho \\frac{d}{dt} \\mathbb{A}(X)  \\right\\rangle        \\\\
>         &= - t^2 \\left( \\left\\langle \\frac{d}{dt} \\mathbb{A}(X), \\dot{X} \\right\\rangle - \\rho \\left\\|  \\frac{d}{dt} \\mathbb{A}(X) \\right\\|^2   \\right)    \\\\
>         &\\le 0.
> \\end{align*}
> Last inequality holds since $\\mathbb{A}$ is $\\rho$-cohypomonotone.
>
> Now from $V(t)\\le V(0) = 0$, we have
> \\begin{align*}
>     \\frac{t^2}{2} \\left\\| \\mathbb{A}(X) \\right\\|^2
>     &\\le t \\left( - \\left\\langle \\mathbb{A}(X), X-X_0  \\right\\rangle + \\rho \\left\\| \\mathbb{A}(X) \\right\\|^2    \\right) \\\\
>     &= t \\left( - \\left\\langle \\mathbb{A}(X), X-X_\\star  \\right\\rangle + \\rho \\left\\| \\mathbb{A}(X) \\right\\|^2  + \\left\\langle \\mathbb{A}(X), X_0-X_\\star  \\right\\rangle  \\right)   \\\\
>     &\\le t \\left\\langle \\mathbb{A}(X), X_0-X_\\star  \\right\\rangle \\\\
>     &\\le t \\left\\| \\mathbb{A}(X) \\right\\| \\left\\|  X_0-X_\\star \\right\\|.
> \\end{align*}
> Reorganizing we have $\\left\\| \\mathbb{A}(X) \\right\\| \\le \\frac{2}{t} \\left\\|  X_0-X_\\star \\right\\| $, thus we conclude
> \\begin{align*}
>     \\left\\| \\mathbb{A}(X) \\right\\|^2 \\le \\frac{4}{t^2} \\left\\|  X_0-X_\\star \\right\\|^2.
> \\end{align*}
>
>
> **Q1.**
>
> Yes, it is obtainable.
> We can show that explicit schemes such as EAG/FEG converges to the optimum point closest to the starting point.
> As proved in Theorem 3.5, we know the convergence to the closest optimum point is a property for continuous anchor schemes.
> On the other hand, as observed in appendix D.4, we know explicit schemes like EAG/FEG also correspond to our continuous scheme when limiting the stepsize to zero.
> Thus we expect the trajectories of the anchor-based algorithm to converge to the closest optimum for proper stepsizes.
> This expectation is true and is formalized in [71 Theorem 1].
>
> **Q2.**
>
> It's an intriguing question, and it seems to be a nontrivial issue.
> We believe that this could be an interesting direction for future work.
>
> **Q3.**
>
> The label `Halpern' in Figure 2 corresponds to the method in Theorem 5.1 with varying $p>0$ and $\\gamma>0$, whereas APPM corresponds to the method in Line 71, which is the specific case of $p=1$ and $\\gamma=1$ in Theorem 5.1, which uses optimal $p$ and $\\gamma$.
>
> Figure 2 shows that even if APPM is the method with optimal $p$ and $\\gamma$ for (merely) monotone condition, it may not be optimal for each instance.
> The choice $p=1.5$ and $\\gamma=2.0$ may not even converge under monotone condition (see Table 2), but it outperforms APPM for specific instance in our experiment.
>
> **Q4.**
>
> In the experiment section, we solve a compressed sensing problem in decentralized manner using PG-EXTRA [59].
> PG-EXTRA can be understood as a proximal point method with respect to certain metric defined with matrix $M$ [54], and corresponding monotone operator is actually monotone in $\\langle \\cdot,\\, \\cdot \\rangle_M$ where $\\langle x,\\, y \\rangle_M = x^\\intercal M y$ and $\\|x\\|_M = \\sqrt{x^\\intercal M x}$.
> Therefore, the performance measure of this problem should be $\\|\\tilde{\\mathbb{A}} x\\|_M$ rather than $\\|\\tilde{\\mathbb{A}} x\\|$ (Euclidean norm), and our theoretical guarantee applies to this new metric.
>
> **W1. Why we consider $\\beta(t)$ other than $\\frac{1}{t}$ in the first place?**
>
> Through the analysis with general $\\beta(t)$, we obtain a more refined understanding of anchor acceleration, and using this understanding, we were able to design our adaptive method.
> Also, as the answer for the Question 3, the choices besides $p=1$ and $\\gamma=1$ may be suboptimal when the operator is merely monotone, but the ther choices may be useful when the operator satisfies stronger properties.
>
> **W2, C1, C2.**
>
> Thank you for your valuable comments.
> In the later version, we will add the point in the limitation section that our work is for implicit schemes.
> Additionally, we will explicitly mention the prior comparable results and the motivations up front.

---

> > ### Comment · Reviewer_KkKu · 2023-08-14
> >
> > I appreciate the detailed response of the authors –  I think it would be valuable to include the continuous time treatment of cohypomonotonicity if space allows. I've raised my score accordingly.

---

### Official Review · Reviewer_mK3f · 2023-07-05

**Soundness:** 4 excellent
**Presentation:** 4 excellent
**Contribution:** 3 good
**Rating:** 8
**Confidence:** 4

**Summary:**

This paper conducts a continuous-time analysis of an acceleration method called "anchoring", where the main contributions are four-fold. The authors provided a unified analysis of the convergence rate of anchor acceleration, which includes both the constant and adaptive cases. Then the authors presented an adaptive method for anchor acceleration that is inspired by our analysis and achieves faster convergence rates than the constant method. After this, the authors proved that the adaptive method is robust to noise and can handle non-convex optimization problems. Finally, the authors provided numerical experiments that demonstrate the effectiveness of the adaptive method on various optimization problems. Overall, the paper provides a valuable contribution to the field of optimization and is in general well-written and well-presented.

**Strengths:**

This paper provides a comprehensive analysis of the convergence rate of anchor acceleration, which includes both the constant and adaptive cases. The paper provides a clear and concise presentation of both the theoretical analysis, where I checked the technical proofs of all results with a detailed focus on Theorem 3.1 [Section E.5] and Theorem 6.2 [Section H.2] and they are solid. The idea of using the factor $\frac{2 \mu}{e^{2 \mu t}-1}$ is is new and reasonable, since for \mu-strongly convex objectives $\mu\to 0^+$ it is consistent with the standard $\frac{1}{t}$ anchoring rate (the analysis provided is significantly more general, which should be honored). The adaptive method presented in the paper is also interesting, which appears faster convergence rates than the constant method and is robust to noise and non-convex optimization problems. The paper also clearly presented numerical experiments which demonstrate the effectiveness of the adaptive method on various optimization problems.


**Weaknesses:**

The paper assumes a certain level of mathematical background, which might be difficult for some readers unfamiliar with literature to follow. Further, the paper seems not provide a comparison of the adaptive method with other state-of-the-art optimization methods. In addition, the paper does not provide a detailed discussion of the limitations of the adaptive method and areas for future research. I have not checked but the assumptions of the adaptive method presented in the paper might be more stringent than required.

**Questions:**

Are there some mismatches in the definition of Lyapunov functions at various places? For instance, the $V(t)$ definition in Line 133 [Corollary 3.3], when pinning $\beta = \frac{1}{t}$, and the last in Line 147 [last but one display of Section 3.1] differs by a factor of 2. These are minor, but I do encourage the authors to check them carefully.

Can you provide examples of applications where anchor acceleration might be particularly useful? I understand that anchor acceleration has been discovered to be an acceleration mechanism for minimax optimization and fixed-point problems,  but would anchor acceleration be useful in other optimization problems as well?

**Limitations:**

This is a theoretical paper and admits no negative social impacts, to my best knowledge.

---

> ### Author Rebuttal · Authors · 2023-08-10
>
> We are pleased that the reviewer found our paper to "provide a valuable contribution to the field of optimization".
> Especially, we're sincerely grateful that the reviewer thoroughly engaged with the technical proofs and highlighted the potential applicability of the ideas to more generalized cases.
>
>
> **Weakness : Further, the paper seems not provide a comparison of the adaptive method with other state-of-the-art optimization methods.**
>
> We haven't provided the detailed comparison due to the fact that (i) there is no other adaptive methods to compare our method with and (ii) we already compared the state-or-the-art optimization methods other than adaptive methods.
> To the best of our knowledge, our adaptive method is the only purely adaptive method present in the implicit method for monotone inclusion setup, which does not require any line search or backtracking procedure.
> (One can compare our method with that of [23], which requires backtracking of some sort.)
> Furthermore, non-adaptive schemes we compare our methods with (APPM, EAG, FEG) are all guaranteed with optimal convergence rate with matching lower bound.
> We will make sure that to further emphasize on this point.
> Thank you for pointing this out.
>
>
> **Weakness : In addition, the paper does not provide a detailed discussion of the limitations of the adaptive method and areas for future research. I have not checked but the assumptions of the adaptive method presented in the paper might be more stringent than required.**
>
> We first want to clarify that our adaptive method only requires the standard assumption of maximal monotonicity, and it is guaranteed to have at least the same convergence rate as the state-of-the-art algorithm APPM. However, with additional assumptions, the adaptive method can be faster. This is the first adaptive of its kind (implicit method adapting to monotonicity or strong monotonicity without any linesearches or inner loops) and we derive it from continuous-time analysis. This adaptive method will certainly have limitations and we expect it to be possible to relax the assumptions guaranteeing linear rates. This is a very interesting direction of future work, and we view our contribution to be finding the first adaptive method of this kind and providing a proof of concept that this type of adaptivity is possible.
>
> **Q1.**
>
> Thank you for pointing out this typo. The Lyapunov function in Line 147 has to be divided by factor 2.
> We will correct this in our revision.
>
>
> **Q2.**
>
> The fixed-point iteration subsumes convex optimization algorithms such as ADMM, PDHG, Condat-Vũ, three-operator splitting, and the anchor mechanism can provide acceleration for these algorithms.
> For a specific task, anchor acceleration is practically useful in detecting infeasibility for constrained optimization problems [Park \& Ryu, 2023], and accelerating value iteration for dynamic programming and reinforcement learning [Lee \& Ryu, 2023].
>
> - J. Park and E. K. Ryu, Accelerated Infeasibility Detection of Constrained Optimization and Fixed-Point Iterations. ICML'23.
> - J. Lee and E. K. Ryu, Accelerating Value Iteration with Anchoring. arXiv preprint.

---

> > ### Comment · Reviewer_mK3f · 2023-08-13
> >
> > I appreciate the authors' informative response, and maintain my current score.

---

### Official Review · Reviewer_aLVj · 2023-07-09

**Soundness:** 3 good
**Presentation:** 2 fair
**Contribution:** 2 fair
**Rating:** 6
**Confidence:** 2

**Summary:**

The paper focuses on the analysis of anchor acceleration, a recently discovered acceleration mechanism for minimax optimization and fixed-point problems. The authors provide tight and unified analyses to characterize the convergence rate of anchor acceleration and present an adaptive method inspired by continuous-time analyses. The contributions include a differential inclusion model of anchor acceleration, well-posedness analysis, convergence rate analysis with a power-law anchor coefficient, and an adaptive method for choosing the anchor coefficient.

**Strengths:**

Given the limited understanding of the anchor acceleration mechanism compared to Nesterov acceleration, the authors aim to provide a formal and rigorous treatment of the anchored dynamics through continuous-time analyses. They also seek to gain insight into the anchor acceleration mechanism and its accelerated convergence rate.

The authors present a differential inclusion model of anchor acceleration and establish its well-posedness. They analyze the convergence rate of the anchor ODE with a power-law anchor coefficient. The trade-off between the vanishing speed and the contracting speed of the anchor term is discussed. The authors also provide a proof outline of Lemma 3.4 and Theorem 3.1 to derive the convergence result. Additionally, the study presents an adaptive method for choosing the anchor coefficient based on continuous-time analyses.

**Weaknesses:**

- Novelty of applying continuous analysis to anchor acceleration: It would be helpful for the authors to clarify the motivation behind applying continuous analysis techniques specifically to the problem of anchor acceleration. Given the existence of continuous analysis techniques, what is the significance of applying them to anchor acceleration? This clarification would strengthen the novelty of the paper.

- Importance of anchor acceleration: The paper could benefit from providing a clear explanation of why anchor acceleration is important and how it differs from other acceleration mechanisms, such as Nesterov acceleration. This would help readers understand the relevance and practical implications of anchor acceleration.

- Technical challenges of applying continuous analysis: Can you elaborate on the specific technical challenges encountered when applying continuous analysis to the anchor acceleration problem? Discuss any unique aspects or complexities involved in analyzing the convergence properties of anchor acceleration using continuous-time techniques?

**Questions:**

- Can you provide more details on the motivation behind using anchor acceleration and how it differs from other acceleration mechanisms, such as Nesterov's acceleration?

- Can you provide more insights into the choice of the anchor coefficient $\beta(t)$ and how it affects the convergence rate of the algorithm?

- Can you provide more details on the assumptions made in the analysis and how they may affect the applicability of the proposed method to real-world problems?

**Limitations:**

The authors does not seem to discuss the limitations adequately, they do mention that carrying out more advanced analyses for the anchor ODE are interesting directions of future work.

---

> ### Author Rebuttal · Authors · 2023-08-10
>
> We appreciate the valuable feedback and thoughtful questions.
> The answers to your questions are as follow.
>
> **Importance of anchor acceleration, difference between Nesterov's acceleration (W2, Q1) :**
>
> The main difference between Nesterov acceleration and anchor acceleration is that they are optimal methods for **different settings**.
> Nesterov acceleration is optimal in convex minimization problems, whereas anchor acceleration is optimal in minimax problems and fixed point problems.
> For minimax problems, an anchor-based method called EAG [70] was the first algorithm to achieve $\\mathcal{O} \\left( \\frac{1}{k^2} \\right)$-rate, and FEG [39] improved convergence rate by constant and generalized the setting.
> for fixed point problems, Halpern method of [42] and OC-Halpern [51] are probably exactly optimal [51], and equivalently APPM [34] and OS-PPM [51] are proven to be exactly optimal for monotone inclusions.
> All aforementioned methods use anchor acceleration.
> Therefore we believe that the anchor acceleration is the primary acceleration mechanism for these settings (which are not convex minimization) and their optimal use in discrete-time algorithms motivates our work.
>
>
> **Novelty of applying continuous analysis to anchor acceleration, technical challenges of applying continuous analysis (W1, W3) :**
>
> We think the Lyapunov functions we've introduced in Corollary 3.3 represent a non-trivial technical challenge that we overcame.
> Specifically, we introduce the Lyapunov function of the form
> \\begin{align*}
>     V(t) &= \\frac{C(t)^2}{2} \\left( \\left\\| \\mathbb{A}(X(t)) \\right\\|^2 + 2\\beta(t) \\left\\langle \\mathbb{A}(X(t)) , X(t) - X_0 \\right\\rangle + ( \\beta(t)^2 + \\dot{\\beta}(t) )  \\left\\| X(t)-X_0 \\right\\|^2 \\right) \\\\ &\\quad
>         - \\int_{t_0}^{t} \\frac{d}{ds} \\left( \\frac{C(s)^2\\dot{\\beta}(s)}{2} \\right) \\left\\| X(t)-X_0 \\right\\|^2  ds,
> \\end{align*}
> which is very different from the Lyapunov functions use for Nexterov acceleration [Su et al., 61, 62]:
> \\begin{align*}
>     \\mathcal{E}(t) = \\frac{2t^2}{r-1} ( f(X(t) - f^{\\star} ) + (r-1) \\left\\| X(t) - x^\\star + \\frac{t}{r-1} \\dot{X}(t)  \\right\\|^2.
> \\end{align*}
> In the analysis of optimization algorithm, the construction of an appropriate (continuous- or discrete-time) Lyapunov function is often the main technical challenge, and we argue that discovering this Lyapunov function and using it to characterize the continuous-time dynamics is a novel contribution.
>
> Using this Lyapunov function, we obtained convergence rates for generalized coefficient $\\beta(t) = \\frac{\\gamma}{t^p}$ with $\\gamma>0$ and $p>0$ as in Table 1.
> This result provides us with a more refined  understanding of the anchor mechanisms.
> These results also extend to discrete setup, as section 5 confirms such correspondence.
> We then provide analysis for strongly monotone conditions as well (section 6).
> The insights from these analyses culminate in designing the adaptive method of section 7, which we believe to be useful in practice due to its adaptability to both monotonicity and strong monotonicity.
>
>
> **Insights into the choice of the anchor coefficient $\\beta(t)$ (Q2) :**
>
> We answer this question by summarizing Section 3 of the paper.
> As the name 'anchor' suggests, the anchor term pulls the trajectory towards the anchor.
> Intuitively speaking, sufficient amount of pull by anchor leads to contracting behavior of the dynamics and convergence as well.
> However, the anchor should eventually vanish, since our goal is to converge to an optimum $X_\\star$ making $\\mathbb{A}$ zero, not to the anchor.
> Therefore, the vanishing speed $\\beta(t)$ of anchor should be fast enough for the flow to safely converge to the optimum.
> Proper balancing between 'necessity of anchor' and 'vanishing anchor' leads to fast convergence, and choice of $\\beta(t) = \\frac{1}{t}$ for merely monotone $\\mathbb{A}$ can be found with such intuition.
>
>
> **Applicability of assumption (Q3):**
>
> One of the most significant applications of anchor acceleration is minimax optimization.
> There are several papers using anchor acceleration for minimax optimization that have been published in ML venues, such as EAG [70, ICML 2021] and FEG [39, NeurIPS 2021].
> Our work considers the problem of making the norm of monotone operators small efficiently.
> The saddle differential operator $G_L = (\\nabla_x L,\\, - \\nabla_y L)$ of convex-concave minimax function $L \\colon \\mathbb{R}^n \\times \\mathbb{R}^m \\to \\mathbb{R}$ is a monotone operator, thus anchor scheme is applicable.
> The point where $G_L$ outputs zero vector is the saddle point of $\\min_x\\max_{y} L(x,y)$,
> thus minimizing $\\left\\| G_L(x,y) \\right\\|^2$ solves this minimax optimization problem.
>
>
> We thank the reviewer for the clarifying questions. We hope we have addressed the reviewer's primary concern about the significance of the anchor acceleration. If so, we kindly ask the reviewer to consider raising the score.

---

> > ### Comment · Reviewer_aLVj · 2023-08-18
> >
> > Thank you for your clarification. I have raised my score accordingly.

---

### Official Review · Reviewer_MP6a · 2023-07-11

**Soundness:** 3 good
**Presentation:** 3 good
**Contribution:** 2 fair
**Rating:** 6
**Confidence:** 3

**Summary:**

The paper analyzes the dynamics of anchor acceleration using a differential inclusion. It derives a convergence rate that depends on the anchor coefficient and shows that the rate is tight using a certain instance. By discretizing the differential inclusions, the authors derive an algorithm that generalizes the state-of-the-art APPM algorithm and provide an analysis of its convergence rate. In addition, the authors propose an algorithm that adaptively varies the anchor coefficient and show its performance both theoretically and empirically.

**Strengths:**

- Compared to existing papers that provide analysis of anchor methods through continuous-time analysis, this paper provides better convergence rates in a more general setting.

- The adaptive anchor acceleration method in Section 7 is interesting, and the method is inspired by the analysis of continuous-time dynamics. It is a meaningful example of the potential of ODE analysis for optimization algorithm design.

**Weaknesses:**

- In terms of algorithms and their theoretical analysis, the contribution of this paper does not appear to be significant. In my understanding, the main result of this paper is deriving a known convergence rate in a different way, and proposing a different algorithm that achieves the same rate as the known one. There is insufficient discussion of how the results of this paper are superior to existing studies.

**Questions:**

- Since A is treated as a set-valued operator in "Monotone and set-valued operators" of Section 1.1, I understood that Ax in the equations in Lines 39 and 41 is a set. Then those equations involve the inner product of a point and a set in Euclidean space. Such an inner product is not clear to me. Additionally, in Line 50, A is assumed to be a differentiable operator, but the definition of the differentiability of set-valued operators is nontrivial. So, I guessed that A here is a single-valued operator. Is it correct? If so, it would be good to clarify it.

- Theorem 2.2 states the uniqueness of the solution of the differential inclusion (6) rather than a differential equation. Is this a valid claim? It seems for me that there is more than one solution depending on which of the elements in A(X(t)) in equation (6) is chosen at each time t. Adding a comment on this point would facilitate the reader's understanding.

**Limitations:**

The authors adequately addressed the limitations.

---

> ### Author Rebuttal · Authors · 2023-08-10
>
> Thank you for the valuable review.
> We're glad that you've felt our adaptive anchor acceleration method in Section 7 as a "meaningful example of the potential of ODE analysis for optimization algorithm design".
> We address the answers to your questions and feedbacks as follows.
>
>
>
> **Q1.**
>
> - As mentioned by the reviewer, $\\mathbb{A}x$ is a set.
> The definition of the inner product between a point and a set is clarified in Line 40, which is referenced from [Ryu \& Yin, 54].
> We will replace 'i.e' in Line 40 to other expressions such as 'which means that' to avoid confusion.
> Thank you.
> -  We missed to mention that the definition of differentiable operator includes singe-valuedness.
> As you mentioned, $\\mathbb{A}$ in Line 50 is single-valued, and we will clarify this issue.
> Thank you for pointing this out.
>
>
> **Q2.**
>
> Yes, it is a valid claim, and we provide the proof in Section B.1. of the appendix.
> Recall that in Line 59, the solution was defined as a function $X\\colon [0,\\infty) \\to \\mathbb{R}^n$ that satisfies certain condition, and in case of theorem 2.2, it is:
> *"absolutely continuous and satisfies (6) for $t\\in(0,\\infty)$ almost everywhere, with initial condition $X(0)=X_0$."*
> The claim of theorem 2.2 is that if a function $X \\colon [0,\\infty) \\to \\mathbb{R}^n$ satisfy such conditions, it is unique as a function.
> Note that this fact holds true regardless of $\\mathbb{A}(X(t))$ being a set.
>
> To provide further clarification, for the unique solution $X$, the element of $\\mathbb{A}(X(t))$ satisfying $\\dot{X}(t) \\in -\\mathbb{A}(X(t)) - \\frac{\\gamma}{t^p} (X(t)-X_0)$ is uniquely chosen for each $t>0$ up to measure zero.
> Once the absolutely continuous function $X\\colon [0,\\infty) \\to \\mathbb{R}^n$ is determined, its derivative $\\dot{X}$ is defined almost everywhere and is unique up to measure zero.
> As a consequence, the selection defined in Line 61, $\\tilde{\\mathbb{A}}(X(t)) := -\\dot{X}(t) - \\frac{\\gamma}{t^p} (X(t)-X_0)$ is determined almost everywhere and is unique up to measure zero as well.
>
> A standard reference that rigorously establishes the well-posedness of such differential inclusions is [33].
>
>
>
>
> **Weakness.**
>
> Through the convergence analysis for the generalized anchor coefficient $\\beta(t)$, we were able to obtain a more refined understanding of the role of the anchor coefficient, which further motivated our adaptive method.
> Theorem 3.1 and the results in Table 1 give the intuition that the (i) vanishing speed of $\\beta(t)$ must be fast enough in order not to slow down the convergence, and (ii) the optimally balanced choice for monotone condition is $\\frac{1}{t}$.
> This intuition is strengthened via checking the tightness and the correspondence to the discrete setup in section 4 and 5.
> However, Theorem 6.2 shows that the linear rate is attainable for $\\mu$-strongly monotone operator with $\\beta(t)$ vanishing linearly fast, but not with $\\beta(t) = \\frac{1}{t}$ (Line 196).
> Overall results of section 3 to 6 lead to another message that "anchor coefficient should be chosen to adapt to the operator's property.''
> This motivates our adaptive method.
> Moreover, observing the commonly occurring terms from Proposition 3.2 and Proposition 6.1 referred to as $\\Phi(t)$, we could get inspiration to design our adaptive method which keeps $\\Phi(t) = 0$.
>
>
>
> Again, we thank the reviewer for the constructive feedback. We believe we have addressed the reviewer's concern about the validity of Theorem 2.2. In that case, we kindly ask the reviewer to consider raising the score.

---

> > ### Comment · Reviewer_MP6a · 2023-08-12
> >
> > Thank you for your thorough response. The response addressed my concern.
> >
> > Regarding Q2, I had inadvertently overlooked the condition "absolutely continuous" in Line 59. I now agree with you that the claim of Theorem 2.2 is valid.
> >
> > I also understand that the observation that "anchor coefficient should be chosen to adapt to the operator's property" was obtained through the ODE analysis, and this observation is also the contribution of the paper.
> >
> > My score has been modified.

---

### Author Rebuttal · Authors · 2023-08-10

# Common Response

First and foremost, we thank all reviewers for their time and feedback on our paper. We are pleased to see that most of the reviewers found our contributions to be valuable, especially the adaptive anchor method that we obtained using the insight gained through the continuous-time analyses. Although one reviewer expressed concerns regarding novelty, we believe this is perhaps due to a misunderstanding, and we clarify that Nesterov and anchor accelerations are accelerations applied to different/disjoint settings, making them not in competition with each other. We provide further details of the distinction in the individual response, and we hope this resolves the misunderstanding.

---

### Decision · Program_Chairs · 2023-09-21

**Decision:**

Accept (poster)

**Comment:**

The paper conducts an analysis of the continuous time equivalent of the so-called "anchor point acceleration" technique for finding zero points of monotone operators (e.g. stationary points in min-max problems). The analysis sheds light on the choice of anchor point regularization schedule and motivates a novel adaptive version of this schedule. There is a consensus for acceptance across reviewers, with which I concur.